# Benign Overfitting in Single-Head Attention

**Roey Magen**[*]
Weizmann Institute of Science
roey.magen@weizmann.ac.il

**Shuning Shang**[*†]
Princeton University
shuning@princeton.edu

**Zhiwei Xu**
University of Michigan
zhiweixu@umich.edu

**Spencer Frei**[‡]
UC Davis
sfrei@ucdavis.edu

**Wei Hu**[§]
University of Michigan
vvh@umich.edu

**Gal Vardi**[§]
Weizmann Institute of Science
gal.vardi@weizmann.ac.il

## Abstract

The phenomenon of *benign overfitting*, where a trained neural network perfectly fits noisy training data but still achieves near-optimal test performance, has been extensively studied in recent years for linear models and fully-connected/convolutional networks. In this work, we study benign overfitting in a single-head softmax attention model, which is the fundamental building block of Transformers. We prove that under appropriate conditions, the model exhibits benign overfitting in a classification setting already after two steps of gradient descent. Moreover, we show conditions where a minimum-norm/maximum-margin interpolator exhibits benign overfitting. We study how the overfitting behavior depends on the signal-to-noise ratio (SNR) of the data distribution, namely, the ratio between norms of signal and noise tokens, and prove that a sufficiently large SNR is both necessary and sufficient for benign overfitting.

## 1 Introduction

Neural networks often exhibit a remarkable phenomenon, known as *benign overfitting*, where they achieve a perfect fit to noisy training examples and still generalize well to unseen data [1, 2]. This phenomenon contradicts classical wisdom in machine learning, and has become a central research question in the theory of deep learning. Existing works on benign overfitting study under what conditions the phenomenon occurs in different architectures. These works focus on linear models, and on shallow fully-connected and convolutional neural networks.

In recent years, Transformers [3] have emerged as a leading neural network architecture, with impactful applications across a wide range of domains such as natural language processing and computer vision. The fundamental building block of Transformers is the attention mechanism, which allows them to process sequences and focus on different parts of the input. Despite the central role of the attention mechanism, we currently do not understand its overfitting behavior and the conditions under which it exhibits benign overfitting.

In this work, we show benign-overfitting results for the attention mechanism. We consider classification with a single-head softmax attention model, and study the conditions that allow for benign overfitting. In our results, the data distribution consists of multiple tokens: a *signal token*, which can

---

[*]Equal contribution
[†]Work performed while visiting the University of Michigan
[‡]Now at Google DeepMind
[§]Equal advising

39th Conference on Neural Information Processing Systems (NeurIPS 2025).

be used for correctly classifying clean test examples, and *noisy tokens*, which are independent of the label but can be used for interpolating (i.e., perfectly fitting) noisy training examples. We study the signal-to-noise ratio (SNR), namely, the expected ratio between the norms of signal and noise tokens, that allows for benign overfitting.

Below we summarize our main contributions:

- In Theorem 3.3 (Section 3) we show that under appropriate conditions, gradient descent with the logistic loss exhibits benign overfitting already after two iterations. This result holds when the SNR is $\Omega(1/\sqrt{n})$, where $n$ is the number of training samples.

- We then turn to consider other natural learning rules, which allow for benign overfitting under the same requirement on the SNR. In Theorems 4.2 and 4.4 (Section 4), we prove that minimum-norm (i.e., maximum-margin) interpolators exhibit benign overfitting when the SNR is $\Omega(1/\sqrt{n})$.

- In Theorem 4.6 (Section 4), we prove that the above requirement on the SNR is tight. Namely, if the SNR is smaller than it, then the min-norm interpolator exhibits harmful overfitting, where it fits the training data but has poor generalization performance.

- In Section 6, we complement our theoretical results with an empirical study. We show that sufficiently large SNR and input dimension are necessary and sufficient to achieve benign overfitting.

The paper is structured as follows. In Section 2, we provide some preliminaries and define the data distribution and the single-head attention model. In Sections 3 and 4 we state our main results on benign overfitting with gradient descent and with min-norm interpolators. In Section 5 we discuss the main proof ideas, with all formal proofs deferred to the appendix. Finally, in Section 6 we show empirical results.

## 1.1 Related Work

**Optimization in Transformers.** Li et al. [4] provided a theoretical analysis of training a shallow Vision Transformer (ViT) for a classification task. They showed that the sample complexity required to achieve a zero generalization error is correlated with the inverse of the fraction of label-relevant tokens, the token noise level, and the initial model error. Ataee Tarzanagh et al. [5] showed that optimizing the attention layer via gradient descent leads to convergence to an Support Vector Machine (SVM) solution, where the implicit bias of the attention mechanism depends on whether the parameters are represented as a product of key-query matrices or directly as a combined matrix, with different norm-minimization objectives in each case. Ataee Tarzanagh et al. [6] provided a regularization path analysis and proved that the attention weights converge in direction to a max-margin solution that separates locally optimal tokens from non-optimal. They also showed that gradient descent with a specific initialization direction and without optimizing the attention head converges in direction to the same max-margin solution. [7] expanded on their findings by identifying non-trivial data settings for which the convergence of GD is provably global, i.e., without requiring assumptions about the initialization direction. They also provided convergence rate bounds and analysis for optimizing both the attention weights and the attention head, although they did not consider the case of noisy data labels, as we do in our work. Another line of work looks at the learning dynamics of single-layer linear attention models trained on linear regression tasks [8–10]. Additional works that consider optimization dynamics in Transformers include [11, 12].

**Benign overfitting.** A significant body of research has explored why neural networks (NNs) that perfectly interpolate the training data can still generalize well [1, 2]. This has sparked substantial interest in studying overfitting and generalization in NNs trained to fit datasets with noisy labels. The literature on benign overfitting is broad and cannot be reasonably covered here. We refer the reader to the surveys Bartlett et al. [13], Belkin [14]. Most relevant to our work are Cao et al. [15], Kou et al. [16], Meng et al. [17] that studied benign overfitting in convolutional neural networks. Their data distribution resembles ours, as we discuss in Section 2.1. Benign overffiting in fully-connected two-layer neural network classification was studied in Frei et al. [18, 19], Xu et al. [20], Xu and Gu [21], Kornowski et al. [22], George et al. [23], Karhadkar et al. [24] for various activation functions, data distributions and loss functions (both the logistic and the hinge losses). Recently, Jiang et al. [25] studied benign overfitting in a simplified transformer model. However, in contrast to our work,

they do not allow for label-flipping noise, which is a fundamental aspect for understanding whether interpolation is compatible with generalization. Indeed, including label noise is the common setting in the literature on benign overfitting [5], and it plays a key role in our analysis. Concurrent with our study, Sakamoto and Sato [26] also examined benign overfitting (with label noise) in a similar model. They showed that, depending on the step size, there exists a time step at which benign overfitting occurs. However, their approach differs significantly from ours: they do not optimize the attention head and instead assume a strong condition, namely that the angle between the fixed attention head and the signal is bounded below by a constant (see Assumption 3.3 in their paper). Notably, this assumption is highly restrictive; if the attention head is drawn from a standard $d$-dimensional Gaussian, the probability of satisfying this condition decreases exponentially with $d$. In contrast, we optimize both the attention head and the softmax weights, and we show that both learn different patterns for clean and noisy examples. Additionally, we provide an asymptotic analysis.

## 2  Preliminaries

**Notations.** We use bold-face letters to denote vectors and matrices, and let $[m]$ be shorthand for $\{1, 2, \ldots, m\}$. Given a vector $\boldsymbol{x}$, we denote by $x_j$ its $j$-th coordinate. Let $\boldsymbol{I}_d$ be the $d \times d$ identity matrix, and let $\boldsymbol{0}_d$ (or just $\boldsymbol{0}$, if $d$ is clear from the context) denote the zero vector in $\mathbb{R}^d$. We let $\|\cdot\|$ denote the Euclidean norm. We denote a multivariate Gaussian distribution with mean vector $\boldsymbol{\mu}$ and covariance matrix $\boldsymbol{\Sigma}$ by $N(\boldsymbol{\mu}, \boldsymbol{\Sigma})$. We use standard big-Oh notation, with $\Theta(\cdot), \Omega(\cdot), O(\cdot)$ hiding universal constants and $\widetilde{\Theta}(\cdot), \widetilde{\Omega}(\cdot), \widetilde{O}(\cdot)$ hiding constants and factors that are polylogarithmic in the problem parameters. We use $\mathbb{I}(\cdot)$ to denote the indicator variable of an event. For a finite set $\mathcal{A}$, denote the uniform distribution over $\mathcal{A}$ by $\mathsf{Unif}(\mathcal{A})$ and let $|\mathcal{A}|$ be its cardinality.

### 2.1  Data Generation Setting

In this work we focus on the following data distribution:

**Definition 2.1** (clean data distribution). Let $\boldsymbol{\mu}_1, \boldsymbol{\mu}_2 \in \mathbb{R}^d$ such that $\|\boldsymbol{\mu}_1\| = \|\boldsymbol{\mu}_2\| = \rho$ for some $\rho > 0$ and $\langle \boldsymbol{\mu}_1, \boldsymbol{\mu}_2 \rangle = 0$, be two fixed orthogonal vectors representing the signal contained in each data point. Define $\mathcal{D}_{\text{clean}}$ as the distribution over $\mathbb{R}^{T \times d} \times \{\pm 1\}$ of labelled data such that a data point $(\boldsymbol{X}, \widetilde{y})$ is generated by the following procedure:

1. Sample the label $\widetilde{y} \sim \mathsf{Unif}\{\pm 1\}$.

2. Generate a vector $\boldsymbol{u}$, which represents the signal, as follows: If $\widetilde{y} = +1$, set $\boldsymbol{u} = \boldsymbol{\mu}_1$; and if $\widetilde{y} = -1$, set $\boldsymbol{u} = \boldsymbol{\mu}_2$.

3. Generate i.i.d vectors $\boldsymbol{\xi}_2, \ldots, \boldsymbol{\xi}_T$, which represents the noise, from the Gaussian distribution $\boldsymbol{\xi}_\tau \sim \mathcal{N}(\boldsymbol{0}, \boldsymbol{I}_d - \boldsymbol{\mu}_1 \boldsymbol{\mu}_1^\top / \rho^2 - \boldsymbol{\mu}_2 \boldsymbol{\mu}_2^\top / \rho^2)$ for any $\tau \in \{2, \ldots, T\}$.

4. Denote $\boldsymbol{X} = (\boldsymbol{x}^{(1)}, \boldsymbol{x}^{(2)}, \ldots, \boldsymbol{x}^{(T)})^\top$. Select $k \sim \mathsf{Unif}\{1, \ldots, T\}$ and set $\boldsymbol{x}^{(k)} = \boldsymbol{u}$. Set the other tokens $(\boldsymbol{x}^{(1)}, \ldots, \boldsymbol{x}^{(k-1)}, \boldsymbol{x}^{(k+1)} \ldots, \boldsymbol{x}^{(T)})^\top$ to be $\boldsymbol{\xi}_2, \ldots, \boldsymbol{\xi}_T$.

To study the overfitting behavior we also need to introduce label-flipping noise:

**Definition 2.2** (noisy data distribution). Let $\eta \in [0, 1/2)$ be the label flipping probability. We define $\mathcal{D}$ as the distribution over $\mathbb{R}^{T \times d} \times \{\pm 1\}$ which is the $\eta$-label-flipped version of $\mathcal{D}_{\text{clean}}$. Namely, to generate $(\boldsymbol{X}, y) \sim \mathcal{D}$, first generate $(\boldsymbol{X}, \widetilde{y}) \sim \mathcal{D}_{\text{clean}}$, then let $y = \widetilde{y}$ with probability $1 - \eta$ and $y = -\widetilde{y}$ with probability $\eta$.

Our data distribution resembles the distributions considered by Kou et al. [16], Cao et al. [15], Meng et al. [17]. They proved benign overfitting in two-layer convolutional neural networks, and in their setting each data point consists of two patches $\boldsymbol{x}^{(1)}, \boldsymbol{x}^{(2)}$ (rather than $T$ tokens in our setting). Since our single-head attention model is invariant to the order of the tokens, we assume without loss of generality throughout this work that $\boldsymbol{x}^{(1)}$ is the signal token and $\boldsymbol{x}^{(2)}, \ldots \boldsymbol{x}^{(T)}$ are the noisy tokens in all data points. Note that the noise token $\boldsymbol{x}^{(\tau)} = \boldsymbol{\xi}_\tau$ is independent of the label, and that it is

---

[5]Without label noise, many existing benign-overfitting results can be trivially explained through standard uniform convergence arguments (e.g., the classical result of Bartlett et al. [2] on benign overfitting in linear regression).

generated from $\mathcal{N}(\mathbf{0}, \boldsymbol{I}_d - \boldsymbol{\mu}_1\boldsymbol{\mu}_1^\top/\rho^2 - \boldsymbol{\mu}_2\boldsymbol{\mu}_2^\top/\rho^2)$, ensuring that it is orthogonal to the signal vector. Note that when the dimension $d$ is large, $\|\boldsymbol{\xi}_i\| \approx \sqrt{d-2} \approx \sqrt{d}$ by standard concentration bounds. Therefore, we denote the signal-to-noise ratio (SNR) as $\mathrm{SNR} = \|\boldsymbol{\mu}\|/\sqrt{d} = \rho/\sqrt{d}$.

We consider a training dataset $\{(\boldsymbol{X}_i, y_i)\}_{i=1}^n$ of $n$ samples generated i.i.d. from the distribution $\mathcal{D}$. Denote the index set of data whose labels are not flipped by $\mathcal{C} = \{i : \widetilde{y}_i = y_i\}$ ("clean examples"), and the index set of data whose labels are flipped by $\mathcal{N} = \{i : \widetilde{y}_i = -y_i\}$ ("noisy examples"). For indices in $\mathcal{C}$, we further denote $\mathcal{C}_1 := \mathcal{C} \cap \{i : \boldsymbol{x}_i^{(1)} = \boldsymbol{\mu}_1\}, \mathcal{C}_2 := \mathcal{C} \cap \{i : \boldsymbol{x}_i^{(1)} = \boldsymbol{\mu}_2\}$, and define the subsets $\mathcal{N}_1, \mathcal{N}_2$ of $\mathcal{N}$ analogously.

## 2.2 Single-Head Attention Model

Self-attention serves the core building block of transformers. Given an input consisting of $T$ tokens $\boldsymbol{X} = (\boldsymbol{x}^{(1)}, \boldsymbol{x}^{(2)}, \ldots, \boldsymbol{x}^{(T)})^\top \in \mathbb{R}^{T \times d}$, self-attention with key-query matrix $\boldsymbol{W} \in R^{d \times d}$, and value matrix $\boldsymbol{V} \in \mathbb{R}^{d \times k}$, the self-attention model is defined as follows:

$$f(\boldsymbol{X}) = \mathbb{S}(\boldsymbol{X}\boldsymbol{W}\boldsymbol{X}^\top)\boldsymbol{X}\boldsymbol{V},$$

where $\mathbb{S} : \mathbb{R}^d \to \mathbb{R}^d$ is the softmax function. In practice, additional tokens are often appended to the raw input features $\boldsymbol{X}$, and this position is used for the model prediction. For example, a [CLS] token is added for classification purposes [27], and prompt vectors can be appended to adapt pretrained models to new tasks. Let $\boldsymbol{q} \in \mathbb{R}^d$ denote the tunable token ([CLS] token or prompt vector) and concatenate it to $\boldsymbol{X}$ to form $\boldsymbol{X}_q := [\boldsymbol{q} \ \boldsymbol{X}^\top]^\top \in \mathbb{R}^{(T+1) \times d}$. The cross-attention features derived from $\boldsymbol{X}_q$ and $\boldsymbol{X}$ are given by:

$$f(\boldsymbol{X}) = \mathbb{S}(\boldsymbol{X}_q\boldsymbol{W}\boldsymbol{X}^\top)\boldsymbol{X}\boldsymbol{V} = \begin{bmatrix} \mathbb{S}(\boldsymbol{q}^\top\boldsymbol{W}\boldsymbol{X}^\top) \\ \mathbb{S}(\boldsymbol{X}\boldsymbol{W}\boldsymbol{X}^\top) \end{bmatrix} \boldsymbol{X}\boldsymbol{V}, \tag{1}$$

Then we can use the upper term for classification, set $k = 1$ and denote $\boldsymbol{v} = \boldsymbol{V} \in \mathbb{R}^d$. This brings us to our attention model of interest:

$$f(\boldsymbol{X}; \boldsymbol{W}, \boldsymbol{v}) = \boldsymbol{v}^\top\boldsymbol{X}^\top\mathbb{S}(\boldsymbol{X}\boldsymbol{W}^\top\boldsymbol{q}), \tag{2}$$

Here the trained parameters are $\boldsymbol{W}$ and $\boldsymbol{v}$. We note that self-attention with respect to such a tunable token was considered in several other theoretical works (see, e.g., [6, 26]).

In this work, we follow Ataee Tarzanagh et al. [6] and consider the following model:

$$f(\boldsymbol{X}; \boldsymbol{v}, \boldsymbol{p}) = \boldsymbol{v}^\top\boldsymbol{X}^\top\mathbb{S}(\boldsymbol{X}\boldsymbol{p}). \tag{3}$$

Here, the trained parameters are $\boldsymbol{v}, \boldsymbol{p} \in \mathbb{R}^d$. Note that our model corresponds to fixing $\boldsymbol{q} = (1, 0, \ldots, 0)^\top$ in Eq. (2). We note that Ataee Tarzanagh et al. [6] showed that in the model from Eq. (2), gradient iterations on $\boldsymbol{W}$ (with fixed $\boldsymbol{q}$) and on $\boldsymbol{q}$ (after setting $\boldsymbol{W} = \boldsymbol{I}_d$) admit a one-to-one mapping (see Lemma 1 from their paper), and hence the dynamics in models (2) and (3) are essentially similar for any choice of a fixed $\boldsymbol{q}$. Thus, instead of the key-query matrix $\boldsymbol{W}$ we have a vector $\boldsymbol{p}$ that controls the attention. We denote the output of the softmax layer $\mathbb{S}(\boldsymbol{X}_i\boldsymbol{p})$ by $\boldsymbol{s}_i = (s_{i,1}, s_{i,2}, \ldots, s_{i,T})^\top$, and denote the output of the attention layer $\boldsymbol{X}_i^\top\boldsymbol{s}_i$ by $\boldsymbol{r}_i = s_{i,1}\boldsymbol{\mu}_i + s_{i,2}\boldsymbol{\xi}_{i,2} + \cdots + s_{i,T}\boldsymbol{\xi}_{i,T}$, where $0 \leq s_{i,1}, \ldots, s_{i,\tau} \leq 1, s_{i,1} + \cdots + s_{i,\tau} = 1$ are the attention on $T$ tokens of the $i$-th sample.

## 3 Benign Overfitting with Gradient Descent

In this section, we study the joint optimization of the head $\boldsymbol{v}$ and attention weights $\boldsymbol{p}$ using the logistic loss function. We show that the model exhibits benign overfitting after just two iterations of gradient descent (GD). Formally, for a training dataset $\{(\boldsymbol{X}_i, y_i)\}_{i=1}^n$ we define the empirical risk as

$$\mathcal{L}(\boldsymbol{v}, \boldsymbol{p}) = \frac{1}{n}\sum_{i=1}^n \ell(y_i \cdot f(\boldsymbol{X}_i; \boldsymbol{v}, \boldsymbol{p})),$$

where $\ell(z) = \log(1 + \exp(-z))$ is the logistic loss function, and $f$ is the model from Eq. (3). We consider GD optimization. Starting from $\boldsymbol{p}_0 = \mathbf{0}$ and $\boldsymbol{v}_0 = \mathbf{0}$, we have

$$\boldsymbol{v}_{t+1} = \boldsymbol{v}_t - \beta\nabla_{\boldsymbol{v}}\mathcal{L}(\boldsymbol{v}_t, \boldsymbol{p}_t) \text{ and } \boldsymbol{p}_{t+1} = \boldsymbol{p}_t - \beta\nabla_{\boldsymbol{p}}\mathcal{L}(\boldsymbol{v}_t, \boldsymbol{p}_t),$$

where $\beta$ is the step size. When we discuss some fixed $t$, we sometimes write in the subscript "$t = \cdot$", e.g., $\boldsymbol{p}_{t=2}$ instead of $\boldsymbol{p}_2$.

We make the following assumptions:

**Assumption 3.1** (Assumptions for GD with SNR $= \Omega(1/\sqrt{n})$). Let $\delta > 0$ be a desired probability of failure. Let $c_\rho \geq 6(T-1)$ be a parameter that controls the signal strength. For any constant $c_T \geq 2$, there exists a sufficiently large constant $C = C(c_T)$ that may depend on $c_T$, such that the following conditions hold:

1. Number of samples $n$ should be sufficiently large: $n \geq C \log(1/\delta)$.

2. Dimension $d$ should be sufficiently large: $d \geq C \cdot n^2 \log(n/\delta)$.

3. Signal strength is: $\rho = c_\rho \sqrt{d/n}$.

4. Label flipping rate $\eta$: $0 \leq \eta \leq 1/C$.

5. The step size $\beta$ satisfies: $\beta \in [z, 1.02 \cdot z]$ for $z = c_\beta \cdot n/d$, where $c_\beta := c_\beta(\eta, c_\rho, T)$.

6. Initialization at zero: $\|\boldsymbol{v}_0\| = \|\boldsymbol{p}_0\| = 0$.

7. The number of token $T$ satisfies: $2 \leq T \leq c_T$.

Item 1 is required to estimate the number of clean examples compared to noisy examples. The assumption of high dimensionality (Item 2) is important for enabling benign overfitting (see Figure 4 in the appendix), and implies that noise tokens from different training samples are nearly-orthogonal. This assumption appears in many prior works on benign overfitting in neural network classification (e.g., Cao et al. [15], Kou et al. [16], Meng et al. [17], Frei et al. [18, 19], Xu et al. [20], Kornowski et al. [22], Xu and Gu [21]). Item 3 states that the signal-to-noise ratio (SNR) is $\frac{\rho}{\sqrt{d}} = \Omega(1/\sqrt{n})$. In Section 5 we will discuss how the SNR affects the dynamics of GD.

Interestingly, SNR of $\Omega(1/\sqrt{n})$ matches the tight lower bound of the required SNR that allows for benign overfitting with the min-norm (i.e. max-margin) learning rule that we will study in Section 4. Item 4 ensures the flipping rate is small enough to allow the model to learn the signal token. Item 5 is required to achieve benign overfitting after two iterations; with a smaller step size, the model will need more iterations to fit the noisy samples, which we will demonstrate empirically in Section 6. Item 7 ensures that the number of tokens is constant.

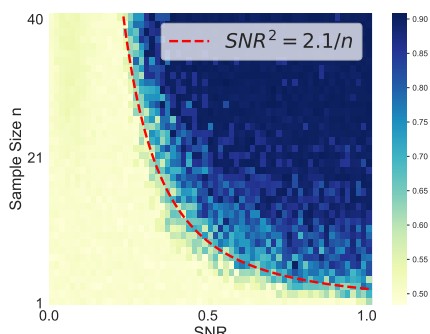

Figure 1: A heatmap of the test accuracy (averaged over 5 runs) after achieving training accuracy $100\%$, plotted across varying signal-to-noise ratios (SNR) and sample sizes ($n$). The red curves represent the expression $\text{SNR}^2 = 2.1/n$. This validates our tight bound of SNR $= \Omega(1/\sqrt{n})$ to achieve benign overfitting, and with a smaller SNR the model exhibits harmful overfitting. Parameters: $d = 900, T = 5, \beta = 0.015, \eta = 0.1$, test sample size $= 2000$.

*Remark* 3.2 (random initialization). The assumption of zero initialization (Item 6) is without loss of generality, as the model is smooth around $\mathbf{0}$. Indeed, the gradient with respect to the softmax weight $\boldsymbol{p}$ is Lipschitz (as shown in Lemma 6 of Ataee Tarzanagh et al. [6]), and the same also holds for the gradient with respect to the attention head $\boldsymbol{v}$, since $f(\boldsymbol{X}; \boldsymbol{v}, \boldsymbol{p})$ is linear in $\boldsymbol{v}$. Consequently, the loss and gradient for any sample under random initialization with zero expectation and sufficiently small variance closely resemble the loss and gradient under zero initialization. Thus, our result can be easily extended to small random initialization. This is also demonstrated empirically in Figure 8 in the appendix.

We now state our main result on benign overfitting with GD:

**Theorem 3.3.** *Suppose that Assumption 3.1 holds. Then, with probability at least $1 - \delta$ over the training dataset, after two iterations of GD we have:*

- *The classifier $\boldsymbol{X} \mapsto \mathrm{sign}(f(\boldsymbol{X}; \boldsymbol{v}_{t=2}, \boldsymbol{p}_{t=2}))$ correctly classifies all training data points:*

$$y_i = \mathrm{sign}(f(\boldsymbol{X}_i; \boldsymbol{v}_{t=2}, \boldsymbol{p}_{t=2})), \ \forall i \in [n].$$

- *The classifier $\boldsymbol{X} \mapsto \mathrm{sign}(f(\boldsymbol{X}; \boldsymbol{v}_{t=2}, \boldsymbol{p}_{t=2})$ generalizes well:*

$$\mathop{\mathbb{P}}_{(\boldsymbol{X},y) \sim \mathcal{D}} (y \neq \mathrm{sign}(f(\boldsymbol{X}; \boldsymbol{v}_{t=2}, \boldsymbol{p}_{t=2}))) \leq \eta + \exp(-\sqrt{n}/2) + \exp\left(-C_1 \eta d/(c_\rho^{0.1} n^{1.5})\right),$$

*where $C_1 := C_1(c_T) > 0$ is a constant. We can also conclude that for the clean-labeled distribution $\mathcal{D}_{clean}$ we have*

$$\mathop{\mathbb{P}}_{(\boldsymbol{X},y) \sim \mathcal{D}_{clean}} (y \neq \mathrm{sign}(f(\boldsymbol{X}; \boldsymbol{v}_{t=2}, \boldsymbol{p}_{t=2}))) \leq \exp(-\sqrt{n}/2) + \exp\left(-C_1 \eta d/(c_\rho^{0.1} n^{1.5})\right),$$

*which approaches zero as $d$ and $n$ grow (see items 1, 2 in assumption 3.1).*

- *High softmax probability for "optimal" tokens:*

$$\forall i \in \mathcal{C}: \ s_{i,1}^{t=2} \geq \frac{1}{1 + (T-1)c_\rho^{2.1}}, \quad \forall i \in \mathcal{N}: \ \sum_{\tau=2}^{T} s_{i,\tau}^{t=2} \geq 1 - \frac{1}{1 + (T-1)c_\rho^2},$$

*where $s_{i,j}^t$ is the softmax probability of the $j^{th}$ token in the $i^{th}$ sample at time $t$.*

The third item in Theorem 3.3 provides insight into how benign overfitting occurs in attention mechanisms. After two iterations of gradient descent, the model assigns enough attention to the signal tokens for clean examples and to the noise tokens for noisy examples. This enables the model to interpolate noisy training examples using the noise tokens while still achieving good generalization performance through the signal tokens.

*Remark* 3.4. When $c_\rho$ is a constant (i.e., the constant $C$ in Assumption 3.1 may also depend on $c_\rho$), the bounds on the attention probabilities can be improved to $s_{i,1}^{t=2} \geq \frac{1}{T}$ for all $i \in \mathcal{C}$.

## 4   Benign Overfitting of Max-Margin Solution

In the previous section we showed that GD exhibits benign overfitting in a setting where the SNR is $\Omega(1/\sqrt{n})$. We now turn to study the overfitting behavior of single-head attention models, when using another learning rule, which returns solutions that interpolate the training data with large margin while keeping the parameters norms small. As we will show, such a learning rule allows us to obtain benign overfitting under the same requirement on the SNR.

We note that learning rules that return min-norm (or max-margin) solutions are considered natural, and hence understanding properties of min-norm interpolators has attracted much interest in recent years, even in settings where the implicit bias of GD does not necessarily lead to a min-norm solution (see, e.g., Savarese et al. [28], Ongie et al. [29], Ergen and Pilanci [30], Hanin [31], Debarre et al. [32], Boursier and Flammarion [33]). More directly related to our work, min-norm interpolation with Transformers has been studied in Ataee Tarzanagh et al. [6, 5], and benign/tempered overfitting in min-norm univariate neural network interpolators has been studied in Joshi et al. [34].

The motivation for analyzing min-norm solutions also arises since they roughly correspond to training using GD with weight decay (which encourages norm minimization). Thus, while in the previous section we showed that GD exhibits benign overfitting after two iterations, in this section our results suggest that GD with weight decay may exhibit benign overfitting also after long training.

We first consider the following learning rule:

$$(\boldsymbol{v}_{(r,R)}, \boldsymbol{p}_{(r,R)}) = \mathop{\mathrm{argmax}}_{\|\boldsymbol{v}\| \leq r, \|\boldsymbol{p}\| \leq R} \min_{i \in [n]} y_i \cdot f(\boldsymbol{X}_i; \boldsymbol{v}, \boldsymbol{p}), \tag{4}$$

where $f$ is the model from (3). The learning rule returns a solution that maximizes the margin $\min_{i \in [n]} y_i \cdot f(\boldsymbol{X}_i; \boldsymbol{v}, \boldsymbol{p})$ under a restriction on the norms. We make the following assumption:

**Assumption 4.1** (Assumptions for max-margin with SNR $= \Omega(1/\sqrt{n})$)**.** Let $\delta \in (0, 0.5)$ be a desired probability of failure. For any constant $c_T \geq 2$ there exists a sufficiently large constant $C = C(c_T)$ that may depend on $c_T$, such that the following conditions hold:

1. Dimension $d$ is sufficiently large: $d \geq Cn^2 \log(n/\delta)$.

2. Number of samples $n$ is sufficiently large: $n \geq C \log(1/\delta)$.

3. Signal strength: $\rho \geq C\sqrt{d/n}$.

4. Label flipping rate: $0 \leq \eta \leq 1/C$.

5. Norm constraint of $\boldsymbol{p}$ satisfies: $R \geq C\sqrt{\eta n/d + 1/\rho^2} \log(T\rho n)$.

6. Number of tokens: $2 \leq T \leq C_T$.

Items 1, 2 and 4 are similar to Assumption 3.1. Item 3 requires $\text{SNR} \geq \Omega(1/\sqrt{n})$, as in Assumption 3.1. We will show later a lower bound on the required SNR for benign overfitting, implying that the $\Omega(1/\sqrt{n})$ bound is tight. Item 5 provides the lower bound for the norm constraint of $\boldsymbol{p}$ so that the model can allocate enough attention on signal tokens to achieve benign overfitting. Note that the norm constraint $r$ for $\boldsymbol{v}$ can take any positive value. Intuitively, since the model is linear in $\boldsymbol{v}$, once $\boldsymbol{p}$ is properly learned, $\boldsymbol{v}$ can achieve accurate classification even with a small norm.

With these assumptions in place, we give our result on benign overfitting with the learning rule (4).

**Theorem 4.2.** *Suppose that Assumption 4.1 holds, and consider the classifier $\boldsymbol{X} \rightarrow \text{sign}(f(\boldsymbol{X}; \boldsymbol{v}_{(r,R)}, \boldsymbol{p}_{(r,R)}))$, where $(\boldsymbol{v}_{(r,R)}, \boldsymbol{p}_{(r,R)})$ is a solution to Problem (4). Then, with probability at least $1 - \delta$ over the training dataset, we have:*

- *The classifier $\text{sign}(f(\boldsymbol{X}; \boldsymbol{v}_{(r,R)}, \boldsymbol{p}_{(r,R)}))$ correctly classifies all training data points:*

$$y_i = \text{sign}(f(\boldsymbol{X}_i; \boldsymbol{v}_{(r,R)}, \boldsymbol{p}_{(r,R)})), \ \forall i \in [n].$$

- *The classifier $\text{sign}(f(\boldsymbol{X}; \boldsymbol{v}_{(r,R)}, \boldsymbol{p}_{(r,R)}))$ generalizes well on test data:*

$$\underset{(\boldsymbol{X},y)\sim\mathcal{D}}{\mathbb{P}}(y \neq \text{sign}(f(\boldsymbol{X}; \boldsymbol{v}_{(r,R)}, \boldsymbol{p}_{(r,R)})))$$

$$\leq \eta + \exp(-\Omega(d/n^2)) + \exp\Big(-\Omega\big(\frac{(1-\zeta)}{\varphi} - \frac{\log(n)}{R}\big)^2\Big),$$

*where $\varphi = \sqrt{\eta n/d + 1/\rho^2}$, $\zeta = \Theta(\varphi \log(T\rho n)/R)$.*

*Remark* 4.3. To see why Theorem 4.2 implies benign overfitting, consider the limit $R \rightarrow \infty$. Then, the upper bound for test error becomes $\eta + \exp(-\Omega(d/n^2)) + \exp(-\Theta((1/\rho^2 + \eta n/d)^{-1}))$, which can be arbitrarily close to $\eta$ if $d$ is large (see Assumption 4.1, item 1).

Next, we consider the following learning rule, which explicitly requires to minimize the parameters norms while allowing interpolation with margin at least $\gamma$:

$$(\boldsymbol{v}_\gamma, \boldsymbol{p}_\gamma) = \underset{\|\boldsymbol{p}\|^2 + \|\boldsymbol{v}\|^2}{\text{argmin}} \ \text{s.t.} \ \min_{i\in[n]} y_i f(\boldsymbol{X}_i; \boldsymbol{v}, \boldsymbol{p}) \geq \gamma \,, \tag{5}$$

where $f$ is the model from Eq. (3). We show that under Assumption 4.1, the solution $(\boldsymbol{v}_\gamma, \boldsymbol{p}_\gamma)$ exhibits benign overfitting for large enough $\gamma$ and $d$:

**Theorem 4.4.** *Suppose that Assumption 4.1 (Items 1 through 4, and 6) holds, and consider the classifier $\boldsymbol{X} \rightarrow \text{sign}(f(\boldsymbol{X}; \boldsymbol{v}_\gamma, \boldsymbol{p}_\gamma))$, where $(\boldsymbol{v}_\gamma, \boldsymbol{p}_\gamma)$ is a solution of Problem (5). Then there exists $\gamma_0$ such that for any $\gamma \geq \gamma_0$, with probability at least $1 - \delta$ over the training dataset, we have:*

- *The classifier $\text{sign}(f(\boldsymbol{X}; \boldsymbol{v}_\gamma, \boldsymbol{p}_\gamma))$ correctly classifies all training data points:*

$$y_i = \text{sign}(f(\boldsymbol{X}_i; \boldsymbol{v}_\gamma, \boldsymbol{p}_\gamma)), \ \forall i \in [n].$$

- *The classifier $\text{sign}(f(\boldsymbol{X}; \boldsymbol{v}_\gamma, \boldsymbol{p}_\gamma))$ generalizes well on test data:*

$$\underset{(\boldsymbol{X},y)\sim\mathcal{D}}{\mathbb{P}}(y \neq \text{sign}(f(\boldsymbol{X}; \boldsymbol{v}_\gamma, \boldsymbol{p}_\gamma))) \leq \eta + \exp(-\Omega(d/n^2)) + \exp(-\Theta((1/\rho^2 + \eta n/d)^{-1})).$$

Thus, for large enough $\gamma$, the theorem implies that the trained model interpolates the training data, and the test error approaches $\eta$ as $d \rightarrow \infty$.

Note that Theorems 4.2 and 4.4 hold only when $\mathrm{SNR} = \Omega(1/\sqrt{n})$. This raises the question: what is the overfitting behavior of min-norm interpolators when the SNR is smaller? We now consider the two-token case where $\rho \leq \sqrt{1/Cn}$ for some sufficiently large universal constant $C$. We will show that in this case, although the model can correctly classify all training samples, the test error of learning rule (4) is at least a universal constant, indicating that benign overfitting does not happen. Formally, we make the following assumptions:

**Assumption 4.5** (Assumptions for max-margin with $\mathrm{SNR} = O(1/\sqrt{n})$)**.** Let $\delta \in (0, 0.5)$ be a desired probability of failure. Consider the case where every sample is composed of two tokens, $\boldsymbol{X}_i = (\boldsymbol{\mu}_i, \boldsymbol{\xi}_i)^\top$. There exists a sufficiently large constant $C$ such that the following hold:

1. Dimension $d$ is sufficiently large: $d \geq Cn^2 \log(n/\delta)$

2. Number of samples $n$ is sufficiently large: $n \geq C \log(1/\delta)$.

3. Signal strength: $\rho \leq \sqrt{d/Cn}$.

4. Label flipping rate is a constant: $\eta \in (0, 1/2)$.

5. The norm of $\boldsymbol{p}$ should be sufficiently large: $R \geq C\sqrt{\frac{n}{d}} \log\left(\frac{n\rho}{d}\right)$.

Compared with Assumption 4.1, the main difference is in the second item, namely that $\mathrm{SNR} \leq O(1/\sqrt{n})$. Additionally, the condition on $\eta$ is relaxed, as in our analysis clean and noisy samples can be treated equivalently when the norm of the signal token is sufficiently small. With these assumptions in place, we can state the following theorem which characterizes the training error and test error of the single-head attention model when the SNR is small:

**Theorem 4.6.** *Suppose that Assumption 4.5 holds, and consider the classifier $\boldsymbol{X} \rightarrow \mathrm{sign}(f(\boldsymbol{X}; \boldsymbol{v}_{(r,R)}, \boldsymbol{p}_{(r,R)}))$, where $(\boldsymbol{v}_{(r,R)}, \boldsymbol{p}_{(r,R)})$ is a solution of Problem (4). Then, with probability at least $1 - \delta$ over the training data, we have:*

- *The classifier $\mathrm{sign}(f(\boldsymbol{X}; \boldsymbol{v}_{(r,R)}, \boldsymbol{p}_{(r,R)}))$ correctly classifies all training data points:*

$$y_i = \mathrm{sign}(f(\boldsymbol{X}_i; \boldsymbol{v}_{(r,R)}, \boldsymbol{p}_{(r,R)})), \ \forall i \in [n].$$

- *The classifier $\mathrm{sign}(f(\boldsymbol{X}; \boldsymbol{v}_{(r,R)}, \boldsymbol{p}_{(r,R)}))$ does not generalize well on test data:*

$$\mathbb{P}_{(\boldsymbol{X},y) \sim \mathcal{D}_{clean}}(y \neq \mathrm{sign}(f(\boldsymbol{X}; \boldsymbol{v}_{(r,R)}, \boldsymbol{p}_{(r,R)}))) \geq \frac{1}{16}.$$

## 5 Proof ideas

**Proof ideas for Section 3.** Here, we discuss the main proof idea of Theorem 3.3. Since the initialization is at zero, $\boldsymbol{v}_t$ is a linear combination of the training data tokens. Specifically, we can express $\boldsymbol{v}_{t=1}$ as $\lambda_1^{t=1} \boldsymbol{\mu}_1 + \lambda_2^{t=1} \boldsymbol{\mu}_2 + \sum_{i=1}^{n} y_i \theta_i^{t=1} \sum_{\tau=2}^{T} \boldsymbol{\xi}_{i,\tau}$, where $\lambda_1^{t=1} > 0, \lambda_2^{t=1} < 0$. Note that $\lambda_1^t > 0, \lambda_2^t < 0$ holds since $|\mathcal{C}| > |\mathcal{N}|$. We begin by analyzing the first step of GD. We show that after one step, the coefficients of $\boldsymbol{v}_{t=1}$ can be estimated as $|\lambda_k^{t=1}| \approx \frac{\beta}{4T}(1 - 2\eta), k \in [2]$ and $\theta_i^{t=1} = \frac{\beta}{2Tn}, i \in [n]$. Moreover, we have $\boldsymbol{p}_{t=1} = 0$, and hence for a training sample $(\boldsymbol{X}_j = (\boldsymbol{\mu}_k, \boldsymbol{\xi}_2, \ldots, \boldsymbol{\xi}_T), y_j)$, the margin is:

$$y_j f(\boldsymbol{X}_j; \boldsymbol{v}_{t=1}, \boldsymbol{p}_{t=1}) = \frac{1}{T} y_j \boldsymbol{v}_{t=1}^\top (\boldsymbol{x}_j^{(1)} + \cdots + \boldsymbol{x}_j^{(T)}) \approx \frac{1}{T} y_j \lambda_k^{t=1} \|\boldsymbol{\mu}_k\|^2 + \frac{1}{T} \theta_j^{t=1} \sum_{\tau=2}^{T} \|\boldsymbol{\xi}_{j,\tau}\|^2,$$

where in the last approximate equality we use the high dimensional setting (i.e. by item 2 in our assumption $d \gg n^2 \log(n)$) to neglect the $\sum_{i,\tau,\tau':(i,\tau)\neq(j,\tau')} y_i y_j \theta_j^{t=1} \boldsymbol{\xi}_{i,\tau}^\top \boldsymbol{\xi}_{j,\tau'}$ term, since it is much smaller (in absolute value) than the other terms. Indeed, we have w.h.p. that $|\boldsymbol{\xi}_{i,\tau}^\top \boldsymbol{\xi}_{j,\tau'}| \leq \sqrt{d} \log(n)$, $\|\boldsymbol{\xi}_{j,\tau}\|^2 \approx d$ and recall that $\|\boldsymbol{\mu}_k\|^2 = c_\rho^2(d/n)$ (item 3 in our assumption). Therefore, for a clean sample $j \in \mathcal{C}$ and large enough $c_\rho$, the margin is $y_j f(\boldsymbol{X}_j; \boldsymbol{v}_{t=1}, \boldsymbol{p}_{t=1}) \approx c_\beta > 0$, where $c_\beta$ is a parameter that controls the step size $\beta$. On the other hand, for a noisy sample $j \in \mathcal{N}$, we have $y_j f(\boldsymbol{X}_j; \boldsymbol{v}_{t=1}, \boldsymbol{p}_{t=1}) \approx -c_\beta < 0$. This implies that after one iteration of GD the classifier $\mathrm{sign}(f(\boldsymbol{X}; \boldsymbol{v}_{t=1}, \boldsymbol{p}_{t=1}))$ does not correctly classify noisy training samples, but still correctly classifies

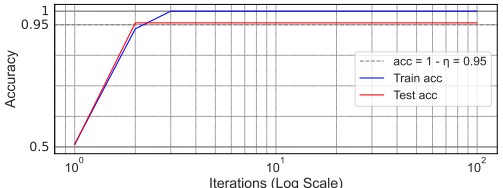 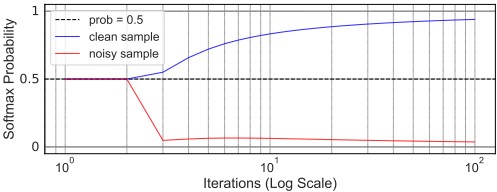

(a) Train and test accuracies.      (b) Softmax probabilities.

Figure 2: The top panel shows the train and test accuracies during training. It shows that benign overfitting occurs after 2 iterations. After the first iteration, the model correctly classifies the clean training examples, but not the noisy ones. In the bottom panel, we show the softmax probability of the signal token for clean and noisy samples (average of the softmax probabilities $s_{j,1}^t$ over $\mathcal{C}$ and $\mathcal{N}$ respectively). We see that after 2 iterations, the attention focuses on signal tokens for clean examples, and on noise tokens for noisy examples. This aligns with Theorem 3.3 and Remark 3.4. Parameters: $n = 200, d = 40000, T = 2, \beta = 0.025, \rho = 30, \eta = 0.05,$ test sample size $= 2000$.

clean training samples. Together with $\boldsymbol{p}_{t=1} = \boldsymbol{0}$, the classifier $\text{sign}(f(\boldsymbol{X}; \boldsymbol{v}_{t=1}, \boldsymbol{p}_{t=1}))$ will also correctly classify, with high probability, a clean test sample. Moreover, since the loss function $\ell$ is decreasing, the loss of noisy samples, denoted $\ell_{t=1,j}, j \in \mathcal{N}$, dominates the loss of clean samples $\ell_{t=1,i}, i \in \mathcal{C}$. This implies that after two iterations, the coefficients $|\theta_j^{t=2}|, j \in \mathcal{N}$, of the noisy tokens in $\boldsymbol{v}_{t=2}$, corresponding to noisy samples, grow faster than the coefficients $|\lambda_i^{t=2}|$ of the first (signal) tokens. This property is important to allow for interpolation of noisy examples. We also show that $\boldsymbol{p}_{t=2}$ focuses on optimal tokens, namely, on noisy tokens for noisy samples (i.e. $\sum_{\tau=2}^T s_{i,\tau}^{t=2} \geq 1/1 + (T-1)c_\rho^2, \forall i \in \mathcal{N}$), and on the signal token for clean training and test samples. Using this property we conclude that the model parameterized by $(\boldsymbol{v}_{t=2}, \boldsymbol{p}_{t=2})$ exhibits benign overfitting.

*Remark* 5.1. Note that our proof implies the following behavior of GD. After the first iteration, the model correctly classifies only the clean training samples, resulting in an expected training accuracy of $1 - \eta$. Additionally, the model successfully classifies a clean test sample w.h.p., leading to the same expected test accuracy. After the second iteration, the model interpolates the training data, achieving a training accuracy of $1$. This is shown empirically in Figure 2. When using a smaller step size, we empirically observe a similar trend: after the first iteration, the model learns the signal tokens, and with more iterations, it captures the noisy tokens of the noisy samples and fits the entire dataset. This behavior is shown in Figure 3.

**Proof ideas for Section 4.** We now discuss proof idea for Theorem 4.2. Consider the behavior of $\boldsymbol{p}_{(r,R)}$ and $\boldsymbol{v}_{(r,R)}$ as $R, r \to \infty$. The main insight is that $\boldsymbol{p}$ converges to a direction that focuses on signal tokens for clean samples, and $\boldsymbol{v}$ approximates the corresponding max-margin classifier over the attention outputs. First, we consider the attention output $\boldsymbol{r}_i = \boldsymbol{X}_i^\top \mathbb{S}(\boldsymbol{X}_i \boldsymbol{p})$ as a selection of signal and noise tokens based on softmax probabilities. We define a learning rule that selects the signal token $\boldsymbol{\mu}_k$ for clean samples and a fixed noise token $\boldsymbol{\xi}_{i,2}$ for noisy ones. Then we show that any $\boldsymbol{p}$ which is not aligned with this rule (that is, does not select clean tokens for clean samples) leads to a strictly smaller margin. This implies that the optimal solution also tends to select signal tokens for clean samples. Since test data shares the same signal tokens, the attention output on test samples also concentrates on the signal token when $R$ is large. Combined with a near-maximal margin vector $\boldsymbol{v}_{(r,R)}$, the model predicts correctly on test samples with high probability.

## 6 Experiments

We complement our theoretical results with an empirical study on benign overfitting in single-head softmax attention. We trained single-head softmax attention models (Eq. (3)) on data generated as specified in Section 2.1 using GD with a fixed step size and the logistic loss function. In all figures, the x-axis corresponds to the time and has a log scale. We add $1$ to the time so that the initialization $t = 0$ can be shown in the log scale (i.e. iteration $10^0$ is the initialization).

In Figure 2, we consider a setting similar to Theorem 3.3, and demonstrate that benign overfitting occurs after two iterations, and that the behavior of GD aligns with our discussion in Remark 5.1. We also plot how the softmax probabilities evolve during training, and see after two iterations a behavior similar to the last item of Theorem 3.3 and Remark 3.4. In Figure 3 (Appendix A.4), we

consider a similar setting, but with a smaller step size. Here, benign overfitting occurs after about 150 iterations. In Figure 1, we present a heatmap of the test accuracy across varying SNR and sample sizes, validating the SNR threshold of $\Theta(1/\sqrt{n})$ established in this work. Additional experiments are provided in Section A.4, including investigation of the overfitting behavior for different dimensions, self-attention w.r.t. the first token, multi-layer transformers, GD with weight decay (which encourages norm minimization, as in our learning rule from Section 4), and experiments with real-world datasets (MNIST and CIFAR-10). These experiments demonstrate that our results capture the overfitting behavior also in more complex settings.

## Acknowledgments and Disclosure of Funding

GV is supported by the Israel Science Foundation (grant No. 2574/25), by a research grant from Mortimer Zuckerman (the Zuckerman STEM Leadership Program), and by research grants from the Center for New Scientists at the Weizmann Institute of Science, and the Shimon and Golde Picker – Weizmann Annual Grant. Part of this work was done while ZX, SF, and WH were visiting the Simons Institute for the Theory of Computing.

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

# A  Appendix

*Remark* A.1. Throughout our proofs, we assume without loss of generality that $\boldsymbol{\mu}_1 = (\rho, 0, 0, ..., 0)^\top$, $\boldsymbol{\mu}_2 = (0, \rho, 0, ..., 0)^\top$ and $\boldsymbol{\xi}_i = (0, 0, \boldsymbol{\xi}^\top)$ for $\boldsymbol{\xi} \sim \mathcal{N}(\mathbf{0}, \boldsymbol{I}_{d-2})$. Indeed, since $\boldsymbol{\mu}_1$ and $\boldsymbol{\mu}_2$ are orthogonal, we can find orthogonal matrix $\boldsymbol{A} \in \mathbb{R}^{d \times d}$ such that $\boldsymbol{A}\boldsymbol{\mu}_1 = (\rho, 0, 0, ..., 0)^\top, \boldsymbol{A}\boldsymbol{\mu}_2 = (0, \rho, 0, ..., 0)^\top$ and $\boldsymbol{A}\boldsymbol{\xi}_i \sim \mathcal{N}(\boldsymbol{A}\mathbf{0}, \boldsymbol{A}(\boldsymbol{I}_d - \boldsymbol{\mu}_1\boldsymbol{\mu}_1^\top/\rho^2 - \boldsymbol{\mu}_2\boldsymbol{\mu}_2^\top/\rho^2)\boldsymbol{A}^\top)$, which mean that $\boldsymbol{A}\boldsymbol{\xi}_i = (0, 0, \boldsymbol{\xi}^\top)$ for $\boldsymbol{\xi} \sim \mathcal{N}(\mathbf{0}, \boldsymbol{I}_{d-2})$. We emphasize that an orthogonal transformation does not affect our results.

## A.1   Proofs for Sec. 3

### A.1.1   Notations for Sec. 3.

Given $a, b, c \in \mathbb{R}$, we denote by $c(a \pm b)$ the close segment $[c(a-b), c(a+b)]$. Given vector $\boldsymbol{x}$, we denote by $\boldsymbol{x}[i]$ the $i^{\text{th}}$ coordinate of $\boldsymbol{x}$, and $\boldsymbol{x}[i : j]$ denotes the subvector containing the elements from the $i^{\text{th}}$ to the $j^{\text{th}}$, inclusive. We also list some key notations used in this section for convenience.

Table 1: Usefull notation.

| | |
|---|---|
| $\boldsymbol{x}_{i,j}$ | $j^{\text{th}}$ token in the $i^{\text{th}}$ sample |
| $\gamma_{i,j}^t$ | $y_i \boldsymbol{v}_t^\top \boldsymbol{x}_{i,j}$ i.e. $j^{\text{th}}$ token score in time $t$ |
| $\alpha_{i,j}^t$ | softmax probability of the $j^{\text{th}}$ token in the $i^{\text{th}}$ sample in time $t$ |
| $\ell_{t,i}$ | $\ell(\boldsymbol{X}_i; \boldsymbol{v}_t, \boldsymbol{p}_t)$ |

We remind that $\mathcal{C}, \mathcal{N} \subseteq [n]$ denotes the indices of clean and noisy training examples, and $\mathcal{C}_k, \mathcal{N}_k$ denotes the clean and noisy examples from cluster $k \in \{1, 2\}$. For example if $i \in \mathcal{C}_1$, then $x_{i,1} = \mu_1$ and $y_1 = 1$, and for $j \in \mathcal{N}_1$ we have that $x_{j,1} = \mu_1$ and $y_1 = -1$. Let $\mathbb{S}'(\boldsymbol{v}) := \nabla\mathbb{S}(\boldsymbol{v}) = \text{diag}(\mathbb{S}(\boldsymbol{v})) - \mathbb{S}(\boldsymbol{v})\mathbb{S}(\boldsymbol{v})^\top$ denote the Jacobian of the softmax function $\mathbb{S}(\boldsymbol{v})$ at $\boldsymbol{v} \in \mathbb{R}^d$.

### A.1.2 Additional Lemmas & Definitions for Sec 3.

The following equations will be useful throughout the proof:

$$\nabla_{\boldsymbol{v}}\mathcal{L}(\boldsymbol{v},\boldsymbol{p}) = \frac{1}{n}\sum_{i=1}^{n}\ell_i' \cdot y_i \boldsymbol{X}_i^\top \mathbb{S}(X_i\boldsymbol{p}) \tag{6}$$

$$\nabla_{\boldsymbol{p}}\mathcal{L}(\boldsymbol{v},\boldsymbol{p}) = \frac{1}{n}\sum_{i=1}^{n}\ell_i' \cdot \boldsymbol{X}_i^\top \mathbb{S}'(X_i\boldsymbol{p})\boldsymbol{\gamma}_i, \ \text{ where } \ \boldsymbol{\gamma}_i = y_i\boldsymbol{v}^\top \boldsymbol{X}_i \tag{7}$$

$$\ell'(x) = -1/(1+\exp(x)) \tag{8}$$

$$\mathbb{S}'(\boldsymbol{v}) = \mathrm{diag}(\mathbb{S}(\boldsymbol{v})) - \mathbb{S}(\boldsymbol{v})\mathbb{S}(\boldsymbol{v})^\top \tag{9}$$

**Definition A.2** (Good Training Set). We say that a training set $(\boldsymbol{X}_1,\ldots,\boldsymbol{X}_n)$ is *good* if exsists some universal constant $c_D$ (that may depends just on the number of tokens $T$) s.t.

- $\|\boldsymbol{\xi}_{i,\tau}\|_2^2 \in (1 \pm o_n(1))d$, for all $i \in [n], \tau \in \{2,\ldots,T\}$.

- $|\langle\boldsymbol{\xi}_{i,\tau},\boldsymbol{\xi}_{j,\tau'}\rangle| \le c_D \cdot \sqrt{d\log(n/\delta)}$, for any $i,j \in [n], \tau,\tau' \in \{2,\ldots,T\}$ such that $(i,\tau) \neq (j,\tau')$.

- $|\mathcal{N}_k| \in \frac{n}{2}(\eta \pm o_n(1))$ and $|\mathcal{C}_k| = \frac{n}{2}(1 - \eta \pm o_n(1))$, for $k \in \{1,2\}$.

**Definition A.3** (Good Test Sample). We say that a test sample $(\boldsymbol{X} = (\boldsymbol{x}_1,\boldsymbol{x}_2,\ldots,\boldsymbol{x}_T),y)$ is *good* w.r.t. a training set $(\boldsymbol{X}_1,\ldots,\boldsymbol{X}_n)$ and $C_1$ if

$$|\langle\boldsymbol{x}_{i,\tau},\boldsymbol{x}_{\tau'}\rangle| \le \frac{d}{C_1\sqrt{n}}, \quad \forall i \in [n], \tau,\tau' \in \{2,\ldots,T\} \ \text{ s.t. } \ \tau \neq \tau'$$

Next we write Lemma A.68 slightly different, and also add a formal proof for completeness:

**Lemma A.4.** *Let $\delta > 0$ and $C > 0$. Suppose that Assumption 3.1 (item 1) holds with constant $C$, then with probability at least $1 - \delta/2$ we have that*

$$|\mathcal{C}_k| \in \frac{n}{2}(1 - \eta \pm \sqrt{2/C}), \quad |\mathcal{N}_k| \in \frac{n}{2}(\eta \pm \sqrt{2/C}), \quad \forall k \in \{1,2\}.$$

*Moreover, we have*

$$|\mathcal{C}_k| \in \frac{n}{2}(1 - \eta \pm o_n(1)), \quad |\mathcal{N}_k| \in \frac{n}{2}(\eta \pm o_n(1)), \quad \forall k \in \{1,2\}.$$

*Proof.* By Hoeffding's inequality,

$$\mathbb{P}\left(\left||\mathcal{C}_j| - \frac{n}{2}(1-\eta)\right| \ge \sqrt{n\log(16/\delta)/2}\right) \le \delta/8,$$

which means that with probability at least $1 - \delta/8$ we have that $|\mathcal{C}_j| \in \frac{n}{2}(1 - \eta \pm c_n)$, where $c_n = \sqrt{2n\log(16/\delta)}/n$. Hence, if $n \ge C\log(16/\delta)$, then $c_n = \sqrt{2\log(16/\delta)}/\sqrt{n} \le \sqrt{2/C}$. Similarly, we can estimate $|\mathcal{N}_k|$ for $k \in \{1,2\}$, and by union bound, the result follows. $\square$

The next lemma A.5 allows us to analyze $\nabla_{\boldsymbol{p}}\mathcal{L}$ as a function of the score gap.

**Lemma A.5.** *Let $\boldsymbol{z},\boldsymbol{\gamma},\boldsymbol{p} \in \mathbb{R}^T$ and let $\boldsymbol{\alpha} = \mathbb{S}(\boldsymbol{p})$. Define $\gamma_{min} := \min_{\tau \ge 2}\gamma_\tau$, $\gamma_{max} := \max_{\tau \ge 2}\gamma_\tau$, $\gamma := (\gamma_{min} + \gamma_{max})/2$ and $\epsilon := (\gamma_{max} - \gamma_{min})/2$. Then*

$$\boldsymbol{z}^T\mathbb{S}'(\boldsymbol{p})\boldsymbol{\gamma} \in (\gamma_1 - \gamma)(1-\alpha_1)\alpha_1\left(z_1 - \frac{\sum_{i=2}^{T}z_i\alpha_i}{1-\alpha_1}\right) \pm \epsilon\left(2\sum_{i=2}^{T}z_i\alpha_i + \alpha_1\sum_{i=2}^{T}z_i\alpha_i + (1-\alpha_1)\alpha_1 z_1\right)$$

*Proof.* Observe that $\sum_{i=1}^{T} \alpha_i = 1$. Therefore,

$$\boldsymbol{z}^T \mathbb{S}'(\boldsymbol{p})\boldsymbol{\gamma} = \boldsymbol{z}^T \mathrm{diag}(\boldsymbol{\alpha})\boldsymbol{\gamma} - \boldsymbol{z}^T \boldsymbol{\alpha}\boldsymbol{\alpha}^\top \boldsymbol{\gamma} = \sum_{i=1}^{T} z_i \gamma_i \alpha_i - \sum_{i=1}^{T} z_i \alpha_i \sum_{i=1}^{T} \gamma_i \alpha_i$$

$$\in z_1 \gamma_1 \alpha_1 + (\gamma \pm \epsilon)\sum_{i=2}^{T} z_i \alpha_i - \left( z_1 \alpha_1 + \sum_{i=2}^{T} z_i \alpha_i \right)\left( \gamma_1 \alpha_1 + (\gamma \pm \epsilon)\sum_{i=2}^{T} \alpha_i \right)$$

$$= \left( (\gamma \pm \epsilon) - \left( \alpha_1 \gamma_1 + (\gamma \pm \epsilon)\sum_{i=2}^{T} \alpha_i \right) \right)\sum_{i=2}^{T} z_i \alpha_i + \left( \gamma_1 - \left( \alpha_1 \gamma_1 + (\gamma \pm \epsilon)\sum_{i=2}^{T} \alpha_i \right) \right)\alpha_1 z_1$$

$$= \left( (\gamma \pm \epsilon) - (\alpha_1 \gamma_1 + (\gamma \pm \epsilon)(1 - \alpha_1)) \right)\sum_{i=2}^{T} z_i \alpha_i + \left( \gamma_1 - (\alpha_1 \gamma_1 + (\gamma \pm \epsilon)(1 - \alpha_1)) \right)\alpha_1 z_1$$

$$= (\alpha_1(\gamma \pm \epsilon) - \alpha_1 \gamma_1 \pm 2\epsilon)\sum_{i=2}^{T} z_i \alpha_i + (1 - \alpha_1)(\gamma_1 - \gamma \pm \epsilon)\alpha_1 z_1$$

$$= \alpha_1 (\gamma - \gamma_1)\sum_{i=2}^{T} z_i \alpha_i + (1 - \alpha_1)(\gamma_1 - \gamma)\alpha_1 z_1 \pm \epsilon \left( 2\sum_{i=2}^{T} z_i \alpha_i + \alpha_1 \sum_{i=2}^{T} z_i \alpha_i + (1 - \alpha_1)\alpha_1 z_1 \right)$$

$$= (\gamma_1 - \gamma)(1 - \alpha_1)\alpha_1 \left( z_1 - \frac{\sum_{i=2}^{T} z_i \alpha_i}{1 - \alpha_1} \right) \pm \epsilon \left( 2\sum_{i=2}^{T} z_i \alpha_i + \alpha_1 \sum_{i=2}^{T} z_i \alpha_i + (1 - \alpha_1)\alpha_1 z_1 \right)$$

$\square$

We will show that in our setting the score difference between noisy tokens (i.e. $\epsilon$ from Lemma A.5) is relatively small and thus the second term in Lemma A.5 is negligible compare to the first term.

*Remark* A.6. To prove Thm. 3.3, we demonstrate that $\nabla_{\boldsymbol{p}}\mathcal{L}$ can be expressed as a function of the score gap between the optimal token to the noisy tokens. Specifically as a function of $\gamma_{i,1} - \gamma_{i,\tau}$, where $\boldsymbol{\gamma}_i := y_i \boldsymbol{v}^\top \boldsymbol{X}_i$ is the vector score of the $i$ sample, and with some additive term that depends on the score gap between any two distinct noisy tokens, defined as $\epsilon_i := \max_{\tau \neq \tau'} |\gamma_{i,\tau} - \gamma_{i,\tau'}|$ (see Lemma A.5). We establish that in our case $\epsilon_i$ is relatively small. This technique is noteworthy on its own, as it enables the analysis of softmax weights in a multiple-token setting without relying on potentially unnatural assumptions, such as all non-optimal tokens having identical scores [6, 7] or the presence of a single noisy token with a larger norm compared to other noisy tokens [25].

**Lemma A.7.** *Let $x_1, x_2, \ldots, x_n$ be independent random variables such that $\mathbb{E}[x_i] = 0$ and $x_i \in [-b, b]$ almost surely. Consider the sum of these random variable $S_n = x_1 + \cdots + x_n$. Then Hoeffding's theorem states that*

$$\Pr[S_n \geq n^{0.75}b] \leq \exp(-2n^{1.5}b^2/4b^2 n) = \exp(-n^{0.5}/2)$$

### A.1.3 Proof of Thm. 3.3

*Proof.* To simplify the proof, we express the step size $\beta$ in an alternative form:

$$c_\beta := C_1 \sqrt{\log(c_\rho)/\eta} \ \text{ where } \ C_1 \in \left[ \sqrt{\frac{16T^3}{0.998(T-1)}}, 1.02 \cdot \sqrt{\frac{16T^3}{0.998(T-1)}} \right]$$

$$\beta = c_\beta \cdot n/(c_\rho^2 \cdot d), \tag{10}$$

which is equivalent to Item 5. We emphasize that $c_\beta$ can be arbitrarily larger than any constant whenever $\eta$ is small enough i.e. $C$ from Assumption 3.1 is large enough.

Next, under Assumption 3.1, we argue that with probability at least $1 - \delta$ the training set is good (Def. A.2) i.e.:

- $|\mathcal{C}_k| \in \frac{n}{2}(\eta \pm o_n(1))$ and $\mathcal{N}_k \in \frac{n}{2}(1 - \eta \pm o_n(1))$, for $k \in \{1, 2\}$.

- $\|\boldsymbol{\xi}_{i,\tau}\|_2^2 \in (1 \pm o_n(1))d$, for any $i \in [n], \tau \in \{2, \ldots, T\}$.

- $|\langle \boldsymbol{\xi}_{i,\tau}, \boldsymbol{\xi}_{j,\tau'} \rangle| \le c_D \cdot \sqrt{d \log(n/\delta)}$, for any $i, j \in [n], \tau, \tau' \in \{2, \dots, T\}$ such that $(i, \tau) \neq (j, \tau')$,

where $c_D$ is some universal constant. Indeed, this holds by Lemma A.65, Lemma A.4, and the union bound. We emphasize that the notation $o_n(1)$ represents a term that becomes arbitrarily small as $n$ increases, and thus it can be bounded by a small constant if $C$ from Assumption 1 is large enough.

Next, we show that under a good training set, the model exhibits benign overfitting, already after two iterations. See Remark A.1 for the data setting used throughout the proof.

**GD after 1 iteration.** We start by analyzing the first coordinate of $\boldsymbol{v}_1$ (i.e. $\boldsymbol{v}$ after one iteration of GD). By assumption 3.1 (item 6), we have that $\boldsymbol{p}_0 = \boldsymbol{v}_0 = \boldsymbol{0}$, which implies that $\ell'_{0,i} = -1/2$, for any $i \in [n]$. Hence

$$
\begin{aligned}
-\beta \nabla_{\boldsymbol{v}} \mathcal{L}(\boldsymbol{v}_0, \boldsymbol{p}_0)[1] = -\frac{\beta}{Tn} \sum_{i=1}^{n} \ell'_{0,i} \cdot y_i \boldsymbol{x}_{i,1}[1] &= \frac{\beta}{2Tn} \sum_{i \in \mathcal{C}_1} y_i \rho + \frac{\beta}{2Tn} \sum_{i \in \mathcal{N}_1} y_i \rho \\
&= \frac{\beta}{2Tn}(|\mathcal{C}_1| - |\mathcal{N}_1|)\rho \\
&\in \frac{\beta}{4T}(1 - 2\eta \pm o_n(1))\rho \qquad \text{``good'' training set}
\end{aligned}
$$

In the same way, we can estimate the second coordinate of $\boldsymbol{v}_{t=1}$:

$$
\boldsymbol{v}_{t=1}[2] = \frac{\beta}{2Tn} \sum_{i \in \mathcal{C}_2} y_i \rho + \frac{\beta}{2Tn} \sum_{i \in \mathcal{N}_2} y_i \rho \in -\frac{\beta}{4T}(1 - 2\eta \pm o_n(1))\rho,
$$

where we remind that $y_i = -1$, when $i \in \mathcal{C}_2$, hence $\boldsymbol{v}_{t=1}[2]$ has the same bounds as $\boldsymbol{v}_{t=1}[1]$, just with opposite sign. We move to analyze the rest of the coordinates of $\boldsymbol{v}_{t=1}$:

$$
\boldsymbol{v}_t[3 : d] = \frac{\beta}{2Tn} \sum_{i=1}^{n} y_i \sum_{\tau=2}^{T} \boldsymbol{\xi}_{i,\tau}.
$$

Overall, we can write $\boldsymbol{v}_{t=1}$ as $\lambda_1^{t=1} \boldsymbol{\mu}_1 + \lambda_2^{t=1} \boldsymbol{\mu}_2 + \sum_{i=1}^{n} y_i \theta_i^{t=1} \sum_{\tau=2}^{T} \boldsymbol{\xi}_{i,\tau}$ with

$$
\lambda_1^{t=1} \in \frac{\beta}{4T}(1 - 2\eta \pm o_n(1)), \quad \lambda_2^{t=1} \in -\frac{\beta}{4T}(1 - 2\eta \pm o_n(1)), \quad \theta_i^{t=1} = \frac{\beta}{2Tn}. \qquad (11)
$$

Moreover, since $\boldsymbol{\gamma}_i^{t=0} = \boldsymbol{0}$ for every $i \in [n]$, we have that $\boldsymbol{p}_1 = \boldsymbol{0}$ (see Eq. 7).

**Preparation for next iteration.** To estimate $(\boldsymbol{v}_{t=2}, \boldsymbol{p}_{t=2})$, we first need to estimate the loss for clean/noisy samples and the score $\gamma_{i,\tau}$ (see Table 1).

We remind that $\|\boldsymbol{\mu}_j\|^2 = \rho^2 = c_\rho^2 d/n$ (Assumption 3.1 (item 3)). For $j \in \mathcal{C}_k$, where $k \in \{1, 2\}$ we have that

$$
\begin{aligned}
y_j f(\boldsymbol{X}_j; \boldsymbol{v}_{t=1}, \boldsymbol{p}_{t=1}) &= \frac{1}{T} \cdot y_j \boldsymbol{v}_{t=1}^\top \sum_{\tau=1}^{T} \boldsymbol{x}_{j,\tau} \qquad \text{since } \boldsymbol{p}_1 = \boldsymbol{0} \\
&\in \frac{1}{T} |\lambda_k^{t=1}| \|\boldsymbol{\mu}_k\|^2 + \frac{1}{T} \theta_j^{t=1} \sum_{\tau=1}^{T-1} \|\boldsymbol{\xi}_{j,\tau}\|^2 \pm \frac{\beta}{n} o_n(d) \quad y_j \lambda_k^{t=1} > 0 \quad (12)
\end{aligned}
$$

where the last inequality holds since the training set is "good" and $T$ is a constant i.e.

$$
\sum_{i,\tau,\tau':(i,\tau) \neq (j,\tau')} \boldsymbol{\xi}_{i,\tau}^\top \boldsymbol{\xi}_{j,\tau'} \in \pm o_n(1) \cdot d.
$$

Since the training set is "good" then by Eq. 11, we can bound $y_j f(\boldsymbol{X}_j; \boldsymbol{v}_{t=1}, \boldsymbol{p}_{t=1})$ as follows:

$$
\begin{aligned}
y_j f(\boldsymbol{X}_j; \boldsymbol{v}_{t=1}, \boldsymbol{p}_{t=1}) &\leq \frac{\beta}{4T^2}(1 - 2\eta + o_n(1)) \cdot c_\rho^2 \cdot \frac{d}{n} + \frac{\beta(T-1)}{2T^2 n}d(1 + o_n(1)) + \frac{\beta}{n} \cdot o_n(d) \\
&\leq \left( \frac{c_\rho^2(1 - 2\eta) + 2(T-1) + o_n(1)}{4T^2} \right) \cdot \frac{\beta d}{n} \qquad \text{Assumption 3.1 (item 2)} \\
&= c_\beta \cdot \left( \frac{(1 - 2\eta) + 2(T-1)/c_\rho^2 + o_n(1)}{4T^2} \right) \qquad \text{Eq. 10} \\
&\leq \frac{1.1 c_\beta}{4T^2},
\end{aligned}
\tag{13}
$$

where the last inequality holds since $c_\rho \geq 5(T-1)$, which implies that $2(T-1)/c_\rho^2 + o_n(1) \leq 0.1$. Similarly, we have that

$$
\begin{aligned}
y_j f(\boldsymbol{X}_j; \boldsymbol{v}_{t=1}, \boldsymbol{p}_{t=1}) &\geq \frac{\beta}{4T^2}(1 - 2\eta - o_n(1)) \cdot c_\rho^2 \cdot \frac{d}{n} + \frac{\beta}{2T^2 n}d(1 - o_n(1)) - \frac{\beta}{n}o(d) \\
&\geq \left( \frac{c_\rho^2(1 - 2\eta) + 2(T-1) - o_n(1)}{4T^2} \right) \cdot \frac{\beta d}{n} \\
&= c_\beta \cdot \left( \frac{(1 - 2\eta) + 2(T-1)/c_\rho^2 - o_n(1)}{4T^2} \right) \\
&\geq \frac{0.9 c_\beta}{4T^2}
\end{aligned}
\tag{14}
$$

For $j \in \mathcal{N}_k$, where $k \in \{1, 2\}$ we have that

$$
\begin{aligned}
y_j f(\boldsymbol{X}_j; \boldsymbol{v}_{t=1}, \boldsymbol{p}_{t=1}) &= \frac{1}{T} \cdot y_j \boldsymbol{v}_{t=1}^\top \sum_{\tau=1}^{T} \boldsymbol{x}_{j,\tau} \qquad\qquad\qquad \text{since } \boldsymbol{p}_1 = \boldsymbol{0} \\
&\in -\frac{1}{T}|\lambda_k^{t=1}| \|\boldsymbol{\mu}_k\|^2 + \frac{1}{T}\theta_j^{t=1} \sum_{\tau=1}^{T-1} \|\boldsymbol{\xi}_{j,\tau}\|^2 \pm \frac{\beta}{n}o(d) \quad y_j \lambda_k^{t=1} > 0 \quad (15)
\end{aligned}
$$

where the last inequality holds since that the training set is "good" and $T$ is a constant. Since the training set is "good" then by Eq. 11, we can bound $y_j f(\boldsymbol{X}_j; \boldsymbol{v}_{t=1}, \boldsymbol{p}_{t=1})$ as follows:

$$
\begin{aligned}
y_j f(\boldsymbol{X}_j; \boldsymbol{v}_{t=1}, \boldsymbol{p}_{t=1}) &\leq -\frac{\beta}{4T^2}(1 - 2\eta - o_n(1)) \cdot c_\rho^2 \cdot \frac{d}{n} + \frac{\beta(T-1)}{2T^2 n}d(1 + o_n(1)) + \frac{\beta}{n} \cdot o(d) \\
&\leq \left( \frac{-c_\rho^2(1 - 2\eta) + 2(T-1) + o_n(1)}{4T^2} \right) \cdot \frac{\beta d}{n} \qquad \text{Assumption 3.1 (item 2)} \\
&\leq \frac{-0.9 c_\beta}{4T^2},
\end{aligned}
\tag{16}
$$

where the last inequality holds since $c_\rho \geq 5(T-1)$, which implies that $2(T-1)/c_\rho^2 + 2\eta + o_n(1) \leq 0.1$. Similarly, we have that

$$
\begin{aligned}
y_j f(\boldsymbol{X}_j; \boldsymbol{v}_{t=1}, \boldsymbol{p}_{t=1}) &\geq -\frac{\beta}{4T^2}(1 - 2\eta + o_n(1)) \cdot c_\rho^2 \cdot \frac{d}{n} + \frac{\beta}{2T^2 n}d(1 - o_n(1)) - \frac{\beta}{n}o(d) \\
&\geq \left( \frac{-c_\rho^2(1 - 2\eta) + 2(T-1) - o_n(1)}{4T^2} \right) \cdot \frac{\beta d}{n} \\
&= c_\beta \cdot \left( \frac{-(1 - 2\eta) + 2(T-1)/c_\rho^2 - o_n(1)}{4T^2} \right) \\
&\geq \frac{-1.1 c_\beta}{4T^2}
\end{aligned}
\tag{17}
$$

We remind that $-\ell'_{1,j} = 1/(1 + \exp(y_i f(\boldsymbol{X}_i; \boldsymbol{v}_{t=1}, \boldsymbol{p}_{t=1})))$ and that $\beta = c_\beta \cdot n/(dc_\rho^2)$ (Eq. 10). Combine with Eqs. 13 and 14, we have that

$$i \in \mathcal{C}, \quad -\ell'_{t=1,i} \geq 1/(1 + \exp(1.1 c_\beta/4T^2)) := m_\mathcal{C}^{t=1} > 0 \tag{18}$$

$$i \in \mathcal{C}, \quad -\ell'_{t=1,i} \leq 1/(1 + \exp(0.9 c_\beta/4T^2)) := M_\mathcal{C}^{t=1} \leq 1/(4(T-1)c_\rho^2), \tag{19}$$

where the last inequality holds since $c_\beta \geq \log(c_\rho)/\sqrt{\eta}$ and since $1 + \exp(0.9 c_\rho) \geq 4c_\rho^2$ for any $c_\rho \geq 6$.

Moreover, by Eqs. 16 and 17, we have that

$$j \in \mathcal{N}, \quad -\ell'_{t=1,j} \geq 1/(1 + \exp(-0.9 c_\beta/4T^2)) := m_\mathcal{N}^{t=1} \geq 0.99 \tag{20}$$

$$j \in \mathcal{N}, \quad -\ell'_{t=1,j} \leq 1/(1 + \exp(-1.1 c_\beta/4T^2)) := M_\mathcal{N}^{t=1} \leq 1 \tag{21}$$

The notations $M_\mathcal{C}^t$ and $m_\mathcal{C}^t$ ($M_\mathcal{N}^t$ and $m_\mathcal{N}^t$) denote the upper and lower bounds, respectively, on the derivative of the loss for clean (noisy) samples at time $t$, and we use them throughout the proof. We remind that $\gamma_{i,\tau}^t = y_i \boldsymbol{v}_t^\top \boldsymbol{x}_{i,\tau}$. Then by Eq. 11, for $i \in \mathcal{C}_k$ we have that

$$\gamma_{i,1}^{t=1} \in \frac{\beta}{4T}(1 - 2\eta \pm o_n(1))\rho^2 = \frac{c_\beta}{4T}(1 - 2\eta \pm o_n(1))$$

$$\gamma_{i,\tau}^{t=1} \in \frac{\beta}{2Tn} \cdot d(1 \pm o_n(1)) = \frac{c_\beta}{2T}(1/c_\rho^2 \pm o_n(1)), \forall \tau \in \{2, \ldots, T\}$$

$$\gamma_{i,1}^{t=1} - \gamma_{i,2}^{t=2} \in \frac{c_\beta}{4T}(1 - 2/c_\rho^2 - 2\eta \pm o_n(1)) . \tag{22}$$

where in the calculation of $\gamma_{i,\tau}^{t=1}$ we use $\sum_{i \in [n]} y_i y_j \sum_{\tau \neq \tau'} \boldsymbol{\xi}_{i,\tau}^\top \boldsymbol{\xi}_{j,\tau'} \in \pm o_n(1) \cdot d$, which holds since the training set is good. For $i \in \mathcal{N}_k$, we have that

$$\gamma_{i,1}^{t=1} \in -\frac{\beta}{4T}(1 - 2\eta \pm o_n(1))\rho^2 = -\frac{c_\beta}{4T}(1 - 2\eta \pm o_n(1))$$

$$\gamma_{i,\tau}^{t=1} \in \frac{\beta}{2Tn} \cdot d(1 \pm o_n(1)) = \frac{c_\beta}{2T}(1/c_\rho^2 \pm o_n(1)), \forall \tau \in \{2, \ldots, T\}$$

$$\gamma_{i,2}^{t=1} - \gamma_{i,1}^{t=2} \in \frac{c_\beta}{4T}(1 + 2/c_\rho^2 - 2\eta \pm o_n(1)) . \tag{23}$$

**GD after 2 iterations.**
**Analysis of $\boldsymbol{v}_{t=2}$.**
Observe that

$$-\beta \nabla_{\boldsymbol{v}} \mathcal{L}(\boldsymbol{v}_1, \boldsymbol{p}_1) = -\frac{\beta}{n} \sum_{i=1}^n \ell'_{1,i} \cdot y_i \boldsymbol{X}_i^\top \mathbb{S}(X_i \boldsymbol{p}_1) = -\frac{\beta}{Tn} \sum_{i=1}^n \ell'_{1,i} \cdot y_i \sum_{\tau=1}^T \boldsymbol{x}_i.$$

We start by analyzing the first coordinate of $\nabla_{\boldsymbol{v}} \mathcal{L}(\boldsymbol{v}_1, \boldsymbol{p}_1)$.

$$-\beta \nabla_{\boldsymbol{v}} \mathcal{L}(\boldsymbol{v}_1, \boldsymbol{p}_1)[1] = \frac{\beta}{Tn} \sum_{i \in \mathcal{C}_1} -\ell'_{1,i} \cdot y_i \boldsymbol{x}_{i,1}[1] + \frac{\beta}{Tn} \sum_{i \in \mathcal{N}_1} -\ell'_{1,i} \cdot y_i \boldsymbol{x}_{i,1}[1]$$

$$= \frac{\beta}{Tn} \sum_{i \in \mathcal{C}_1} -\ell'_{1,i} \cdot \rho - \frac{\beta}{Tn} \sum_{i \in \mathcal{N}_1} -\ell'_{1,i} \cdot \rho$$

$$= \frac{\beta}{Tn} \left( \sum_{i \in \mathcal{C}_1} -\ell'_{1,i} - \sum_{j \in \mathcal{N}_1} -\ell'_{1,j} \right) \cdot \rho . \tag{24}$$

Observe that

$$\sum_{i \in \mathcal{C}_1} -\ell'_{1,i} - \sum_{j \in \mathcal{N}_1} -\ell'_{1,j} \geq -\frac{n}{2}(\eta + o_n(1)) \cdot M_\mathcal{N} \qquad \text{good training set}$$

$$> -\frac{n}{2}(\eta + o_n(1)) \qquad \text{Eq. 21,}$$

Substituting it into Eq. 24, we obtain that

$$-\beta\nabla_{\boldsymbol{v}}\mathcal{L}(\boldsymbol{v}_1,\boldsymbol{p}_1)[1] > -\frac{\beta}{2T}(\eta + o_n(1))\rho.$$

On the other hand, by Eq. 24, we can upper bound the first coordinate of the gradient of $\boldsymbol{v}$ by

$$-\beta\nabla_{\boldsymbol{v}}\mathcal{L}(\boldsymbol{v}_1,\boldsymbol{p}_1)[1] \leq \frac{\beta}{Tn}\left(\sum_{i\in\mathcal{C}_1}-\ell'_{1,i}\right)\cdot\rho$$

$$\leq \frac{\beta}{17T}\cdot\rho \qquad\qquad -\ell'_{1,i} < 1/17, \text{Eq. 19}.$$

Similarly, we can estimate the second coordinate of $\nabla_{\boldsymbol{v}}\mathcal{L}(\boldsymbol{v}_1,\boldsymbol{p}_1)$:

$$\frac{\beta}{2T}(\eta + o_n(1))\rho \geq -\beta\nabla_{\boldsymbol{v}}\mathcal{L}(\boldsymbol{v}_1,\boldsymbol{p}_1)[2] \geq -\frac{\beta}{17T}\cdot\rho.$$

Write $\boldsymbol{v}_{t=2} = \lambda_1^{t=2}\boldsymbol{\mu}_1 + \lambda_2^{t=2}\boldsymbol{\mu}_2 + \sum_{i=1}^n y_i\theta_i^{t=2}\sum_{\tau=1}^{T-1}\boldsymbol{\xi}_i$. Together with Eq. 11, we get that

$$\lambda_1^{t=2} = \lambda_1^{t=1} - \beta\nabla_{\boldsymbol{v}}\mathcal{L}(\boldsymbol{v}_1,\boldsymbol{p}_1)[1]/\rho \leq \frac{\beta}{4T}(1+o_n(1)) + \frac{\beta}{17T} \leq \frac{5\beta}{16T} \tag{25}$$

$$\lambda_1^{t=2} \geq \frac{\beta}{4T}(1 - 4\eta - o_n(1)) \tag{26}$$

$$\lambda_2^{t=2} = \lambda_2^{t=1} - \beta\nabla_{\boldsymbol{v}}\mathcal{L}(\boldsymbol{v}_1,\boldsymbol{p}_1)[2] \geq -\frac{\beta}{4T}(1+o_n(1)) - \frac{\beta}{17T} \geq -\frac{5\beta}{16} \tag{27}$$

$$\lambda_2^{t=2} \leq -\frac{\beta}{4T}(1 - 4\eta - o_n(1)). \tag{28}$$

Next, we analyze the rest of the coordinates of $\nabla_{\boldsymbol{v}}\mathcal{L}(\boldsymbol{v}_1,\boldsymbol{p}_1)$.

$$-\beta\nabla_{\boldsymbol{v}}\mathcal{L}(\boldsymbol{v}_1,\boldsymbol{p}_1)[3:d] = \frac{\beta}{Tn}\sum_{i\in\mathcal{C}}-\ell'_{1,i}\cdot y_i\sum_{\tau=2}^T\boldsymbol{\xi}_i + \frac{\beta}{Tn}\sum_{j\in\mathcal{N}}-\ell'_{1,j}\cdot y_j\sum_{\tau=2}^T\boldsymbol{\xi}_j,$$

and use it to analyze the coefficients of the noise (second) tokens in $\boldsymbol{v}_{t=2}$, i.e., $\theta_i^{t=2}$. Indeed, for $i\in\mathcal{C}$ we have that

$$\theta_i^{t=2} = \theta_i^{t=1} - \frac{\beta}{Tn}\ell'_{1,i} = \frac{\beta}{Tn}(-\ell'_{1,i} + 0.5) \qquad \text{Eq. 11}$$

$$\in \left[\frac{\beta}{Tn}(m_{\mathcal{C}} + 0.5), \frac{\beta}{Tn}(M_{\mathcal{C}} + 0.5)\right]. \tag{29}$$

For $j\in\mathcal{N}$ we have that

$$\theta_j^{t=2} = \theta_j^{t=1} - \frac{\beta}{Tn}\ell'_{1,j} = \frac{\beta}{Tn}(-\ell'_{1,j} + 0.5) \qquad \text{Eq. 11}$$

$$\in \left[\frac{\beta}{Tn}(m_{\mathcal{N}} + 0.5), \frac{\beta}{Tn}(M_{\mathcal{N}} + 0.5)\right]. \tag{30}$$

Next we move to analyze $\boldsymbol{p}_{t=2}$.

**$\boldsymbol{p}_{t=2}$ focuses on noisy tokens for noisy samples**.

Define $\gamma_{i,min} := \min_{\tau\geq2}\gamma_{i,\tau}$, $\gamma_{i,max} := \max_{\tau\geq2}\gamma_{i,\tau}$, $\gamma_i := (\gamma_{min} + \gamma_{max})/2$ and $\epsilon_i := (\gamma_{max} - \gamma_{min})/2$. In words $\gamma_{i,min}, \gamma_{i,max}$ and $\epsilon_i$ are the maximun, minimum and the gap among the scores of the noisy tokens of the $i^{\text{th}}$ sample respectively. By Eqs. 22 and 23 we have that $\epsilon_i = o_n(1)\cdot c_\beta$ for any $i\in[n]$. Observe that $\boldsymbol{p}_2 = -\beta\nabla_{\boldsymbol{p}}\mathcal{L}(\boldsymbol{v}_1,\boldsymbol{p}_1)$. Therefore, for any $j\in\mathcal{N}_k$ and $\tau\in\{2,\dots,T\}$ we have:

$$\boldsymbol{p}_2^\top(\boldsymbol{x}_{j,1} - \boldsymbol{x}_{j,\tau})$$

$$= -(\boldsymbol{x}_{j,1} - \boldsymbol{x}_{j,\tau})^\top\beta\nabla_{\boldsymbol{p}}\mathcal{L}(\boldsymbol{v}_t,\boldsymbol{p}_t) = (\boldsymbol{x}_{j,1} - \boldsymbol{x}_{j,\tau})^\top\frac{\beta}{n}\sum_{i=1}^n-\ell'_{1,i}\cdot\boldsymbol{X}_i^\top\mathbb{S}'(X_i\boldsymbol{p}_t)\boldsymbol{\gamma}_i^{t=1}$$

$$= \frac{\beta}{n}\sum_{i=1}^n-\ell'_{1,i}\cdot\boldsymbol{x}_{j,1}^\top\boldsymbol{X}_i^\top\mathbb{S}'(X_i\boldsymbol{p}_t)\boldsymbol{\gamma}_i^{t=1} - \frac{\beta}{n}\sum_{i=1}^n-\ell'_{1,i}\cdot\boldsymbol{x}_{j,\tau}^\top\boldsymbol{X}_i^\top\mathbb{S}'(X_i\boldsymbol{p}_t)\boldsymbol{\gamma}_i^{t=1}. \tag{31}$$

Write $\boldsymbol{z}_{1,i,j} := \boldsymbol{X}_i \boldsymbol{x}_{j,1}$. Observe that $\boldsymbol{z}_{1,i,j} = (\boldsymbol{x}_{i,1}^\top \boldsymbol{x}_{j,1}, 0, \ldots, 0)$ for $i \in \mathcal{C}_k \cup \mathcal{N}_k$ and $\boldsymbol{z}_{1,i,j} = \boldsymbol{0}$ otherwise. By Lemma A.5, we can lower bound the first term in Eq. 31 as

$$\frac{\beta}{n} \sum_{i=1}^n -\ell'_{1,i} \cdot \boldsymbol{z}_{1,i,j}^\top \mathbb{S}'(X_i \boldsymbol{p}_t) \boldsymbol{\gamma}_i^{t=1} \geq \frac{\beta}{n} \sum_{i \in \mathcal{C}_k} -\ell'_{1,i} \cdot (\gamma_{i,1}^{t=1} - \gamma_i^{t=1})(1 - \alpha_{i,1}^{t=1})\alpha_{i,1}^{t=1}(1 - o_n(1)) \boldsymbol{x}_{i,1}^\top \boldsymbol{x}_{j,1}$$
$$- \frac{\beta}{n} \sum_{i \in \mathcal{N}_k} -\ell'_{1,i} \cdot (\gamma_i^{t=1} - \gamma_{i,1}^{t=1})(1 - \alpha_{i,1}^{t=1})\alpha_{i,1}^{t=1}(1 - o_n(1)) \boldsymbol{x}_{i,1}^\top \boldsymbol{x}_{j,1},$$

where the $(1 - o_n(1))$ term is from the second tern in Lemma A.5. Now we move to the second term of Eq. 31. Write $\boldsymbol{z}_{\tau,i,j} := \boldsymbol{X}_i \boldsymbol{x}_{j,\tau}$. Observe that $\boldsymbol{X}_i \boldsymbol{x}_{j,\tau} = (0, \boldsymbol{x}_{i,1}^\top \boldsymbol{x}_{j,\tau}, \ldots, \boldsymbol{x}_{i,\tau}^\top \boldsymbol{x}_{j,\tau}, \ldots, \boldsymbol{x}_{i,T}^\top \boldsymbol{x}_{j,\tau})$. By Lemma A.5, we can lower bound the second term in Eq. 31 as

$$- \frac{\beta}{n} \sum_{i=1}^n -\ell'_{1,i} \cdot \boldsymbol{z}_{\tau,i,j}^\top \mathbb{S}'(X_i \boldsymbol{p}_t) \boldsymbol{\gamma}_i^{t=1}$$

$$= \frac{\beta}{(T-1)n} \sum_{i=1}^n -\ell'_{1,i} \cdot (\gamma_i^{t=1} - \gamma_{i,1}^{t=1})(1 - \alpha_{i,1}^{t=1})\alpha_{i,1}^{t=1}(1 - o_n(1)) \sum_{\tau'=1}^T \boldsymbol{x}_{i,\tau'}^\top \boldsymbol{x}_{j,\tau}$$

$$\geq \frac{\beta}{(T-1)n} (-\ell'_{1,j}) \cdot (\gamma_{j,1}^{t=1} - \gamma_j^{t=1})(1 - \alpha_{j,1}^{t=1})\alpha_{j,1}^{t=1}(1 - o_n(1)) \left( \|\boldsymbol{x}_{j,\tau}\|^2 + \sum_{\tau' \neq \tau}^T \boldsymbol{x}_{j,\tau'}^\top \boldsymbol{x}_{j,\tau} \right)$$

$$- \frac{\beta}{(T-1)n} \sum_{i \in n : i \neq j} -\ell'_{1,i} \cdot |\gamma_i^{t=1} - \gamma_{i,1}^{t=1}| \cdot (1 - \alpha_{i,1}^{t=1})\alpha_{i,1}^{t=1}(1 + o_n(1)) \sum_{\tau'=2}^T \boldsymbol{x}_{i,\tau'}^\top \boldsymbol{x}_{j,\tau},$$

where once again where the $(1 - o_n(1))$ term is from the second tern in Lemma A.5. Overall,

$$\boldsymbol{p}_2^\top (\boldsymbol{x}_{j,\tau} - \boldsymbol{x}_{j,1})$$
$$\geq \frac{\beta}{n} (-\ell'_{1,j})(\gamma_j^{t=1} - \gamma_{j,1}^{t=1})(1 - \alpha_{j,1})\alpha_{j,1}(1 - o_n(1)) \left( \|\boldsymbol{x}_{j,1}\|^2 + \|\boldsymbol{x}_{j,\tau}\|^2 / (T-1) \right)$$

$$- \frac{\beta}{n} \sum_{i \in \mathcal{C}_k : i \neq j} -\ell'_{1,i} \cdot (\gamma_{i,1}^{t=1} - \gamma_i^{t=1})(1 - \alpha_{i,1}^{t=1})(1 + o_n(1))\alpha_{i,1}^{t=1}(\boldsymbol{x}_{j,1}^\top \boldsymbol{x}_{i,1})$$

$$+ \frac{\beta}{n} \sum_{i \in \mathcal{N}_k : i \neq j} -\ell'_{1,i} \cdot (\gamma_i^{t=1} - \gamma_{i,1}^{t=1})(1 - \alpha_{i,1}^{t=1})\alpha_{i,1}^{t=1}(1 - o_n(1))(\boldsymbol{x}_{j,1}^\top \boldsymbol{x}_{i,1})$$

$$- \frac{\beta}{(T-1)n} \sum_{i \in [n] : i \neq j} -\ell'_{1,i} \cdot |\gamma_{i,1}^{t=1} - \gamma_{i,2}^{t=1}| \cdot (1 - \alpha_{i,1}^{t=1})\alpha_{i,1}^{t=1}(1 + o_n(1)) \sum_{\tau'=2}^T (\boldsymbol{x}_{j,\tau}^\top \boldsymbol{x}_{i,\tau'}) .$$

Observe that $\alpha_{i,1}^{t=1} = 1/T$ and that $(1 - \alpha_{i,1})\alpha_{i,1} = (T-1)/T^2$ for any $i \in [n]$. In Eqs. 22 and 23 we calculate the score (e.g. $\gamma_{i,\tau}^{t=1}$). Overall, we can lower bound the above equation by:

$$\geq \frac{\beta}{T^2 n} \left( m_{\mathcal{N}} \cdot \frac{c_\beta}{4T}(1 + 2/c_\rho^2 - 2\eta - o_n(1)) \cdot d(1 - o_n(1)) \right)$$

$$- \frac{(T-1)\beta}{T^2 n} \left( (1 - \eta - o_n(1)) \cdot \frac{n}{2} \cdot M_{\mathcal{C}} \frac{c_\beta}{4T}(1 - 2/c_\rho^2 - 2\eta - o_n(1)) \frac{d}{n} c_\rho^2 \right)$$

$$+ \frac{(T-1)\beta}{T^2 n} \left( |\mathcal{N}_k| \cdot m_{\mathcal{N}} \frac{c_\beta}{4T}(1 + 2/c_\rho^2 - 2\eta + o_n(1)) \frac{d}{n} c_\rho^2 \right)$$

$$- \frac{(T-1)\beta}{T^2 n} \left( n \cdot M_{\mathcal{N}} \frac{c_\beta}{4T}(1 + 2/c_\rho^2 - 2\eta + o_n(1)) c_D \sqrt{d \log(n/\delta)} \right) .$$

By Assumption 3.1 (item 2, $d \gg n^2 \log(n)$), the first term dominates the last term. Then we can lower-bound the above term by

$$-\frac{(T-1)\beta}{T^2 n}\left((1-\eta-o_n(1))\cdot\frac{n}{2}\cdot M_{\mathcal{C}}\frac{c_\beta}{4T}(1-2/c_\rho^2-2\eta-o_n(1))\frac{d}{n}c_\rho^2\right)$$

$$+\frac{(T-1)\beta}{T^2 n}\left(|\mathcal{N}_k|\cdot m_{\mathcal{N}}\frac{c_\beta}{4T}(1+2/c_\rho^2-2\eta+o_n(1))\frac{d}{n}c_\rho^2\right)$$

$$\geq-\frac{(T-1)c_\beta^2}{2T^2}\left((1-\eta-o_n(1))\cdot M_{\mathcal{C}}\frac{1}{4T}(1-2/c_\rho^2-2\eta-o_n(1))\right)$$

$$+\frac{(T-1)c_\beta^2}{2T^2}\left(\eta\cdot(1-o_n(1))\cdot m_{\mathcal{N}}\frac{1}{4T}(1+2/c_\rho^2-2\eta+o_n(1))\right),$$

where in the last inequality we use $\beta=(c_\beta\cdot n)/(dc_\rho^2)$. Next, we argue that the second term in the above Eq. is at least 1000 times the absolute value of the first term (i.e. the second term dominates the other terms). Indeed for $c_\beta > 1/\eta$ we have that $M_{\mathcal{C}} < 0.0002\eta$ for small enough $\eta$ (Eqs. 19 and 21). This means that $m_{\mathcal{N}}\cdot\eta\geq0.0002M_{\mathcal{C}}$. Then we can conclude that the second term is at least 1000 times bigger than the absolute value of the first term. Overall, for any $i\in\mathcal{N}$ we have that:

$$\boldsymbol{p}_2^\top(\boldsymbol{x}_{j,\tau}-\boldsymbol{x}_{j,1})\geq0.999\cdot\frac{(T-1)c_\beta^2}{2T^2}\left(\eta\cdot(1-o_n(1))\cdot m_{\mathcal{N}}\frac{1}{4T}(1+2/c_\rho^2-2\eta+o_n(1))\right)$$

$$\geq0.999\cdot c_\beta^2\eta\cdot\left(\frac{T-1}{8T^3}\right)(m_{\mathcal{N}}\cdot(1-2\eta-o_n(1))$$

$$\geq0.998\cdot c_\beta^2\eta\cdot\left(\frac{T-1}{8T^3}\right)$$

$$\geq2\log(c_\rho),$$

where that last inequality holds since $c_\beta^2\geq2\log(c_\rho)\eta^{-1}\cdot\left(\frac{8T^3}{0.998(T-1)}\right)$ (Eq. 10). We conclude that,

$$\alpha_{i,1}^{t=2}=\frac{1}{1+\sum_{\tau=2}^T\exp(\boldsymbol{p}_2^\top(\boldsymbol{x}_{j,\tau}-\boldsymbol{x}_{j,1}))}\leq\frac{1}{1+(T-1)\exp(\log(c_\rho^2))}=\frac{1}{1+(T-1)c_\rho^2}$$

$$\leq\frac{1}{(T-1)c_\rho^2}. \tag{32}$$

We also conclude that for any $j\in\mathcal{N}$ we have that

$$\alpha_{j,1}^{t=2}\leq\frac{1}{1+(T-1)c_\rho^2},\quad\sum_{\tau=2}^T\alpha_{j,\tau}^{t=2}\geq1-\frac{1}{1+(T-1)c_\rho^2}. \tag{33}$$

In the next part, we assume that $C$ from Assumption 3.1 may depend on $c_\rho$ and we show that $\boldsymbol{p}_{t=2}$ focuses on the signal tokens for clean sample.

$\boldsymbol{p}_{t=2}$ **focuses on the signal tokens for clean sample.** Similarly to the previous case, for any $j\in\mathcal{C}_k$ and $\tau\in\{2,\ldots,T\}$ we have

$$\boldsymbol{p}_2^\top(\boldsymbol{x}_{j,1}-\boldsymbol{x}_{j,\tau})$$

$$\geq\frac{\beta}{n}(-\ell_{1,j}')(\gamma_{j,1}^{t=1}-\gamma_j^{t=1})(1-\alpha_{j,1})\alpha_{j,1}(1-o_n(1))\left(\|\boldsymbol{x}_{j,1}\|^2+\|\boldsymbol{x}_{j,\tau}\|^2/(T-1)\right)$$

$$+\frac{\beta}{n}\sum_{i\in\mathcal{C}_k:i\neq j}-\ell_{1,i}'\cdot(\gamma_{i,1}^{t=1}-\gamma_i^{t=1})(1-\alpha_{i,1}^{t=1})(1-o_n(1))\alpha_{i,1}^{t=1}(\boldsymbol{x}_{j,1}^\top\boldsymbol{x}_{i,1})$$

$$-\frac{\beta}{n}\sum_{i\in\mathcal{N}_k:i\neq j}-\ell_{1,i}'\cdot(\gamma_i^{t=1}-\gamma_{i,1}^{t=1})(1-\alpha_{i,1}^{t=1})\alpha_{i,1}^{t=1}(1-o_n(1))(\boldsymbol{x}_{j,1}^\top\boldsymbol{x}_{i,1})$$

$$-\frac{\beta}{(T-1)n}\sum_{i\in[n]:i\neq j}\sum_{\tau':}-\ell_{1,i}'\cdot|\gamma_{i,1}^{t=1}-\gamma_{i,2}^{t=1}|\cdot(1-\alpha_{i,1}^{t=1})\alpha_{i,1}^{t=1}(1+o_n(1))\sum_{\tau'=2}^T(\boldsymbol{x}_{j,\tau}^\top\boldsymbol{x}_{i,\tau'}).$$

Observe that $\alpha_{i,1}^{t=1} = 1/T$ and that $(1 - \alpha_{i,1})\alpha_{i,1} = (T-1)/T^2$ for any $i \in [n]$. In Eqs. 22 and 23 we calculate the score (i.e. $\gamma_{i,\tau}^{t=1}$). Overall, we can lower bound the above equation by:

$$\geq \frac{\beta}{T^2 n} \left( m_{\mathcal{C}} \cdot \frac{c_\beta}{4T}(1 - 2/c_\rho^2 - 2\eta - o_n(1)) \cdot d(1 - o_n(1)) \right)$$
$$+ \frac{(T-1)\beta}{T^2 n} \left( (1 - \eta - o_n(1)) \cdot \frac{n}{2} \cdot m_{\mathcal{C}} \frac{c_\beta}{4T}(1 - 2/c_\rho^2 - 2\eta - o_n(1))\frac{d}{n}c_\rho^2 \right)$$
$$- \frac{(T-1)\beta}{T^2 n} \left( (\eta + o_n(1)) \cdot \frac{n}{2} \cdot M_{\mathcal{N}} \frac{c_\beta}{4T}(1 + 2/c_\rho^2 - 2\eta + o_n(1))\frac{d}{n}c_\rho^2 \right)$$
$$- \frac{(T-1)\beta}{T^2 n} \left( n \cdot M_{\mathcal{N}} \frac{c_\beta}{4T}(1 + 2/c_\rho^2 - 2\eta + o_n(1))c_D \sqrt{d \log(n/\delta)} \right). \tag{34}$$

Observe that the first two terms are non-negative. Then we can lower-bound the above term by

$$- \frac{(T-1)\beta}{T^2 n} \left( (\eta + o_n(1)) \cdot \frac{n}{2} \cdot M_{\mathcal{N}} \frac{c_\beta}{4T}(1 + 2/c_\rho^2 - 2\eta + o_n(1))\frac{d}{n}c_\rho^2 \right)$$
$$- \frac{(T-1)\beta}{T^2 n} \left( n \cdot M_{\mathcal{N}} \frac{c_\beta}{4T}(1 + 2/c_\rho^2 - 2\eta + o_n(1))c_D \sqrt{d \log(n/\delta)} \right)$$
$$= - \frac{(T-1)c_\beta^2}{T^2} \left( (\eta + o_n(1)) \cdot \frac{1}{2} \cdot M_{\mathcal{N}} \frac{1}{4T}(1 + 2/c_\rho^2 - 2\eta + o_n(1)) \right)$$
$$- \frac{(T-1)c_\beta^2}{T^2} \left( M_{\mathcal{N}} \frac{1}{4T}(1 + 2/c_\rho^2 - 2\eta + o_n(1))c_D n \cdot \frac{\sqrt{\log(n/\delta)}}{\sqrt{d}c_\rho^2} \right)$$

By Assumption 3.1 (item 2) the first term dominates the second term whenever $C$ from that assumption is large enough, which means that we can lower bound the displayed equation by 1.0002 the first term i.e. by

$$- 1.0002 \cdot \frac{(T-1)c_\beta^2}{T^2} \left( (\eta + o_n(1)) \cdot \frac{1}{2} \cdot M_{\mathcal{N}} \frac{1}{4T}(1 + 2/c_\rho^2 - 2\eta + o_n(1)) \right)$$
$$\geq -1.0002 \cdot \frac{(T-1)c_\beta^2}{8T^3} \left( (\eta + o_n(1)) \cdot M_{\mathcal{N}}(1 + 2/c_\rho^2 - 2\eta + o_n(1)) \right)$$
$$\geq -1.0001 \cdot \frac{(T-1)}{8T^3} \cdot c_\beta^2 \eta$$
$$\geq -2.1 \log(c_\rho),$$

where the last inequality holds by Eq. 10. We conclude that

$$\alpha_{i,1}^{t=2} = \frac{1}{1 + \sum_{\tau=2}^{T} \exp(\boldsymbol{p}_2^\top(\boldsymbol{x}_{j,\tau} - \boldsymbol{x}_{j,1}))} \geq \frac{1}{1 + (T-1)\exp(2.1 \log(c_\rho))} \tag{35}$$
$$= \frac{1}{1 + (T-1)c_\rho^{2.1}}$$

Together with Eq. 33, this proves the last part of the Thm.

$\boldsymbol{p}_{t=2}$ **focuses more on the signal tokens for clean sample when** $c_\rho$ **is a constant**. In this part, we prove Remark 3.4. In this case, $C$ from Assumption 3.1 may depend on $c_\rho$. We can start directly from Eq. 34 that states that for any $j \in \mathcal{C}_k$ and $\tau \in \{2, \ldots, T\}$ we have

$$\boldsymbol{p}_2^\top(\boldsymbol{x}_{j,1} - \boldsymbol{x}_{j,\tau}) \tag{36}$$
$$\geq \frac{\beta}{T^2 n} \left( m_{\mathcal{C}} \cdot \frac{c_\beta}{4T}(1 - 2/c_\rho^2 - 2\eta - o_n(1)) \cdot d(1 - o_n(1)) \right)$$
$$+ \frac{(T-1)\beta}{T^2 n} \left( (1 - \eta - o_n(1)) \cdot \frac{n}{2} \cdot m_{\mathcal{C}} \frac{c_\beta}{4T}(1 - 2/c_\rho^2 - 2\eta - o_n(1))\frac{d}{n}c_\rho^2 \right)$$
$$- \frac{(T-1)\beta}{T^2 n} \left( (\eta + o_n(1)) \cdot \frac{n}{2} \cdot M_{\mathcal{N}} \frac{c_\beta}{4T}(1 + 2/c_\rho^2 - 2\eta + o_n(1))\frac{d}{n}c_\rho^2 \right)$$
$$- \frac{(T-1)\beta}{T^2 n} \left( n \cdot M_{\mathcal{N}} \frac{c_\beta}{4T}(1 + 2/c_\rho^2 - 2\eta + o_n(1))c_D \sqrt{d \log(n/\delta)} \right).$$

Since $C$ from Assumption 3.1 may depend on $c_\rho$. Then $M_\mathcal{N}, m_\mathcal{C}$ are also constants. Overall, the first term dominates the last term since $d \gg n\sqrt{d\log(n/\delta)}$ (see Assumption 3.1 (item 2)). The second term dominates the third term for small enough $\eta$ (see Assumption 4). Overall, we obtain that for any $\tau \in \{2, \ldots, T\}$ that

$$\boldsymbol{p}_2^\top (\boldsymbol{x}_{j,1} - \boldsymbol{x}_{j,\tau}) > 0, \tag{37}$$

which means that for any $i \in \mathcal{C}$ we have:

$$\alpha_{i,1}^{t=2} = \frac{1}{1 + \sum_{\tau=2}^T \exp(\boldsymbol{p}_2^\top(\boldsymbol{x}_{j,\tau} - \boldsymbol{x}_{j,1}))} > \frac{1}{T}. \tag{38}$$

Next, we move to our original assumption and don't assume that $C$ from Assumption 3.1 may depend on $c_\rho$. We show that:
**The classifier** $\operatorname{sign}(f(\boldsymbol{X}; \boldsymbol{v}_{t=2}, \boldsymbol{p}_{t=2}))$ **classifies correctly clean training samples.** Let $(\boldsymbol{X}_j = (\boldsymbol{x}_{j,1}, \ldots, \boldsymbol{x}_{j,T}), y_j)$ for $j \in \mathcal{C}$. We remind that $\boldsymbol{x}_{j,1} = \boldsymbol{\mu}_k$ for $k \in \{1,2\}$ and $\boldsymbol{x}_{j,\tau} = \boldsymbol{\xi}_{j,\tau}$. we have that,

$$f(\boldsymbol{X}_j; \boldsymbol{v}_{t=2}, \boldsymbol{p}_{t=2}) = \alpha_{j,1}^{t=2}\boldsymbol{v}_2^\top \boldsymbol{x}_{j,1} + \sum_{\tau=2}^T \alpha_{j,\tau}^{t=2}\boldsymbol{v}_2^\top \boldsymbol{x}_{j,\tau},$$

and it suffices to prove that

$$y_j(f(\boldsymbol{X}_j; \boldsymbol{v}_2, \boldsymbol{p}_2)) > 0.$$

Indeed,

$$y_j f(\boldsymbol{X}_j; \boldsymbol{v}, \boldsymbol{p}) = y_j \alpha_{j,1}^{t=2}\boldsymbol{v}_2^\top \boldsymbol{x}_{j,1} + y_j \sum_{\tau=2}^T \alpha_{j,\tau}^{t=2}\boldsymbol{v}_2^\top \boldsymbol{x}_{j,\tau}$$

$$= \alpha_{j,1}^{t=2}|\lambda_k|\,\|\boldsymbol{\mu}_k\|^2 + \sum_{\tau=2}^T \alpha_{j,\tau}^{t=2}\theta_j\,\|\boldsymbol{\xi}_{j,\tau}\|^2 + \sum_{\tau=2}^T \alpha_{j,\tau}^{t=2}y_j \sum_{i\in[n],\tau':i\neq j\vee\tau\neq\tau'} y_i\theta_i^{t=2}\boldsymbol{\xi}_{i,\tau}^\top\boldsymbol{\xi}_{j,\tau'} \qquad y_j\lambda_k > 0$$

$$\geq \max_{\tau\in[T]}\alpha_{j,\tau}\min\{|\lambda_k|\,\|\boldsymbol{\mu}_k\|^2, \theta_j\,\|\boldsymbol{\xi}_{j,\tau}\|^2\} + \sum_{\tau=2}^T \alpha_{j,\tau}^{t=2}y_j \sum_{i\in[n],\tau':i\neq j\vee\tau\neq\tau'} y_i\theta_i^{t=2}\boldsymbol{\xi}_{i,\tau}^\top\boldsymbol{\xi}_{j,\tau'}$$

$$\geq \frac{1}{T}\cdot\min\left(\frac{\beta}{5T}\cdot\frac{d}{n}c_\rho^2, \frac{\beta}{2nT}\cdot d(1-o_n(1))\right) - nT^2\frac{\beta}{Tn}(M_\mathcal{N}+0.5)c_D\sqrt{d\log(n/\delta)} \qquad \text{Eqs. 30, 26 and 28}$$

$$> 0, \qquad\qquad\qquad\qquad\qquad\qquad\qquad\qquad\qquad\qquad\qquad\qquad\qquad\qquad\qquad d \gg nc_D\sqrt{d\log(n/\delta)}$$

as required.
**The classifier** $\operatorname{sign}(f(\boldsymbol{X}; \boldsymbol{v}_{t=2}, \boldsymbol{p}_{t=2}))$ **classifies correctly noisy training samples.** Let $(\boldsymbol{X}_j = (\boldsymbol{x}_{j,1}, \ldots, \boldsymbol{x}_{j,T}), y_j)$ for $j \in \mathcal{N}$. We remind that $\boldsymbol{x}_{j,1} = \boldsymbol{\mu}_k$ for $k \in \{1,2\}$ and $\boldsymbol{x}_{j,\tau} = \boldsymbol{\xi}_{j,\tau}$. we have that,

$$f(\boldsymbol{X}_j; \boldsymbol{v}_{t=2}, \boldsymbol{p}_{t=2}) = \alpha_{j,1}^{t=2}\boldsymbol{v}_2^\top \boldsymbol{x}_{j,1} + \sum_{\tau=2}^T \alpha_{j,\tau}^{t=2}\boldsymbol{v}_2^\top \boldsymbol{x}_{j,\tau},$$

and it suffices to prove that

$$y_j(f(\boldsymbol{X}_j; \boldsymbol{v}_2, \boldsymbol{p}_2)) > 0.$$

Indeed,

$$
y_j f(\boldsymbol{X}_j; \boldsymbol{v}, \boldsymbol{p}) = y_j \alpha_{j,1}^{t=2} \boldsymbol{v}_2^\top \boldsymbol{x}_{j,1} + y_j \sum_{\tau=2}^{T} \alpha_{j,\tau}^{t=2} \boldsymbol{v}_2^\top \boldsymbol{x}_{j,\tau}
$$

$$
= -\alpha_{j,1}^{t=2} |\lambda_k| \, \|\boldsymbol{\mu}_k\|^2 + \sum_{\tau=2}^{T} \alpha_{j,\tau}^{t=2} \theta_j \, \|\boldsymbol{\xi}_{j,\tau}\|^2 + \sum_{\tau=2}^{T} \alpha_{j,\tau}^{t=2} y_j \sum_{i\in[n],\tau':i\neq j \vee \tau\neq\tau'} y_i \theta_i^{t=2} \boldsymbol{\xi}_{i,\tau}^\top \boldsymbol{\xi}_{j,\tau'} \quad y_j\lambda_k < 0
$$

$$
\geq -\alpha_{j,1}^{t=2}\left(\frac{5\beta}{16T}\right)\frac{d}{n}c_\rho^2 + \sum_{\tau=2}^{T}\alpha_{j,\tau}^{t=2}\frac{\beta}{Tn}(m_\mathcal{N}+0.5)d(1-o_n(1))
$$

$$
- \sum_{\tau=2}^{T-1}\alpha_{j,\tau}^{t=2}n(T-1)\frac{\beta}{Tn}(M_\mathcal{N}+0.5)c_D\sqrt{d\log(n/\delta)} \qquad\qquad \text{Eqs. 30, 25 and 27}
$$

$$
\geq \left(\frac{1}{c_\rho^2}\right)\left(\frac{5\beta}{16T}\right)\frac{d}{n}c_\rho^2 + \left(1-\frac{1}{c_\rho^2}\right)\frac{\beta}{Tn}(m_\mathcal{N}+0.5)d(1-o_n(1))
$$

$$
- n(T-1)\frac{\beta}{Tn}(M_\mathcal{N}+0.5)c_D\sqrt{d\log(n/\delta)} \qquad\qquad\qquad \text{Eq. 33}
$$

$$
> 0, \qquad\qquad\qquad\qquad\qquad\qquad\qquad\qquad\qquad\qquad d \gg nc_D\sqrt{d\log(n/\delta)}
$$

as required.

**The classifier** $\text{sign}(f(\boldsymbol{X}; \boldsymbol{v}_{t=2}, \boldsymbol{p}_{t=2}))$ **classifies correctly clean test samples.**

Let $(\boldsymbol{X} = (\boldsymbol{x}_1,\ldots,\boldsymbol{x}_T), y)$ be a fresh clean sample i.e. $(\boldsymbol{X}, y) \sim \mathcal{D}_{\text{clean}}$. Observe that $\boldsymbol{x}_1 = \boldsymbol{\mu}_k$ for some $k \in \{1, 2\}$ and $y = 1$ iff $k = 1$. By Remark A.66, exists some constant $c_1$ such that with probability at least $1 - n\exp(-d/c_1^2 C_2 n^{1.5})$ for some $C_2 = C_2(c_\rho, 1/\eta)$ that will be chosen later, we have that $(\boldsymbol{X}, y)$ is a good test sample w.r.t. $C_2$ (Def. A.3), i.e. $|\boldsymbol{x}_\tau^\top \boldsymbol{x}_{i,\tau'}| \leq d/C_2 n^{0.75}$. We work under the event that $(\boldsymbol{X}, y)$ is a good test sample and show that $y = \text{sign}(f(\boldsymbol{X}; \boldsymbol{v}_{t=2}, \boldsymbol{p}_{t=2})$. Recall that $\boldsymbol{p}_2 = -\beta\nabla_{\boldsymbol{p}}\mathcal{L}(\boldsymbol{v}_1, \boldsymbol{p}_1)$ and therefore (similar to the clean sample case) for any $\tau \in \{2, \ldots, T\}$:

$$
\boldsymbol{p}_2^\top(\boldsymbol{x}_1 - \boldsymbol{x}_\tau)
$$

$$
\geq \frac{\beta}{n}\sum_{i\in\mathcal{C}_k} -\ell'_{1,i}\cdot(\gamma_{i,1}^{t=1}-\gamma_i^{t=1})(1-\alpha_{i,1}^{t=1})\alpha_{i,1}^{t=1}(1-o_n(1))(\boldsymbol{x}_1^\top\boldsymbol{x}_{i,1})
$$

$$
- \frac{\beta}{n}\sum_{i\in\mathcal{N}_k} -\ell'_{1,i}\cdot(\gamma_i^{t=1}-\gamma_{i,1}^{t=1})(1-\alpha_{i,1}^{t=1})\alpha_{i,1}^{t=1}(1-o_n(1))(\boldsymbol{x}_1^\top\boldsymbol{x}_{i,1})
$$

$$
- \frac{\beta}{(T-1)n}\sum_{i\in[n]}\sum_{\tau=2}^{T} -\ell'_{1,i}\cdot|\gamma_{i,1}^{t=1}-\gamma_i^{t=1}|\cdot(1-\alpha_{i,1}^{t=1})\alpha_{i,1}^{t=1}(1+o_n(1))\sum_{\tau'=2}^{T}(\boldsymbol{x}_\tau^\top\boldsymbol{x}_{i,\tau'})\,.
$$

Observe that $\alpha_{i,1}^{t=1} = 1/T$ and that $(1-\alpha_{i,1})\alpha_{i,1} = (T-1)/T^2$ for any $i \in [n]$. In Eqs. 22 and 23 we calculate the score (e.g. $\gamma_{i,\tau}^{t=1}$). Overall, we can lower bound the above equation by:

$$
+ \frac{(T-1)\beta}{T^2 n}\left((1-\eta-o_n(1))\cdot\frac{n}{2}\cdot m_\mathcal{C}\frac{c_\beta}{4T}(1-2/c_\rho^2-2\eta-o_n(1))\frac{d}{n}c_\rho^2\right)
$$

$$
- \frac{(T-1)\beta}{T^2 n}\left((\eta+o_n(1))\cdot\frac{n}{2}\cdot M_\mathcal{N}\frac{c_\beta}{4T}(1+2/c_\rho^2-2\eta+o_n(1))\frac{d}{n}c_\rho^2\right)
$$

$$
- \frac{(T-1)\beta}{T^2 n}\left(n^{0.75}\cdot M_\mathcal{N}\frac{c_\beta}{4T}(1+2/c_\rho^2-2\eta+o_n(1))\frac{d}{C_2 n^{0.75}}\right).
$$

Observe that the first term is positive, and the second term is 100 times smaller than the last term whenever $C_2 \geq 201/(\eta c_\rho^2)$. Then we can lower bound the above equation by 1.01 times the second

term i.e. by

$$-\frac{1.01 \cdot (T-1)\beta}{T^2 n} \left( (\eta + o_n(1)) \cdot \frac{n}{2} \cdot M_{\mathcal{N}} \frac{c_\beta}{4T} (1 + 2/c_\rho^2 - 2\eta + o_n(1)) \frac{d}{n} c_\rho^2 \right)$$

$$= -\frac{1.01 \cdot (T-1)c_\beta^2}{T^2 n} \left( (\eta + o_n(1)) \cdot \frac{n}{2} \cdot M_{\mathcal{N}} \frac{1}{4T} (1 + 2/c_\rho^2 - 2\eta + o_n(1)) \right)$$

$$\geq -\frac{1.02 \cdot (T-1)}{8T^3} \cdot \eta \cdot c_\beta^2$$

$$\geq -2.1 \log(c_\rho),$$

where the last inequality holds for $c_\beta^2 \leq 2.1\eta^{-1}\log(c_\rho)\left(\frac{8T^3}{1.02\cdot(T-1)}\right)$ (see Eq. 10 and note that $8 \cdot 2.1/1.02 < 1.02^2 \cdot 16/0.98$). We conclude that

$$\alpha_{i,1}^{t=2} = \frac{1}{1 + \sum_{\tau=2}^{T} \exp(\boldsymbol{p}_2^\top (\boldsymbol{x}_\tau - \boldsymbol{x}_1))} \geq \frac{1}{1 + (T-1)\exp(2.1\log(c_\rho))} \tag{39}$$

$$= \frac{1}{1 + (T-1)c_\rho^{2.1}}$$

Let $\boldsymbol{x}_1 = \boldsymbol{\mu}_k$ for $k \in \{1, 2\}$ and $\boldsymbol{x}_\tau = \boldsymbol{\xi}_\tau$ for $\tau \in \{2, \ldots, T\}$. We have that,

$$f(\boldsymbol{X}; \boldsymbol{v}_{t=2}, \boldsymbol{p}_{t=2}) = \alpha_1^{t=2} \boldsymbol{v}_2^\top \boldsymbol{x}_1 + \sum_{\tau=2}^{T} \alpha_\tau^{t=2} \boldsymbol{v}_2^\top \boldsymbol{x}_\tau,$$

and it suffices to prove that

$$y(f(\boldsymbol{X}; \boldsymbol{v}_2, \boldsymbol{p}_2)) > 0.$$

Indeed,

$$yf(\boldsymbol{X}_j; \boldsymbol{v}, \boldsymbol{p}) = y_j \alpha_1^{t=2} \boldsymbol{v}_2^\top \boldsymbol{x}_1 + y \sum_{\tau=2}^{T} \alpha_\tau^{t=2} \boldsymbol{v}_2^\top \boldsymbol{x}_\tau$$

$$= \alpha_1^{t=2} |\lambda_k| \|\boldsymbol{\mu}_k\|^2 + \sum_{\tau=2}^{T} \alpha_\tau^{t=2} y \sum_{i \in [n], \tau'} y_i \theta_i^{t=2} \boldsymbol{\xi}_{i,\tau}^\top \boldsymbol{\xi}_{\tau'} \qquad y\lambda_k > 0$$

Note that $\theta_i^{t=2}$ is independent of $\boldsymbol{\xi}_{\tau'}$. Moreover, since $yy_i \boldsymbol{\xi}_{i,\tau}^\top \boldsymbol{\xi}_{\tau'}$ is a symmetric random variable with $|\boldsymbol{\xi}_{i,\tau}^\top \boldsymbol{\xi}_{\tau'}| \leq d/C_2 n^{0.75}$ (assuming the test sample is good with respect to $C_2$), by Lemma A.7 with probability at least $1 - \exp(-n^{0.5}/2)$ we have that $y \sum_{i \in [n], \tau'} y_i \theta_i^{t=2} \boldsymbol{\xi}_{i,\tau}^\top \boldsymbol{\xi}_{\tau'} \geq -n^{0.75} \max_i |\theta_i^{t=2}| \cdot d/(C_2 n^{0.75})$. Overall, we can lower bound the displayed equation by

$$\geq \alpha_1^{t=2} |\lambda_k| \|\boldsymbol{\mu}_k\|^2 - \sum_{\tau=2}^{T-1} \alpha_\tau^{t=2} (T-1) \max_i |\theta_i| \frac{d}{C_2}$$

$$\geq \alpha_1^{t=2} \left( \frac{\beta}{4T+1} \right) \frac{d}{n} c_\rho^2 - (T-1)^2 \frac{\beta}{Tn} (M_{\mathcal{N}} + 0.5) \frac{d}{C_2} \qquad \text{Eqs. 30, 26 and 28}$$

$$\geq \left( \frac{1}{1 + (T-1)c_\rho^{2.1}} \right) \left( \frac{\beta}{4T+1} \right) \frac{d}{n} c_\rho^2 - (T-1)^2 \frac{\beta}{Tn} (M_{\mathcal{N}} + 0.5) \frac{d}{C_2}$$

$$> 0,$$

where the last inequality holds for $C_2 = C_3 \cdot c_\rho^{0.1}$, where $C_3$ is large enough constant which depends on $T$. Overall by choosing $C_2 = max(C_3 c_\rho^{0.1}, 1/\eta)$ and union bound, we have that

$$\mathbb{P}_{(\boldsymbol{X}, y) \sim \mathcal{D}}(y \neq \text{sign}(f(\boldsymbol{X}; \boldsymbol{v}_{t=2}, \boldsymbol{p}_{t=2})))$$

$$\leq \eta + \mathbb{P}_{(\boldsymbol{X}, y) \sim \mathcal{D}_{\text{clean}}}(y \neq \text{sign}(f(\boldsymbol{X}; \boldsymbol{v}_{t=2}, \boldsymbol{p}_{t=2})))$$

$$\leq \eta + \exp(-\sqrt{n}/2) + \exp(-d\eta/C_3 c_\rho^{0.1} n^{1.5} + \log(n)).$$

This proves the last part of the theorem. $\qquad\qquad \square$

## A.2 Proofs for Sec. 4

### A.2.1 Proof Idea for Section 4

We first provide the proof sketch for Theorem 4.2. Our key proposition is that $\boldsymbol{p}_{(r,R)}$ will converge to a direction that focuses on the signal token for clean samples, and $\boldsymbol{v}_{(r,R)}$ will converge to the corresponding max-margin solution. To begin, consider the output of the attention layer $\boldsymbol{r}_i = \boldsymbol{X}_i^{\top} \mathbb{S}(\boldsymbol{X}_i \boldsymbol{p})$ which is a combination of signal and noise tokens. This can be considered as a "token selection" based on softmax probabilities. Consider the following **second-token selection** rule:

$$\boldsymbol{r}_i^{\text{sec}} = \boldsymbol{x}_i^{(1)} = \boldsymbol{\mu}_k, \ i \in \mathcal{C}_k, k \in \{1, 2\}$$
$$\boldsymbol{r}_i^{\text{sec}} = \boldsymbol{x}_i^{(2)} = \boldsymbol{\xi}_{i,2}, \ i \in \mathcal{N}.$$

This selects the signal token for all clean samples and the first noise token for all noisy samples. Following this token selection rule, we define the corresponding max-margin solution as $\boldsymbol{p}_{\text{sec}}$ and $\boldsymbol{v}_{\text{sec}}$:

**Definition A.8** (p-SVM for Second-token Selection)**.**

$$\boldsymbol{p}_{\text{sec}} = \operatorname*{argmin}_{\boldsymbol{p} \in \mathbb{R}^d} \|\boldsymbol{p}\| \quad \text{subject to:}$$
$$\boldsymbol{p}^{\top}(\boldsymbol{x}_i^{(\alpha_i)} - \boldsymbol{x}_i^{(t)}) \geq 1, \ \alpha_i = 1 \ \text{for} \ i \in \mathcal{C}$$
$$\text{and} \ \alpha_i = 2 \ \text{for} \ i \in \mathcal{N}, \ t \in \{1, \ldots, T\} \setminus \{\alpha_i\}.$$

Let $\Xi := 1/\|\boldsymbol{p}_{\text{sec}}\|$ be the margin induced by $\boldsymbol{p}_{\text{sec}}$.

**Definition A.9** (v-SVM for Second-token Selection)**.**

$$\boldsymbol{v}_{\text{sec}} := \operatorname*{argmin}_{\boldsymbol{v} \in \mathbb{R}^d} \|\boldsymbol{v}\| \ \text{s.t.} \ y_i \cdot \boldsymbol{v}^{\top} \boldsymbol{r}_i^{\text{sec}} \geq 1, \ \text{for all} \ i \in [n] \ ,$$

Let $\Gamma_{\text{sec}} := 1/\|\boldsymbol{v}_{\text{sec}}\|$ be the label margin induced by $\boldsymbol{v}_{\text{sec}}$.

We prove that $\boldsymbol{p}_{(r,R)}$ has a direction that selects the signal token for every clean sample, since otherwise it will induce a max-margin at most $\Gamma_{\text{sec}} - \frac{C}{\|\boldsymbol{v}_{\text{sec}}\|_2^3 n \rho^2} \cdot \max_{i \in \mathcal{C}} (1 - s_{i,1})$, where $s_{i,1}$ is the attention probability on signal tokens. This is strictly smaller than $\Gamma_{\text{sec}}$.

Then, we show that when jointly optimizing $\boldsymbol{p}$ and $\boldsymbol{v}$ for (4), we obtain solutions that induce similar max-margin as $\boldsymbol{p}_{\text{sec}}$ and $\boldsymbol{v}_{\text{sec}}$ as $R, r \to \infty$. To be specific, we have

- $\min_{\tau \in [2,T]} \boldsymbol{p}_{(r,R)}^{\top}(\boldsymbol{\mu}_k - \boldsymbol{\xi}_{i,\tau}) \geq (1 - \zeta) \Xi R$ for all $i \in \mathcal{C}_k, k \in [2]$, where $\Xi$ is the margin induced by $\boldsymbol{p}_{\text{sec}}$.
- The label margin for clean samples induced by $\boldsymbol{v}_{(r,R)}/r$ in *SVM* is at least $(1 - \gamma) \Gamma_{\text{sec}}$.

Here, $\zeta, \gamma$ are some small value quantifying the difference between $(\boldsymbol{p}_{(r,R)}, \boldsymbol{v}_{(r,R)})$ and $(\boldsymbol{p}_{\text{sec}}, \boldsymbol{v}_{\text{sec}})$. As $R \to \infty$, both $\zeta$ and $\gamma$ converge to 0. Thus, for sufficiently large $R$, we conclude that $\boldsymbol{p}_{(r,R)}^{\top}(\boldsymbol{\mu}_k - \boldsymbol{\xi}_i)$ becomes large for $i \in \mathcal{C}_k$.

This ensures that $\boldsymbol{p}_{(r,R)}$ captures sufficient information about signal tokens, which enhances the accuracy of test sample predictions. Since the signal token remains invariant between training and test data, for a given test sample $(\boldsymbol{X}, y)$ with $\boldsymbol{X} = (\boldsymbol{\mu}^{\star}, \boldsymbol{\xi}_2^{\star}, ..., \boldsymbol{\xi}_T^{\star})$, w.h.p. the attention layer $\boldsymbol{p}_{(r,R)}$ will focus on $\boldsymbol{\mu}^{\star}$ when $R$ is sufficiently large. As a result, the signal token $\boldsymbol{\mu}^{\star}$ will dominate the attention layer's output. As $\boldsymbol{v}_{(r,R)}$ converges to the corresponding max-margin solution, it can make accurate predictions on $(\boldsymbol{\mu}^{\star}, y)$. Thus, the component induced by the signal token $y \cdot \langle \boldsymbol{v}_{(r,R)}, \boldsymbol{\mu}^{\star} \rangle$ is large enough to eliminate the randomness introduced by the noise token (denoted by $\Delta(\boldsymbol{\xi}^{\star})$ here) and the model will make an accurate prediction with high probability: $y \cdot f(\boldsymbol{v}_{(r,R)}, \boldsymbol{p}_{(r,R)}; \boldsymbol{X}) \geq y \cdot \boldsymbol{v}_{(r,R)}^{\top} \boldsymbol{\mu}^{\star} - \Delta(\boldsymbol{\xi}^{\star}) \geq 0$.

### A.2.2 Notation for Section 4

We first introduce some additional notations. Denote

$$n_1 = |\mathcal{C}|, \quad n_2 = |\mathcal{N}|; \quad n_{1i} = |\mathcal{C}_i|, \quad n_{2i} = |\mathcal{N}_i| \ \text{for} \ i = 1, 2.$$

Denote the output of the softmax layer $\mathbb{S}(\boldsymbol{X}_i \boldsymbol{p})$ by

$$\boldsymbol{s}_i = (1 - \beta_i, \beta_i)^\top.$$

Denote the output of the attention layer $\boldsymbol{X}_i^\top \boldsymbol{s}_i$ by $\boldsymbol{r}_i = (1 - \beta_i)\boldsymbol{\mu}_i + \beta_i \boldsymbol{\xi}_i$, where $0 \leq \beta_i \leq 1$ is the attention on the noise token of each sample. Then $f(\boldsymbol{X}_i; \boldsymbol{v}, \boldsymbol{p}) = \langle \boldsymbol{v}, \boldsymbol{r}_i \rangle$ can be treated as a linear classifier on $(y_i, \boldsymbol{r}_i)_{i \in [n]}$. Additionally, from the property of log function, item 1 in Assumption 4.1 can be understood as $d \geq Cn^2 \log(\text{poly}(n)/\delta)$ and the same is for item 5. For the proof of this section, we consider the case when $T = 2$ in Assumption 4.1 and have the following theorem.

**Theorem A.10.** *Suppose that Assumption 4.1 holds when $T = 2$, and consider the classifier $\boldsymbol{X} \to \text{sign}(f(\boldsymbol{X}; \boldsymbol{v}_{(r,R)}, \boldsymbol{p}_{(r,R)}))$, where $(\boldsymbol{v}_{(r,R)}, \boldsymbol{p}_{(r,R)})$ is the solution to Problem (4). Then, with probability at least $1 - \delta$ over the training dataset, we have:*

- *The classifier $\text{sign}(f(\boldsymbol{X}; \boldsymbol{v}_{(r,R)}, \boldsymbol{p}_{(r,R)}))$ correctly classifies all training data points:*

$$y_i = \text{sign}(f(\boldsymbol{X}_i; \boldsymbol{v}_{(r,R)}, \boldsymbol{p}_{(r,R)})), \ \forall i \in [n].$$

- *The classifier $\text{sign}(f(\boldsymbol{X}; \boldsymbol{v}_{(r,R)}, \boldsymbol{p}_{(r,R)}))$ generalizes well on test data:*

$$\mathbb{P}_{(\boldsymbol{X}, y) \sim \mathcal{D}}(y \neq \text{sign}(f(\boldsymbol{X}; \boldsymbol{v}_{(r,R)}, \boldsymbol{p}_{(r,R)})))$$

$$\leq \eta + \exp(-\Omega(d/n^2)) + \exp\left(-\Omega\left(\frac{(1 - \zeta)}{\sqrt{\eta n/d + 1/\rho^2}} - \frac{\log(n)}{R}\right)^2\right),$$

*where $\zeta = \Theta(\sqrt{\eta n/d + 1/\rho^2} \log(\rho n)/R)$.*

### A.2.3   Proof of Thm. A.10

**Proof Sketch**

There are two main parts in our proof. In the first part, we prove that only by selecting signal tokens for clean samples and noise tokens for non-clean samples can we reach the maximum margin when doing SVM on $(y_i, \boldsymbol{r}_i)_{i \in [n]}$.

**Definition A.11** (Optimal Token). We define the "optimal token" for sample $(\boldsymbol{X}_i, y_i)$ as

$$\boldsymbol{r}_i^\star = \boldsymbol{\mu}_i, \ i \in \mathcal{C}$$
$$\boldsymbol{r}_i^\star = \boldsymbol{\xi}_i, \ i \in \mathcal{N} \tag{40}$$

Next we define the respective max-margin solution for $\boldsymbol{p}$ and $\boldsymbol{v}$. We will show that when jointly optimizing parameters $\boldsymbol{p}$ and $\boldsymbol{v}$ for (4), they will converge to their respective max-margin solutions as $R, r \to \infty$, which are $\boldsymbol{p}_{mm}$ and $\boldsymbol{v}_{mm}$ defined as follows.

**Definition A.12.** (p-SVM)

$$\boldsymbol{p}_{mm} = \operatorname*{argmin}_{\boldsymbol{p}} \|\boldsymbol{p}\|$$

subjected to

$$\boldsymbol{p}^\top(\boldsymbol{\mu}_i - \boldsymbol{\xi}_i) \geq 1, i \in \mathcal{C}$$
$$\boldsymbol{p}^\top(\boldsymbol{\xi}_i - \boldsymbol{\mu}_i) \geq 1, i \in \mathcal{N} \tag{41}$$

for all $i \in [n]$. $\Xi = 1/\|\boldsymbol{p}_{mm}\|$ is the margin induced by $\boldsymbol{p}_{mm}$.

Then for a given $\boldsymbol{p}$, we define $\boldsymbol{v}(\boldsymbol{p})$ as the standard max-margin classifier on $(y_i, \boldsymbol{r}_i)_{i \in [n]}$ and $\boldsymbol{v}_{mm}$ as the standard max-margin classifier on $(y_i, \boldsymbol{r}_i^\star)_{i \in [n]}$ which can be understood as the limit scenario when $\boldsymbol{p} = \boldsymbol{p}_{mm}$ and $R \to +\infty$ .

**Definition A.13.** (v-SVM)

$$\boldsymbol{v}(\boldsymbol{p}) = \operatorname*{argmin}_{\boldsymbol{v} \in \mathbb{R}^d} \|\boldsymbol{v}\| \text{ s.t. } y_i \cdot \boldsymbol{v}^\top \boldsymbol{r}_i \geq 1, \quad \text{for all } i \in [n]. \tag{42}$$

$\Gamma(\boldsymbol{p}) = 1/\|\boldsymbol{v}(\boldsymbol{p})\|$ is the **label margin** induced by $\boldsymbol{v}$ and $\boldsymbol{p}$. When $\boldsymbol{r}_i = \boldsymbol{r}_i^\star, i \in [n]$,

$$\boldsymbol{v}_{mm} = \operatorname*{argmin}_{\boldsymbol{v} \in \mathbb{R}^d} \|\boldsymbol{v}\| \text{ s.t. } y_i \cdot \boldsymbol{v}^\top \boldsymbol{r}_i^\star \geq 1, \quad \text{for all } i \in [n]. \tag{43}$$

$\Gamma = 1/\|\boldsymbol{v}_{mm}\|$ is the label margin induced by $\boldsymbol{v}_{mm}$.

After proving the convergence direction of $\boldsymbol{p}_{(r,R)}$ and $\boldsymbol{v}_{(r,R)}$, we can utilize their properties similar to $\boldsymbol{p}_{mm}$ and $\boldsymbol{v}_{mm}$ to proceed with the training and test error analysis. Therefore proving that the model exhibits benign-overfitting.

It is worth noting that in the first part, we show the optimality of the token selection in (40) is strict in the sense that mixing other tokens in $\boldsymbol{r}_i$ will shrink the label margin. We formalize this into the following proposition:

**Proposition A.14** (Optimal Token Condition). *Under Condition 4.1, for all $\boldsymbol{p}$, the token selection under $\boldsymbol{p}$ results in a label margin of at most $\Gamma - \frac{C}{\|\boldsymbol{v}_{mm}\|^3 n \rho^2} \cdot \max_{i \in [n]}(1 - s_{i\alpha_i})$ in (A.13) where*

$$\alpha_i = \mathbb{I}(i \in \mathcal{C}) + 2\mathbb{I}(i \in \mathcal{N}) \text{ and } C > 0 \text{ is some constant.}$$

We now highlight some aspects of the technical novelty of our work compared to Ataee Tarzanagh et al. [6], Jiang et al. [25]. Unlike Ataee Tarzanagh et al. [6], our results are in a non-asymptotic setting, while their work focuses on the case where both $R, r \to \infty$. Additionally, we do not rely on any additional, unnatural assumptions about the data distribution. In contrast, Ataee Tarzanagh et al. [6] specifies the optimal token indices that achieve the maximum margin, and Jiang et al. [25] assumes that for each sample, the first noise token has a much larger norm than the other noise tokens.

We give detailed proof in the following.

**Optimal Token Condition**
Since $\boldsymbol{v}_{mm}$ satisfies the KKT conditions of the max-margin problem (42), by the stationarity condition, we can represent $\boldsymbol{v}_{mm}$ as

$$\boldsymbol{v}_{mm} = \lambda_1 \boldsymbol{\mu}_1 + \lambda_2 \boldsymbol{\mu}_2 + \sum_{i \in [n]} y_i \theta_i \boldsymbol{\xi}_i. \tag{44}$$

Note that the conditions in (42) can be written as:
*Condition* 1 (Optimal tokens).

$$\begin{cases} \boldsymbol{v}^\top \boldsymbol{\mu}_1 \geq 1 \\ -\boldsymbol{v}^\top \boldsymbol{\mu}_2 \geq 1 \\ y_i \boldsymbol{v}^\top \boldsymbol{\xi}_i \geq 1, i \in \mathcal{N} \end{cases}$$

Plugging (44) in the condition 1, we can rewrite these conditions as:

$$\begin{cases} \lambda_1 \cdot \|\boldsymbol{\mu}_1\|^2 \geq 1 \\ -\lambda_2 \cdot \|\boldsymbol{\mu}_2\|^2 \geq 1 \\ \theta_i \cdot \|\boldsymbol{\xi}_i\|^2 + y_i y_{i'} \sum_{i' \neq i} \theta_{i'} \langle \boldsymbol{\xi}_i, \boldsymbol{\xi}_{i'} \rangle \geq 1, i \in \mathcal{N} \end{cases}$$

Then we introduce a lemma to estimate the coefficients $\theta_i$ of $\boldsymbol{v}_{mm}$ under this condition:

**Lemma A.15** (balanced noise factor for KKT points). *Suppose that Assumption 4.1 holds, under Condition 1, we have that for $\boldsymbol{v}_{mm}$,*

$$\theta_i = 0, \quad i \in \mathcal{C}; \tag{45}$$

$$\theta_i \in \left[ \frac{(1-\kappa)d - 4n_2\sqrt{d\log(6n^2/\delta)}}{(1+\kappa)d((1-\kappa)d - 2n_2\sqrt{d\log(6n^2/\delta)})}, \frac{1}{(1-\kappa)d - 2n_2\sqrt{d\log(6n^2/\delta)}} \right], \quad i \in \mathcal{N}. \tag{46}$$

*Proof of Lemma A.15.* Note that Condition 1 does not have any constraint for samples with $i \in \mathcal{C}$. Thus we have $\theta_i = 0$ for any $i \in \mathcal{C}$ in the representation (44). For $\theta_i$ with $i \in \mathcal{N}$, we first prove the upper bound by contradiction. Denote $j = \operatorname*{argmax}_{i \in \mathcal{N}} \theta_i$. Then we have

$$y_j \boldsymbol{v}^\top \boldsymbol{\xi}_j = \sum_{i \in \mathcal{N}} y_i y_j \theta_i \langle \boldsymbol{\xi}_i, \boldsymbol{\xi}_j \rangle = \theta_j \|\boldsymbol{\xi}_j\|_2^2 + \sum_{i \neq j, i \in \mathcal{N}} y_i y_j \theta_i \langle \boldsymbol{\xi}_i, \boldsymbol{\xi}_j \rangle$$

$$\geq \theta_j \cdot (1-\kappa)d - n_2 \theta_j \cdot 2\sqrt{d\log(6n^2/\delta)},$$

where the inequality is from Lemma A.65 and the definition of $j$. Consider the contrary case when $\theta_j > \frac{1}{(1-\kappa)d - 2n_2\sqrt{d\log(6n^2/\delta)}}$, we have

$$y_j \boldsymbol{v}^\top \boldsymbol{\xi}_j > \frac{1}{(1-\kappa)d - 2n_2\sqrt{d\log(6n^2/\delta)}} \cdot \left((1-\kappa)d - n_2 \cdot 2\sqrt{d\log(6n^2/\delta)}\right) = 1.$$

By the complementary slackness, if $y_j \boldsymbol{v}^\top \boldsymbol{\xi}_j > 1$, then we must have $\theta_j = 0$, and thus we reach a contradiction.

Then we prove for the lower bound. For $\forall j \in \mathcal{N}$ we have

$$1 \le \theta_j \|\boldsymbol{\xi}_j\|_2^2 + \sum_{i \ne j, i \in \mathcal{N}} y_i y_j \theta_i \langle \boldsymbol{\xi}_i, \boldsymbol{\xi}_j \rangle$$

$$\le \theta_j \cdot (1+\kappa)d + n_2 \max_{i \in \mathcal{N}} \theta_i \cdot 2\sqrt{d\log(6n^2/\delta)}$$

$$\le \theta_j \cdot (1+\kappa)d + \frac{n_2}{(1-\kappa)d - 2n_2\sqrt{d\log(6n^2/\delta)}} \cdot 2\sqrt{d\log(6n^2/\delta)}.$$

The second inequality is due to Lemma A.65 and the last inequality is from the upper bound we just get. Therefore, we have

$$\theta_j \ge \frac{(1-\kappa)d - 4n_2\sqrt{d\log(6n^2/\delta)}}{(1+\kappa)d((1-\kappa)d - 2n_2\sqrt{d\log(6n^2/\delta)})}.$$

This completes the proof. $\qquad\square$

Then we introduce a lemma to estimate $\|\boldsymbol{v}_{mm}\|$:

**Lemma A.16** (Norm of $\boldsymbol{v}_{mm}$). *Suppose that Assumption 4.1 holds, for the solution $\boldsymbol{v}_{mm}$ of (42) under the token selection (40), we have*

$$\frac{2}{\rho^2} + \frac{\eta n}{2d} \le \|\boldsymbol{v}_{mm}\|^2 \le \frac{2}{\rho^2} + \frac{5\eta n}{d}.$$

*This implies*

$$\|\boldsymbol{v}_{mm}\| = \Theta\left(\sqrt{\frac{1}{\rho^2} + \frac{\eta n}{d}}\right).$$

*Proof of Lemma A.16.* As $\boldsymbol{v}_{mm}$ is the max-margin solution and satisfies KKT condition, it can be represented as

$$\boldsymbol{v}_{mm} = \lambda_1 \boldsymbol{\mu}_1 + \lambda_2 \boldsymbol{\mu}_2 + \sum_{i \in \mathcal{C}} y_i \theta_i \boldsymbol{\xi}_i + \sum_{i \in \mathcal{N}} y_i \theta_i \boldsymbol{\xi}_i. \qquad (47)$$

As $\boldsymbol{v}_{mm}$ satisfies Condition 1, we have $\lambda_1 \ge 1/\rho^2$ and $\lambda_2 \le -1/\rho^2$. So we could lower bound $\|\boldsymbol{v}_{mm}\|$ as

$$\|\boldsymbol{v}_{mm}\|^2 \ge \lambda_1^2 \|\boldsymbol{\mu}_1\|^2 + \lambda_2^2 \|\boldsymbol{\mu}_2\|^2 + \sum_{i \in \mathcal{N}} \theta_i^2 \|\boldsymbol{\xi}_i\|^2 + \sum_{i \in \mathcal{N}}\sum_{j \in \mathcal{N}} y_i y_j \theta_i \theta_j \langle \boldsymbol{\xi}_i, \boldsymbol{\xi}_j \rangle$$

$$\ge \frac{2}{\rho^2} + \frac{n_2(1-\kappa)}{d} + O\left(\frac{\eta^2 n^2}{d^{3/2}}\right) \ge \frac{2}{\rho^2} + \frac{\eta n}{2d}.$$

The second inequality is from Lemma A.15 that $\theta_i = \Theta(1/d)$ for $i \in \mathcal{N}$ and the last inequality is from Assumption 4.1.

Then to upper bound $\|\boldsymbol{v}_{mm}\|$, consider the following possible solution $\widetilde{\boldsymbol{v}}$

$$\widetilde{\boldsymbol{v}} = \rho^{-2}\boldsymbol{\mu}_1 - \rho^{-2}\boldsymbol{\mu}_2 + \sum_{i \in \mathcal{N}} 2y_i \boldsymbol{\xi}_i / d.$$

For $i \in \mathcal{C}$, we have

$$y_i \widetilde{\boldsymbol{v}}^\top \boldsymbol{r}_i = y_i \widetilde{\boldsymbol{v}}^\top \boldsymbol{\mu}_i \ge 1.$$

And for $i \in \mathcal{N}$, we have

$$y_i \widetilde{\boldsymbol{v}}^\top \boldsymbol{r}_i = y_i \widetilde{\boldsymbol{v}}^\top \boldsymbol{\xi}_i = 2\|\boldsymbol{\xi}_i\|^2/d + \sum_{j \in \mathcal{N}, j \neq i} 2y_i y_j \langle \boldsymbol{\xi}_i, \boldsymbol{\xi}_j \rangle/d$$

$$\geq 2(1-\kappa) - 2n_2\sqrt{\log(6n^2/\delta)/d} \geq 1.$$

The first inequality is from Lemma A.65 and the second inequality is from Assumption 4.1. Therefore, $\widetilde{\boldsymbol{v}}$ is a possible solution of SVM problem A.13 when $\boldsymbol{p}$ converges to $\boldsymbol{p}_{mm}$. So we have

$$\|\boldsymbol{v}_{mm}\|^2 \leq \|\widetilde{\boldsymbol{v}}\|^2 = 2/\rho^2 + \sum_{i \in \mathcal{N}} 4\|\boldsymbol{\xi}_i\|^2/d^2 + \sum_{i \in \mathcal{N}} \sum_{j \in \mathcal{N}} 4y_i y_j \langle \boldsymbol{\xi}_i, \boldsymbol{\xi}_j \rangle/d^2 \leq \frac{2}{\rho^2} + \frac{5\eta n}{d}.$$

The last inequality is from Lemma A.65, Lemma A.68 and Assumption 4.1. Combine the results above, we have $\|\boldsymbol{v}_{mm}\|^2 = \Theta(\frac{1}{\rho^2} + \frac{\eta n}{d})$. □

Based on the lemmas above, we introduce our main proposition in this section:

**Proposition A.17** (Optimal Token Condition). *Under Condition 4.1, for all $\boldsymbol{p}$, the token selection under $\boldsymbol{p}$ results in a label margin of at most $\Gamma - \frac{C}{\|\boldsymbol{v}_{mm}\|^3 n \rho^2} \cdot \max_{i \in [n]} (1 - s_{i\alpha_i})$ in (A.13) where* $\alpha_i = \mathbb{I}(i \in \mathcal{C}) + 2\mathbb{I}(i \in \mathcal{N})$ *and $C > 0$ is some constant.*

*Proof of Proposition A.14.* The main idea is to show the optimality of the token selection rule in the sense that mixing any other tokens will shrink the label margin. For a given $\boldsymbol{p}$, we say a sample $\boldsymbol{x}_i$ is a "mixed sample" if $\boldsymbol{r}_i \neq \boldsymbol{r}_i^\star$. We say $\boldsymbol{r}_i$ is a mixture of optimal token and non-optimal token in this case. Note that for any $\boldsymbol{p}$ with finite norm, $\boldsymbol{r}_i \neq \boldsymbol{r}_i^\star$. This notation is introduced for the clearness of the proof.

We use contradiction to prove Proposition A.14 by showing that any token selection different from (40) can only result in a strictly smaller label margin than that for the max-margin problem (42). Since $\boldsymbol{v}$ satisfies the KKT conditions of the max-margin problem, we can write $\boldsymbol{v}$ as

$$\boldsymbol{v} = \lambda_1 \boldsymbol{\mu}_1 + \lambda_2 \boldsymbol{\mu}_2 + \sum_{i \in \mathcal{C}} y_i \theta_i \boldsymbol{\xi}_i + \sum_{i \in \mathcal{N}} y_i \theta_i \boldsymbol{\xi}_i. \tag{48}$$

For a given $\boldsymbol{p}$, denote $\boldsymbol{v}'$ as the max-margin solution in (42), and $\Gamma' = 1/\|\boldsymbol{v}'\|$ as the new label margin. According to Lemma A.16, we have

$$\|\boldsymbol{v}_{mm}\|^2 = \Theta\left(\frac{1}{\rho^2} + \frac{\eta n}{d}\right) = \Omega(1/\rho^2).$$

Then we have

$$\Gamma - \frac{C}{\|\boldsymbol{v}_{mm}\|^3 n \rho^2} \cdot \max_{i \in [n]} (1 - s_{i\alpha_i}) \geq \Gamma - \frac{C}{\|\boldsymbol{v}_{mm}\|^3 n \rho^2} \geq \frac{\Gamma}{2}$$

for sufficiently large $d$. Here the last inequality uses $\|\boldsymbol{v}_{mm}\|^2 = \Omega(1/\rho^2)$. Thus we only need consider the case when the new label margin $\Gamma' \geq \Gamma/2$, or equivalently,

$$\|\boldsymbol{v}'\| \leq 2\|\boldsymbol{v}_{mm}\|. \tag{49}$$

Assume that there are $k$ samples ($0 < k \leq n$) that violate the token selection rule (40) and among them, $p$ samples are from clean set $\mathcal{C}$ and $k - p$ samples are from label-flipped set $\mathcal{N}$. Denote the indices of the $k$ samples as $I_v$. Then we consider the following three scenarios:

1. $p \neq 0, k - p = 0$. (All mixed samples come from $\mathcal{C}$)

2. $p = 0, k - p \neq 0$. (All mixed samples come from $\mathcal{N}$)

3. $p \neq 0, k - p \neq 0$. (Mixed samples are from both sets)

We will separately discuss each scenario and show that Proposition A.14 holds in all cases.
**Case 1:** $p \neq 0, k - p = 0$

Under this scenario, we have:

$$I_v \cap \mathcal{C} = I_v; \quad I_v \cap \mathcal{N} = \varnothing.$$

We proceed to analyze this scenario by dividing it into three distinct subcases.

- $p < n_1, I_v \cap \mathcal{C}_1 \neq \varnothing, I_v \cap \mathcal{C}_2 \neq \varnothing$

- $p < n_1, I_v \cap \mathcal{C}_i \neq \varnothing, I_v \cap \mathcal{C}_{i'} = \varnothing, (i, i' \in [2], i \neq i')$

- $p = n_1$

***Case 1.1*** $p < n_1, I_v \cap \mathcal{C}_1 \neq \varnothing, I_v \cap \mathcal{C}_2 \neq \varnothing$

In this case, both clusters exist clean samples that are not mixed. Denote the index of mixed samples $I_v$ as $\{k_1, k_2, ..., k_p\}$. For every mixed sample $k_i$, we have $\boldsymbol{r}_{k_i} = \beta_{k_i}\boldsymbol{\mu}_{k_i} + (1 - \beta_{k_i})\boldsymbol{\xi}_{k_i}$. Then the conditions under *Case 1.1* become

*Condition* 2 ($p$ clean samples violating optimal token selection).

$$\begin{cases} \boldsymbol{v}^\top \boldsymbol{\mu}_1 \geq 1 \\ -\boldsymbol{v}^\top \boldsymbol{\mu}_2 \geq 1 \\ y_i \boldsymbol{v}^\top \boldsymbol{\xi}_i \geq 1, i \in \mathcal{N} \\ y_i \boldsymbol{v}^\top \boldsymbol{r}_i \geq 1, i \in I_v \end{cases}$$

From the condition above, we could see that in this case, mixing one more clean sample is equal to adding one more constraint. Therefore, mixing $p$ samples will not result in a better solution than only mixing one sample, i.e. larger max-margin in our setting. So we can reduce this case to mixing only one clean sample with index $k^\star = \underset{i \in I_v}{\arg\min}\,\beta_i$. Denote $\boldsymbol{r}_{k^\star} = \beta\boldsymbol{\mu}_{k^\star} + (1 - \beta)\boldsymbol{\xi}_{k^\star}$ for some $\beta \in [0, 1)$.

Without loss of generality, we assume $\boldsymbol{\mu}_{k^\star} = \boldsymbol{\mu}_1, y_{k^\star} = +1$. Then the conditions become:

*Condition* 3 (one clean sample violating optimal token selection).

$$\begin{cases} \boldsymbol{v}^\top \boldsymbol{\mu}_1 \geq 1 \\ -\boldsymbol{v}^\top \boldsymbol{\mu}_2 \geq 1 \\ y_i \boldsymbol{v}^\top \boldsymbol{\xi}_i \geq 1, i \in \mathcal{N} \\ y_{k^\star} \boldsymbol{v}^\top \boldsymbol{r}_{k^\star} \geq 1 \end{cases}$$

Denote $\boldsymbol{v}'$ as the optimal solution under this condition. $\boldsymbol{v}'$ can also be written in the form of (48) with coefficients denoted as $\lambda'_1, \lambda'_2$ and $\theta'_i, i \in [n]$. Plugging this representation into the condition 3, we have:

$$\begin{cases} \lambda'_1 \cdot \|\boldsymbol{\mu}_1\|^2 \geq 1 \\ -\lambda'_2 \cdot \|\boldsymbol{\mu}_2\|^2 \geq 1 \\ \theta'_i \cdot \|\boldsymbol{\xi}_i\|^2 + \sum_{i' \neq i} y_i y_{i'} \theta'_{i'} \langle \boldsymbol{\xi}_i, \boldsymbol{\xi}_{i'} \rangle \geq 1, i \in \mathcal{N} \\ \beta\lambda'_1 \cdot \|\boldsymbol{\mu}_1\|^2 + (1 - \beta)(\theta'_{k^\star}\|\boldsymbol{\xi}_{k^\star}\|^2 + \sum_{i \neq k^\star} y_{k^\star} y_i \theta'_i \langle \boldsymbol{\xi}_i, \boldsymbol{\xi}_{k^\star} \rangle) \geq 1 \end{cases}$$

First, we introduce another lemma similar to Lemma A.15 to characterize the scale of $\theta'_i, i \in [n]$ in this case.

**Lemma A.18.** *Suppose that Assumption 4.1 holds, under Condition 3, we have*

$$\theta'_i = 0, \quad i \in \mathcal{C}\backslash\{k^\star\};$$

$$\theta_i \in \left[\frac{(1 - \kappa)d - 4n_2\sqrt{d\log(6n^2/\delta)}}{(1 + \kappa)d((1 - \kappa)d - 2n_2\sqrt{d\log(6n^2/\delta)})}, \frac{1}{(1 - \kappa)d - 2n_2\sqrt{d\log(6n^2/\delta)}}\right], \quad i \in \mathcal{N}.$$

*Proof of Lemma A.18.* Same as Condition 1, Condition 3 does not have any constraint for samples with $i \in \mathcal{C}\backslash\{k^\star\}$. Thus we have $\theta'_i = 0$ for any $i \in \mathcal{C}\backslash\{k^\star\}$.

Meanwhile, Condition 3 introduces an additional constraint compared to Condition 1. Consequently, the feasible region for $\{\theta'_i\}_{i \in \mathcal{N}}$ under Condition 3 is a subset of the feasible region for $\{\theta_i\}_{i \in \mathcal{N}}$ under Condition 1. Therefore, the bounds established in Lemma A.15 remain applicable to $\{\theta'_i\}_{i \in \mathcal{N}}$. $\square$

From this lemma, We can see that $\theta'_i = \Theta(1/d)$ for $i \in \mathcal{N}$. To proceed, we introduce a crucial lemma:

**Lemma A.19.** *Suppose that Assumption 4.1 holds, denote $\boldsymbol{v}$ and $\boldsymbol{v}'$ as the optimal solutions under condition 1 and condition 3 respectively. We have*

$$\|\boldsymbol{v}'\|_2^2 - \|\boldsymbol{v}_{mm}\|_2^2 \geq \frac{C_1(1 - \beta\lambda_1'\rho^2)^2}{(1-\beta)^2(1+\kappa)d} + \widetilde{O}\Big(\frac{\eta n}{d^{3/2}}\Big).$$

*where $0 < C_1 \leq 1$ is a constant.*

*Proof of Lemma A.19.* We consider two cases under this scenario:

- $\theta_k' = 0$ in $\boldsymbol{v}'$

  In this case, from Lemma A.18 we have $\beta\lambda_1' \geq (1 + o(1))/\rho^2$ and all other conditions are the same as the optimal selection. In order to get $\min \|\boldsymbol{v}\|$, we have $\lambda_1' = (1 + o(1))/\beta\rho^2$. Consider another solution $\boldsymbol{v}_0$ which has parameters $\lambda_{01} = 1/\rho^2$, $\lambda_{02} = \lambda_2'$, $\theta_{0i} = \theta_i'(i \in [n])$. As $\boldsymbol{v}_0$ satisfies all the inequities under Condition 1, we have $\Gamma_0 \leq \Gamma$ So we have

  $$\Gamma^2 - \Gamma'^2 \geq \Gamma_0^2 - \Gamma'^2 = \frac{1}{\|\boldsymbol{v}_0\|^2} - \frac{1}{\|\boldsymbol{v}'\|^2} = \frac{(\lambda_{01}^2 - \lambda_1'^2) \cdot \|\boldsymbol{\mu}_1\|^2}{\|\boldsymbol{v}_0\|^2 \cdot \|\boldsymbol{v}'\|^2}$$

  $$= \frac{(1 + o(1))/\beta^2 - 1}{\|\boldsymbol{v}_0\|^2 \cdot \|\boldsymbol{v}'\|^2} = \frac{(1 + \beta)(1 - \beta) + o(1)}{\beta^2\|\boldsymbol{v}_0\|^2 \cdot \|\boldsymbol{v}'\|^2} \geq \frac{1 - \beta}{\|\boldsymbol{v}_0\|^2 \cdot \|\boldsymbol{v}'\|^2}.$$

  Therefore,

  $$\Gamma - \Gamma' \geq \frac{1 - \beta}{(\Gamma_0 + \Gamma')\|\boldsymbol{v}_0\|^2 \cdot \|\boldsymbol{v}'\|^2} \geq \frac{1 - \beta}{2\Gamma_0\|\boldsymbol{v}_0\|^2 \cdot \|\boldsymbol{v}'\|^2}.$$

  Set $c = \frac{1}{2\Gamma_0\|\boldsymbol{v}_0\|^2 \cdot \|\boldsymbol{v}'\|^2} = \frac{1}{2\|\boldsymbol{v}_0\|\|\boldsymbol{v}'\|^2}$. we have $\Gamma' \leq \Gamma - c(1 - \beta)$. Moreover, we could upper bound $c$ as

  $$c = \frac{1}{2\|\boldsymbol{v}_0\|\|\boldsymbol{v}'\|^2} \leq \frac{1}{2r_{mm}^3}.$$

  The last inequality is from $\|\boldsymbol{v}'\| \geq \|\boldsymbol{v}_0\| \geq r_{mm}$.

- $\theta_k' \neq 0$ in $\boldsymbol{v}'$

  From KKT condition, we have

  $$\theta_{k^*}' \cdot \Big[\beta\lambda_1' \cdot \|\boldsymbol{\mu}_1\|^2 + (1 - \beta)(\theta_{k^*}'\|\boldsymbol{\xi}_{k^*}\|^2 + \sum_{i \neq k^*} y_{k^*} y_i \theta_i'\langle\boldsymbol{\xi}_i, \boldsymbol{\xi}_{k^*}\rangle) - 1\Big] = 0.$$

  As $\theta_{k^*}' > 0$, we have

  $$\beta\lambda_1' \cdot \|\boldsymbol{\mu}_1\|^2 + (1 - \beta)(\theta_{k^*}'\|\boldsymbol{\xi}_{k^*}\|^2 + \sum_{i \in \mathcal{N}} y_{k^*} y_i \theta_i'\theta_i'\langle\boldsymbol{\xi}_i, \boldsymbol{\xi}_{k^*}\rangle) = 1.$$

  So we can estimate $\theta_{k^*}'$ as

  $$\theta_{k^*}'\|\boldsymbol{\xi}_{k^*}\|^2 = \frac{1 - \beta\lambda_1'\rho^2}{1 - \beta} - \sum_{i \in \mathcal{N}} y_{k^*} y_i \theta_i'\theta_i'\langle\boldsymbol{\xi}_i, \boldsymbol{\xi}_{k^*}\rangle \leq \frac{1 - \beta\lambda_1'\rho^2}{1 - \beta} + 2n_2 \max_{i \in \mathcal{N}} \theta_i' \sqrt{d\log(6n^2/\delta)}$$

  $$= \frac{1 - \beta\lambda_1'\rho^2}{1 - \beta} + \frac{2n_2\sqrt{d\log(6n^2/\delta)}}{(1 - \kappa)d - 2n_2\sqrt{d\log(6n^2/\delta)}}. \tag{50}$$

  The first inequality is from Lemma A.65 and the last equality is from Lemma A.18. We can also lower bound it as

  $$\theta_{k^*}'\|\boldsymbol{\xi}_{k^*}\|^2 = \frac{1 - \beta\lambda_1'\rho^2}{1 - \beta} - \sum_{i \in \mathcal{N}} y_{k^*} y_i \theta_i'\theta_i'\langle\boldsymbol{\xi}_i, \boldsymbol{\xi}_{k^*}\rangle \geq \frac{1 - \beta\lambda_1'\rho^2}{1 - \beta} - 2n_2 \max_{i \in \mathcal{N}} \theta_i' \sqrt{d\log(6n^2/\delta)}$$

  $$= \frac{1 - \beta\lambda_1'\rho^2}{1 - \beta} - \frac{2n_2\sqrt{d\log(6n^2/\delta)}}{(1 - \kappa)d - 2n_2\sqrt{d\log(6n^2/\delta)}}. \tag{51}$$

The first inequality is from Lemma A.65 and the last equality is from Lemma A.18. Therefore, we have $\theta'_{k^*} = \Theta(\frac{1-\beta\lambda'_1\rho^2}{(1-\beta)d}) \pm O(\frac{\eta n}{d^{3/2}})$.

Then from the third inequality in Condition 3, we have

$$\theta'_i \cdot \|\boldsymbol{\xi}_i\|^2 + \sum_{i'\in\mathcal{N},i'\neq i} y_i y_{i'}\theta'_{i'}\langle\boldsymbol{\xi}_i,\boldsymbol{\xi}_{i'}\rangle \geq 1 - y_i y_{k^*}\theta'_{k^*}\langle\boldsymbol{\xi}_i,\boldsymbol{\xi}_{k^*}\rangle$$

$$\geq 1 - \left[\frac{1-\beta\lambda'_1\rho^2}{(1-\beta)(1+\kappa)d} + O\left(\frac{\eta n}{d^{3/2}}\right)\right]\cdot|\langle\boldsymbol{\xi}_i,\boldsymbol{\xi}_{k^*}\rangle|$$

$$\geq 1 - \frac{2(1-\beta\lambda'_1\rho^2)\sqrt{\log(6n^2/\delta)}}{(1-\beta)(1+\kappa)\sqrt{d}} - \widetilde{O}\left(\frac{\eta n}{d}\right)$$

$$\geq 1 - \frac{2\sqrt{\log(6n^2/\delta)}}{\sqrt{d}} - \widetilde{O}\left(\frac{\eta n}{d}\right)$$

$$= 1 - \frac{3\sqrt{\log(6n^2/\delta)}}{\sqrt{d}}. \tag{52}$$

The second inequality is from (50); The third inequality is from Lemma A.65 and the last inequality is from the first inequality in Condition 3 that $\lambda'_1\rho^2 \geq 1$.

Consider $\widetilde{\boldsymbol{v}} = \widetilde{\lambda}_1\boldsymbol{\mu}_1 + \widetilde{\lambda}_2\boldsymbol{\mu}_2 + \sum_{i\in[n]} y_i\widetilde{\theta}_i\boldsymbol{\xi}_i$, which has $\widetilde{\lambda}_1 = \lambda'_1, \widetilde{\lambda}_2 = \lambda'_2, \widetilde{\theta}_i = \theta'_i/(1 - \frac{3\sqrt{\log(6n^2/\delta)}}{\sqrt{d}})$ for $i \in \mathcal{N}$ and $\widetilde{\theta}'_i = 0$ for $i \in \mathcal{C}$. We can verify that $\widetilde{\boldsymbol{v}}$ satisfies all conditions for $\boldsymbol{v}_{mm}$. For $\forall i \in \mathcal{N}$, we have

$$\widetilde{\theta}_i \cdot \|\boldsymbol{\xi}_i\|^2 + \sum_{i'\in\mathcal{N},i'\neq i} y_i y_{i'}\widetilde{\theta}_{i'}\langle\boldsymbol{\xi}_i,\boldsymbol{\xi}_{i'}\rangle$$

$$= \left[\theta'_i \cdot \|\boldsymbol{\xi}_i\|^2 + \sum_{i'\in\mathcal{N},i'\neq i} y_i y_{i'}\theta'_{i'}\langle\boldsymbol{\xi}_i,\boldsymbol{\xi}_{i'}\rangle\right]/\left(1 - \frac{3\sqrt{\log(6n^2/\delta)}}{\sqrt{d}}\right) \geq 1.$$

The last inequality is from (52). Meanwhile, we have $\widetilde{\lambda}_1\|\boldsymbol{\mu}_1\|^2 = \lambda'_1\|\boldsymbol{\mu}_1\|^2 \geq 1$, $-\widetilde{\lambda}_2\|\boldsymbol{\mu}_2\|^2 = -\lambda'_2\|\boldsymbol{\mu}_2\|^2 \geq 1$. So $\widetilde{\boldsymbol{v}}$ is a possible solution for Condition 3, which implies $\|\boldsymbol{v}_{mm}\| \leq \|\widetilde{\boldsymbol{v}}\|$.

Next we estimate the difference between $\|\boldsymbol{v}'\|^2$ and $\|\widetilde{\boldsymbol{v}}\|^2$. We write the expansion of $\|\widetilde{\boldsymbol{v}}\|^2$ and $\|\boldsymbol{v}'\|^2$:

$$\|\widetilde{\boldsymbol{v}}\|^2 = \widetilde{\lambda}_1^2\|\boldsymbol{\mu}_1\|^2 + \widetilde{\lambda}_2^2\|\boldsymbol{\mu}_2\|^2 + \sum_{i\in\mathcal{N}}\widetilde{\theta}_i^2\|\boldsymbol{\xi}_i\|^2 + \sum_{i,j\in\mathcal{N};i\neq j} y_i y_j\widetilde{\theta}_i\widetilde{\theta}_j\langle\boldsymbol{\xi}_i,\boldsymbol{\xi}_j\rangle,$$

$$\|\boldsymbol{v}'\|^2 = \lambda_1'^2\|\boldsymbol{\mu}_1\|^2 + \lambda_2'^2\|\boldsymbol{\mu}_2\|^2 + \sum_{i\in\mathcal{N}\cup\{k^\star\}}\theta_i'^2\|\boldsymbol{\xi}_i\|^2 + \sum_{i,j\in\mathcal{N}\cup\{k^\star\};i\neq j} y_i y_j\theta'_i\theta'_j\langle\boldsymbol{\xi}_i,\boldsymbol{\xi}_j\rangle.$$

From the construction of $\widetilde{\boldsymbol{v}}$, we have $\lambda'_1 = \lambda_1, \lambda'_2 = \lambda_2$. So we have

$$\|\boldsymbol{v}'\|^2 - \|\widetilde{\boldsymbol{v}}\|^2 \geq \theta_{k^*}'^2\|\boldsymbol{\xi}_{k^*}\|^2 + \underbrace{\sum_{i\in\mathcal{N}}(\theta_i'^2 - \widetilde{\theta}_i^2)\|\boldsymbol{\xi}_i\|^2}_{I_1} + \underbrace{\sum_{i\in\mathcal{N}\cup\{k^\star\}}\sum_{j\in\mathcal{N}\cup\{k^\star\}\setminus\{i\}} y_i y_j\theta'_i\theta'_j\langle\boldsymbol{\xi}_i,\boldsymbol{\xi}_j\rangle}_{I_2}$$

$$- \underbrace{\sum_{i\in\mathcal{N}}\sum_{j\in\mathcal{N}\setminus\{i\}} y_i y_j\widetilde{\theta}_i\widetilde{\theta}_j\langle\boldsymbol{\xi}_i,\boldsymbol{\xi}_j\rangle}_{I_3}.$$

From (51), we have

$$\theta'_{k^\star}\|\boldsymbol{\xi}_{k^\star}\| \geq \frac{1-\beta\lambda'_1\rho^2}{(1-\beta)\sqrt{(1+\kappa)d}} - \widetilde{O}\left(\frac{\eta n}{d}\right).$$

We then bound the last three terms respectively. First we have

$$|I_1| = \sum_{i \in \mathcal{N}} (\widetilde{\theta}_i^2 - \theta_i'^2) \|\boldsymbol{\xi}_i\|^2 \le \left( \frac{1}{(1 - \widetilde{O}(1/\sqrt{d}))^2} - 1 \right) \cdot \sum_{i \in \mathcal{N}} \theta_i'^2 \|\boldsymbol{\xi}_i\|^2$$

$$\le \frac{\widetilde{O}(1/\sqrt{d})}{(1 - \widetilde{O}(1/\sqrt{d}))^2} \cdot \frac{n_2(1 + \kappa)d}{\left( (1 - \kappa)d - 2n_2\sqrt{d \log(6n^2/\delta)} \right)^2}$$

$$= \widetilde{O}\left( \frac{\eta n}{d^{3/2}} \right).$$

The first inequality is from the definition of $\widetilde{\theta}_i$; The second inequality is from Lemma A.15 and Lemma A.65.

Then we bound $|I_2 - I_3|$ as:

$$|I_2 - I_3| = \sum_{i \in \mathcal{N}} \sum_{j \in \mathcal{N} \setminus \{i\}} (\widetilde{\theta}_i \widetilde{\theta}_j - \theta_i' \theta_j') \cdot |\langle \boldsymbol{\xi}_i, \boldsymbol{\xi}_j \rangle| + \theta_k' \sum_{i \in \mathcal{N}} \theta_i' |\langle \boldsymbol{\xi}_{k^*}, \boldsymbol{\xi}_i \rangle|$$

$$\le \left( \frac{1}{(1 - \widetilde{O}(1/\sqrt{d}))^2} - 1 \right) \sum_{i \in \mathcal{N}} \sum_{j \in \mathcal{N} \setminus \{i\}} \theta_i' \theta_j' \cdot |\langle \boldsymbol{\xi}_i, \boldsymbol{\xi}_j \rangle| + n_2 \theta_{k^*}' \cdot \max_{i \in \mathcal{N}} \theta_i' \cdot |\langle \boldsymbol{\xi}_{k^*}, \boldsymbol{\xi}_i \rangle|$$

$$\le \frac{\widetilde{O}(1/\sqrt{d})}{(1 - \widetilde{O}(1/\sqrt{d}))^2} \cdot \frac{(n_2)^2 2\sqrt{d \log(6n^2/\delta)}}{\left( (1 - \kappa)d - 2\eta n \sqrt{d \log(6n^2/\delta)} \right)^2} + \theta_{k^*}' \cdot \Theta\left( \frac{\eta n}{\sqrt{d}} \right)$$

$$= \widetilde{O}\left( \frac{\eta^2 n^2}{d^2} \right) + \Theta\left( \frac{\eta n}{d^{3/2}} \right)$$

$$= \widetilde{O}\left( \frac{\eta n}{d^{3/2}} \right).$$

The first inequality is from the definition of $\widetilde{\theta}_i$; The second inequality is from Lemma A.15 and Lemma A.65. Combining the above results, we finally have

$$\|\boldsymbol{v}'\|_2^2 - \|\boldsymbol{v}_{mm}\|_2^2 \ge \frac{C_1(1 - \beta\lambda_1'\rho^2)^2}{(1 - \beta)^2(1 + \kappa)d} + \widetilde{O}\left( \frac{\eta n}{d^{3/2}} \right).$$

$\square$

Now we can prove the main proposition in this case.

*Proof of Proposition A.14 in Case 1.1.* From Lemma A.19 we have

$$\|\boldsymbol{v}'\|_2^2 - \|\boldsymbol{v}_{mm}\|_2^2 \ge \frac{C_1(1 - \beta\lambda_1'\rho^2)^2}{(1 - \beta)^2(1 + \kappa)d} + o\left( \frac{1}{d} \right) \ge \frac{C_1(1 - \beta\lambda_1'\rho^2)^2}{(1 + \kappa)d}(1 - \beta) = T(1 - \beta).$$

In the last equation we substitute $T = \frac{C_1(1 - \beta\lambda_1'\rho^2)^2}{(1 + \kappa)d} \ge 0$. Then we have

$$\Gamma^2 - \Gamma'^2 = \frac{1}{\|\boldsymbol{v}_{mm}\|^2} - \frac{1}{\|\boldsymbol{v}'\|^2} = \frac{\|\boldsymbol{v}'\|^2 - \|\boldsymbol{v}_{mm}\|^2}{\|\boldsymbol{v}_{mm}\|^2 \cdot \|\boldsymbol{v}'\|^2} \ge \frac{T(1 - \beta)}{\|\boldsymbol{v}_{mm}\|^2 \cdot \|\boldsymbol{v}'\|^2}.$$

Therefore,

$$\Gamma - \Gamma' \ge \frac{T(1 - \beta)}{(\Gamma + \Gamma')\|\boldsymbol{v}_{mm}\|^2 \cdot \|\boldsymbol{v}'\|^2} \ge \frac{T(1 - \beta)}{2\Gamma\|\boldsymbol{v}_{mm}\|^2 \cdot \|\boldsymbol{v}'\|^2} = \frac{T(1 - \beta)}{2\|\boldsymbol{v}_{mm}\|\|\boldsymbol{v}'\|^2} \ge \frac{T(1 - \beta)}{2\|\boldsymbol{v}'\|^3}.$$

The last inequality is from $\|\boldsymbol{v}'\| \ge \|\boldsymbol{v}_{mm}\|$. This implies

$$\Gamma' \le \Gamma - \frac{T(1 - \beta)}{2\|\boldsymbol{v}'\|^3} \le \Gamma - \frac{C_1}{\|\boldsymbol{v}_{mm}\|^3 n\rho^2}(1 - \beta).$$

The last inequality is from our assumption that $\|\boldsymbol{v}'\| \le 2\|\boldsymbol{v}_{mm}\|$ and $\rho^2 = \Omega(d/n)$. $\square$

Next we consider the other case.

***Case 1.2*** $p = n_1$

Next we consider the case when all clean samples are mixed. In this case, all samples in clean set are mixed, so the first two inequalities in Condition 3 do not hold, which means that $\lambda'_1$ may be smaller than $\lambda_1$. But we could still prove that Lemma A.19 holds. We first write down the condition in this case:

*Condition* 4 (All clean samples violate optimal token selection rule).

$$\begin{cases} y_i \boldsymbol{v}^\top \boldsymbol{\xi}_i \geq 1, i \in \mathcal{N} \\ y_i \boldsymbol{v}^\top \boldsymbol{r}_i \geq 1, i \in \mathcal{C} \end{cases}$$

Plugging the representation (48) into the condition, we have:

$$\begin{cases} \theta'_i \cdot \|\boldsymbol{\xi}_{i'}\|^2 + \sum\limits_{i' \neq i} y_i y_{i'} \theta'_{i'} \langle \boldsymbol{\xi}_i, \boldsymbol{\xi}_{i'} \rangle \geq 1, i \in \mathcal{N} \\ \beta_i \lambda'_i \cdot \|\boldsymbol{\mu}_i\|^2 + (1-\beta_i)(\theta'_i \cdot \|\boldsymbol{\xi}_i\|^2 + \sum\limits_{j \neq i} y_i y_j \theta'_i \langle \boldsymbol{\xi}_i, \boldsymbol{\xi}_j \rangle) \geq 1, i \in \mathcal{C} \end{cases}$$

*Proof of Lemma A.19.* First we assume that $\max\{\lambda'_1 \cdot \|\boldsymbol{\mu}_1\|^2, -\lambda'_2 \cdot \|\boldsymbol{\mu}_2\|^2\} = q$ in optimal $\boldsymbol{v}'$. If $q \geq 1$, this is the same as *Case 1.3*. So we assume that $q \leq 1$. Denote $k^\star = \underset{i \in \mathcal{C}}{\operatorname{argmin}} \frac{1-\beta_i q}{1-\beta_i}$ and $\beta = \beta_{k^\star}$, consider the following condition

*Condition* 5 (Relaxed version of Condition 4).

$$\begin{cases} \theta'_i \cdot \|\boldsymbol{\xi}_{i'}\|^2 + \sum\limits_{i' \neq i} y_i y_{i'} \theta'_{i'} \langle \boldsymbol{\xi}_i, \boldsymbol{\xi}_{i'} \rangle \geq 1, i \in \mathcal{N} \\ \theta'_i \cdot \|\boldsymbol{\xi}_{i'}\|^2 + \sum\limits_{i' \neq i} y_i y_{i'} \theta'_{i'} \langle \boldsymbol{\xi}_i, \boldsymbol{\xi}_{i'} \rangle \geq \frac{1-\beta q}{1-\beta}, i \in \mathcal{C} \end{cases}$$

Compared with Condition 4, the second inequality is relaxed for $i \in \mathcal{C}$. Therefore, denote the max-margin solution as $\widehat{\boldsymbol{v}}$ under Condition 5, we must have $\|\widehat{\boldsymbol{v}}\| \leq \|\boldsymbol{v}'\|$. Then we will prove that Lemma A.19 still holds between $\|\boldsymbol{v}_{mm}\|$ and $\|\widehat{\boldsymbol{v}}\|$, which indicates $\|\boldsymbol{v}'\|_2^2 - \|\boldsymbol{v}_{mm}\|_2^2 \geq \|\widehat{\boldsymbol{v}}\|_2^2 - \|\boldsymbol{v}_{mm}\|_2^2 \geq \frac{C_1(1-\beta\lambda'_1\rho^2)^2}{(1-\beta)^2(1+\kappa)d} + o(\frac{1}{d})$. Denote the parameters in $\widehat{\boldsymbol{v}}$ are $\widehat{\lambda}_1, \widehat{\lambda}_2$ and $\widehat{\theta}_i$, we first introduce the following lemma to estimate $\widehat{\theta}_i$. Here we denote $\alpha = \frac{1-\beta q}{1-\beta}$ for convenience.

**Lemma A.20.** *Suppose that Assumption 4.1 holds, under Condition 5, we have*

$$\widehat{\theta}_i \in \left[ \frac{\alpha}{(1+\kappa)d} \left( 1 - \frac{2n\sqrt{d\log(6n^2/\delta)}}{(1-\kappa)d - 2n\sqrt{d\log(6n^2/\delta)}} \right), \frac{\alpha}{((1-\kappa)d - 2n\sqrt{d\log(6n^2/\delta)})} \right], i \in \mathcal{C},$$

$$\widehat{\theta}_i \in \left[ \frac{1}{(1+\kappa)d} \left( 1 - \frac{2\alpha n\sqrt{d\log(6n^2/\delta)}}{(1-\kappa)d - 2n\sqrt{d\log(6n^2/\delta)}} \right), \frac{\alpha}{((1-\kappa)d - 2n\sqrt{d\log(6n^2/\delta)})} \right], i \in \mathcal{N}.$$

*Proof of Lemma A.20.* Denote $j = \underset{i \in [n]}{\operatorname{argmax}} \widehat{\theta}_i$, we have

$$\widehat{\theta}_i \cdot \|\boldsymbol{\xi}_i\|^2 + \sum_{j \neq i} y_i y_j \widehat{\theta}_i \langle \boldsymbol{\xi}_i, \boldsymbol{\xi}_j \rangle \geq \widehat{\theta}_j \|\boldsymbol{\xi}_j\|^2 - n\widehat{\theta}_j \sqrt{d\log(6n^2/\delta)}$$

$$\geq \widehat{\theta}_j ((1-\kappa)d - 2n\sqrt{d\log(6n^2/\delta)}).$$

The two inequalities are from Lemma A.65 and our definition of j. Consider the contrary case when $\widehat{\theta}_j > \frac{\alpha}{((1-\kappa)d - 2n\sqrt{d\log(6n^2/\delta)})}$, we have

$$y_j \widehat{\boldsymbol{v}}^\top \boldsymbol{\xi}_j > \alpha.$$

By the complementary slackness condition, if $y_j \widehat{\boldsymbol{v}}^\top \boldsymbol{\xi}_j > \alpha \geq 1$, then we must have $\widehat{\theta}_j = 0$, and thus we reach a contradiction.

Then we lower bound $\widehat\theta_i$, for $i \in \mathcal{C}$ we have

$$\alpha \le \widehat\theta_i \cdot \|\boldsymbol\xi_i\|^2 + \sum_{j\ne i} y_i y_j \widehat\theta_i \langle \boldsymbol\xi_i, \boldsymbol\xi_j \rangle \le \widehat\theta_i(1+\kappa)d + 2n \max_{i\in[n]} \widehat\theta_i \sqrt{d\log(6n^2/\delta)}$$

$$\le \widehat\theta_i(1+\kappa)d + \frac{2\alpha n\sqrt{d\log(6n^2/\delta)}}{(1-\kappa)d - 2n\sqrt{d\log(6n^2/\delta)}}.$$

The second inequality is from Lemma A.65 and the last inequality is from the upper bound of $\widehat\theta_i$ we just derived. Therefore, we have

$$\widehat\theta_i \ge \frac{\alpha}{(1+\kappa)d}\left(1 - \frac{2n\sqrt{d\log(6n^2/\delta)}}{(1-\kappa)d - 2n\sqrt{d\log(6n^2/\delta)}}\right).$$

Similarly, for $i \in \mathcal{N}$, we have

$$\widehat\theta_i \ge \frac{1}{(1+\kappa)d}\left(1 - \frac{2\alpha n\sqrt{d\log(6n^2/\delta)}}{(1-\kappa)d - 2n\sqrt{d\log(6n^2/\delta)}}\right).$$

$\square$

Note that we only consider the case when $\|\widehat{\boldsymbol v}\| \le \|\boldsymbol v'\| \le 2\|\boldsymbol v_{mm}\|$. And from Lemma A.20 we have $\widehat\theta_i = \Theta(\alpha/d)$ for $i \in \mathcal{C}$. So we must have $\alpha = O(\log n)$ is some constant. Otherwise, for $i \in \mathcal{C}$ we have

$$\widehat\theta_i \|\boldsymbol\xi_i\|^2 \ge \alpha - \sum_{i'\ne i} y_i y_{i'} \widehat\theta_i \langle \boldsymbol\xi_i, \boldsymbol\xi_{i'} \rangle = \Omega(\alpha).$$

It further yields that

$$\|\widehat{\boldsymbol v}\|^2 = \Omega(\frac{1}{\rho^2}) + \Omega(\frac{\eta n}{d}) + \sum_{i\in\mathcal{C}} \widehat\theta_i^2 \|\boldsymbol\xi_i\|^2 = \Omega(\frac{1}{\rho^2} + \frac{\eta n}{d} + \frac{n\alpha^2}{d}) = \Omega(\frac{n\log^2 n}{d}), \qquad (53)$$

which contradicts with $\|\boldsymbol v''\| = \Theta(\sqrt{1/\rho^2 + \eta n/d})$.

Then the difference between $\|\boldsymbol v_{mm}\|_2^2$ and $\|\widehat{\boldsymbol v}\|_2^2$ becomes

$$\|\widehat{\boldsymbol v}\|^2 - \|\boldsymbol v_{mm}\|^2 \ge \sum_{i\in\mathcal{C}} \widehat\theta_i^2 \|\boldsymbol\xi_i\|^2 - 2/\rho^2 + \underbrace{\sum_{i\in\mathcal{N}} (\widehat\theta_i^2 - \theta_i^2)\|\boldsymbol\xi_i\|^2}_{I_1} + \underbrace{\sum_{i\in[n]}\sum_{j\in[n]\setminus\{i\}} y_i y_j \widehat\theta_i \widehat\theta_j \langle \boldsymbol\xi_i, \boldsymbol\xi_j \rangle}_{I_2}$$

$$- \underbrace{\sum_{i\in\mathcal{N}}\sum_{j\in\mathcal{N}\setminus\{i\}} y_i y_j \theta_i \theta_j \langle \boldsymbol\xi_i, \boldsymbol\xi_j \rangle}_{I_3}.$$

We will bound every term sequentially. For $i \in \mathcal{C}$, we have

$$\widehat\theta_i \|\boldsymbol\xi_i\|^2 \ge \alpha - \sum_{i'\in[n], i'\ne i} y_i \widehat\theta_{i'} \langle \boldsymbol\xi_i, \boldsymbol\xi_{i'} \rangle \ge \alpha - n\max_{i\in[n]} \widehat\theta_i \cdot 2\sqrt{d\log(6n^2/\delta)}$$

$$= \alpha - \frac{2\alpha n\sqrt{\log(6n^2/\delta)}}{(1-\kappa)\sqrt{d} - 2n\sqrt{\log(6n^2/\delta)}} = \alpha - \widetilde{O}\left(\frac{n}{\sqrt{d}}\right).$$

The second inequality is from Lemma A.65; The first equality is from Lemma A.18 and the last equality is from Assumption 4.1. This implies

$$\sum_{i\in\mathcal{C}} \widehat\theta_i^2 \|\boldsymbol\xi_i\|^2 - 2/\rho^2 \ge \frac{n_1\alpha^2}{(1+\kappa)d} - \frac{2}{\rho^2} - \widetilde{O}\left(\frac{n}{d^{3/2}}\right) \ge \frac{C_2 n_1\alpha^2}{(1+\kappa)d} - \widetilde{O}\left(\frac{n}{d^{3/2}}\right).$$

The second inequality is due to the SNR condition $\rho/\sqrt{d} = \Omega(1/\sqrt{n})$ so there exists a constant $C_2$ that $\frac{2}{\rho^2} \le \frac{(1-C_2)n_1\alpha^2}{(1+\kappa)d}$.

Then for $|I_1|$ we have

$$|I_1| \leq (\max_{i \in \mathcal{N}} \theta_i^2 - \min_{i \in \mathcal{N}} \widehat{\theta}_i^2) \sum_{i \in \mathcal{N}} \|\boldsymbol{\xi}_i\|^2$$

$$\leq \left( \left( \frac{1}{(1-\kappa)d - 2\eta n\sqrt{d \log(6n^2/\delta)}} \right)^2 - \left( \frac{1}{(1+\kappa)d} \left( 1 - \frac{2n\sqrt{d \log(6n^2/\delta)}}{(1-\kappa)d - 2n\sqrt{d \log(6n^2/\delta)}} \right) \right)^2 \right) \cdot n_2(1+\kappa)d$$

$$\leq \left( \frac{\sqrt{(1+\kappa)d}}{(1-\kappa)d - 2\eta n\sqrt{d \log(6n^2/\delta)}} \right)^2 \left( 1 - \left( \frac{(1-\kappa)d - 4\eta n\sqrt{d \log(6n^2/\delta)}}{(1+\kappa)d} \right)^2 \right) \cdot n_2$$

$$= \Theta\left( \frac{1}{d} \right) \cdot \Theta\left( \frac{\eta n\sqrt{\log(6n^2/\delta)}}{\sqrt{d}} \right) \cdot n_2$$

$$= \widetilde{O}\left( \frac{\eta^2 n^2}{d^{3/2}} \right).$$

The second inequality is from Lemma A.15 and Lemma A.20; The third inequality is from the fact that $\eta < 1$.

As for the last two terms, we bound them respectively, for $I_2$ we have

$$|I_2| \leq \sum_{i \in [n]} \sum_{j \in [n] \setminus \{i\}} |y_i y_j \widehat{\theta}_i \widehat{\theta}_j \langle \boldsymbol{\xi}_i, \boldsymbol{\xi}_j \rangle| \leq n^2 \max_{i \in [n]} \widehat{\theta}_i^2 \cdot 2\sqrt{d \log(6n^2/\delta)}$$

$$\leq n^2 \frac{\alpha^2}{((1-\kappa)d - 2n\sqrt{d \log(6n^2/\delta)})^2} \cdot 2\sqrt{d \log(6n^2/\delta)}$$

$$= \widetilde{O}\left( \frac{n^2}{d^{3/2}} \right).$$

The first inequality is from triangle inequality; The second inequality is from Lemma A.65; The third inequality is from Lemma A.18. Last for $I_3$, we have

$$|I_3| \leq \sum_{i \in \mathcal{N}} \sum_{j \in \mathcal{N} \setminus \{i\}} |y_i y_j \theta_i \theta_j \langle \boldsymbol{\xi}_i, \boldsymbol{\xi}_j \rangle| \leq (n_2)^2 \max_{i \in \mathcal{N}} \theta_i^2 \cdot 2\sqrt{d \log(6n^2/\delta)}$$

$$\leq (n_2)^2 \frac{1}{((1-\kappa)d - 2\eta n\sqrt{d \log(6n^2/\delta)})^2} \cdot 2\sqrt{d \log(6n^2/\delta)}$$

$$= \widetilde{O}\left( \frac{\eta^2 n^2}{d^{3/2}} \right).$$

The first inequality is from triangle inequality; The second inequality is from Lemma A.65; The third inequality is from Lemma A.15. Combining the results above, we have

$$\|\boldsymbol{v}'\|^2 - \|\boldsymbol{v}_{mm}\|^2 \geq \frac{C_2 n_1 (1-\beta q)^2}{(1-\beta)^2 (1+\kappa)d} + \widetilde{O}\left( \frac{n^2}{d^{3/2}} \right) \geq \frac{C_1 (1-\beta q)^2}{(1-\beta)^2 (1+\kappa)d}.$$

Therefore, we could then use the same method as above to prove that Proposition A.14 also holds in this case.

***Case 1.3*** $p < n_1$, $I_v \cap \mathcal{C}_i \neq \varnothing$, $I_v \cap \mathcal{C}_{i'} = \varnothing$

For the case when only one of the clusters in clean sets are all mixed, we can follow similar method in *Case 1.2* to prove that Lemma A.19 still holds. Without losing generality, assume all clean samples with label $y_i = +1$ violate optimal token selection while only part of clean samples with label $y_i = -1$ violate. we have

*Condition* 6 (One cluster and a clean sample in the opposite cluster violating optimal token selection).

$$\begin{cases} -\boldsymbol{v}^\top \boldsymbol{\mu}_2 \geq 1 \\ y_i \boldsymbol{v}^\top \boldsymbol{\xi}_i \geq 1, i \in \mathcal{N} \\ y_i \boldsymbol{v}^\top \boldsymbol{r}_i \geq 1, i \in \mathcal{C}_{+1} \\ y_i \boldsymbol{v}^\top \boldsymbol{r}_i \geq 1, i \in \mathcal{C}_{-1} \cap I_v \end{cases}$$

Similar to previous analysis, mixing multiple samples with label $-1$ will not result in a better solution than only mixing one sample with label $-1$. Thus we can reduce this case to mixing only one clean sample and denote this mixed sample as $k_{-1}$. Therefore, we have

$$
\begin{cases}
-\lambda'_2 \cdot \|\boldsymbol{\mu}_2\|^2 \geq 1 \\
\theta'_i \cdot \|\boldsymbol{\xi}_{i'}\|^2 + \sum\limits_{i' \neq i} y_i y_{i'} \theta'_{i'} \langle \boldsymbol{\xi}_i, \boldsymbol{\xi}_{i'} \rangle \geq 1, i \in \mathcal{N} \\
y_{k_{-1}} \beta \lambda'_2 \cdot \|\boldsymbol{\mu}_2\|^2 + (1-\beta)(\theta'_{k_{-1}} \cdot \|\boldsymbol{\xi}_{k_{-1}}\|^2 + \sum\limits_{i \neq k_{-1}} y_{k_{-1}} y_i \theta'_i \langle \boldsymbol{\xi}_i, \boldsymbol{\xi}_{k_{-1}} \rangle) \geq 1 \\
\beta \lambda'_1 \cdot \|\boldsymbol{\mu}_1\|^2 + (1-\beta)(\theta'_{k_i} \cdot \|\boldsymbol{\xi}_{k_i}\|^2 + \sum\limits_{i \neq k_i} y_{k_i} y_i \theta'_i \langle \boldsymbol{\xi}_i, \boldsymbol{\xi}_{k_i} \rangle) \geq 1, i \in \mathcal{C}_{+1}
\end{cases}
$$

Denote $q = \lambda'_1 \cdot \|\boldsymbol{\mu}_1\|^2$ and $q \leq 1$. Denote $k^\star = \operatorname*{argmin}\limits_{i \in \mathcal{C}_{+1}} \frac{1 - \beta_i q}{1 - \beta_i}$ and $\beta = \beta_{k^\star}$, we can further reduce the condition to

*Condition* 7 (Relaxed version of Condition 6).

$$
\begin{cases}
\theta'_i \cdot \|\boldsymbol{\xi}_{i'}\|^2 + \sum\limits_{i' \neq i} y_i y_{i'} \theta'_{i'} \langle \boldsymbol{\xi}_i, \boldsymbol{\xi}_{i'} \rangle \geq 1, i \in \mathcal{N} \\
\theta'_i \cdot \|\boldsymbol{\xi}_{i'}\|^2 + \sum\limits_{i' \neq i} y_i y_{i'} \theta'_{i'} \langle \boldsymbol{\xi}_i, \boldsymbol{\xi}_{i'} \rangle \geq \frac{1 - \beta q}{1 - \beta}, i \in \mathcal{C}_{+1}
\end{cases}
$$

Condition 7 relax the constraints in Condition 6. Meanwhile, it differs from Condition 4 only in that the last inequality holds for clean samples with label $+1$. Therefore, we can follow the proof above to show that Lemma A.19 still holds in this case.

$\square$

Then we consider the second scenario.

**Case 2:** $p = 0, k - p \neq 0$

Similar to the previous part, there are two cases we need to consider under this scenario:

1. $k - p < n_2$.

2. $k - p = n_2$.

We will go over every case sequentially.

***Case 2.1*** $k - p < n_2$

In this case, part of noisy samples are mixed. Denote the mixed samples as $k_1, k_2, ..., k_{k-p}$. And for every mixed sample $k_i$, we have $\boldsymbol{r}_i = \beta_i \boldsymbol{\xi}_{k_i} + (1 - \beta_i) \boldsymbol{\mu}_{k_i}$. Then the conditions under *Case 2.1* become:

*Condition* 8 ($k - p$ noisy samples violating optimal token selection rule).

$$
\begin{cases}
\boldsymbol{v}^\top \boldsymbol{\mu}_1 \geq 1 \\
-\boldsymbol{v}^\top \boldsymbol{\mu}_2 \geq 1 \\
y_i \boldsymbol{v}^\top \boldsymbol{\xi}_i \geq 1, i \in \mathcal{N}, i \notin [k - p] \\
y_{k_i} \boldsymbol{v}^\top \boldsymbol{r}_{k_i} \geq 1, i \in [k - p]
\end{cases}
$$

We could also write the last inequality as

$$
y_{k_i} \beta_i \boldsymbol{v}^\top \boldsymbol{\xi}_{k_i} + y_{k_i} (1 - \beta_i) \boldsymbol{v}^\top \boldsymbol{\mu}_{k_i} \geq 1, i \in [k - p].
$$

Therefore,

$$
y_{k_i} \boldsymbol{v}^\top \boldsymbol{\xi}_{k_i} \geq (1 - y_{k_i}(1 - \beta_i) \boldsymbol{v}^\top \boldsymbol{\mu}_{k_i}) / \beta_i, i \in [k - p].
$$

For noisy samples, we have $y_i = -1$ when $\boldsymbol{\mu}_i = \boldsymbol{\mu}_1$ and $y_i = 1$ when $\boldsymbol{\mu}_i = \boldsymbol{\mu}_2$, so $y_{k_i} \boldsymbol{v}^\top \boldsymbol{\mu}_{k_i} \leq 0$ and thus $(1 - y_{k_i}(1 - \beta_i) \boldsymbol{v}^\top \boldsymbol{\mu}_{k_i}) / \beta_i \geq 1$. Compared to the constraint in Condition 1 that $y_{k_i} \boldsymbol{v}^\top \boldsymbol{\mu}_{k_i} \geq 1, i \in \mathcal{N}$, the new condition is strengthened. So mixing 1 more noisy samples is equal to strengthening

1 constraint in the original setting. Therefore, mixing $k - p$ samples will not result in a better solution than only mixing 1 noisy sample. Similarly, we can simplify this case to mixing only 1 noisy sample and denote this sample as $k_*$. We have $\boldsymbol{r}_{k^*} = \beta\boldsymbol{\xi}_{k^*} + (1-\beta)\boldsymbol{\mu}_{k^*}$ and assume that $\boldsymbol{\xi}_{k^*} = \boldsymbol{\mu}_1$.

Denote $\boldsymbol{v}''$ is the optimal solution under this condition, and the parameters in $\boldsymbol{v}''$ are $\lambda_1''$, $\lambda_2''$ and $\theta_i''$. Then the conditions become:

*Condition* 9 (1 noisy sample violating optimal token selection rule).

$$\begin{cases} \boldsymbol{v}^\top \boldsymbol{\mu}_1 \geq 1 \\ -\boldsymbol{v}^\top \boldsymbol{\mu}_2 \geq 1 \\ y_i \boldsymbol{v}^\top \boldsymbol{\xi}_i \geq 1, i \in \mathcal{N}, i \neq k^\star \\ y_{k^\star} \boldsymbol{v}^\top \boldsymbol{r}_{k^\star} \geq 1 \end{cases}$$

Plugging the representation (48) into the condition, we have:

$$\begin{cases} \lambda_1'' \cdot \|\boldsymbol{\mu}_1\|^2 \geq 1 \\ -\lambda_2'' \cdot \|\boldsymbol{\mu}_2\|^2 \geq 1 \\ \theta_i'' \cdot \|\boldsymbol{\xi}_i\|^2 + \sum_{i' \neq i} y_i y_{i'} \theta_{i'}'' \langle \boldsymbol{\xi}_i, \boldsymbol{\xi}_{i'} \rangle \geq 1, i \in \mathcal{N}, i \neq k^\star \\ -(1-\beta)\lambda_1'' \cdot \|\boldsymbol{\mu}_1\|^2 + \beta(\theta_{k^\star}'' \cdot \|\boldsymbol{\xi}_{k^\star}\|^2 + \sum_{i \neq k^\star} y_{k^\star} y_i \theta_i'' \langle \boldsymbol{\xi}_i, \boldsymbol{\xi}_{k^\star} \rangle) \geq 1 \end{cases}$$

We first introduce the following lemma which estimates the parameters of the noises. We define

$$\alpha = \frac{1 + (1-\beta)\lambda_1''\|\boldsymbol{\mu}_1\|^2}{\beta}$$

for the convenience of the following proof.

**Lemma A.21.** *Suppose that Assumption 4.1 holds, under Condition 9, we have*

$$\theta_{k^\star}'' \leq \frac{\alpha}{(1-\kappa)d - 2n_2\sqrt{d\log(6n^2/\delta)}}$$

$$\theta_{k^\star}'' \geq \frac{\alpha}{(1+\kappa)d}\left(1 - \frac{2n_2\sqrt{d\log(6n^2/\delta)}}{(1-\kappa)d - 2n_2\sqrt{d\log(6n^2/\delta)}}\right)$$

$$\max_{i \in \mathcal{N}, i \neq k^\star} \theta_i'' \leq \frac{(1-\kappa)d + 2(\alpha - n_2)\sqrt{d\log(6n^2/\delta)}}{((1-\kappa)d - 2n_2\sqrt{d\log(6n^2/\delta)})^2}$$

$$\min_{i \in \mathcal{N}, i \neq k^\star} \theta_i'' \geq \frac{1}{(1+\kappa)d} \cdot \left(1 - \frac{2\alpha n_2\sqrt{d\log(6n^2/\delta)}}{(1-\kappa)d - 2n_2\sqrt{d\log(6n^2/\delta)}}\right).$$

*Proof of Lemma A.20.* From the last inequality in Condition 9 we have

$$\theta_{k_*}''\|\boldsymbol{\xi}_{k_*}\|^2 + \sum_{i \in \mathcal{N}, i \neq k_*} y_i y_{k_*} \theta_i'' \langle \boldsymbol{\xi}_i, \boldsymbol{\xi}_{k_*} \rangle \geq \alpha > 1.$$

The last inequality is because $\lambda_1''\|\boldsymbol{\mu}_1\|^2 \geq 1$ and $0 < \beta < 1$. Denote $j = \operatorname*{argmax}_{i \in [n]} \theta_i''$, we have

$$\begin{aligned} y_j \boldsymbol{v}''^\top \boldsymbol{\xi}_j &= \theta_j''\|\boldsymbol{\xi}_j\|^2 + \sum_{i \in \mathcal{N}, i \neq j} y_i y_j \theta_i'' \langle \boldsymbol{\xi}_i, \boldsymbol{\xi}_j \rangle \\ &\geq \theta_j''(1-\kappa)d - n_2 \max_{i \in [n]} \theta_i'' \cdot 2\sqrt{d\log(6n^2/\delta)} \\ &= \theta_j''((1-\kappa)d - n_2 \cdot 2\sqrt{d\log(6n^2/\delta)}) \end{aligned}$$

The first inequality is due to Lemma A.65 and the last equation is from our definition of j. Consider the contrary case when $\theta_j'' > \frac{\alpha}{(1-\kappa)d - 2n_2\sqrt{d\log(6n^2/\delta)}}$, we have

$$y_j \boldsymbol{v}''^\top \boldsymbol{\xi}_j > \alpha.$$

By the complementary slackness condition, if $y_j \boldsymbol{v}''^\top \boldsymbol{\xi}_j > \frac{1+\lambda_1''(1-\beta)\|\boldsymbol{\mu}_1\|^2}{\beta}$ then we must have $\theta_j'' = 0$, and thus we reach a contradiction. Therefore, we have $\theta_{k^\star}'' \le \theta_j'' \le \frac{\alpha}{(1-\kappa)d-2n_2\sqrt{d\log(6n^2/\delta)}}$.

Then denote $j' = \underset{i\in[n], i\neq k^\star}{\operatorname{argmax}} \theta_i''$, we have

$$
\begin{aligned}
y_{j'} \boldsymbol{v}''^\top \boldsymbol{\xi}_{j'} &= \theta_{j'}'' \|\boldsymbol{\xi}_{j'}\|^2 + \sum_{i\in\mathcal{N}, i\neq j'} y_i y_{j'} \theta_i'' \langle \boldsymbol{\xi}_i, \boldsymbol{\xi}_{j'}\rangle \\
&\ge \theta_{j'}''(1-\kappa)d - n_2 \max_{i\in[n], i\neq j'} \theta_i'' \cdot 2\sqrt{d\log(6n^2/\delta)} - \theta_{k^\star}'' \sqrt{d\log(6n^2/\delta)} \\
&\ge \theta_j''\big((1-\kappa)d - n_2 \cdot 2\sqrt{d\log(6n^2/\delta)}\big) - \frac{2\alpha\sqrt{d\log(6n^2/\delta)}}{(1-\kappa)d - 2n_2\sqrt{d\log(6n^2/\delta)}}.
\end{aligned}
$$

The first inequality is from Lemma A.65 and the second inequality is from the upper bound of $\theta_{k^\star}''$ we just get. Consider the case when $\theta_{j'}'' > \frac{(1-\kappa)d+2(\alpha-n_2)\sqrt{d\log(6n^2/\delta)}}{((1-\kappa)d-2n_2\sqrt{d\log(6n^2/\delta)})^2}$, we have

$$
y_{j'} \boldsymbol{v}''^\top \boldsymbol{\xi}_{j'} > 1.
$$

By the complementary slackness condition, if $y_{j'} \boldsymbol{v}''^\top \boldsymbol{\xi}_{j'} > 1$ then we must have $\theta_{j'}'' = 0$, and thus we reach a contradiction.

Then we estimate the lower bound of $\theta_j''$ when $j \neq k_*$. We have

$$
\begin{aligned}
1 \le y_j \boldsymbol{v}''^\top \boldsymbol{\xi}_j &= \theta_j'' \|\boldsymbol{\xi}_j\|^2 + \sum_{i\in[n], i\neq j} y_i y_j \theta_i'' \langle \boldsymbol{\xi}_i, \boldsymbol{\xi}_j\rangle \le \theta_j''(1+\kappa)d + n_2 \max_{i\in[n]} \theta_i'' \cdot 2\sqrt{d\log(6n^2/\delta)} \\
&\le \theta_j''(1+\kappa)d + \frac{1+\lambda_1''(1-\beta)\|\boldsymbol{\mu}_1\|^2}{\beta((1-\kappa)d - 2n_2\sqrt{d\log(6n^2/\delta)}} \cdot 2n_2\sqrt{d\log(6n^2/\delta)},
\end{aligned}
$$

where the last inequality is from the upper bound we just get. Therefore, we have

$$
\theta_j'' \ge \frac{1}{(1+\kappa)d} \cdot \left(1 - \frac{2n_2\sqrt{d\log(6n^2/\delta)}}{(1-\kappa)d - 2n_2\sqrt{d\log(6n^2/\delta)}} \cdot \frac{1+\lambda_1''(1-\beta)\|\boldsymbol{\mu}_1\|^2}{\beta}\right)
$$

for all $j \in \mathcal{N}$ and $j \neq k_*$.

Lastly we lower bound $\theta_{k_*}''$. We have

$$
\frac{1+(1-\beta)\lambda_1''\|\boldsymbol{\mu}_1\|^2}{\beta} \le y_{k_*} \boldsymbol{v}''^\top \boldsymbol{\xi}_{k_*} = \theta_{k_*}''(1+\kappa)d + n_2 \max_{i\in[n]} \theta_i'' \cdot 2\sqrt{d\log(6n^2/\delta)}.
$$

Similarly, we have

$$
\theta_{k_*}'' \ge \frac{1}{(1+\kappa)d} \cdot \frac{1+(1-\beta)\lambda_1''\|\boldsymbol{\mu}_1\|^2}{\beta} \left(1 - \frac{2n_2\sqrt{d\log(6n^2/\delta)}}{(1-\kappa)d - 2n_2\sqrt{d\log(6n^2/\delta)}}\right).
$$

$\square$

After getting the bound of parameters, we could derive the norm difference as above

**Lemma A.22.** *Suppose that Assumption 4.1 holds, denote $\boldsymbol{v}$ and $\boldsymbol{v}''$ as the optimal solutions under condition 1 and condition 9 respectively. We have*

$$
\|\boldsymbol{v}''\|_2^2 - \|\boldsymbol{v}_{mm}\|_2^2 \ge \frac{C_3(1-\beta)}{d},
$$

*where $C_3 = \Theta(1)$.*

*Proof of Lemma A.22.* From the third inequality in Condition 9, for $i \in \mathcal{N}, i \neq k^\star$ we have

$$
\theta_i'' \cdot \|\boldsymbol{\xi}_i\|^2 + \sum_{i'\neq i, k^\star} y_i y_{i'} \theta_{i'}'' \langle \boldsymbol{\xi}_i, \boldsymbol{\xi}_{i'}\rangle \ge 1 - y_i y_{k^\star} \theta_{k^\star}'' \langle \boldsymbol{\xi}_i, \boldsymbol{\xi}_{k^\star}\rangle.
$$

Then we add $y_i y_{k^\star} w \langle \boldsymbol{\xi}_i, \boldsymbol{\xi}_{k^\star} \rangle$ on both sides, where we set $w = \theta''_{k^\star} - \frac{\alpha-1}{(1+\kappa)d-2\sqrt{d\log(6n^2/\delta)}} \le \theta''_{k^\star}$. Then we have

$$\theta''_i \cdot \|\boldsymbol{\xi}_{i'}\|^2 + \sum_{i' \neq i, k^\star} y_i y_{i'} \theta''_{i'} \langle \boldsymbol{\xi}_i, \boldsymbol{\xi}_{i'} \rangle + y_i y_{k^\star} w \langle \boldsymbol{\xi}_i, \boldsymbol{\xi}_{k^\star} \rangle \ge 1 - y_i y_{k^\star} (\theta''_{k^\star} - w) \langle \boldsymbol{\xi}_i, \boldsymbol{\xi}_{k^\star} \rangle$$

$$\ge 1 - 2(\theta''_{k^\star} - w)\sqrt{d\log(6n^2/\delta)}$$

$$= \frac{(1+\kappa)d - 2\alpha\sqrt{d\log(6n^2/\delta)}}{(1+\kappa)d - 2\sqrt{d\log(6n^2/\delta)}}. \quad (54)$$

The second inequality is from Lemma A.65. Now consider a new $\overline{v} = \overline{\lambda}_1 \boldsymbol{\mu}_1 + \overline{\lambda}_2 \boldsymbol{\mu}_2 + \sum_{i \in [n]} y_i \overline{\theta}_i \boldsymbol{\xi}_i$ with

$$\overline{\lambda}_1 = \lambda''_1; \quad \overline{\lambda}_2 = \lambda''_2;$$

$$\overline{\theta}_i = \theta''_i / (1 - 2(\theta''_{k^\star} - w)\sqrt{d\log(6n^2/\delta)}) \text{ for } i \in [n], i \neq k^\star$$

and

$$\overline{\theta}_{k^\star} = \frac{w}{1 - 2(\theta''_{k^\star} - w)\sqrt{d\log(6n^2/\delta)}}.$$

We can prove that $\overline{v}$ satisfies all constraints for $\boldsymbol{v}_{mm}$.

From the first two inequalities in Condition 9, we have $\overline{\lambda}_1 \|\boldsymbol{\mu}_1\|^2 = \lambda''_1 \|\boldsymbol{\mu}_1\|^2 \ge 1$, $-\overline{\lambda}_2 \|\boldsymbol{\mu}_2\|^2 = -\lambda''_2 \|\boldsymbol{\mu}_2\|^2 \ge 1$. Then by dividing $1 - 2(\theta''_{k^\star} - w)\sqrt{d\log(6n^2/\delta)}$ on both sides of (54), for $\forall i \in \mathcal{N}, i \neq k^\star$ we have

$$\overline{\theta}_i \cdot \|\boldsymbol{\xi}_i\|^2 + \sum_{i' \neq i} y_i y_{i'} \overline{\theta}_i \langle \boldsymbol{\xi}_i, \boldsymbol{\xi}_{i'} \rangle \ge 1.$$

Lastly we prove that $\overline{\theta}_{k^\star} \|\boldsymbol{\xi}_{k^\star}\|^2 + \sum_{i \neq k^\star} y_i y_{k^\star} \overline{\theta}_i \langle \boldsymbol{\xi}_i, \boldsymbol{\xi}_{k^\star} \rangle \ge 1$. From the last inequality in Condition 9 we have

$$\theta''_{k^\star} \cdot \|\boldsymbol{\xi}_{k^\star}\|^2 + \sum_{i \neq k^\star} y_{k^\star} y_i \theta''_i \langle \boldsymbol{\xi}_i, \boldsymbol{\xi}_{k^\star} \rangle \ge \alpha.$$

Dividing $1 - 2(\theta''_{k^\star} - w)\sqrt{d\log(6n^2/\delta)}$ on both sides, we get

$$\frac{\theta''_{k^\star} \|\boldsymbol{\xi}_{k^\star}\|^2}{1 - 2(\theta''_{k^\star} - w)\sqrt{d\log(6n^2/\delta)}} + \sum_{i \neq k^\star} y_i y_{k^\star} \overline{\theta}_i \langle \boldsymbol{\xi}_i, \boldsymbol{\xi}_{k^\star} \rangle \ge \frac{\alpha}{1 - 2(\theta''_{k^\star} - w)\sqrt{d\log(6n^2/\delta)}}.$$

Therefore we have

$$\overline{\theta}_{k^\star} \|\boldsymbol{\xi}_{k^\star}\|^2 + \sum_{i \neq k^\star} y_i y_{k^\star} \overline{\theta}_i \langle \boldsymbol{\xi}_i, \boldsymbol{\xi}_{k^\star} \rangle \ge \frac{\alpha - (\theta''_{k^\star} - w)\|\boldsymbol{\xi}_{k^\star}\|^2}{1 - 2(\theta''_{k^\star} - w)\sqrt{d\log(6n^2/\delta)}} \ge \frac{\alpha - (\theta''_{k^\star} - w)(1+\kappa)d}{1 - 2(\theta''_{k^\star} - w)\sqrt{d\log(6n^2/\delta)}} = 1.$$

The second inequality is from Lemma A.65 and the last equality is by our definition $\theta''_{k^\star} - w = \frac{\alpha-1}{(1+\kappa)d-2\sqrt{d\log(6n^2/\delta)}}$. Thus, $\overline{v}$ is a possible solution under Condition 1 and $\|\overline{v}\| \ge \|\boldsymbol{v}_{mm}\|$.

Next we estimate the difference between $\|\boldsymbol{v}''\|^2$ and $\|\overline{v}\|^2$. The expansion of $\|\boldsymbol{v}''\|^2$ and $\|\overline{v}\|^2$ are:

$$\|\boldsymbol{v}''\|^2 = \lambda''^2_1 \|\boldsymbol{\mu}_1\|^2 + \lambda''^2_2 \|\boldsymbol{\mu}_2\|^2 + \sum_{i \in \mathcal{N}} \theta''^2_i \|\boldsymbol{\xi}_i\|^2 + \sum_{i \in \mathcal{N}} \sum_{j \in \mathcal{N}} y_i y_j \theta''_i \theta''_j \langle \boldsymbol{\xi}_i, \boldsymbol{\xi}_j \rangle,$$

$$\|\overline{v}\|^2 = \overline{\lambda}^2_1 \|\boldsymbol{\mu}_1\|^2 + \overline{\lambda}^2_2 \|\boldsymbol{\mu}_2\|^2 + \sum_{i \in \mathcal{N}} \overline{\theta}^2_i \|\boldsymbol{\xi}_i\|^2 + \sum_{i \in \mathcal{N}} \sum_{j \in \mathcal{N}} y_i y_j \overline{\theta}_i \overline{\theta}_j \langle \boldsymbol{\xi}_i, \boldsymbol{\xi}_j \rangle.$$

According to the condition (49), we have $\|\boldsymbol{v}''\| \le 2\|\boldsymbol{v}_{mm}\| = \Theta(\sqrt{1/\rho^2 + \eta n/d})$, which implies that $\alpha = O(\sqrt{n}\log n)$. Otherwise, we have

$$\theta''_{k^\star} \|\boldsymbol{\xi}_{k^\star}\|^2 \ge \alpha - \sum_{i \neq k^\star} y_{k^\star} y_i \theta''_i \langle \boldsymbol{\xi}_i, \boldsymbol{\xi}_{k^\star} \rangle = \Omega(\alpha).$$

It further yields that

$$\|\boldsymbol{v}''\|^2 = \Omega(\frac{1}{\rho^2}) + \Omega(\frac{\eta n}{d}) + \theta_{k^\star}''^2 \|\boldsymbol{\xi}_{k^\star}\|^2 = \Omega(\frac{1}{\rho^2} + \frac{\eta n}{d} + \frac{\alpha 2}{d}) = \Omega(\frac{n \log^2 n}{d}),$$

which contradicts with $\|\boldsymbol{v}''\| = \Theta(\sqrt{1/\rho^2 + \eta n/d})$. We decompose the difference between $\|\boldsymbol{v}''\|^2$ and $\|\overline{\boldsymbol{v}}\|^2$ into four terms:

$$\|\boldsymbol{v}''\|^2 - \|\overline{\boldsymbol{v}}\|^2 = \underbrace{(\theta_{k^\star}''^2 - \overline{\theta}_{k^\star}^2)\|\boldsymbol{\xi}_{k^\star}\|^2}_{I_1} + \underbrace{\sum_{i \in \mathcal{N}, i \neq k^\star} (\theta_i''^2 - \overline{\theta}_i^2)\|\boldsymbol{\xi}_i\|^2}_{I_2} - \underbrace{\sum_{i \in \mathcal{N}} \sum_{j \in \mathcal{N}} y_i y_j \overline{\theta}_i \overline{\theta}_j \langle \boldsymbol{\xi}_i, \boldsymbol{\xi}_j \rangle}_{I_3}$$

$$+ \underbrace{\sum_{i \in \mathcal{N}} \sum_{j \in \mathcal{N}} y_i y_j \theta_i'' \theta_j'' \langle \boldsymbol{\xi}_i, \boldsymbol{\xi}_j \rangle}_{I_4}.$$

We now estimate $I_1$ to $I_4$ sequentially. For the first term,

$$I_1 \geq (\theta_{k^\star}''^2 - \overline{\theta}_{k^\star}^2)(1 - \kappa)d = (\theta_{k^\star}'' - \overline{\theta}_{k^\star})(\theta_{k^\star}'' + \overline{\theta}_{k^\star})(1 - \kappa)d$$

$$= \frac{(\alpha - 1)(1 - 2\theta_{k^\star}'' \sqrt{d \log(6n^2/\delta)})}{(1 + \kappa)d - 2\sqrt{d \log(6n^2/\delta)}} \cdot \Omega\left(\frac{1}{d}\right) \cdot (1 - \kappa)d$$

$$= \Omega\left(\frac{\alpha - 1}{d}\right),$$

where the first inequality is from Lemma A.65; the second equality is from Lemma A.20; and the last equality uses the fact that $\alpha = O(\sqrt{n} \log n)$. Then we can further upper bound $\max_{i \in \mathcal{N}, i \neq k^\star} \theta_i''$ as

$$\max_{i \in \mathcal{N}, i \neq k^\star} \theta_i'' \leq \frac{(1 - \kappa)d + 2(\alpha - n_2)\sqrt{d \log(6n^2/\delta)}}{((1 - \kappa)d - 2n_2 \sqrt{d \log(6n^2/\delta)})^2} = O(\frac{1}{d}). \tag{55}$$

For the second term $I_2$, we have

$$|I_2| \leq \sum_{i \in \mathcal{N}, i \neq k^\star} (\overline{\theta}_i^2 - \theta_i''^2)(1 + \kappa)d$$

$$\leq \left(\frac{1}{(1 - (\theta_{k^\star}'' - w)\sqrt{d \log(6n^2/\delta)})^2} - 1\right) \max_{i \in \mathcal{N}, i \neq k^\star} \theta_i''^2 \cdot \eta n(1 + \kappa)d$$

$$= \frac{(\alpha - 1)\sqrt{d \log(6n^2/\delta)}}{(1 + \kappa)d - \sqrt{d \log(6n^2/\delta)}} \cdot O(\frac{\eta n}{d}) = \widetilde{O}\left(\frac{(\alpha - 1)\eta n}{d^{3/2}}\right).$$

The second inequality is from Lemma A.20. The first equality is from (55) and the last equality is from Assumption 4.1.

Then we bound $|-I_3 + I_4|$ as:

$$|-I_3 + I_4| \leq \sum_{i \in \mathcal{N}} \sum_{j \in \mathcal{N} \setminus \{i\}} |\overline{\theta}_i \overline{\theta}_j - \theta_i'' \theta_j''| \cdot |\langle \boldsymbol{\xi}_i, \boldsymbol{\xi}_j \rangle|$$

$$\leq \sum_{i \in \mathcal{N} \setminus \{k^\star\}} \sum_{j \in \mathcal{N} \setminus \{k^\star, i\}} |\overline{\theta}_i \overline{\theta}_j - \theta_i'' \theta_j''| \cdot |\langle \boldsymbol{\xi}_i, \boldsymbol{\xi}_j \rangle| + 2 \sum_{t \in \mathcal{N} \setminus \{k^\star\}} |\overline{\theta}_{k^\star} \overline{\theta}_t - \theta_{k^\star}'' \theta_t''| \cdot |\langle \boldsymbol{\xi}_{k^\star}, \boldsymbol{\xi}_t \rangle|$$

$$\leq (\eta n)^2 \left(\frac{1}{(1 - (\theta_{k^\star}'' - w)\sqrt{d \log(6n^2/\delta)})^2} - 1\right) \max_{i \in \mathcal{N}, i \neq k^\star} \theta_i''^2 \cdot 2\sqrt{d \log(6n^2/\delta)}$$

$$+ \eta n \left(\theta_{k^\star}'' - \frac{\overline{\theta}_{k^\star}}{1 - 2(\theta_{k^\star}'' - w)\sqrt{d \log(6n^2/\delta)}}\right) \max_{i \in \mathcal{N}, i \neq k^\star} \theta_i'' 4\sqrt{d \log(6n^2/\delta)}$$

$$\leq \frac{(\alpha - 1)\sqrt{d \log(6n^2/\delta)}}{(1 + \kappa)d - \sqrt{d \log(6n^2/\delta)}} \cdot O(\frac{(\eta n)^2(1 + \kappa)}{d^{3/2}}) + \frac{\alpha - 1}{d} \cdot O(\eta n \frac{c_1}{d}) \cdot 2\sqrt{d \log(6n^2/\delta)}$$

$$= O\left(\frac{(\alpha - 1)\eta^2 n^2}{d^2} + \frac{(\alpha - 1)\eta n}{d^{3/2}}\right).$$

The third inequality is from Lemma A.15 and Lemma A.20; The fourth inequality is from the fact that

$$\theta_{k^\star}'' - \frac{\overline{\theta}_{k^\star}}{1 - 2(\theta_{k^\star}'' - w)\sqrt{d\log(6n^2/\delta)}} = \frac{\theta_{k^\star}'' - \overline{\theta}_{k^\star} - 2\theta_{k^\star}''(\theta_{k^\star}'' - w)\sqrt{d\log(6n^2/\delta)}}{1 - 2(\theta_{k^\star}'' - w)\sqrt{d\log(6n^2/\delta)}}$$

$$= \frac{\Omega(\frac{\alpha-1}{d}) - O(\frac{\alpha(\alpha-1)}{d^{3/2}})}{1 - 2(\theta_{k^\star}'' - w)\sqrt{d\log(6n^2/\delta)}} > 0$$

So we have $\theta_{k^\star}'' - \frac{\overline{\theta}_{k^\star}}{1 - 2(\theta_{k^\star}'' - w)\sqrt{d\log(6n^2/\delta)}} \leq \theta_{k^\star}'' - \overline{\theta}_{k^\star}$; The last equality is from Assumption 4.1.

Combining the above results, we have

$$\|\boldsymbol{v}''\|_2^2 - \|\boldsymbol{v}_{mm}\|_2^2 \geq \Theta\left(\frac{\alpha-1}{d}\right) + O\left(\frac{(\alpha-1)\eta n}{d^{3/2}}\right) \geq \frac{C_3(1-\beta)}{d}.$$

Here $C_3 = \Theta(1)$ is a constant. $\qquad\square$

Now we can prove the main proposition in this case.

*Proof of Proposition A.14 under Case 2.1.* From Lemma A.22 we have

$$\|\boldsymbol{v}''\|_2^2 - \|\boldsymbol{v}_{mm}\|_2^2 \geq \frac{C_3(1-\beta)}{d} = T'(1-\beta).$$

Here we substitute $T' = \frac{C_3}{d} \geq 0$. Then we have

$$\Gamma^2 - \Gamma''^2 = \frac{1}{\|\boldsymbol{v}_{mm}\|^2} - \frac{1}{\|\boldsymbol{v}''\|^2} = \frac{\|\boldsymbol{v}''\|^2 - \|\boldsymbol{v}_{mm}\|^2}{\|\boldsymbol{v}''\|^2 \cdot \|\boldsymbol{v}_{mm}\|^2} \geq \frac{T'(1-\beta)}{\|\boldsymbol{v}''\|^2 \cdot \|\boldsymbol{v}_{mm}\|^2}.$$

Therefore,

$$\Gamma - \Gamma'' \geq \frac{T'(1-\beta)}{(\Gamma + \Gamma'')\|\boldsymbol{v}_{mm}\|^2 \cdot \|\boldsymbol{v}'\|^2} \geq \frac{T'(1-\beta)}{2\Gamma\|\boldsymbol{v}_{mm}\|^2 \cdot \|\boldsymbol{v}''\|^2} = \frac{T'(1-\beta)}{2\|\boldsymbol{v}_{mm}\|\|\boldsymbol{v}''\|^2} \geq \frac{T'(1-\beta)}{2\|\boldsymbol{v}''\|^3}.$$

The last inequality is from $\|\boldsymbol{v}''\| \geq \|\boldsymbol{v}_{mm}\|$. This implies

$$\Gamma'' \leq \Gamma - \frac{T'(1-\beta)}{2\|\boldsymbol{v}''\|^3} \leq \Gamma - \frac{C_1}{\|\boldsymbol{v}_{mm}\|^3 n\rho^2}(1-\beta).$$

The last inequality is from our assumption that $\|\boldsymbol{v}''\| \leq 2\|\boldsymbol{v}_{mm}\|$ and $\rho^2 = \Omega(d/n)$. $\qquad\square$

Then we consider the other case.

***Case 2.2*** $k - p = n_2$

In this case, all noisy samples are mixed. From previous analysis, this is equivalent to strengthening all conditions $y_i \boldsymbol{v}^\top \boldsymbol{\xi}_i \geq 1$ while other conditions remain the same. As mixing $k - p$ samples will not result in a better solution than only mixing 1 noisy sample, the proof is the same as *Case 2.1* and we omit it for convenience.

Finally, we consider the last scenario.

**Case 3:** $p \neq 0, k - p \neq 0$

This scenario is more complex as both clean and noisy sets are mixed. There are four cases to consider

1. $p < n_1, k - p < n_2$. (Both clean and noisy sets are partially mixed)

2. $p < n_1, k - p = n_2$ (Clean set is partially mixed, noisy set is all mixed)

3. $p = n_1, k - p < n_2$ (Clean set is all mixed, noisy set is partially mixed)

4. $p = n_1, k - p = n_2$ (Both clean and noisy sets are all mixed)

We will go over every case to prove Proposition A.14 holds.

### Case 3.1 $p < n_1, k - p < n_2$

This case is simple because from the analysis above, mixing 1 more clean sample is equivalent to adding 1 more constraint and mixing 1 more noisy sample is equivalent to strengthening 1 original constraint. So mixing both sets will not result in a better solution than only mixing 1 clean sample. Therefore, the proof is the same as *Case 1.1* and we omit is for convenience.

### Case 3.2 $p < n_1, k - p = n_2$

In this case, all noisy samples and part of clean samples are mixed. We can consider this case as an extension of *Case 2.2* by mixing some clean samples. From previous analysis, mixing 1 more clean sample is equivalent to adding 1 more constraint. So this case will not result in a better solution than *Case 2.2*. The following proof is the same as *Case 2.2* and we omit it for convenience.

### Case 3.3 $p = n_1, k - p < n_2$

In this case, all clean samples and part of noisy samples are mixed. We can consider this case as an extension of *Case 1.2* by mixing some noisy samples. From previous analysis, mixing 1 more noisy sample is equivalent to strengthening 1 original constraint. So this case will not result in a better solution than *Case 1.2*. The following proof is the same as *Case 1.2* and we omit it for convenience.

### Case 3.4 $p = n_1, k - p = n_2$

This case is more complex. We cannot simply consider it as an extension of *Case 2.2* because the analysis of *Case 2.2* is based on the condition that there exist clean samples that follow optimal token selection rule. Denote $r_i = \beta_i \boldsymbol{\mu}_i + (1 - \beta_i)\boldsymbol{\xi}_i$ for $i \in \mathcal{C}$ and $r_i = (1 - \beta_i)\boldsymbol{\mu}_i + \beta_i \boldsymbol{\xi}_i$ for $i \in \mathcal{N}$. The condition in this case becomes

*Condition* 10 (All samples are mixed).

$$y_i \boldsymbol{v}''^{\top} \boldsymbol{r}_i \geq 1.$$

This indicates

$$
\begin{cases}
\beta_i y_i \lambda_i'' \|\boldsymbol{\mu}_i\|^2 + (1 - \beta_i)(\theta_i'' \|\boldsymbol{\xi}_i\|^2 + \sum\limits_{j \neq i} y_i y_j \theta_j'' \langle \boldsymbol{\xi}_i, \boldsymbol{\xi}_j \rangle) \geq 1, i \in \mathcal{C}, \\
(1 - \beta_i) y_i \lambda_i'' \|\boldsymbol{\mu}_i\|^2 + \beta_i(\theta_i'' \|\boldsymbol{\xi}_i\|^2 + \sum\limits_{j \neq i} y_i y_j \theta_j'' \langle \boldsymbol{\xi}_i, \boldsymbol{\xi}_j \rangle) \geq 1, i \in \mathcal{N}.
\end{cases}
$$

Assume that $\min\{\lambda_1'' \cdot \|\boldsymbol{\mu}_1\|^2, -\lambda_2'' \cdot \|\boldsymbol{\mu}_2\|^2\} = q$ in optimal $\boldsymbol{v}''$. If $q \geq 1$, we can directly follow the proof in *Case 2.2*. Otherwise, denote $\alpha = \frac{1 - \beta_i q}{1 - \beta_i}$. We have $\alpha > 1$ due to $q < 1$ and $0 \leq \beta_i < 1$. Without losing generality, we assume $\lambda_1'' \cdot \|\boldsymbol{\mu}_1\|^2 = q < 1$. Then consider the following relaxed condition

*Condition* 11 (Relaxed version of constraints in Condition 10).

$$\theta_i'' \|\boldsymbol{\xi}_i\|^2 + \sum_{j \neq i} y_i y_j \theta_j'' \langle \boldsymbol{\xi}_i, \boldsymbol{\xi}_j \rangle \geq \alpha, i \in \mathcal{C}_1.$$

Denote the optimal solution under Condition 11 as $\breve{\boldsymbol{v}}$ and the corresponding coefficients in $\breve{\boldsymbol{v}}$ as $\breve{\lambda}_1, \breve{\lambda}_2$ and $\breve{\theta}_i$, i.e.

$$\breve{\boldsymbol{v}} = \breve{\lambda}_1 \boldsymbol{\mu}_1 + \breve{\lambda}_2 \boldsymbol{\mu}_2 + \sum_{i \in [n]} \breve{\theta}_i \boldsymbol{\xi}_i.$$

Since the constraints in Condition 11 is a subset of the constraints in Condition 10, we have $\|\breve{\boldsymbol{v}}\| \leq \|\boldsymbol{v}''\|$. Meanwhile, we have the following lemma to estimate $\breve{\theta}_i$:

**Lemma A.23.** *Suppose that Assumption 4.1 holds, under Condition 11, we have*

$$\breve{\theta}_i = 0, i \in [n] \backslash \mathcal{C}_1;$$

$$\breve{\theta}_i \in \left[ \frac{\alpha}{(1 + \kappa)d}\left(1 - \frac{n\sqrt{d\log(6n^2/\delta)}}{(1 - \kappa)d - n\sqrt{d\log(6n^2/\delta)}}\right), \frac{\alpha}{((1 - \kappa)d - 2n_{11}\sqrt{d\log(6n^2/\delta)}} \right], i \in \mathcal{C}_1.$$

*Proof of Lemma A.23.* Note that Condition 11 does not have any constraint for samples with $i \in [n]\backslash\mathcal{C}_1$. Thus we have $\breve{\theta}_i = 0$ for any $i \in [n]\backslash\mathcal{C}_1$ in the representation (44). Denote $j = \underset{i \in \mathcal{C}_1}{\operatorname{argmax}}\,\breve{\theta}_i$, then we have

$$\breve{\theta}_j \cdot \|\boldsymbol{\xi}_j\|^2 + \sum_{k \neq j} y_k y_j \breve{\theta}_k \langle \boldsymbol{\xi}_i, \boldsymbol{\xi}_j \rangle \geq \breve{\theta}_j \|\boldsymbol{\xi}_j\|^2 - 2\breve{\theta}_j n_{11} \sqrt{d\log(6n^2/\delta)} \geq \breve{\theta}_j((1-\kappa)d - 2n_{11}\sqrt{d\log(6n^2/\delta)}).$$

The two inequalities are from Lemma A.65 and our definition of $j$. Consider the contrary case when $\breve{\theta}_j > \frac{\alpha}{((1-\kappa)d - 2n_{11}\sqrt{d\log(6n^2/\delta)})}$, we have

$$y_j \breve{\boldsymbol{v}}^\top \boldsymbol{\xi}_j > \alpha.$$

By the complementary slackness condition, if $y_j \breve{\boldsymbol{v}}^\top \boldsymbol{\xi}_j > \alpha$, then we must have $\breve{\theta}_j = 0$, and thus we reach a contradiction.
Then we lower bound $\breve{\theta}_i$. For $\forall i \in \mathcal{C}_1$ we have

$$\alpha \leq \breve{\theta}_i \cdot \|\boldsymbol{\xi}_i\|^2 + \sum_{j \neq i} y_i y_j \breve{\theta}_i \langle \boldsymbol{\xi}_i, \boldsymbol{\xi}_j \rangle \leq \breve{\theta}_i(1+\kappa)d + 2n_{11} \max_{i \in [n]} \breve{\theta}_i \sqrt{d\log(6n^2/\delta)}$$

$$\leq \breve{\theta}_i(1+\kappa)d + \frac{2\alpha n_{11}\sqrt{d\log(6n^2/\delta)}}{(1-\kappa)d - 2n_{11}\sqrt{d\log(6n^2/\delta)}}.$$

The second inequality is from Lemma A.65 and the last inequality is from the upper bound of $\breve{\theta}_i$ we just derived. Therefore, we have

$$\breve{\theta}_i \geq \frac{\alpha}{(1+\kappa)d}\left(1 - \frac{2n_{11}\sqrt{d\log(6n^2/\delta)}}{(1-\kappa)d - 2n_{11}\sqrt{d\log(6n^2/\delta)}}\right).$$

$\square$

From this Lemma we have $\breve{\theta}_i = \Theta(\alpha/d)$ for $i \in \mathcal{C}_1$. Similar as (53), under our assumption $\|\breve{\boldsymbol{v}}\| \leq 2\|\boldsymbol{v}_{mm}\|$, we have $\alpha = O(\log(n))$. Next we estimate the difference between $\|\breve{\boldsymbol{v}}\|^2$ and $\|\boldsymbol{v}_{mm}\|^2$. We can prove that Lemma A.22 still holds in this case.

*Proof of Lemma A.22.* Under this case, the difference between $\|\breve{\boldsymbol{v}}\|_2^2$ and $\|\boldsymbol{v}_{mm}\|_2^2$ becomes

$$\|\breve{\boldsymbol{v}}\|^2 - \|\boldsymbol{v}_{mm}\|^2 \geq \underbrace{\sum_{i \in [n]}(\breve{\theta}_i^2 - \theta_i^2)\|\boldsymbol{\xi}_i\|^2 - (\lambda_1^2 - \breve{\lambda}_1^2)\|\boldsymbol{\mu}_1\|^2 - (\lambda_2^2 - \breve{\lambda}_2^2)\|\boldsymbol{\mu}_2\|^2}_{I_1}$$

$$\underbrace{-\sum_{i \in \mathcal{N}}\sum_{j \in \mathcal{N}\backslash\{i\}} y_i y_j \theta_i \theta_j \langle \boldsymbol{\xi}_i, \boldsymbol{\xi}_j \rangle}_{I_2} + \underbrace{\sum_{i \in \mathcal{C}_1}\sum_{j \in \mathcal{C}_1\backslash\{i\}} y_i y_j \breve{\theta}_i \breve{\theta}_j \langle \boldsymbol{\xi}_i, \boldsymbol{\xi}_j \rangle}_{I_3}$$

We then bound $I_1 \sim I_3$ respectively. For $I_1$ we have

$$|I_1| \geq \sum_{i \in \mathcal{C}_1} \breve{\theta}_i^2 \|\boldsymbol{\xi}_i\|^2 - \sum_{i \in \mathcal{N}} \theta_i^2\|\boldsymbol{\xi}_i\|^2 - 2/\rho^2 \geq n_{11}\min_{i \in [n]} \breve{\theta}_i^2(1-\kappa)d - n_2\max_{i \in \mathcal{N}} \theta_i^2(1+\kappa)d - 2/\rho^2$$

$$\geq \frac{\alpha^2 n_{11}(1-\kappa)}{(1+\kappa)^2 d}\left(1 - \frac{2\sqrt{d\log(6n^2/\delta)}}{(1-\kappa)d - 2n_{11}\sqrt{d\log(6n^2/\delta)}}\right) - \frac{n_2(1+\kappa)d}{((1-\kappa)d - 2n_2\sqrt{d\log(6n^2/\delta)})^2} - \frac{2}{\rho^2}$$

$$= \Omega\left(\frac{n}{d}\right).$$

The second inequality is from Lemma A.65; The third inequality is from Lemma A.15 and A.23; The last equality is due to the SNR condition $\rho/\sqrt{d} = \Omega(1/\sqrt{n})$ so that $\frac{1}{\rho^2} \leq \frac{n}{4d}$. For $I_2$, we have

$$|I_2| \leq \sum_{i \in \mathcal{N}} \max_{i \in \mathcal{N}} \theta_i^2 \cdot 2\sqrt{d\log(6n^2/\delta)} \leq \frac{2n_2\sqrt{d\log(6n^2/\delta)}}{((1-\kappa)d - 2n_2\sqrt{d\log(6n^2/\delta)})^2} = \widetilde{O}\left(\frac{n}{d^{3/2}}\right).$$

The first inequality is from Lemma A.65; The second inequality is from Lemma A.15. Similarly, for $|I_3|$ we have

$$|I_3| \leq \sum_{i \in \mathcal{C}_1} \max_{i \in \mathcal{C}_1} \breve{\theta}_i^2 \cdot 2\sqrt{d \log(6n^2/\delta)} \leq \frac{2n_{11}\alpha^2 \sqrt{d \log(6n^2/\delta)}}{((1-\kappa)d - 2n_{11}\sqrt{d \log(6n^2/\delta)})^2} = \widetilde{O}\left(\frac{n}{d^{3/2}}\right).$$

The second inequality is from Lemma A.23. Combining the above results, we have

$$\|\boldsymbol{v}''\|_2^2 - \|\boldsymbol{v}\|_2^2 \geq \Theta\left(\frac{n_{11}}{d}\right) - \widetilde{O}\left(\frac{n}{d^{3/2}}\right) \geq \frac{C_3 n(1-\beta)}{d}.$$

The remaining proof is the same as *Case 2.1* and we omit it for convenience. $\qquad\square$

Therefore, we complete the proof for all possible scenarios. $\qquad\square$

**Training and Test Error Analysis**

From Proposition A.14 we can analyze the properties of both parameters to estimate the training and test error. In this section, we first get the convergence direction of parameters $\boldsymbol{p}$ and $\boldsymbol{v}$. The main difference between our setting with Ataee Tarzanagh et al. [6] is that they only consider the infinite case and their results hold only when $R, r \to \infty$. We extend their results to the finite case. Specifically, given fixed upper bound $R$ and $r$ for $\|\boldsymbol{p}\|$ and $\|\boldsymbol{v}\|$ respectively, we denote the solution of the constrained optimization (4) as $(\boldsymbol{v}_{(r,R)}, \boldsymbol{p}_{(r,R)})$ in this section for brevity.

Our main theorem in this section estimates the corresponding deviation of $\boldsymbol{p}_{(r,R)}/R$ and $\boldsymbol{v}_{(r,R)}/r$ from their convergence direction $\boldsymbol{p}_{mm}/\|\boldsymbol{p}_{mm}\|$ and $\boldsymbol{v}_{mm}/\|\boldsymbol{v}_{mm}\|$. For a given $\boldsymbol{p}$, it is elementary that the margin induced by $\boldsymbol{p}$ is $\min_{i,t_i \neq \alpha_i}(\boldsymbol{x}_{i\alpha_i} - \boldsymbol{x}_{it_i})^\top \boldsymbol{p}/\|\boldsymbol{p}\|$, thus when $\|\boldsymbol{p}\| = 1$, the margin becomes $\min_{i,t_i \neq \alpha_i}(\boldsymbol{x}_{i\alpha_i} - \boldsymbol{x}_{it_i})^\top \boldsymbol{p}$. And for a given $\boldsymbol{v}$, the label margin induced by $\boldsymbol{v}$ is $\min_i y_i \boldsymbol{v}^\top \boldsymbol{r}_i/\|\boldsymbol{v}\|$. Recall that the label margin induced by $\boldsymbol{v}_{mm}$ is $\Gamma$ and the margin of *p-SVM* induced by $\boldsymbol{p}_{mm}$ is $\Xi$.

First we introduce a lemma to estimate the norm of $\|\boldsymbol{p}_{mm}\|$. This will benefit our proof of the main theorem.

**Lemma A.24** (Norm of $\boldsymbol{p}_{mm}$). *Suppose that Assumption 4.1 holds, recall that the solution of (p-SVM) is $\boldsymbol{p}_{mm}$. With probability at least $1 - \delta$ on the training dataset we have*

$$\frac{1}{\rho^2} + \frac{\eta n}{d} \leq \|\boldsymbol{p}_{mm}\|^2 \leq \frac{8}{\rho^2} + \frac{17\eta n}{d}.$$

*This implies*

$$\|\boldsymbol{p}_{mm}\| = \Theta\left(\sqrt{\frac{1}{\rho^2} + \frac{\eta n}{d}}\right).$$

*Proof of Lemma A.24.* First we prove the upper bound. Consider the following possible solution $\widetilde{\boldsymbol{p}}$:

$$\widetilde{\boldsymbol{p}} = \frac{2(\boldsymbol{\mu}_1 + \boldsymbol{\mu}_2)}{\rho^2} + \sum_{i \in \mathcal{N}} 4\frac{\boldsymbol{\xi}_i}{d}. \tag{56}$$

We then proved that $\widetilde{\boldsymbol{p}}$ satisfies (41). For $k \in \mathcal{C}$ we have

$$\widetilde{\boldsymbol{p}}^\top(\boldsymbol{\mu}_k - \boldsymbol{\xi}_k) = 2 - \sum_{i \in \mathcal{N}} 4\frac{\langle \boldsymbol{\xi}_i, \boldsymbol{\xi}_k \rangle}{d} \geq 2 - \frac{4n_2 \sqrt{d \log(6n^2/\delta)}}{d} \geq 1.$$

The first inequality is from the definition of $d$ in Lemma A.65 and the second inequality is from Assumption 4.1. And for $k \in \mathcal{N}$, we have

$$\widetilde{\boldsymbol{p}}^\top(\boldsymbol{\xi}_k - \boldsymbol{\mu}_k) = -2 + \sum_{i \in \mathcal{N}} 4\frac{\langle \boldsymbol{\xi}_i, \boldsymbol{\xi}_k \rangle}{d} \geq -2 + 4(1-\kappa) + \sum_{i \in \mathcal{N}, i \neq k} 4\frac{\langle \boldsymbol{\xi}_i, \boldsymbol{\xi}_k \rangle}{d}$$

$$\geq -2 + 4(1-\kappa) + \frac{4n_2 \sqrt{d \log(6n^2/\delta)}}{d} \geq 1.$$

The first and second inequalities are from Lemma A.65; The last inequality is from Assumption 4.1.

Therefore, the max-margin solution $\boldsymbol{p}_{mm}$ must have no greater norm than $\widetilde{\boldsymbol{p}}$. So we can upper bound $\boldsymbol{p}_{mm}$ as

$$\|\boldsymbol{p}_{mm}\|^2 \leq \|\widetilde{\boldsymbol{p}}\|^2 = \frac{8}{\rho^2} + \frac{16}{d^2}\left(\sum_{i\in\mathcal{N}}\|\boldsymbol{\xi}_i\|^2 + \sum_{i,j\in\mathcal{N},i\neq j}\langle\boldsymbol{\xi}_i,\boldsymbol{\xi}_j\rangle\right)$$

$$\leq \frac{8}{\rho^2} + \frac{16}{d^2}\left((1+\kappa)n_2 d + 2n_2^2\sqrt{d\log(6n^2/\delta)}\right) \leq \frac{8}{\rho^2} + \frac{17\eta n}{d}.$$

The second inequality is from Lemma A.65; The last inequality is from the definition of $d$ in Assumption 4.1.

Then we prove for the lower bound. As $\boldsymbol{p}_{mm}$ is the max-margin solution and satisfies KKT condition, it can be expressed as the sum of signal and noise tokens. Then we decompose $\boldsymbol{p}_{mm} = \boldsymbol{p}_{\boldsymbol{\mu}}^{mm} + \boldsymbol{p}_{\boldsymbol{\xi}}^{mm}$ where $\boldsymbol{p}_{\boldsymbol{\mu}}^{mm} = f_1^{mm}\boldsymbol{\mu}_1 + f_2^{mm}\boldsymbol{\mu}_2$ and $\boldsymbol{p}_{\boldsymbol{\xi}}^{mm} = \sum_{i\in[n]}g_i^{mm}\boldsymbol{\xi}_i$. Note that $\boldsymbol{\mu}_j \perp \boldsymbol{\xi}_i$ for all $j \in \{\pm 1\}, i \in [n]$. From Lemma A.29, we have $f_j^{mm} \geq 0.9/\rho^2$, so we can lower bound $\|\boldsymbol{p}_{\boldsymbol{\mu}}^{mm}\|_2^2$ as

$$\|\boldsymbol{p}_{\boldsymbol{\mu}}^{mm}\|_2^2 = f_1^{mm2}\|\boldsymbol{\mu}_1\|^2 + f_2^{mm2}\|\boldsymbol{\mu}_2\|^2 \geq \frac{2\cdot 0.9^2}{\rho^2} \geq \frac{1}{\rho^2}.$$

As for $\|\boldsymbol{p}_{\boldsymbol{\xi}}^{mm}\|_2$, from p-SVM condition, for every noisy sample we have

$$\boldsymbol{p}_{mm}^\top(\boldsymbol{\xi}_i - \boldsymbol{\mu}_i) \geq 1,$$

which indicates

$$\boldsymbol{p}_{\boldsymbol{\xi}}^{mm\top}\boldsymbol{\xi}_i = \boldsymbol{p}_{mm}^\top\boldsymbol{\xi}_i \geq 1 + \boldsymbol{p}_{mm}^\top\boldsymbol{\mu}_i \geq 1.9.$$

The last inequality is from Lemma A.29. Sum up the inequality for all noisy sample, we have

$$\sum_{i\in\mathcal{N}}\boldsymbol{p}_{\boldsymbol{\xi}}^{mm\top}\boldsymbol{\xi}_i \geq 1.9n_2.$$

Thus,

$$\|\boldsymbol{p}_{\boldsymbol{\xi}}^{mm}\| \geq \frac{1.9n_2}{\|\sum_{i\in\mathcal{N}}\boldsymbol{\xi}_i\|} = \frac{1.9n_2}{\sqrt{\sum_{i\in\mathcal{N}}\|\boldsymbol{\xi}_i\|^2 + \sum_{i,j\in\mathcal{N}}\langle\boldsymbol{\xi}_i,\boldsymbol{\xi}_j\rangle}} \geq \frac{1.9n_2}{\sqrt{2\cdot n_2\cdot(1+\kappa)d}} \geq \sqrt{\frac{\eta n}{d}}.$$

The second inequality is from Lemma A.65 and the last inequality is from Assumption 4.1. Therefore,

$$\|\boldsymbol{p}_{mm}\|^2 = \|\boldsymbol{p}_{\boldsymbol{\mu}}^{mm}\|_2^2 + \|\boldsymbol{p}_{\boldsymbol{\xi}}^{mm}\|_2^2 \geq \frac{1}{\rho^2} + \frac{\eta n}{d}.$$

Combining the results above, we have

$$\|\boldsymbol{p}_{mm}\|^2 = \Theta\left(\frac{1}{\rho^2} + \frac{\eta n}{d}\right).$$

$\square$

**Definition A.25.** Let $f : \mathbb{R}^2 \to \mathbb{R}^d$. We say that

$$\lim_{x,y\to\infty} f(x,y) = L$$

iff $\forall\epsilon > 0\ \exists M$ such that $\forall x, y > M$ we have that $\|f(x,y) - L\| < \epsilon$.

*Remark* A.26. Let $g : \mathbb{R} \to \mathbb{R}$ be a function with $\lim_{x\to\infty} g(x) = \infty$. Assume that $\lim_{x,y\to\infty} f(x,y) = L$, then $\lim_{x\to\infty} f(x, g(x)) = L$ and $\lim_{x\to\infty} f(g(x), x) = L$

Now we introduce our key theorem:

**Theorem A.27.** *Suppose that Assumption 4.1 holds, with probability at least $1 - \delta$ on the training dataset, we have*

- *The margin induced by $\boldsymbol{p}_{(r,R)}/R$ in p-SVM is at least $(1 - \zeta)\Xi$, where*

$$\zeta = \frac{\log(4\sqrt{\rho^2 + (1+\kappa)d}\|\boldsymbol{v}_{mm}\|^3 d\rho^2)}{R\Xi}.$$

- *The label margin induced by $\boldsymbol{v}_{(r,R)}/r$ in v-SVM is at least $(1-\gamma)\Gamma$, where $\gamma = \frac{2\sqrt{\rho^2+(1+\kappa)d}}{\Gamma\exp((1-\zeta)R\Xi)}$.*

*Proof of Theorem A.27.* From Proposition A.14, we have that for any $\|\boldsymbol{p}\|$, the label margin $1/\|\boldsymbol{v}(\boldsymbol{p})\|$ is at most

$$\Gamma - \frac{C\max_{i\in[n]}(1-s_{i\alpha_i})}{\|\boldsymbol{v}_{mm}\|^3 n\rho^2},$$

where $\alpha_i = 1$ for $i \in \mathcal{C}$ and $\alpha_i = 2$ for $i \in \mathcal{N}$. Recall that $\boldsymbol{s}_i = \mathbb{S}(\boldsymbol{X}_i\boldsymbol{p})$ is the softmax probability vector. We define $q_i^{\boldsymbol{p}} = 1 - s_{i\alpha_i}$ to measure the amount of non-optimality (attention on non-optimal token).

We first consider the convergence of $\boldsymbol{p}_{(r,R)}$ and use contradiction to prove the first statement. Denote $\boldsymbol{p}_R^{mm} = R\boldsymbol{p}_{mm}/\|\boldsymbol{p}_{mm}\|$ which has the same norm as $\boldsymbol{p}_{(r,R)}$ and the direction of $\boldsymbol{p}_{mm}$. Suppose the margin induced by $\boldsymbol{p}_{(r,R)}/R$ is at most $(1-\zeta)\Xi$, i.e. $\min_{i,t_i\neq\alpha_i}(\boldsymbol{x}_{i\alpha_i} - \boldsymbol{x}_{it_i})^\top\boldsymbol{p}_{(r,R)} \leq (1-\zeta)R\Xi, \forall i \in [n]$. Note that here each sequence only has two tokens, thus $t_i, \alpha_i \in [2]$, and $t_i = 3 - \alpha_i$.

According to Lemma A.24, we have

$$\Xi = \|\boldsymbol{p}_{mm}\|_2^{-1} = \Theta((\eta n/d + 1/\rho^2)^{-1/2}).$$

Following the definition of $q_i^{\boldsymbol{p}}$ above, we set $\widehat{q}_{max} = \sup_{i\in[n]} q_i^{\boldsymbol{p}_{(r,R)}}$ and $q_{max}^* = \sup_{i\in[n]} q_i^{\boldsymbol{p}_R^{mm}}$ to be the worst non-optimality in $\boldsymbol{p}_{(r,R)}$ and $\boldsymbol{p}_R^{mm}$. Then we have

$$q_i^{\boldsymbol{p}_R^{mm}} = \frac{\exp(\boldsymbol{x}_{it_i}^\top\boldsymbol{p}_R^{mm})}{\sum_{t\in[2]}\exp(\boldsymbol{x}_{it}^\top\boldsymbol{p}_R^{mm})} \leq \frac{\exp(\boldsymbol{x}_{it_i}^\top\boldsymbol{p}_R^{mm})}{\exp(\boldsymbol{x}_{i\alpha_i}^\top\boldsymbol{p}_R^{mm})} \leq \exp(-R\Xi).$$

The last inequality is from the definition of $\boldsymbol{p}_{mm}$ that $\boldsymbol{p}_{mm}^\top(\boldsymbol{x}_{i\alpha_i} - \boldsymbol{x}_{it}) \geq 1$, so $\boldsymbol{p}_R^{mm\top}(\boldsymbol{x}_{i\alpha_i} - \boldsymbol{x}_{it}) \geq R/\|\boldsymbol{p}_{mm}\| = R\Xi$. Thus, $q_{max}^* = \sup_{i\in[n]} q_i^{\boldsymbol{p}_{mm}} \leq \exp(-R\Xi)$. Then denote the output of attention layer $\boldsymbol{r}_i = \boldsymbol{X}_i^\top\mathbb{S}(\boldsymbol{X}_i\boldsymbol{p}_R^{mm})$. Define $\epsilon_i = \|\boldsymbol{r}_i - \boldsymbol{x}_{i\alpha_i}\|$, we have $y_i \cdot \boldsymbol{r}_i^\top\boldsymbol{v}_{mm} \geq y_i \cdot \boldsymbol{x}_{i\alpha_i}^\top\boldsymbol{v}_{mm} - \|\boldsymbol{r}_i - \boldsymbol{x}_{i\alpha_i}\| \cdot \|\boldsymbol{v}_{mm}\| \geq 1 - \epsilon_i/\Gamma$. So if we set $\epsilon_{max} = \sup_{i\in[n]}\epsilon_i$, $\boldsymbol{v}_{mm}$ achieves a label margin of at least $\Gamma - \epsilon_{max}$ on $(y_i, \boldsymbol{r}_i)_{i\in[n]}$. To better estimate $\epsilon_{max}$, we define $M = \sup_{i\in[n]}\|\boldsymbol{\mu}_i - \boldsymbol{\xi}_i\| \leq \sqrt{\rho^2 + (1+\kappa)d}$, then we have

$$\epsilon_{max} = M \cdot q_{max}^* \leq M\exp(-R\Xi). \tag{57}$$

This implies the max-margin achieved by $(\boldsymbol{p}_{(r,R)}^{mm}, \boldsymbol{v}_{(r,R)}^{mm})$ is at least

$$y_i f(\boldsymbol{v}_{(r,R)}^{mm}, \boldsymbol{p}_{(r,R)}^{mm}; \boldsymbol{x}_i) = y_i\boldsymbol{v}_r^{mm\top}\boldsymbol{r}_i \geq r\Gamma - r\epsilon_{max} \geq r\Gamma - rM\exp(-R\Xi). \tag{58}$$

The first inequality is from $y_i \cdot \boldsymbol{r}_i^\top\boldsymbol{v}_r^{mm} \geq r(\Gamma - \epsilon_i)$ and the last inequality is from (57).

Then we consider the case when $\min_{i,t_i\neq\alpha_i}(x_{i\alpha_i} - x_{it_i})^\top\boldsymbol{p}_{(r,R)} \leq (1-\zeta)R\Xi$ the minimal margin constraint is $\zeta$-violated by $\boldsymbol{p}_{(r,R)}$. Without losing generality we assume that $1 = \underset{i\in[n]}{\operatorname{argmin}}[(\boldsymbol{x}_{i\alpha_i} - \boldsymbol{x}_{it})^\top\boldsymbol{p}_{(r,R)}]_{t\neq\alpha_i}$. Then we have

$$\widehat{q}_{max} \geq \frac{\exp(\boldsymbol{x}_{1t_1}^\top\boldsymbol{p}_{(r,R)})}{\sum_{t\in[2]}\exp(\boldsymbol{x}_{1t}^\top\boldsymbol{p}_{(r,R)})} \geq \frac{1}{2}\frac{\exp(\boldsymbol{x}_{1t_1}^\top\boldsymbol{p}_{(r,R)})}{\exp(\boldsymbol{x}_{1\alpha_1}^\top\boldsymbol{p}_{(r,R)})} \geq \frac{1}{2\exp((1-\zeta)R\Xi)}.$$

From Proposition A.14, optimizing v-SVM on $(y_i, \widehat{\boldsymbol{r}}_i)_{i\in[n]}$ can achieve the max-margin at most

$$\min_{i\in[n]} y_i f(\boldsymbol{v}_{(r,R)}, \boldsymbol{p}_{(r,R)}; \boldsymbol{x}_i) \leq \Gamma - \frac{C}{2\|\boldsymbol{v}_{mm}\|^3 n\rho^2} \cdot e^{-(1-\zeta)R\Xi}. \tag{59}$$

And from the definition $\zeta = \frac{1}{R\Xi}\log(2M\|\boldsymbol{v}_{mm}\|^3 n\rho^2/C)$, we have

$$\frac{C}{2\|\boldsymbol{v}_{mm}\|^3 n\rho^2}\exp(-(1-\zeta)R\Xi) > M\exp(-R\Xi)$$

for sufficiently large $R$, which implies

$$\min_{i \in [n]} y_i \cdot f(\boldsymbol{v}_{(r,R)}, \boldsymbol{p}_{(r,R)}; \boldsymbol{x}_i) < \min_{i \in [n]} y_i \cdot f(\boldsymbol{p}_R^{mm}, \boldsymbol{v}_r^{mm}; \boldsymbol{x}_i).$$

This contradicts with the problem definition (4) to maximize the margin.

Then we prove for the second statement. When the margin induced by $\boldsymbol{p}_{(r,R)}/R$ in $p$-SVM is less than $(1-\zeta)\Xi$, we can use the proof above to derive a contradiction, so $(\boldsymbol{x}_{i\alpha_1} - \boldsymbol{x}_{it})^\top \boldsymbol{p}_{(r,R)} \geq (1-\zeta)R\Xi$ must hold. Then set $\widehat{\boldsymbol{r}}_i = \boldsymbol{X}_i^\top \mathcal{S}(\boldsymbol{X}_i \boldsymbol{p}_{(r,R)})$, we have that

$$
\begin{aligned}
\min_{i \in [n]} y_i \boldsymbol{v}_{(r,R)}^\top \widehat{\boldsymbol{r}}_i &\leq \min_{i \in [n]} y_i \boldsymbol{v}_{(r,R)}^\top \boldsymbol{x}_{i\alpha_i} + \sup_{i \in [n]} |\boldsymbol{v}_{(r,R)}^\top (\widehat{\boldsymbol{r}}_i - \boldsymbol{x}_{i\alpha_i})| \\
&\leq (1-\gamma)\Gamma r + M \exp(-(1-\zeta)R\Xi)r \\
&\leq (1 - \gamma/2)\Gamma r.
\end{aligned}
$$

The second inequality is from previous analysis that $(\boldsymbol{x}_{i\alpha_i} - \boldsymbol{x}_{it})^\top \boldsymbol{p}_{(r,R)} \geq (1-\zeta)R\Xi$, so $|\widehat{\boldsymbol{r}}_i - \boldsymbol{x}_{i1}| \leq M \exp(-(1-\zeta)R\Xi)$; The last inequality is from our definition $\gamma = \frac{2M}{\Gamma \exp((1-\zeta)R\Xi)}$.

Therefore, combining with (58), we have

$$\gamma \Gamma r/2 > rM \exp(-R\Xi),$$

which implies

$$\min_{i \in [n]} y_i \cdot f(\boldsymbol{v}_{(r,R)}, \boldsymbol{p}_{(r,R)}; \boldsymbol{x}_i) < \min_{i \in [n]} y_i \cdot f(\boldsymbol{v}_R^{mm}, \boldsymbol{p}_r^{mm}; \boldsymbol{x}_i).$$

Again this contradicts with the problem definition (4). $\qquad\square$

Then we have the following lemma to bound the derivation $\zeta$ and $\gamma$:

**Lemma A.28.** *Suppose that Assumption 4.1 holds, consider the same setting in Theorem A.27, we have $\zeta < 0.2$ and $\gamma < 0.1$.*

*Proof of Lemma A.28.* From the definition of $\zeta$ in Theorem A.27, we have

$$
\begin{aligned}
\zeta &= \frac{\log(2M\|\boldsymbol{v}_{mm}\|^3 n\rho^2/C)}{R\Xi} = C_1 \frac{\sqrt{\eta n/d + 1/\rho^2}}{R} \log(M\|\boldsymbol{v}_{mm}\|^3 n\rho^2) \\
&\leq C_2 \frac{\sqrt{\eta n/d + 1/\rho^2}}{R} \log\left(\frac{n^2(\rho^2 + d)(\rho^2 \eta n + d)^3}{\rho^2 d^3}\right) = \frac{C_3 \sqrt{\eta n/d + 1/\rho^2}}{R} \log(\rho n) < 0.2.
\end{aligned}
$$

Here $C_1, C_2, C_3 = \Theta(1)$. The first inequality is from the upper bound of $\|\boldsymbol{v}_{mm}\|$ in Lemma A.16 and the last inequality is from the definition of $R$ in Assumption 4.1. And for $\gamma$, we have

$$\gamma = \frac{2M}{\Gamma \exp((1-\zeta)R\Xi)} = C_1' \frac{M\|\boldsymbol{v}_{mm}\|}{\exp(R/\|\boldsymbol{v}_{mm}\|)} \leq C_2' \frac{\sqrt{(\rho^2 + d)(\eta n/d + 1/\rho^2)}}{\exp(R/\sqrt{\eta n/d + 1/\rho^2})} < 1.$$

Here $C_1', C_2' = \Theta(1)$. The first inequality is from the lower and upper bound of $\|\boldsymbol{v}_{mm}\|$ in Lemma A.16 and the last inequality is from the definition of $R$ in Assumption 4.1. $\qquad\square$

Then we can estimate $\langle \boldsymbol{p}_{(r,R)}, \boldsymbol{\mu} \rangle$ with the following lemma:

**Lemma A.29.** *Suppose that Assumption 4.1 holds, with probability at least $1 - \delta$ on the training dataset, $\boldsymbol{p}_{(r,R)}$ should satisfy*

$$0.5(1-\zeta)R\Xi \leq \langle \boldsymbol{p}_{(r,R)}, \boldsymbol{\mu}_j \rangle \leq R\rho$$

*for $j \in \{1, 2\}$.*

*Proof of Lemma A.29.* The upper bound is given by

$$\langle \boldsymbol{p}_{(r,R)}, \boldsymbol{\mu}_j \rangle \leq \|\boldsymbol{p}_{(r,R)}\| \|\boldsymbol{\mu}_j\| = R\rho.$$

Then we use contradiction to prove for the lower bound. From Theorem A.27, $\boldsymbol{p}_{(r,R)}$ satisfies

$$\boldsymbol{p}_{(r,R)}^\top(\boldsymbol{\mu}_i - \boldsymbol{\xi}_i) \geq (1-\zeta)R\Xi, i \in \mathcal{C}$$
$$\boldsymbol{p}_{(r,R)}^\top(\boldsymbol{\xi}_i - \boldsymbol{\mu}_i) \geq (1-\zeta)R\Xi, i \in \mathcal{N} \qquad (60)$$

If $\langle \boldsymbol{p}_{(r,R)}, \boldsymbol{\mu}_j \rangle \leq 0.5(1-\zeta)R\Xi$, then for every clean sample from cluster j we must have $\langle \boldsymbol{p}_{(r,R)}, \boldsymbol{\xi}_i \rangle \leq -0.5(1-\zeta)R\Xi$ and thus

$$\langle \boldsymbol{p}_{(r,R)}, \sum_{i\in\mathcal{C}_j} \boldsymbol{\xi}_i \rangle = \sum_{i\in\mathcal{C}_j} \langle \boldsymbol{p}_{(r,R)}, \boldsymbol{\xi}_i \rangle \leq -0.5(1-\zeta)R\Xi n_{1j}.$$

So we could estimate $\|\boldsymbol{p}_{(r,R)}\|$ as follows

$$\|\boldsymbol{p}_{(r,R)}\| \geq 0.5(1-\zeta)R\Xi \cdot n_{1j} \frac{1}{\|\sum_{i\in\mathcal{C}_j} \boldsymbol{\xi}_i\|} = 0.5(1-\zeta)R\Xi \cdot n_{1j} \frac{1}{\sqrt{\sum_{i\in\mathcal{C}_j} \|\boldsymbol{\xi}_i\|^2 + \sum_{i,j\in\mathcal{C}_j} \langle \boldsymbol{\xi}_i, \boldsymbol{\xi}_j \rangle}}$$

$$\geq 0.5(1-\zeta)R\Xi \cdot n_{1j} \frac{1}{\sqrt{2 \cdot n_{1j} \cdot (1+\kappa)d}} \geq 0.4R\Xi \cdot \frac{\sqrt{n_{1j}}}{\sqrt{2(1+\kappa)d}}.$$

The first inequality is from the property of innerproduct; The second inequality is from Lemma A.65 and the definition of d in Assumption 4.1; The last inequality is from Lemma A.28. Meanwhile, from Lemma A.24 we have $\|\boldsymbol{p}_{mm}\| \leq \sqrt{8/\rho^2 + 17\eta n/d}$. Recall that $\Xi = \|\boldsymbol{p}_{mm}\|^{-1}$. Therefore, we further have

$$\|\boldsymbol{p}_{(r,R)}\| \geq 0.4R\Xi \cdot \frac{\sqrt{n_{1j}}}{\sqrt{2(1+\kappa)d}} \geq \sqrt{\frac{0.4^2 n_{1j}}{(8/\rho^2 + 17\eta n/d) \cdot 2(1+\kappa)d}} \cdot R$$

$$\geq \sqrt{\frac{0.04(n - \eta n - O(\sqrt{n}))}{(8/\rho^2 + 17\eta n/d) \cdot (1+\kappa)d}} \cdot R > R.$$

The second inequality is from Lemma A.24; The third inequality is from Lemma A.68 and the last inequality is from Assumption 4.1 about SNR and $\eta$. This leads to a contradiction.

$\square$

Now we can estimate the output of attention layer for some test sample $(\boldsymbol{X}, y)$.

**Lemma A.30.** *Suppose that Assumption 4.1 holds, with probability at least $1-\delta$ on the training dataset, for a given a test sample $\boldsymbol{X}, y$, where $\boldsymbol{X} = (\boldsymbol{\mu}^\star, \boldsymbol{\xi}^\star)$, $\boldsymbol{\mu}^\star$ can be $\boldsymbol{\mu}_1$ or $\boldsymbol{\mu}_2$, we have with probability at least $1 - \exp\left(-\frac{1}{2}(\frac{1}{2}(1-\zeta)\Xi - K/R)^2\right)$ that*

$$\langle \boldsymbol{p}_{(r,R)}, \boldsymbol{\mu}^\star \rangle - \langle \boldsymbol{p}_{(r,R)}, \boldsymbol{\xi}^\star \rangle \geq K,$$

*where $K \leq \frac{1}{2}(1-\zeta)R\Xi$ and $\zeta, \Xi$ are defined in Theorem A.27.*

*Proof of Lemma A.30.* Note that $\boldsymbol{p}^\top \boldsymbol{\xi}^\star$ follows Gaussian distribution $\mathcal{N}(0, R^2)$, we have

$$\mathbb{P}(\langle \boldsymbol{p}_{(r,R)}, \boldsymbol{\mu}^\star \rangle - \langle \boldsymbol{p}_{(r,R)}, \boldsymbol{\xi}^\star \rangle < K) = \mathbb{P}(\langle \boldsymbol{p}_{(r,R)}, \boldsymbol{\xi}^\star \rangle > \langle \boldsymbol{p}_{(r,R)}, \boldsymbol{\mu}^\star \rangle - K) \leq \mathbb{P}(\boldsymbol{p}_{(r,R)}^\top \boldsymbol{\xi}^\star > \frac{1}{2}(1-\zeta)R\Xi - K)$$

$$\leq \exp\left(-\frac{1}{2}(\frac{1}{2}(1-\zeta)\Xi - K/R)^2\right).$$

The first inequality is from Lemma A.29 and the second inequality comes from the property of Gaussian tail probability. $\square$

We also have the following lemma to estimate $\boldsymbol{v}_{(r,R)}$. We first prove that $\boldsymbol{v}_{(r,R)}$ can be expressed as the sum of signal and noise tokens.

**Lemma A.31.** *The solution of constrained optimization problem (4) $\boldsymbol{v}_{(r,R)}$ can be expressed in the form that*

$$\boldsymbol{v}_{(r,R)} = \lambda_1 \boldsymbol{\mu}_1 + \lambda_2 \boldsymbol{\mu}_2 + \sum_{i=1}^n \theta_i \boldsymbol{\xi}_i.$$

*Proof of Lemma A.31.* Similar to Theorem A.27, define $\widehat{r}_i = X_i^\top S(X_i p_{(r,R)})$ as the output of attention layer, we have

$$v_{(r,R)} = \underset{\|v\| \le r}{\operatorname{argmax}} \min_{i \in [n]} y_i v^\top r_i. \tag{61}$$

Then denote $s = \min_{i \in [n]} y_i v^\top r_i$ and $s_r = \min_{i \in [n]} y_i v_{(r,R)}^\top r_i$. Then (61) can be written as

$$(v_{(r,R)}, s_r) = \underset{v,s}{\operatorname{argmax}} \, s, \text{ s.t. } y_i v^\top r_i \ge s, \quad 1 \le i \le n$$

$$\|v\| \le r.$$

The corresponding Lagrangian function is

$$L(s, \psi) = -s + \sum_{i=1}^{n} \psi_i y_i (s - y_i v^\top r_i) + \psi_0 (\|v\|^2 - r^2).$$

Take derivative of this function on $(s, v)$, we have

$$-\sum_{i=1}^{n} \psi_i y_i r_i + 2\psi_0 v = 0.$$

Therefore from the last equation we can get

$$v = \frac{1}{2\psi_0} \sum_{i=1}^{n} \psi_i y_i r_i.$$

As $r_i = \beta_i \mu_i + (1 - \beta_i)\xi_i$ for every $i \in [n]$, $v$ can be expressed as the combination of signal and noise token of every sample:

$$v_{(r,R)} = \lambda_1 \mu_1 + \lambda_2 \mu_2 + \sum_{i=1}^{n} \theta_i \xi_i.$$

$\square$

Based on this representation, we can then bound the parameters in $v_{(r,R)}$:

**Lemma A.32.** *Suppose that Assumption 4.1 holds, denote $v_{(r,R)} = \lambda_1 \mu_1 + \lambda_2 \mu_2 + \sum_{i \in [n]} \theta_i \xi_i$. Then with probability at least $1 - \delta$ on the training dataset, we have*

$$\lambda_1 \ge (1 - \gamma)\Gamma r / \rho^2,$$
$$\lambda_2 \le -(1 - \gamma)\Gamma r / \rho^2,$$
$$|\theta_i| \le 2\sqrt{1/\rho^2 + 5\eta n/d} \cdot \Gamma r / \sqrt{d}.$$

*Proof of Lemma A.32.* The first two statements are obvious because from Theorem A.27 we have

$$y_i v_{(r,R)}^\top \mu_i \ge (1 - \gamma)\Gamma r,$$

for $\forall i \in C$. This implies $|\lambda_j| \ge (1 - \gamma)\Gamma r / \rho^2$ for $j \in \{1, 2\}$. Meanwhile, we decompose $v_{(r,R)} = v_\mu + v_\xi$ where $v_\mu = \lambda_1 \mu_1 + \lambda_2 \mu_2$ and $v_\xi = \sum_{i \in [n]} \theta_i \xi_i$. And we can upper bound $\|v_\xi\|$ as

$$\|v_\xi\|^2 = \|v_{(r,R)}\|^2 - \|v_\mu\|^2 \le r^2 - \lambda_1^2 \rho^2 - \lambda_2^2 \rho^2 \le r^2 (1 - 2(1 - \gamma)^2 \Gamma^2 / \rho^2).$$

The first inequality is from $\|v\| \le r$ and the second inequality is from the first two statements we just proved. Therefore, denote $j = \underset{i \in [n]}{\operatorname{argmax}} \, \theta_i$, we have

$$\theta_j^2 \|\xi_j\|^2 \le \|v_\xi\|^2 \le r^2 (1 - 2(1 - \gamma)^2 \Gamma^2 / \rho^2).$$

Then we can upper bound $|\theta_j|$ as

$$\theta_j^2 \leq r^2(1 - 2(1-\gamma)^2\Gamma^2/\rho^2)/\|\boldsymbol{\xi}_j\|^2 \leq r^2(1 - 2(1-\gamma)^2\Gamma^2/\rho^2)/(1-\kappa)d$$

$$= r^2\left(1 - \frac{2(1-\gamma)^2}{\|\boldsymbol{v}_{mm}\|^2\rho^2}\right)/(1-\kappa)d \leq r^2\left(1 - \frac{1}{(2/\rho^2 + 5\eta n/d)\rho^2}\right)/(1-\kappa)d$$

$$= \frac{1 + 5\eta n\rho^2/d}{2 + 5\eta n\rho^2/d} \cdot \frac{r^2}{(1-\kappa)d} \leq \left(\frac{1}{\rho^2} + \frac{5\eta n}{d}\right) \cdot \frac{\Gamma^2 r^2}{2d}.$$

The second inequality is from Lemma A.65; The third inequality is from Lemma A.16 that $\|\boldsymbol{v}_{mm}\| \leq \sqrt{2/\rho^2 + 5\eta n/d}$ and our definition of $\gamma = \frac{2\sqrt{\rho^2 + (1+\kappa)d}}{\Gamma \exp((1-\zeta)R\Xi)}$; The last inequality is from $\Gamma = \|\boldsymbol{v}_{mm}\|^{-1} \geq (2/\rho^2 + 5\eta n/d)^{-1}$. Thus, we can bound $|\theta_j|$ as

$$|\theta_j| \leq 2\sqrt{1/\rho^2 + 5\eta n/d} \cdot \Gamma r/\sqrt{d}.$$

$\square$

Therefore, we can prove the main theorem.

*Proof of Theorem A.10.* First we show that the model can perfectly classify all training samples. From Theorem A.27, we have

$$y_i\boldsymbol{v}_{(r,R)}^\top\boldsymbol{r}_i \geq (1-\gamma)\Gamma r > 0$$

for $\forall i \in [n]$. The last inequality is from Lemma A.28. Thus $y_i = \text{sign}(f(\boldsymbol{X}_i; \boldsymbol{v}_{(r,R)}, \boldsymbol{p}_{(r,R)}))$ for all $i \in [n]$. Then we bound the test error. Given a test sample $\boldsymbol{X}, y$, where $\boldsymbol{X} = (\boldsymbol{\mu}^\star, \boldsymbol{\xi}^\star)$, $\boldsymbol{\mu}^\star$ can be $\boldsymbol{\mu}_1$ or $\boldsymbol{\mu}_2$. From RemarkA.66, with probability at least $1 - 6n\exp(-d/4C_1 n^2)$,

$$|\langle\boldsymbol{\xi}^\star, \boldsymbol{\xi}_i\rangle| \leq \frac{d}{C_1 n}. \tag{62}$$

According to Lemma A.30, with probability at least $1 - \exp\left(-\frac{1}{2}(\frac{1}{2}(1-\zeta)\Xi - K/R)^2\right)$, we have

$$y \cdot f(\boldsymbol{v}_{(r,R)}, \boldsymbol{p}_{(r,R)}; \boldsymbol{X}) \geq \frac{\langle y\boldsymbol{v}_{(r,R)}, e^K\boldsymbol{\mu}^\star + \boldsymbol{\xi}^\star\rangle}{e^K + 1} \geq \frac{e^K(1-\gamma)\Gamma r\|\boldsymbol{\mu}^\star\|^2}{\rho^2(e^K + 1)} - \frac{1}{e^K + 1}\sum_{i\in[n]}|\theta_i| \cdot |\langle\boldsymbol{\xi}_i, \boldsymbol{\xi}^\star\rangle|. \tag{63}$$

Let $K = \log(\sqrt{d}\sqrt{1/\rho^2 + \eta n/d}) + C < \frac{1}{2}(1-\zeta)R\Xi$. By uniform bound, we have that with probability at least $1 - 6n\exp(-d/4C_1 n^2) - \exp\left(-\frac{1}{2}(\frac{1}{2}(1-\zeta)\Xi - K/R)^2\right)$,

$$y \cdot f(\boldsymbol{v}_{(r,R)}, \boldsymbol{p}_{(r,R)}; \boldsymbol{X}) \geq \frac{e^K(1-\gamma)\Gamma r - n \cdot d/(C_1 n) \cdot 2\sqrt{1/\rho^2 + \eta n/d} \cdot \Gamma r/\sqrt{d}}{1 + e^K}$$

$$\geq \frac{0.9e^K\Gamma r - \sqrt{d}/C_1 \cdot 2\sqrt{1/\rho^2 + \eta n/d} \cdot \Gamma r}{1 + e^K}$$

$$> 0,$$

where the first inequality uses (62), (63) and Lemma A.32; The second inequality is from Lemma A.28 and the last inequality is from Assumption 4.1 and our selection of $K$. Therefore,

$$\mathbb{P}(y \neq f(\boldsymbol{v}_{(r,R)}, \boldsymbol{p}_{(r,R)}; \boldsymbol{X})) \leq \exp\left(-\frac{1}{2}\frac{1}{2}(1-\zeta)\Xi - \frac{K}{R})^2\right) + \exp(-\Omega(d/n^2)),$$

where $\zeta = \frac{\log(2M\|\boldsymbol{v}_{mm}\|^3 n\rho^2)}{R\Xi} = \Theta\left(\frac{\sqrt{\eta n/d + 1/\rho^2}}{R}\log(\rho n)\right)$, $K = \log(\sqrt{d}\sqrt{1/\rho^2 + \eta n/d}) + C = \Theta(\log(\sqrt{d/\rho^2 + \eta n}))$ and $\Xi = \|\boldsymbol{p}_{mm}\|_2^{-1} = \Theta((\eta n/d + 1/\rho^2)^{-1/2})$. Plugging in the order of $\Xi$ and

$K$, we have

$$\mathbb{P}_{(\boldsymbol{X},y)\sim\mathcal{D}}(y \neq \mathrm{sign}(f(\boldsymbol{X}; \boldsymbol{v}_{(r,R)}, \boldsymbol{p}_{(r,R)})))$$
$$= \mathbb{P}_{(\boldsymbol{X},y)\sim\mathcal{D}}(y \neq \mathrm{sign}(f(\boldsymbol{X}; \boldsymbol{v}_{(r,R)}, \boldsymbol{p}_{(r,R)})), y = -\widetilde{y})$$
$$+ \mathbb{P}_{(\boldsymbol{X},y)\sim\mathcal{D}}(y \neq \mathrm{sign}(f(\boldsymbol{X}; \boldsymbol{v}_{(r,R)}, \boldsymbol{p}_{(r,R)})), y = \widetilde{y})$$
$$= \eta + \mathbb{P}_{(\boldsymbol{X},y)\sim\mathcal{D}}(y \neq \mathrm{sign}(f(\boldsymbol{X}; \boldsymbol{v}_{(r,R)}, \boldsymbol{p}_{(r,R)})), y = \widetilde{y})$$
$$\leq \eta + \exp(-d/C_1 n^2) + \exp\left(-\Theta\left(\frac{(1-\zeta)}{\sqrt{\eta n/d + 1/\rho^2}} - \frac{\log(\sqrt{d/\rho^2 + \eta n})}{R}\right)^2\right)$$
$$= \eta + \exp\left(-\Omega\left(\frac{d}{n^2}\right)\right) + \exp\left(-\Omega\left(\frac{(1-\zeta)}{\sqrt{\eta n/d + 1/\rho^2}} - \frac{\log(n)}{R}\right)^2\right),$$

where $\zeta = \Theta\left(\frac{\sqrt{\eta n/d + 1/\rho^2}}{R}\log(\rho n)\right)$. This completes the proof. $\qquad\square$

### A.2.4 Proof of Thm. 4.2

In the case of multiple tokens setting, we denote the $\tau$-th token in input $\boldsymbol{x}_i^{(\tau)}$ as $\boldsymbol{\xi}_{i,\tau}$. And we introduce the definition of **Combined Noise Token**.

**Definition A.33** (Combined Noise Token). $\overline{\boldsymbol{\xi}}_i$ is called combined noise token if $\overline{\boldsymbol{\xi}}_i = \sum_{\tau=2}^T t_{i,\tau}\boldsymbol{\xi}_{i,\tau}$ and $t_\tau \in [0,1]$, $\sum_{\tau=2}^T t_\tau = 1$.

Then we have the following lemma to estimate the norm and innerproduct of these combined noises:

**Lemma A.34** (Properties of Combined Noise Tokens). *Suppose that* $\delta > 0$ *and* $\kappa' = O(\sqrt{\log(6n/\delta)/d}) = \widetilde{O}(1/\sqrt{d})$ *.If Lemma A.65 holds, we have*

$$(1-\kappa')d/T \leq \|\overline{\boldsymbol{\xi}}_i\|_2^2 \leq (1+\kappa')d$$
$$|\langle \overline{\boldsymbol{\xi}}_i, \overline{\boldsymbol{\xi}}_j \rangle| \leq 2T^2\sqrt{d\log(6n^2/\delta)}$$

*for any* $i,j \in [n]$.

**Definition A.35.** If the event in Lemma A.65 and A.34 occur, let us say we have a **good run**.

Lemmas A.65 and A.34 show that a good run occurs with probability at least $1 - \delta$. In what follows, we will assume that a good run occurs.

Because the optimal tokens are data specific and not fixed, we consider the following token selection rule:

$$\boldsymbol{r}_i^{sec} = \boldsymbol{x}_i^{(1)} = \boldsymbol{\mu}_k, \ i \in \mathcal{C}_k, k \in \{1,2\}$$
$$\boldsymbol{r}_i^{sec} = \boldsymbol{x}_i^{(2)} = \boldsymbol{\xi}_{i,2}, \ i \in \mathcal{N}. \tag{64}$$

This selects signal token for all clean samples and the first noise token for all noisy sample. Following this new token selection rule, we redefine the p-SVM and v-SVM:

**Definition A.36** (p-SVM for Second-token Selection).

$$\boldsymbol{p}_{sec} = \operatorname*{argmin}_{\boldsymbol{p}\in\mathbb{R}^d} \|\boldsymbol{p}\| \quad \text{subject to:}$$
$$\boldsymbol{p}^\top(\boldsymbol{x}_i^{(\alpha_i)} - \boldsymbol{x}_i^{(t)}) \geq 1, \ \alpha_i = 1 \ \text{for} \ i \in \mathcal{C} \ \text{and} \ \alpha_i = 2 \ \text{for} \ i \in \mathcal{N}, \ t \in [2,T]\backslash\{\alpha_i\}. \tag{65}$$

Let $\Xi := 1/\|\boldsymbol{p}_{sec}\|$ be the margin induced by $\boldsymbol{p}_{sec}$.

Then for a given $\boldsymbol{p}$, we define $\boldsymbol{v}(\boldsymbol{p})$ as the standard max-margin classifier on $(\boldsymbol{r}_i, y_i)_{i\in[n]}$ and $\boldsymbol{v}_{sec}$ as the standard max-margin classifier on $(\boldsymbol{r}_i^{sec}, y_i)_{i\in[n]}$ which represents the limiting case when $\boldsymbol{p} = \boldsymbol{p}_{sec}$ and $R \to +\infty$.

**Definition A.37** (v-SVM for Second-token Selection).

$$\boldsymbol{v}(\boldsymbol{p}) := \operatorname*{argmin}_{\boldsymbol{v}\in\mathbb{R}^d} \|\boldsymbol{v}\| \ \text{s.t.} \ y_i \cdot \boldsymbol{v}^\top \boldsymbol{r}_i \geq 1, \quad \text{for all} \ i \in [n]. \tag{66}$$

$\Gamma(\boldsymbol{p}) := 1/\|\boldsymbol{v}(\boldsymbol{p})\|$ is the **label margin** induced by $\boldsymbol{v}(\boldsymbol{p})$. When $\boldsymbol{r}_i = \boldsymbol{r}_i^{sec}, i \in [n]$, we define

$$\boldsymbol{v}_{sec} := \underset{\boldsymbol{v} \in \mathbb{R}^d}{\operatorname{argmin}} \|\boldsymbol{v}\| \text{ s.t. } y_i \cdot \boldsymbol{v}^\top \boldsymbol{r}_i^{sec} \geq 1, \quad \text{for all } i \in [n]. \tag{67}$$

$\Gamma := 1/\|\boldsymbol{v}_{sec}\|$ is the label margin induced by $\boldsymbol{v}_{sec}$.

Since $\boldsymbol{v}_{sec}$ satisfies the KKT conditions of the max-margin problem (66), by the stationarity condition, we can represent $\boldsymbol{v}_{sec}$ as

$$\boldsymbol{v}_{sec} = \lambda_1 \boldsymbol{\mu}_1 + \lambda_2 \boldsymbol{\mu}_2 + \sum_{i \in [n]} \sum_{\tau=2}^{T} \theta_{i,\tau} \boldsymbol{\xi}_{i,\tau}. \tag{68}$$

Note that the conditions in (66) can be written as:

*Condition* 12 (Second-token Selection).

$$\begin{cases} \boldsymbol{v}^\top \boldsymbol{\mu}_1 \geq 1 \\ -\boldsymbol{v}^\top \boldsymbol{\mu}_2 \geq 1 \\ y_i \boldsymbol{v}^\top \boldsymbol{\xi}_{i,2} \geq 1, i \in \mathcal{N} \end{cases}$$

Plugging (44) in the condition 1, we can rewrite these conditions as:

$$\begin{cases} \lambda_1 \cdot \|\boldsymbol{\mu}_1\|^2 \geq 1 \\ -\lambda_2 \cdot \|\boldsymbol{\mu}_2\|^2 \geq 1 \\ y_i(\theta_{i,2} \cdot \|\boldsymbol{\xi}_{i,2}\|^2 + \sum_{\substack{i' \in [n], \\ \tau' \in [3,T]}} \theta_{i',\tau'} \langle \boldsymbol{\xi}_{i,\tau_i}, \boldsymbol{\xi}_{i',\tau'} \rangle) \geq 1, i \in \mathcal{N} \end{cases}$$

Then we introduce a lemma to estimate the coefficients $\theta_{i,\tau}$ of $\boldsymbol{v}_{sec}$ under this condition:

**Lemma A.38** (balanced noise factor for KKT points). *Under Condition 4.1 and 12, on a good run, we have that for $\boldsymbol{v}_{sec}$,*

$$\theta_{i,\tau} = 0, \quad i \in \mathcal{C}, \tau \in [3, T]; i \in \mathcal{N} \tag{69}$$

$$\theta_{i,2} \in \left[ \frac{(1-\kappa)d - 4n_2\sqrt{d\log(6n^2/\delta)}}{(1+\kappa)d((1-\kappa)d - 2n_2 T\sqrt{d\log(6n^2/\delta)})}, \frac{1}{(1-\kappa)d - 2n_2 T\sqrt{d\log(6n^2/\delta)}} \right], \quad i \in \mathcal{N}. \tag{70}$$

*Proof of Lemma A.38.* Note that Condition 12 does not have any constraint for noise tokens with index $i \in \mathcal{C}, i \in \mathcal{N}, \tau \neq 2$. Thus we have $\theta_{i,\tau} = 0$ for any $i \in \mathcal{C}, i \in \mathcal{N}, \tau \neq 2$ in the representation (68). For $\theta_{i,2}$ with $i \in \mathcal{N}$, we first prove the upper bound by contradiction. Denote $j = \underset{i \in \mathcal{N}}{\operatorname{argmax}} \theta_{i,2}$. Then we have

$$y_j \boldsymbol{v}^\top \boldsymbol{\xi}_{j,2} = \sum_{i \in \mathcal{N}} y_i y_j \theta_{i,2} \langle \boldsymbol{\xi}_{i,2}, \boldsymbol{\xi}_{j,2} \rangle = \theta_{j,2} \|\boldsymbol{\xi}_{j,2}\|_2^2 + \sum_{i \neq j, i \in \mathcal{N}} y_i y_j \theta_{i,2} \langle \boldsymbol{\xi}_{i,2}, \boldsymbol{\xi}_{j,2} \rangle$$

$$\geq \theta_{j,2} \cdot (1-\kappa)d - n_2 T \theta_{j,2} \cdot 2\sqrt{d\log(6n^2/\delta)},$$

where the inequality is from Lemma A.65 and the definition of $j$. Consider the contrary case when $\theta_{j,2} > \frac{1}{(1-\kappa)d - 2n_2 T\sqrt{d\log(6n^2/\delta)}}$, we have

$$y_j \boldsymbol{v}^\top \boldsymbol{\xi}_{j,2} > \frac{1}{(1-\kappa)d - 2n_2 T\sqrt{d\log(6n^2/\delta)}} \cdot \left((1-\kappa)d - n_2 T \cdot 2\sqrt{d\log(6n^2/\delta)}\right) = 1.$$

By the complementary slackness, if $y_j \boldsymbol{v}^\top \boldsymbol{\xi}_{j,2} > 1$, then we must have $\theta_{j,2} = 0$, and thus we reach a contradiction.

Then we prove for the lower bound. For $\forall j \in \mathcal{N}$ we have

$$1 \leq \theta_{j,2} \|\boldsymbol{\xi}_{j,2}\|_2^2 + \sum_{i \neq j, i \in \mathcal{N}} y_i y_j \theta_{i,2} \langle \boldsymbol{\xi}_{i,2}, \boldsymbol{\xi}_{j,2} \rangle$$

$$\leq \theta_{j,2} \cdot (1+\kappa)d + n_2 \max_{i \in \mathcal{N}} \theta_{i,2} \cdot 2\sqrt{d\log(6n^2/\delta)}$$

$$\leq \theta_{j,2} \cdot (1+\kappa)d + \frac{n_2}{(1-\kappa)d - 2n_2 T\sqrt{d\log(6n^2/\delta)}} \cdot 2\sqrt{d\log(6n^2/\delta)}.$$

The second inequality is due to Lemma A.65 and the last inequality is from the upper bound we just get. Therefore, we have

$$\theta_{j,2} \geq \frac{(1-\kappa)d - 4n_2\sqrt{d\log(6n^2/\delta)}}{(1+\kappa)d((1-\kappa)d - 2n_2T\sqrt{d\log(6n^2/\delta)})}.$$

This completes the proof. $\qquad\qquad\square$

Then we introduce a lemma to estimate $\|\boldsymbol{v}_{sec}\|$:

**Lemma A.39** (Norm of $\boldsymbol{v}_{sec}$). *Under Condition 4.1, on a good run, for the solution $\boldsymbol{v}_{sec}$ of* (66) *under the token selection* (64)*, we have*

$$\frac{2}{\rho^2} + \frac{\eta n}{2d} \leq \|\boldsymbol{v}_{sec}\|^2 \leq \frac{2}{\rho^2} + \frac{5\eta n}{d}.$$

*This implies*

$$\|\boldsymbol{v}_{sec}\| = \Theta\left(\sqrt{\frac{1}{\rho^2} + \frac{\eta n}{d}}\right).$$

This lemma is the same as Lemma A.16 except that we substitute $\boldsymbol{\xi}_i$ with $\boldsymbol{\xi}_{i,2}$. So we omit the proof for clarity. The remaining proof idea is similar to that of two-token setting: 1) Prove the (relative) optimality of this second token selection. 2) Prove the convergence of parameters and training/test error.

We first prove the optimal token for clean samples from the max-margin they induced.

**Proposition A.40** (optimal token condition). *Suppose that Assumption 4.1 holds, on a good run, for all $\boldsymbol{p}$ that $\exists i \in \mathcal{C}, \tau \in [2,T], \langle \boldsymbol{p}, \boldsymbol{\mu}_i - \boldsymbol{\xi}_{i,\tau}\rangle < \|\boldsymbol{p}\|_2/\|\boldsymbol{p}_{sec}\|_2$, the token selection under $\boldsymbol{p}$ results in a label margin (Def. A.37) of at most $\Gamma_{sec} - \frac{C}{\|\boldsymbol{v}_{sec}\|_2^3 n\rho^2} \cdot \max_{i \in \mathcal{C}}(1 - s_{i1})$.*

*Proof of Proposition A.40.* As we consider the non-asymptotic case here, $s_{i1}$ cannot be exactly 1 for $\forall i \in [n]$, so we only consider the case that all clean samples are mixed. Denote $\boldsymbol{r}_i = \sum_{\tau=1}^{T} \beta_{i,\tau}\boldsymbol{x}_i^{(\tau)} = \beta_i\boldsymbol{\mu}_i + (1-\beta_i)\overline{\boldsymbol{\xi}}_i$ for $i \in \mathcal{C}$, where $\overline{\boldsymbol{\xi}}_i$ is the combined noise token in Definition A.33. The condition in this case becomes

*Condition* 13 (Mixed clean samples, multiple case).

$$y_i\boldsymbol{v}''^{\top}\boldsymbol{r}_i \geq 1.$$

This indicates

$$\beta_i y_i \lambda_i''\|\boldsymbol{\mu}_i\|^2 + (1-\beta_i)(\theta_i''\|\overline{\boldsymbol{\xi}}_i\|^2 + \sum_{j\neq i} y_iy_j\theta_j''\langle\overline{\boldsymbol{\xi}}_i, \overline{\boldsymbol{\xi}}_j\rangle) \geq 1, i \in \mathcal{C}$$

Assume that $\min\{\lambda_1'' \cdot \|\boldsymbol{\mu}_1\|^2, -\lambda_2'' \cdot \|\boldsymbol{\mu}_2\|^2\} = q$ in optimal $\boldsymbol{v}''$. Similar to the two-token scenario, we consider the case when $q < 1$. Denote $\alpha = \frac{1-\beta_i q}{1-\beta_i}$. We have $\alpha > 1$ due to $q < 1$ and $0 \leq \beta_i < 1$. Without losing generality, we assume $\lambda_1'' \cdot \|\boldsymbol{\mu}_1\|^2 = q < 1$. The special condition $\langle \boldsymbol{p}, \boldsymbol{\mu}_i - \boldsymbol{\xi}_{i,\tau}\rangle < \|\boldsymbol{p}\|_2/\|\boldsymbol{p}_{sec}\|_2$ here is to guarantee that there always exists an upper bound for $\beta_i$. Then consider the following relaxed condition:

*Condition* 14 (Relaxed version of constraints in Condition 13).

$$\theta_i''\|\overline{\boldsymbol{\xi}}_i\|^2 + \sum_{j\neq i} y_iy_j\theta_j''\langle\overline{\boldsymbol{\xi}}_i, \overline{\boldsymbol{\xi}}_j\rangle \geq \alpha, i \in \mathcal{C}_1.$$

Denote the optimal solution under Condition 14 as $\breve{\boldsymbol{v}}$ and the corresponding coefficients in $\breve{\boldsymbol{v}}$ as $\breve{\lambda}_1, \breve{\lambda}_2$ and $\breve{\theta}_i$, i.e.

$$\breve{\boldsymbol{v}} = \breve{\lambda}_1\boldsymbol{\mu}_1 + \breve{\lambda}_2\boldsymbol{\mu}_2 + \sum_{i\in[n]} \breve{\theta}_i\overline{\boldsymbol{\xi}}_i. \qquad\qquad (71)$$

Since the constraints in Condition 14 is a subset of the constraints in Condition 13, we have $\|\breve{\boldsymbol{v}}\| \leq \|\boldsymbol{v}''\|$. Meanwhile, we have the following lemma to estimate $\breve{\theta}_i$:

**Lemma A.41.** *Under Condition 4.1 and 14, on a good run, we have*

$$\breve{\theta}_i = 0, i \in [n]\backslash\mathcal{C}_1;$$

$$\breve{\theta}_i \in \left[\frac{\alpha}{(1+\kappa')d}\left(1 - \frac{2T^2 n_{11}\sqrt{d\log(6n^2/\delta)}}{(1-\kappa')d/T - 2T^2 n_{11}\sqrt{d\log(6n^2/\delta)}}\right), \frac{\alpha}{((1-\kappa')d/T - 2T^2 n_{11}\sqrt{d\log(6n^2/\delta)})}\right], i \in \mathcal{C}_1.$$

*Proof of Lemma A.41.* Note that Condition 14 does not have any constraint for samples with $i \in [n]\backslash\mathcal{C}_1$. Thus we have $\breve{\theta}_i = 0$ for any $i \in [n]\backslash\mathcal{C}_1$ in the representation (71). Denote $j = \underset{i\in\mathcal{C}_1}{\operatorname{argmax}}\,\breve{\theta}_i$, then we have

$$\breve{\theta}_j \cdot \|\overline{\boldsymbol{\xi}}_j\|^2 + \sum_{k\neq j} y_k y_j \breve{\theta}_k \langle \overline{\boldsymbol{\xi}}_i, \overline{\boldsymbol{\xi}}_j\rangle \geq \breve{\theta}_j \|\overline{\boldsymbol{\xi}}_j\|^2 - 2T^2\breve{\theta}_j n_{11}\sqrt{d\log(6n^2/\delta)} \geq \breve{\theta}_j((1-\kappa')d/T - 2T^2 n_{11}\sqrt{d\log(6n^2/\delta)}).$$

The two inequalities are from Lemma A.65 and our definition of $j$. Consider the contrary case when $\breve{\theta}_j > \frac{\alpha}{((1-\kappa')d/T - 2T^2 n_{11}\sqrt{d\log(6n^2/\delta)})}$, we have

$$y_j \breve{\boldsymbol{v}}^\top \boldsymbol{\xi}_j > \alpha.$$

By the complementary slackness condition, if $y_j \breve{\boldsymbol{v}}^\top \boldsymbol{\xi}_j > \alpha$, then we must have $\breve{\theta}_j = 0$, and thus we reach a contradiction.

Then we lower bound $\breve{\theta}_i$. For $\forall i \in \mathcal{C}_1$ we have

$$\alpha \leq \breve{\theta}_i \cdot \|\overline{\boldsymbol{\xi}}_i\|^2 + \sum_{j\neq i} y_i y_j \breve{\theta}_i \langle \overline{\boldsymbol{\xi}}_i, \overline{\boldsymbol{\xi}}_j\rangle \leq \breve{\theta}_i(1+\kappa')d + 2T^2 n_{11} \max_{i\in[n]} \breve{\theta}_i \sqrt{d\log(6n^2/\delta)}$$

$$\leq \breve{\theta}_i(1+\kappa')d + \frac{2T^2\alpha n_{11}\sqrt{d\log(6n^2/\delta)}}{(1-\kappa')d/T - 2T^2 n_{11}\sqrt{d\log(6n^2/\delta)}}.$$

The second inequality is from Lemma A.65 and the last inequality is from the upper bound of $\breve{\theta}_i$ we just derived. Therefore, we have

$$\breve{\theta}_i \geq \frac{\alpha}{(1+\kappa')d}\left(1 - \frac{2T^2 n_{11}\sqrt{d\log(6n^2/\delta)}}{(1-\kappa')d/T - 2T^2 n_{11}\sqrt{d\log(6n^2/\delta)}}\right).$$

$\square$

From this Lemma we have $\breve{\theta}_i = \Theta(\alpha/d)$ for $i \in \mathcal{C}_1$. Similar as (53), under our assumption $\|\breve{\boldsymbol{v}}\| \leq 2\|\boldsymbol{v}_{sec}\|$, we have $\alpha = O(\log(n))$. Next we estimate the difference between $\|\breve{\boldsymbol{v}}\|^2$ and $\|\boldsymbol{v}_{sec}\|^2$. We can prove the following Lemma similar to Lemma A.22:

**Lemma A.42.** *Suppose that Assumption 4.1 holds, denote $\boldsymbol{v}_{sec}$ and $\breve{\boldsymbol{v}}$ as the optimal solutions under condition 12 and condition 14 respectively. We have*

$$\|\breve{\boldsymbol{v}}\|_2^2 - \|\boldsymbol{v}_{sec}\|_2^2 \geq \frac{C_3(1-\beta)}{d},$$

*where $C_3 = \Theta(1)$.*

*Proof of Lemma A.42.* Under this case, the difference between $\|\breve{\boldsymbol{v}}\|_2^2$ and $\|\boldsymbol{v}_{sec}\|_2^2$ becomes

$$\|\breve{\boldsymbol{v}}\|^2 - \|\boldsymbol{v}_{sec}\|^2 \geq \underbrace{\sum_{i\in[n]} \breve{\theta}_i^2 \|\overline{\boldsymbol{\xi}}_i\|^2 - \theta_i^2\|\boldsymbol{\xi}_i\|^2 - (\lambda_1^2 - \breve{\lambda}_1^2)\|\boldsymbol{\mu}_1\|^2 - (\lambda_2^2 - \breve{\lambda}_2^2)\|\boldsymbol{\mu}_2\|^2}_{I_1}$$

$$\underbrace{-\sum_{i\in\mathcal{N}}\sum_{j\in\mathcal{N}\backslash\{i\}} y_i y_j \theta_i \theta_j \langle \boldsymbol{\xi}_i, \boldsymbol{\xi}_j\rangle}_{I_2} + \underbrace{\sum_{i\in\mathcal{C}_1}\sum_{j\in\mathcal{C}_1\backslash\{i\}} y_i y_j \breve{\theta}_i \breve{\theta}_j \langle \overline{\boldsymbol{\xi}}_i, \overline{\boldsymbol{\xi}}_j\rangle}_{I_3}$$

We then bound $I_1 \sim I_3$ respectively. For $I_1$ we have

$$|I_1| \geq \sum_{i \in \mathcal{C}_1} \breve{\theta}_i^2 \|\bar{\boldsymbol{\xi}}_i\|^2 - \sum_{i \in \mathcal{N}} \theta_i^2 \|\boldsymbol{\xi}_i\|^2 - 2/\rho^2 \geq n_{11}\alpha(1 - O(n/\sqrt{d})) \min_{i \in \mathcal{C}_1} \breve{\theta}_i - n_2 \max_{i \in \mathcal{N}} \theta_i^2 (1 + \kappa)d - 2/\rho^2$$

$$\geq \frac{n_{11}\alpha^2}{(1 + \kappa')d}\left(1 - \frac{2T^2 n_{11}\sqrt{d\log(6n^2/\delta)}}{(1 - \kappa')d/T - 2T^2 n_{11}\sqrt{d\log(6n^2/\delta)}}\right) - \frac{n_2(1 + \kappa)d}{((1 - \kappa)d - 2n_2\sqrt{d\log(6n^2/\delta)})^2} - \frac{2}{\rho^2}$$

$$= \Omega\left(\frac{n}{d}\right).$$

The second inequality is from Lemma A.65; The third inequality is from Lemma A.38 and A.41; The last equality is due to the SNR condition $\rho/\sqrt{d} = \Omega(1/\sqrt{n})$ so that $\frac{1}{\rho^2} \leq \frac{n}{4d}$. For $I_2$, we have

$$|I_2| \leq \sum_{i \in \mathcal{N}} \max_{i \in \mathcal{N}} \theta_i^2 \cdot 2\sqrt{d\log(6n^2/\delta)} \leq \frac{2n_2\sqrt{d\log(6n^2/\delta)}}{((1 - \kappa)d - 2n_2\sqrt{d\log(6n^2/\delta)})^2} = \widetilde{O}\left(\frac{n}{d^{3/2}}\right).$$

The first inequality is from Lemma A.65; The second inequality is from Lemma A.38. Similarly, for $|I_3|$ we have

$$|I_3| \leq \sum_{i \in \mathcal{C}_1} \max_{i \in \mathcal{C}_1} \breve{\theta}_i^2 \cdot 2T^2 \sqrt{d\log(6n^2/\delta)} \leq \frac{2T^2 n_{11}\alpha^2 \sqrt{d\log(6n^2/\delta)}}{((1 - \kappa')d/T - 2T^2 n_{11}\sqrt{d\log(6n^2/\delta)})^2} = \widetilde{O}\left(\frac{n}{d^{3/2}}\right).$$

The second inequality is from Lemma A.41. Combining the above results, we have

$$\|\boldsymbol{v}''\|_2^2 - \|\boldsymbol{v}\|_2^2 \geq \Theta\left(\frac{n}{d}\right) - \widetilde{O}\left(\frac{n}{d^{3/2}}\right) \geq \frac{C_3 n(1 - \beta)}{d}.$$

$\square$

The remaining proof is the same as *Case 2.1* for two-token scenario and we omit it for convenience.

$\square$

In this way, we prove the relateve optimality of this second token selection rule. Then we introduce a lemma to estimate the norm of $\|\boldsymbol{p}_{sec}\|$. This will benefit our proof of the main theorem.

**Lemma A.43** (Norm of $\boldsymbol{p}_{sec}$)**.** *Suppose that Assumption 4.1 holds, recall that the solution of (p-SVM) is $\boldsymbol{p}_{sec}$. On a good run, we have*

$$\frac{1}{2\rho^2} + \frac{\eta n}{d} \leq \|\boldsymbol{p}_{sec}\|^2 \leq \frac{8}{\rho^2} + \frac{17\eta n}{d}.$$

*This implies*

$$\|\boldsymbol{p}_{sec}\| = \Theta\left(\sqrt{\frac{1}{\rho^2} + \frac{\eta n}{d}}\right).$$

*Proof of Lemma A.43.* First we prove the upper bound. Consider the following possible solution $\widetilde{\boldsymbol{p}}$:

$$\widetilde{\boldsymbol{p}} = \frac{2(\boldsymbol{\mu}_1 + \boldsymbol{\mu}_2)}{\rho^2} + \sum_{i \in \mathcal{N}} 4\frac{\boldsymbol{\xi}_{i,2}}{d}. \tag{72}$$

We then proved that $\widetilde{\boldsymbol{p}}$ satisfies (65). For $k \in \mathcal{C}, \tau \in [2, T]$ we have

$$\widetilde{\boldsymbol{p}}^\top (\boldsymbol{\mu}_k - \boldsymbol{\xi}_{k,\tau}) = 2 - \sum_{i \in \mathcal{N}} 4\frac{\langle \boldsymbol{\xi}_{i,2}, \boldsymbol{\xi}_{k,\tau} \rangle}{d} \geq 2 - \frac{4n_2\sqrt{d\log(6n^2/\delta)}}{d} \geq 1.$$

The first inequality is from the definition of $d$ in Lemma A.65 and the second inequality is from Assumption 4.1. And for $k \in \mathcal{N}, \tau \in [3, T]$, we have

$$\widetilde{\boldsymbol{p}}^\top (\boldsymbol{\xi}_{k,2} - \boldsymbol{\mu}_k) = -2 + \sum_{i \in \mathcal{N}} 4 \frac{\langle \boldsymbol{\xi}_{i,2}, \boldsymbol{\xi}_{k,2} \rangle}{d} \geq -2 + 4(1 - \kappa) + \sum_{i \in \mathcal{N}, i \neq k} 4 \frac{\langle \boldsymbol{\xi}_{i,2}, \boldsymbol{\xi}_{k,2} \rangle}{d}$$

$$\geq -2 + 4(1 - \kappa) - \frac{4 n_2 \sqrt{d \log(6n^2/\delta)}}{d} \geq 1.$$

$$\widetilde{\boldsymbol{p}}^\top (\boldsymbol{\xi}_{k,2} - \boldsymbol{\xi}_{k,\tau}) = \sum_{i \in \mathcal{N}} 4 \frac{\langle \boldsymbol{\xi}_{i,2}, \boldsymbol{\xi}_{k,2} \rangle}{d} - \sum_{i \in \mathcal{N}} 4 \frac{\langle \boldsymbol{\xi}_{i,2}, \boldsymbol{\xi}_{k,\tau} \rangle}{d} \geq 4(1 - \kappa) + \sum_{i \in \mathcal{N}, i \neq k} 4 \frac{\langle \boldsymbol{\xi}_{i,2}, \boldsymbol{\xi}_{k,2} \rangle}{d} - \sum_{i \in \mathcal{N}} 4 \frac{\langle \boldsymbol{\xi}_{i,2}, \boldsymbol{\xi}_{k,\tau} \rangle}{d}$$

$$\geq 4(1 - \kappa) - \frac{8 n_2 \sqrt{d \log(6n^2/\delta)}}{d} \geq 1.$$

The first and second inequalities are from Lemma A.65; The last inequality is from Assumption 4.1.

Therefore, the max-margin solution $\boldsymbol{p}_{sec}$ must have no greater norm than $\widetilde{\boldsymbol{p}}$. So we can upper bound $\boldsymbol{p}_{sec}$ as

$$\|\boldsymbol{p}_{sec}\|^2 \leq \|\widetilde{\boldsymbol{p}}\|^2 = \frac{8}{\rho^2} + \frac{16}{d^2} \Big( \sum_{i \in \mathcal{N}} \|\boldsymbol{\xi}_{i,2}\|^2 + \sum_{i,j \in \mathcal{N}, i \neq j} \langle \boldsymbol{\xi}_{i,2}, \boldsymbol{\xi}_{j,2} \rangle \Big)$$

$$\leq \frac{8}{\rho^2} + \frac{16}{d^2} \big( (1 + \kappa) n_2 d + 2 n_2^2 \sqrt{d \log(6n^2/\delta)} \big) \leq \frac{8}{\rho^2} + \frac{17 \eta n}{d}.$$

The second inequality is from Lemma A.65; The last inequality is from the definition of $d$ in Assumption 4.1.

Then we prove for the lower bound. As $\boldsymbol{p}_{sec}$ is the max-margin solution and satisfies KKT condition, it can be expressed as the sum of signal and noise tokens. Then we decompose $\boldsymbol{p}_{sec} = \boldsymbol{p}_{\boldsymbol{\mu}}^{sec} + \boldsymbol{p}_{\boldsymbol{\xi}}^{sec}$ where $\boldsymbol{p}_{\boldsymbol{\mu}}^{sec} = f_1^{sec} \boldsymbol{\mu}_1 + f_2^{sec} \boldsymbol{\mu}_2$ and $\boldsymbol{p}_{\boldsymbol{\xi}}^{sec} = \sum_{i \in [n]} g_i^{sec} \boldsymbol{\xi}_i$. Note that $\boldsymbol{\mu}_j \perp \boldsymbol{\xi}_{i,\tau}$ for all $j \in \{\pm 1\}, i \in [n], \tau \in [T]$. We first prove that $\langle \boldsymbol{p}_{sec}, \boldsymbol{\mu}_j \rangle \geq 0.5$ by contradiction.

If $\langle \boldsymbol{p}_{sec}, \boldsymbol{\mu}_j \rangle < 0.5$, then for every clean sample from cluster j we must have $\langle \boldsymbol{p}_{sec}, \boldsymbol{\xi}_{i,2} \rangle \leq -0.5$ and thus

$$\langle \boldsymbol{p}_{(r,R)}, \sum_{i \in \mathcal{C}_j} \boldsymbol{\xi}_{i,2} \rangle = \sum_{i \in \mathcal{C}_j} \langle \boldsymbol{p}_{(r,R)}, \boldsymbol{\xi}_{i,2} \rangle \leq -0.5 n_{1j}.$$

So we could estimate $\|\boldsymbol{p}_{sec}\|$ as follows

$$\|\boldsymbol{p}_{sec}\| \geq 0.5 n_{1j} \frac{1}{\|\sum_{i \in \mathcal{C}_j} \boldsymbol{\xi}_{i,2}\|} = 0.5 n_{1j} \frac{1}{\sqrt{\sum_{i \in \mathcal{C}_j} \|\boldsymbol{\xi}_{i,2}\|^2 + \sum_{i,j \in \mathcal{C}_j, i \neq j} \langle \boldsymbol{\xi}_{i,2}, \boldsymbol{\xi}_{j,2} \rangle}}$$

$$\geq 0.5 n_{1j} \frac{1}{\sqrt{2 \cdot n_{1j} \cdot (1 + \kappa) d}} = \frac{\sqrt{n_{1j}}}{\sqrt{8(1 + \kappa) d}}.$$

The first inequality is from the property of innerproduct; The second inequality is from Lemma A.65 and the definition of d in Assumption 4.1. Meanwhile, we have $\|\widetilde{\boldsymbol{p}}\| \leq \sqrt{8/\rho^2 + 17 \eta n / d}$ which also satisfies (65). Therefore, we further have

$$\|\boldsymbol{p}_{sec}\| \geq \frac{\sqrt{n_{1j}}}{\sqrt{8(1 + \kappa) d}} \geq \sqrt{8/\rho^2 + 17 \eta n / d} \geq \|\widetilde{\boldsymbol{p}}\|.$$

The second inequality is from Assumption 4.1. This leads to a contradiction. So we have $\langle \boldsymbol{p}_{sec}, \boldsymbol{\mu}_j \rangle \geq 0.5$. This directly indicates $f_j^{sec} \geq 0.5/\rho^2$, so we can lower bound $\|\boldsymbol{p}_{\boldsymbol{\mu}}^{sec}\|_2^2$ as

$$\|\boldsymbol{p}_{\boldsymbol{\mu}}^{sec}\|_2^2 = f_1^{sec2} \|\boldsymbol{\mu}_1\|^2 + f_2^{sec2} \|\boldsymbol{\mu}_2\|^2 \geq \frac{2 \cdot 0.5^2}{\rho^2} = \frac{1}{2\rho^2}.$$

As for $\|\boldsymbol{p}_{\boldsymbol{\xi}}^{sec}\|_2$, from p-SVM condition, for every noisy sample we have

$$\boldsymbol{p}_{sec}^\top (\boldsymbol{\xi}_{i,2} - \boldsymbol{\mu}_i) \geq 1,$$

which indicates

$$\boldsymbol{p}_{\boldsymbol{\xi}}^{sec\top}\boldsymbol{\xi}_{i,2} = \boldsymbol{p}_{sec}^{\top}\boldsymbol{\xi}_{i,2} \geq 1 + \boldsymbol{p}_{sec}^{\top}\boldsymbol{\mu}_i \geq 1.5.$$

The last inequality is from Lemma A.29. Sum up the inequality for all noisy sample, we have

$$\sum_{i\in\mathcal{N}} \boldsymbol{p}_{\boldsymbol{\xi}}^{sec\top}\boldsymbol{\xi}_{i,2} \geq 1.5n_2.$$

Thus,

$$\|\boldsymbol{p}_{\boldsymbol{\xi}}^{sec}\| \geq \frac{1.5n_2}{\|\sum_{i\in\mathcal{N}}\boldsymbol{\xi}_{i,2}\|} = \frac{1.5n_2}{\sqrt{\sum_{i\in\mathcal{N}}\|\boldsymbol{\xi}_{i,2}\|^2 + \sum_{i,j\in\mathcal{N},i\neq j}\langle\boldsymbol{\xi}_{i,2},\boldsymbol{\xi}_{j,2}\rangle}} \geq \frac{1.5n_2}{\sqrt{2\cdot n_2\cdot(1+\kappa)d}} \geq \sqrt{\frac{\eta n}{d}}.$$

The second inequality is from Lemma A.65 and the last inequality is from Assumption 4.1. Therefore,

$$\|\boldsymbol{p}_{sec}\|^2 = |\boldsymbol{p}_{\boldsymbol{\mu}}^{sec}\|_2^2 + \|\boldsymbol{p}_{\boldsymbol{\xi}}^{sec}\|_2^2 \geq \frac{1}{2\rho^2} + \frac{\eta n}{d}.$$

Combining the results above, we have

$$\|\boldsymbol{p}_{sec}\|^2 = \Theta\left(\frac{1}{\rho^2} + \frac{\eta n}{d}\right).$$

$\square$

With the above lemmas in place, we can conduct the analysis of the convergence direction of $\boldsymbol{p}_{(r,R)}$ and $\boldsymbol{v}_{(r,R)}$, which is similar to Theorem A.27.

**Theorem A.44.** *Suppose that Assumption 4.1 holds, on a good run, we have*

- $\min_{\tau\in[2,T]} \boldsymbol{p}_{(r,R)}^{\top}(\boldsymbol{\mu}_k - \boldsymbol{\xi}_{i,\tau}) \geq (1-\zeta)\Xi R$ *for* $\forall i \in \mathcal{C}_k, k \in [2]$, *where* $\Xi$ *is the margin induced by* $\boldsymbol{p}_{sec}$ *and* $\zeta = \frac{1}{R\Xi}\log(2T\sqrt{\rho^2+(1+\kappa)Td}\|\boldsymbol{v}_{sec}\|^3 n\rho^2/C)$.

- *the label margin for clean sample induced by* $\boldsymbol{v}_{(r,R)}/r$ *in SVM is at least* $(1-\gamma)\Gamma_{sec}$, *where* $\gamma = \frac{2\sqrt{\rho^2+(1+\kappa)Td}}{\Gamma_{sec}\exp((1-\zeta)R\Xi)}$.

*Proof of Theorem A.44.* From Proposition A.40, we have that for all $\boldsymbol{p}$ that $\exists i \in \mathcal{C}, \tau \in [2,T], \langle\boldsymbol{p}, \boldsymbol{\mu}_i - \boldsymbol{\xi}_{i,\tau}\rangle < \|\boldsymbol{p}\|_2/\|\boldsymbol{p}_{sec}\|_2$, the label margin $1/\|\boldsymbol{v}(\boldsymbol{p})\|$ is at most

$$\Gamma_{sec} - \frac{C}{\|\boldsymbol{v}_{sec}\|_2^3 n\rho^2} \cdot \max_{i\in\mathcal{C}}(1 - s_{i1}).$$

Recall that $\boldsymbol{s}_i = \mathbb{S}(\boldsymbol{X}_i\boldsymbol{p})$ is the softmax probability vector. We define $q_i^{\boldsymbol{p}} = 1 - s_{i1}$ for $i \in \mathcal{C}$ to measure the amount of non-optimality (attention on non-optimal token).

We use contradiction to prove the convergence of $\boldsymbol{p}_{(r,R)}$. Denote $\boldsymbol{p}_R^{sec} = R\boldsymbol{p}_{sec}/\|\boldsymbol{p}_{sec}\|$ which has the same norm as $\boldsymbol{p}_{(r,R)}$ and the direction of $\boldsymbol{p}_{sec}$. Suppose the margin induced by $\boldsymbol{p}_{(r,R)}/R$ is at most $(1-\zeta)\Xi$, i.e. $\min_{\tau\in[2,T]} \boldsymbol{p}_{(r,R)}^{\top}(\boldsymbol{\mu}_k - \boldsymbol{\xi}_{i,\tau}) \leq (1-\zeta)\Xi R, \forall i \in [n]$.

According to Lemma A.43, we have

$$\Xi = \|\boldsymbol{p}_{sec}\|_2^{-1} = \Theta((\eta n/d + 1/\rho^2)^{-1/2}).$$

Following the definition of $q_i^{\boldsymbol{p}}$ above, we set $\widehat{q}_{max} = \sup_{i\in[n]} q_i^{\boldsymbol{p}_{(r,R)}}$ and $q_{max}^* = \sup_{i\in[n]} q_i^{\boldsymbol{p}_R^{sec}}$ to be the worst non-optimality in $\boldsymbol{p}_{(r,R)}$ and $\boldsymbol{p}_R^{sec}$. Then we have

$$q_i^{\boldsymbol{p}_R^{sec}} = \frac{\sum_{t\neq 1}\exp(\boldsymbol{x}_i^{(t)\top}\boldsymbol{p}_R^{sec})}{\sum_{t\in[T]}\exp(\boldsymbol{x}_i^{(t)\top}\boldsymbol{p}_R^{sec})} \leq \frac{\sum_{t\neq 1}\exp(\boldsymbol{x}_i^{(t)\top}\boldsymbol{p}_R^{sec})}{\exp(\boldsymbol{x}_i^{(1)\top}\boldsymbol{p}_R^{sec})} \leq T\exp(-R\Xi).$$

The last inequality is from the definition of $\boldsymbol{p}_{sec}$ that $\boldsymbol{p}_{sec}^\top(\boldsymbol{x}_i^{(1)} - \boldsymbol{x}_i^{(t)}) \geq 1$ for $t \neq 1$, so $\boldsymbol{p}_R^{sec\top}(\boldsymbol{x}_i^{(1)} - \boldsymbol{x}_i^{(t)}) \geq R/\|\boldsymbol{p}_{sec}\| = R\Xi$. Thus, $q_{max}^* = \sup_{i\in[n]} q_i^{\boldsymbol{p}_{sec}} \leq \exp(-R\Xi)$. Then denote the output of attention layer $\boldsymbol{r}_i = \boldsymbol{X}_i^\top \mathbb{S}(\boldsymbol{X}_i \boldsymbol{p}_R^{sec})$. Define $\epsilon_i = \|\boldsymbol{r}_i - \boldsymbol{x}_{i1}\|$, we have $y_i \cdot \boldsymbol{r}_i^\top \boldsymbol{v}_{sec} \geq y_i \cdot \boldsymbol{x}_{i1}^\top \boldsymbol{v}_{sec} - \|\boldsymbol{r}_i - \boldsymbol{x}_{i1}\| \cdot \|\boldsymbol{v}_{sec}\| \geq 1 - \epsilon_i/\Gamma_{sec}$. So if we set $\epsilon_{max} = \sup_{i\in[n]} \epsilon_i$, $\boldsymbol{v}_{sec}$ achieves a label margin of at least $\Gamma_{sec} - \epsilon_{max}$ on $(y_i, \boldsymbol{r}_i)_{i\in[n]}$. To better estimate $\epsilon_{max}$, we define $M = \sup_{i\in[n],t\in[T]} \|\boldsymbol{\mu}_i - \sum_{t\ in[2,T]} \boldsymbol{\xi}_{i,t}\| \leq \sqrt{\rho^2 + (1+\kappa)Td}$, then we have

$$\epsilon_{max} = M \cdot q_{max}^* \leq MT\exp(-R\Xi). \tag{73}$$

This implies the max-margin achieved by $(\boldsymbol{p}_{(r,R)}^{sec}, \boldsymbol{v}_{(r,R)}^{sec})$ is at least

$$y_i f(\boldsymbol{p}_{(r,R)}^{sec}, \boldsymbol{v}_{(r,R)}^{sec}; \boldsymbol{x}_i) = y_i \boldsymbol{v}_r^{sec\top} \boldsymbol{r}_i \geq r\Gamma_{sec} - r\epsilon_{max} \geq r\Gamma_{sec} - rMT\exp(-R\Xi). \tag{74}$$

The first inequality is from $y_i \cdot \boldsymbol{r}_i^\top \boldsymbol{v}_r^{sec} \geq r(\Gamma_{sec} - \epsilon_i)$ and the last inequality is from (57).

Then we consider the case when $\min_{i\in\mathcal{C}_k, \tau\in[2,T]} \boldsymbol{p}_{(r,R)}^\top(\boldsymbol{\mu}_k - \boldsymbol{\xi}_{i,\tau}) \leq (1-\zeta)\Xi R$ the minimal margin constraint is $\zeta$-violated by $\boldsymbol{p}_{(r,R)}$. Without losing generality we assume that $1 = \operatorname{argmin}_{i\in\mathcal{C}_k}[\min_{\tau\in[T]} \boldsymbol{p}_{(r,R)}^\top(\boldsymbol{\mu}_i - \boldsymbol{\xi}_{i,\tau})]$. Then we have

$$1 - \widehat{q}_{max} = \frac{\exp(\boldsymbol{\mu}_1^\top \boldsymbol{p}_{(r,R)})}{\sum_{t\in[T]} \exp(\boldsymbol{x}_1^{(t)\top} \boldsymbol{p}_{(r,R)})} = \frac{1}{1 + \sum_{\tau\in[2,T]} \exp(\langle \boldsymbol{\xi}_{1,\tau} - \boldsymbol{\mu}_1, \boldsymbol{p}_{(r,R)}\rangle)}$$

$$\leq \frac{1}{1 + \exp(\max_{\tau\in[2,T]} \exp(\langle \boldsymbol{\xi}_{1,\tau} - \boldsymbol{\mu}_1, \boldsymbol{p}_{(r,R)}\rangle))} \leq \frac{1}{1 + \exp(-(1-\zeta)R\Xi)}.$$

This indicates $\widehat{q}_{max} \geq \frac{1}{1+\exp((1-\zeta)R\Xi)} \geq \frac{1}{2}\exp(-(1-\zeta)R\Xi)$. From Proposition A.40, optimizing $\nu$-SVM on $(y_i, \widehat{\boldsymbol{r}}_i)_{i\in[n]}$ can achieve the max-margin at most

$$\min_{i\in[n]} y_i f(\boldsymbol{v}_{(r,R)}, \boldsymbol{p}_{(r,R)}; \boldsymbol{x}_i) \leq \Gamma_{sec}r - \frac{Cr}{2\|\boldsymbol{v}_{sec}\|_2^3 n\rho^2} \cdot e^{-(1-\zeta)R\Xi}. \tag{75}$$

And from the definition $\zeta = \frac{1}{R\Xi}\log(2MT\|\boldsymbol{v}_{sec}\|^3 n\rho^2/C)$, we have

$$\frac{C}{2\|\boldsymbol{v}_{sec}\|_2^3 n\rho^2}\exp(-(1-\zeta)R\Xi) > MT\exp(-R\Xi),$$

for sufficiently large $R$, which implies

$$\min_{i\in[n]} y_i \cdot f(\boldsymbol{v}_{(r,R)}, \boldsymbol{p}_{(r,R)}; \boldsymbol{x}_i) < \min_{i\in[n]} y_i \cdot f(\boldsymbol{v}_r^{sec}, \boldsymbol{p}_R^{sec}; \boldsymbol{x}_i).$$

This contradicts with the problem definition (4) to maximize the margin.

Then we prove for the second statement. When $\min_{\tau\in[2,T]} \boldsymbol{p}_{(r,R)}^\top(\boldsymbol{\mu}_i - \boldsymbol{\xi}_{i,\tau}) \leq (1-\zeta)\Xi R, \forall i \in [n]$, we can use the proof above to derive a contradiction, so $(\boldsymbol{\mu}_i - \boldsymbol{\xi}_{i,\tau})^\top \boldsymbol{p}_{(r,R)} \geq (1-\zeta)R\Xi$ must hold for $\forall i \in \mathcal{C}, \tau \in [2,T]$. Then set $\widehat{\boldsymbol{r}}_i = \boldsymbol{X}_i^\top \mathbb{S}(\boldsymbol{X}_i \boldsymbol{p}_{(r,R)})$ for $i \in \mathcal{C}$, assume $\boldsymbol{v}_{(r,R)}$ achieves the label margin at most $(1-\gamma)\Gamma_{sec}r$ on clean samples, we have that

$$\min_{i\in\mathcal{C}} y_i \boldsymbol{v}_{(r,R)}^\top \widehat{\boldsymbol{r}}_i \leq \min_{i\in\mathcal{C}} y_i \boldsymbol{v}_{(r,R)}^\top \boldsymbol{\mu}_i + \sup_{i\in\mathcal{C}} |\boldsymbol{v}_{(r,R)}^\top(\widehat{\boldsymbol{r}}_i - \boldsymbol{\mu}_i)|$$

$$\leq (1-\gamma)\Gamma_{sec}r + M\exp(-(1-\zeta)R\Xi)r$$

$$\leq (1-\gamma/2)\Gamma_{sec}r.$$

The second inequality is from previous analysis that $(\boldsymbol{\mu}_i - \boldsymbol{\xi}_{i,\tau})^\top \boldsymbol{p}_{(r,R)} \geq (1-\zeta)R\Xi$, so $|\widehat{\boldsymbol{r}}_i - \boldsymbol{\mu}_i| \leq M\exp(-(1-\zeta)R\Xi)$; The last inequality is from our definition $\gamma = \frac{2M}{\Gamma_{sec}\exp((1-\zeta)R\Xi)}$.

Therefore, combining with (74), we have

$$\gamma\Gamma_{sec}r/2 > rM\exp(-R\Xi),$$

which implies

$$\min_{i\in[n]} y_i \cdot f(\boldsymbol{v}_{(r,R)}, \boldsymbol{p}_{(r,R)}; \boldsymbol{x}_i) < \min_{i\in[n]} y_i \cdot f(\boldsymbol{p}_R^{sec}, \boldsymbol{v}_r^{sec}; \boldsymbol{x}_i).$$

Again this contradicts with the problem definition (4). $\qquad\square$

Then we have the following lemma to bound the derivation $\zeta$ and $\gamma$:

**Lemma A.45.** *Under Condition 4.1, consider the same setting in Theorem A.44, we have $\zeta <$ $0.2, \gamma \leq 0.1$.*

*Proof of Lemma A.45.* From the definition of $\zeta$ in Theorem A.44, we have

$$\zeta = \frac{\log(2MT\|\boldsymbol{v}_{sec}\|^3 n\rho^2/C)}{R\Xi} = C_1 \frac{\sqrt{\eta n/d + 1/\rho^2}}{R} \log(MT\|\boldsymbol{v}_{sec}\|^3 n\rho^2)$$

$$\leq C_2 \frac{\sqrt{\eta n/d + 1/\rho^2}}{R} \log\left(\frac{n^2 T(\rho^2 + d)(\rho^2 \eta n + d)^3}{\rho^2 d^3}\right) < 0.2.$$

Here $C_1, C_2 = \Theta(1)$. The first inequality is from the upper bound of $\|\boldsymbol{v}_{sec}\|$ in Lemma A.39 and the last inequality is from the definition of $R$ in condition 4.1.

And for $\gamma$, we have

$$\gamma = \frac{2M}{\Gamma_{sec} \exp((1-\zeta)R\Xi)} = C_1' \frac{M\|\boldsymbol{v}_{sec}\|}{\exp(R/\|\boldsymbol{v}_{sec}\|)} \leq C_2' \frac{\sqrt{(\rho^2 + d)(\eta n/d + 1/\rho^2)}}{\exp(R/\sqrt{\eta n/d + 1/\rho^2})} < 0.1.$$

Here $C_1', C_2' = \Theta(1)$. The first inequality is from the lower and upper bound of $\|\boldsymbol{v}_{sec}\|$ in Lemma A.39 and the last inequality is from the definition of $R$ in condition 4.1. $\square$

Then we can estimate $\langle \boldsymbol{p}_{(r,R)}, \boldsymbol{\mu} \rangle$ with the following lemma:

**Lemma A.46.** *Suppose that Assumption 4.1 holds, on a good run, $\boldsymbol{p}_{(r,R)}$ should satisfy*

$$0.5(1-\zeta)R\Xi \leq \langle \boldsymbol{p}_{(r,R)}, \boldsymbol{\mu}_j \rangle \leq R\rho$$

*for $j \in \{1,2\}$.*

*Proof of Lemma A.46.* The upper bound is given by

$$\langle \boldsymbol{p}_{(r,R)}, \boldsymbol{\mu}_j \rangle \leq \|\boldsymbol{p}_{(r,R)}\| \|\boldsymbol{\mu}_j\| = R\rho.$$

Then we use contradiction to prove for the lower bound. From Theorem A.44, $\boldsymbol{p}_{(r,R)}$ satisfies

$$\min_{\tau \in [2,T]} \boldsymbol{p}_{(r,R)}^\top (\boldsymbol{\mu}_i - \boldsymbol{\xi}_{i,\tau}) \geq (1-\zeta)R\Xi, i \in \mathcal{C}, t \in [2,T] \tag{76}$$

If $\langle \boldsymbol{p}_{(r,R)}, \boldsymbol{\mu}_j \rangle \leq 0.5(1-\zeta)R\Xi$, denote $\tau_i = \operatorname*{argmin}_{\tau \in [2,T]} \boldsymbol{p}_{(r,R)}^\top (\boldsymbol{\mu}_i - \boldsymbol{\xi}_{i,\tau})$, then for every clean sample from cluster j we must have $\langle \boldsymbol{p}_{(r,R)}, \boldsymbol{\xi}_{i,\tau_i} \rangle \leq -0.5(1-\zeta)R\Xi$ and thus

$$\langle \boldsymbol{p}_{(r,R)}, \sum_{i \in \mathcal{C}_j} \boldsymbol{\xi}_i \rangle = \sum_{i \in \mathcal{C}_j} \langle \boldsymbol{p}_{(r,R)}, \boldsymbol{\xi}_{i,\tau_i} \rangle \leq -0.5(1-\zeta)R\Xi n_{1j}.$$

So we could estimate $\|\boldsymbol{p}_{(r,R)}\|$ as follows

$$\|\boldsymbol{p}_{(r,R)}\| \geq 0.5(1-\zeta)R\Xi \cdot n_{1j} \frac{1}{\|\sum_{i \in \mathcal{C}_j} \boldsymbol{\xi}_{i,\tau_i}\|} = 0.5(1-\zeta)R\Xi \cdot n_{1j} \frac{1}{\sqrt{\sum_{i \in \mathcal{C}_j} \|\boldsymbol{\xi}_{i,\tau_i}\|^2 + \sum_{i,j \in \mathcal{C}_j} \langle \boldsymbol{\xi}_{i,\tau_i}, \boldsymbol{\xi}_{j,\tau_j} \rangle}}$$

$$\geq 0.5(1-\zeta)R\Xi \cdot n_{1j} \frac{1}{\sqrt{2 \cdot n_{1j} \cdot (1+\kappa')d}} \geq 0.4R\Xi \cdot \frac{\sqrt{n_{1j}}}{\sqrt{2(1+\kappa')d}}.$$

The first inequality is from the property of innerproduct; The second inequality is from Lemma A.34 and the definition of d in Assumption 4.1; The last inequality is from Lemma A.45. Meanwhile, from Lemma A.24 we have $\|\boldsymbol{p}_{sec}\| \leq \sqrt{8/\rho^2 + 17\eta n/d}$. Recall that $\Xi = \|\boldsymbol{p}_{sec}\|^{-1}$. Therefore, we further have

$$\|\boldsymbol{p}_{(r,R)}\| \geq 0.4R\Xi \cdot \frac{\sqrt{n_{1j}}}{\sqrt{2(1+\kappa')d}} \geq \sqrt{\frac{0.4^2 n_{1j}}{(8/\rho^2 + 17\eta n/d) \cdot 2(1+\kappa')d}} \cdot R$$

$$\geq \sqrt{\frac{0.04(n - \eta n - O(\sqrt{n}))}{(8/\rho^2 + 17\eta n/d) \cdot (1+\kappa')d}} \cdot R > R.$$

The second inequality is from Lemma A.43; The third inequality is from Lemma A.68 and the last inequality is from Assumption 4.1 about SNR and $\eta$. This leads to a contradiction. $\square$

Now we can estimate the output of attention layer for some test sample $(\boldsymbol{X}, y)$.

**Lemma A.47.** *Under Condition 4.1, for the test sample $(\boldsymbol{X}, y) \sim \mathcal{D}_{clean}$, $\boldsymbol{X} = (\boldsymbol{\mu}', \boldsymbol{\xi}')$, with probability at least $1 - \exp\left(-\frac{1}{2}(\frac{1}{2}(1-\zeta)\Xi - K/R)^2\right)$ we have*

$$\langle \boldsymbol{p}_{(r,R)}, \boldsymbol{\mu}' \rangle - \langle \boldsymbol{p}_{(r,R)}, \boldsymbol{\xi}' \rangle \geq K,$$

*where $K \leq \frac{1}{2}(1-\zeta)R\Xi$ and $\zeta, \Xi$ are defined in Theorem A.44.*

*Proof of Lemma A.47.* Note that $\boldsymbol{p}^\top \boldsymbol{\xi}'$ follows Gaussian distribution $\mathcal{N}(0, R^2)$, we have

$$\mathbb{P}(\langle \boldsymbol{p}_{(r,R)}, \boldsymbol{\mu}' \rangle - \langle \boldsymbol{p}_{(r,R)}, \boldsymbol{\xi}' \rangle < K) = \mathbb{P}(\langle \boldsymbol{p}_{(r,R)}, \boldsymbol{\xi}' \rangle > \langle \boldsymbol{p}_{(r,R)}, \boldsymbol{\mu}' \rangle - K) \leq \mathbb{P}(\boldsymbol{p}_{(r,R)}^\top \boldsymbol{\xi}' > \frac{1}{2}(1-\zeta)R\Xi - K)$$

$$\leq \exp\left(-\frac{1}{2}(\frac{1}{2}(1-\zeta)\Xi - K/R)^2\right).$$

The first inequality is from Lemma A.46 and the second inequality comes from the property of Gaussian tail probability. $\square$

Then we can follow the proof of Lemma A.31 to prove that $\boldsymbol{v}_{(r,R)}$ can be expressed as the sum of signal and noise tokens.

**Lemma A.48.** *The solution of constrained optimization problem (4) $\boldsymbol{v}_{(r,R)}$ can be expressed in the form that*

$$\boldsymbol{v}_{(r,R)} = \lambda_1 \boldsymbol{\mu}_1 + \lambda_2 \boldsymbol{\mu}_2 + \sum_{i=1}^{n} \sum_{\tau \in [2,T]} \theta_{i,\tau} \boldsymbol{\xi}_{i,\tau}.$$

Based on this representation, we can then bound the parameters in $\boldsymbol{v}_{(r,R)}$:

**Lemma A.49.** *Under Condition 4.1, denote $\boldsymbol{v}_{(r,R)} = \lambda_1 \boldsymbol{\mu}_1 + \lambda_2 \boldsymbol{\mu}_2 + \sum_{i=1}^{n} \sum_{\tau \in [2,T]} \theta_{i,\tau} \boldsymbol{\xi}_{i,\tau}$. Then we have*

$$\lambda_1 \geq (1-\gamma)\Gamma r/\rho^2,$$
$$\lambda_2 \leq -(1-\gamma)\Gamma r/\rho^2,$$
$$|\theta_{i,\tau}| \leq 2\sqrt{1/\rho^2 + 5\eta n/d} \cdot \Gamma_{sec} r/\sqrt{d}.$$

*Proof of Lemma A.49.* The first two statements are obvious because from Theorem A.44 we have

$$y_i \boldsymbol{v}_{(r,R)}^\top \boldsymbol{\mu}_i \geq (1-\gamma)\Gamma_{sec} r,$$

for $\forall i \in \mathcal{C}$. This implies $|\lambda_j| \geq (1-\gamma)\Gamma_{sec} r/\rho^2$ for $j \in \{1,2\}$. Meanwhile, we decompose $\boldsymbol{v}_{(r,R)} = \boldsymbol{v}_{\boldsymbol{\mu}} + \boldsymbol{v}_{\boldsymbol{\xi}}$ where $\boldsymbol{v}_{\boldsymbol{\mu}} = \lambda_1 \boldsymbol{\mu}_1 + \lambda_2 \boldsymbol{\mu}_2$ and $\boldsymbol{v}_{\boldsymbol{\xi}} = \sum_{i \in [n]} \theta_i \boldsymbol{\xi}_i$. And we can upper bound $\|\boldsymbol{v}_{\boldsymbol{\xi}}\|$ as

$$\|\boldsymbol{v}_{\boldsymbol{\xi}}\|^2 = \|\boldsymbol{v}_{(r,R)}\|^2 - \|\boldsymbol{v}_{\boldsymbol{\mu}}\|^2 \leq r^2 - \lambda_1^2 \rho^2 - \lambda_2^2 \rho^2 \leq r^2(1 - 2(1-\gamma)^2 \Gamma_{sec}^2/\rho^2).$$

The first inequality is from $\|\boldsymbol{v}\| \leq r$ and the second inequality is from the first two statements we just proved. Therefore, denote $j, \tau_j = \operatorname*{argmax}_{i \in [n], \tau \in [2,T]} \theta_{i,\tau}$, we have

$$\theta_{j,\tau_j}^2 \|\boldsymbol{\xi}_{j,\tau_j}\|^2 \leq \|\boldsymbol{v}_{\boldsymbol{\xi}}\|^2 \leq r^2(1 - 2(1-\gamma)^2 \Gamma_{sec}^2/\rho^2).$$

Then we can upper bound $|\theta_{j,\tau_j}|$ as

$$\theta_{j,\tau_j}^2 \leq r^2(1 - 2(1-\gamma)^2 \Gamma_{sec}^2/\rho^2)/\|\boldsymbol{\xi}_{j,\tau_j}\|^2 \leq r^2(1 - 2(1-\gamma)^2 \Gamma_{sec}^2/\rho^2)/(1-\kappa)d$$

$$= r^2\left(1 - \frac{2(1-\gamma)^2}{\|\boldsymbol{v}_{sec}\|^2 \rho^2}\right)/(1-\kappa)d \leq r^2\left(1 - \frac{1}{(2/\rho^2 + 5\eta n/d)\rho^2}\right)/(1-\kappa)d$$

$$= \frac{1 + 5\eta n\rho^2/d}{2 + 5\eta n\rho^2/d} \cdot \frac{r^2}{(1-\kappa)d} \leq \left(\frac{1}{\rho^2} + \frac{5\eta n}{d}\right) \cdot \frac{\Gamma_{sec}^2 r^2}{2d}.$$

The second inequality is from Lemma A.65; The third inequality is from Lemma A.39 that $\|\boldsymbol{v}_{sec}\| \leq \sqrt{2/\rho^2 + 5\eta n/d}$ and our definition of $\gamma = \frac{2\sqrt{\rho^2 + (1+\kappa)Td}}{\Gamma_{sec}\exp((1-\zeta)R\Xi)}$; The last inequality is from $\Gamma_{sec} = \|\boldsymbol{v}_{sec}\|^{-1} \geq (2/\rho^2 + 5\eta n/d)^{-1}$. Thus, we can bound $|\theta_{j,\tau_j}|$ as

$$|\theta_{j,\tau_j}| \leq 2\sqrt{1/\rho^2 + 5\eta n/d} \cdot \Gamma_{sec} r/\sqrt{d}.$$

$\square$

Therefore, we can prove the main theorem.

*Proof of Theorem 4.2.* First we show that the model can perfectly classify all training samples. From the construction of $\boldsymbol{p}_{sec}, \boldsymbol{v}_{sec}$, we have

$$y_i \boldsymbol{v}_{(r,R)}^\top \boldsymbol{r}_i \geq y_i \langle \boldsymbol{v}_r^{sec}, \boldsymbol{r}_i^{sec} \rangle > 0$$

for $\forall i \in [n]$. Here $\boldsymbol{r}^{sec}$ is the token selected by $\boldsymbol{p}_R^{sec}$. Thus $y_i = \mathrm{sign}(f(\boldsymbol{X}_i; \boldsymbol{v}_{(r,R)}, \boldsymbol{p}_{(r,R)}))$ for all $i \in [n]$. Then we bound the test error. Given a test sample $\boldsymbol{X}, y$, where $\boldsymbol{X} = (\boldsymbol{\mu}^\star, \boldsymbol{\xi}^\star)$, $\boldsymbol{\mu}^\star$ can be $\boldsymbol{\mu}_1$ or $\boldsymbol{\mu}_2$. With probability at least $1 - \delta$ a good run will occur. Similar to the proof of Lemma A.65, with probability at least $1 - \exp(-\Omega(d/n^2))$,

$$|\langle \boldsymbol{\xi}^\star, \boldsymbol{\xi}_{i,\tau} \rangle| \leq 2\sqrt{d\log(6n^2/\delta)}. \tag{77}$$

According to Lemma A.47, with probability at least $1 - \exp\left(-\frac{1}{2}(\frac{1}{2}(1-\zeta)\Xi - K/R)^2\right)$, we have

$$y \cdot f(\boldsymbol{v}_{(r,R)}, \boldsymbol{p}_{(r,R)}; \boldsymbol{X}) \geq \frac{\langle y\boldsymbol{v}_{(r,R)}, e^K\boldsymbol{\mu}^\star + \boldsymbol{\xi}^\star \rangle}{e^K + 1} \geq \frac{e^K(1-\gamma)\Gamma_{sec}r\|\boldsymbol{\mu}^\star\|^2}{\rho^2(e^K + 1)} - \frac{1}{e^K + 1}\sum_{i \in [n]}\sum_{\tau \in [2,T]} |\theta_{i,\tau}| \cdot |\langle \boldsymbol{\xi}_{i,\tau}, \boldsymbol{\xi}^\star \rangle|. \tag{78}$$

Let $K = \log(nT\sqrt{1/\rho^2 + \eta n/d}) + \log(\log(6n^2/\delta)) + C < \frac{1}{2}(1-\zeta)R\Xi$. By uniform bound, we have that with probability at least $1 - \exp(-\Omega(d/n^2)) - \exp\left(-\frac{1}{2}(\frac{1}{2}(1-\zeta)\Xi - K/R)^2\right)$,

$$y \cdot f(\boldsymbol{v}_{(r,R)}, \boldsymbol{p}_{(r,R)}; \boldsymbol{X}) \geq \frac{e^K(1-\gamma)\Gamma_{sec}r - nT\sqrt{d\log(6n^2/\delta)} \cdot 2\sqrt{1/\rho^2 + \eta n/d} \cdot \Gamma_{sec}r/\sqrt{d}}{1 + e^K}$$

$$\geq \frac{0.9e^K\Gamma_{sec}r - nT\sqrt{d\log(n^2/\delta)} \cdot 2\sqrt{1/\rho^2 + \eta n/d} \cdot \Gamma_{sec}r/\sqrt{d}}{1 + e^K}$$

$$> 0,$$

where the first inequality uses (77), (78) and Lemma A.49; The second inequality is from Lemma A.45 and the last inequality is from Assumption 4.1 and our selection of $K$. Therefore,

$$\mathbb{P}(y \neq f(\boldsymbol{v}_{(r,R)}, \boldsymbol{p}_{(r,R)}; \boldsymbol{X})) \leq \exp\left(-\frac{1}{2}(\frac{1}{2}(1-\zeta)\Xi - \frac{K}{R})^2\right) + \exp(-\Omega(d/n^2)),$$

where $\zeta = \frac{\log(2MT\|\boldsymbol{v}_{sec}\|^3 n\rho^2/C)}{R\Xi} = \Theta\left(\frac{\sqrt{\eta n/d + 1/\rho^2}}{R}\log\left(\frac{n^2T(\eta n + d/\rho^2)^3(\rho^2 + d)}{\rho^2 d^3}\right)\right)$, $K = \log(nT\sqrt{1/\rho^2 + \eta n/d}) + \log(\log(6n^2/\delta)) + C = \Theta(\log(nT\sqrt{1/\rho^2 + \eta n/d}\log(6n^2/\delta))$ and $\Xi = \|\boldsymbol{p}_{sec}\|_2^{-1} = \Theta((\eta n/d + 1/\rho^2)^{-1/2})$. Plugging in the order of $\Xi$ and $K$, we have

$$\mathbb{P}_{(\boldsymbol{X},y)\sim\mathcal{D}_{clean}}(y \neq \mathrm{sign}(f(\boldsymbol{X}; \boldsymbol{v}_{(r,R)}, \boldsymbol{p}_{(r,R)}))) \leq \exp(-\Omega(d/n^2)) + \exp\left(-\Omega\left(\frac{(1-\zeta)}{\sqrt{\eta n/d + 1/\rho^2}} - \frac{\log(n)}{R}\right)^2\right),$$

where $\zeta = \Theta\left(\frac{\sqrt{\eta n/d + 1/\rho^2}}{R}\log\left(\frac{n^2T(\eta n + d/\rho^2)^3(\rho^2 + d)}{\rho^2 d^3}\right)\right)$. This completes the proof. $\square$

### A.2.5 Proof of Thm. 4.4

**Lemma A.50.** *Consider the next joint-constrained max margin solution:*

$$(\boldsymbol{v}_t, \boldsymbol{p}_t) = \operatorname*{argmax}_{\|\boldsymbol{v}\|^2 + \|\boldsymbol{p}\|^2 \leq t} \min_i y_i f(\boldsymbol{X}_i; \boldsymbol{v}, \boldsymbol{p}). \tag{79}$$

*Let $r_t := \|\boldsymbol{v}_t\|$ and $R_t := \|\boldsymbol{v}_t\|$, then $(\boldsymbol{v}_t, \boldsymbol{p}_t) = (\boldsymbol{v}_{(r_t, R_t)}, \boldsymbol{p}_{(r_t, R_t)})$, where $(\boldsymbol{v}_{(r_t, R_t)}, \boldsymbol{p}_{(r_t, R_t)})$ is a solution to Problem 4. Moreover, under Assumption 4.1 (items 1-3), with probability at least $1 - \delta$ over the random data generation, we have that $r_t \to \infty, R_t \to \infty$ as $t \to \infty$.*

*Proof.* By Proposition A.40, with probability at least $1 - \delta$, for all $\boldsymbol{p} \in \mathbb{R}^d$, the token selection under $\boldsymbol{p}$ results in a label margin of at most $\Gamma - c \cdot \max_{i \in \mathcal{C}}(1 - s_{i1}^{\boldsymbol{p}})$ in A.13 (with $\boldsymbol{r}_i = \boldsymbol{X}_i^\top \boldsymbol{S}(\boldsymbol{X}_i \boldsymbol{p})$), where $\boldsymbol{s}_i^{\boldsymbol{p}} = \mathbb{S}(\boldsymbol{X}_i \boldsymbol{p})$ is the softmax probabilities, and $c := C/|\boldsymbol{v}_{sec}|^3 n \rho^2$ is some constant (which may depends on $n$ and $d$, but not in $t$).

Observe that as the norm of $\boldsymbol{v}$ increases, the margin increases; thus, it's easy to verify that $\|\boldsymbol{v}_t\| \to \infty$ as $t \to \infty$. We argue that also $\|\boldsymbol{p}_t\| \to \infty$ as $t \to \infty$. To see that, assume by contradiction that $\|\boldsymbol{p}_t\| \le R_0$ for some arbitrary large $t$ that will be determined later. Set $\Gamma = 1/\|\boldsymbol{v}_{sec}\|$, $\|\boldsymbol{v}_t\| = r_t$, $\widetilde{\boldsymbol{v}}_{sec} = (r_t - 1)\Gamma \boldsymbol{v}_{sec}$. Hence $t = r_t^2 + R_0^2$ and $\|\widetilde{\boldsymbol{v}}_{sec}\|^2 = (r - 1)^2$. The idea is that by decreasing $\|\boldsymbol{v}_t\|$ by 1, we can choose $\boldsymbol{p}$ with $\|\boldsymbol{p}\|^2 + (r_t - 1)^2 = t = r_t^2 + R_0^2$, i.e., $\|\boldsymbol{p}\|^2 = 2r_t - 1 + R_0^2$, which can be arbitrary large for large enough $t$. Set $\Pi := 1/\|\boldsymbol{p}_{sec}\|$ and $\widetilde{\boldsymbol{p}}_{sec} := \sqrt{2r_t - 1 + R_0^2}\Pi \boldsymbol{p}_{sec}$. The proof strategy is obtaining a contradiction by proving that $(\widetilde{\boldsymbol{v}}_{sec}, \widetilde{\boldsymbol{p}}_{sec})$ is a strictly better solution compared to $(\boldsymbol{v}_t, \boldsymbol{p}_t)$. Define $q_i^{\boldsymbol{p}} = 1 - s_{i1}$ for $i \in \mathcal{C}$ to be the amount of non-optimality softmax probability. Then we have that

$$\max_i q_i^{\boldsymbol{p}_t} \ge \kappa$$

where $\kappa > 0$ is a constant that depends just on $R_0$ and data parameters (e.g. $n, d, \rho, \delta$). On the other hand, for every $\epsilon > 0$, we have that

$$q^* = \max_i q_i^{\widetilde{\boldsymbol{p}}_{sec}} \le \epsilon,$$

for large enough $r_t$ i.e. large enough $t$. Therefore, By Proposition A.14 (see the first paragraph in the proof), we can upper bound the margin induced by $\boldsymbol{v}_t$ on $(Y_i, \boldsymbol{r}_i)$ for $\boldsymbol{r}_i = \boldsymbol{X}_i^\top \mathbb{S}(\boldsymbol{X}_i \boldsymbol{p}_t)$ by

$$\min_{i \in [n]} y_i \boldsymbol{v}_t^\top \boldsymbol{r}_i \le r_t(\Gamma - c\kappa),$$

for some constant $c > 0$. On the other hand, the margin induced by $\widetilde{\boldsymbol{v}}_{sec}$ on $(Y_i, \boldsymbol{r}_i)$ for $\boldsymbol{r}_i = \boldsymbol{x}_{i\alpha_i}$ is $(r_t - 1)\Gamma$. This means that we margin induced by $\widetilde{\boldsymbol{v}}_{sec}$ on $(y_i, \boldsymbol{r}_i)$ for $\boldsymbol{r}_i = \boldsymbol{X}_i^\top \mathbb{S}(\boldsymbol{X}_i \widetilde{\boldsymbol{p}}_{sec})$ is at least

$$\min_i y_i r_i^\top \widetilde{\boldsymbol{v}}_{sec} \ge \min_i y_i x_{i\alpha_i}^\top \widetilde{\boldsymbol{v}}_{sec} - q^* \left\| \boldsymbol{x}_i^{(1)} - \boldsymbol{x}_i^{(2)} \right\| \|\widetilde{\boldsymbol{v}}_{sec}\|$$
$$\ge (\boldsymbol{r}_t - 1)(\Gamma - M\epsilon),$$

where $M = \sup_{i \in n} \left\| \boldsymbol{x}_i^{(1)} - \boldsymbol{x}_i^{(2)} \right\|$. Observe that this lower bound is bigger than the previous upper bound when

$$(r_t - 1)(\Gamma - M\epsilon) > r_t(\Gamma - c\kappa)$$
$$M\epsilon < -(\Gamma - M\epsilon)/r_t + c\kappa.$$

Choose large enough $t$ such that $(\Gamma - M\epsilon)/r_t < c\kappa/2$ and $M\epsilon < c\kappa/2$, gives us the desired contradiction. Recall that $R_t := \|p_t\|$ and $r_t := \|v_t\|$. Since $r_t^2 + R_t^2 \le t$, we have that $(\boldsymbol{v}_t, \boldsymbol{p}_t$ is a solution to Problem 4 with $r = r_t$, $R = R_t$, and $(\boldsymbol{v}_{(r_t, R_t)}, \boldsymbol{p}_{(r_t, R_t)})$ is a solution to Problem 79.

$\square$

*Proof of Thm. 4.4.* By Thm. 4.2, with probability at least $1 - \delta$, the training set is feasible, i.e. exists $(\boldsymbol{v}, \boldsymbol{p})$ such that $\min_{i \in [n]} y_i f(\boldsymbol{X}_i; \boldsymbol{v}, \boldsymbol{p}) > 0$. Therefore, for any $\gamma > 0$, with probability at least $1 - \delta$, we have that $\min_{i \in [n]} y_i f(\boldsymbol{X}_i; \boldsymbol{v}_\gamma, \boldsymbol{p}_\gamma) \ge \gamma > 0$, which proves the first part of the Thm. Next, we show that the classifier $\text{sign}(f(\boldsymbol{X}; \boldsymbol{v}_\gamma, \boldsymbol{p}_\gamma))$ generalizes well, for large enough $\gamma$. Recall the next joint-constrained max margin solution:

$$(v_t, p_t) = \underset{\|\boldsymbol{v}\|^2 + \|\boldsymbol{p}\|^2 \le t}{\text{argmax}} \min_i y_i f(\boldsymbol{X}_i; \boldsymbol{v}, \boldsymbol{p}), \tag{80}$$

which was introduced in Lemma A.50. Fix $\gamma > 0$, and let $(\boldsymbol{v}_\gamma, \boldsymbol{p}_\gamma)$ be the solution of Problem 5. Define $t(\gamma) := \|\boldsymbol{v}_\gamma\|^2 + \|\boldsymbol{p}_\gamma\|^2$. We argue that $(\boldsymbol{v}_\gamma, \boldsymbol{p}_\gamma)$ is a solution to Problem 80 for $t = t(\gamma)$. Indeed, let

$$m := \max_{\|\boldsymbol{v}\|^2 + \|\boldsymbol{p}\|^2 \le t(\gamma)} \min_{i \in [n]} y_i f(\boldsymbol{X}_i; \boldsymbol{v}, \boldsymbol{p})$$

be the maximum margin for Problem 80 with $t = t(\gamma)$. Assume by contradiction that

$$\min_{i \in [n]} y_i f(\boldsymbol{X}_i; \boldsymbol{p}_\gamma, \boldsymbol{v}_\gamma) < m,$$

which implies that

$$\gamma \leq \min_{i \in [n]} y_i f(\boldsymbol{X}_i; \boldsymbol{p}_\gamma, \boldsymbol{v}_\gamma) < m.$$

Let $(\boldsymbol{v}^*, \boldsymbol{p}^*)$ be a solution to Problem 80 with $t = t(\gamma)$ i.e. $\|\boldsymbol{v}^*\|^2 + \|\boldsymbol{p}^*\|^2 = t(\gamma)$ and $\min_{i \in [n]} y_i f(\boldsymbol{X}_i; \boldsymbol{p}^*, \boldsymbol{v}^*) = m > \gamma$. Write $\boldsymbol{v}' := (\gamma/m) \cdot \boldsymbol{v}^*$. We remind that $f(\boldsymbol{X}; \boldsymbol{v}, \boldsymbol{p}) = \boldsymbol{v}^\top \boldsymbol{X}^\top \mathbb{S}(\boldsymbol{X}\boldsymbol{p})$ and overall we get that

- $\|\boldsymbol{v}'\|^2 + \|\boldsymbol{p}^*\|^2 = (\gamma/m)^2 \|\boldsymbol{v}^*\|^2 + \|\boldsymbol{p}^*\|^2 < \|\boldsymbol{v}^*\|^2 + \|\boldsymbol{p}^*\|^2 = t(\gamma)$

- $\min_{i \in [n]} y_i f(\boldsymbol{X}_i; \boldsymbol{p}^*, \boldsymbol{v}') = \frac{\gamma}{m} \min_{i \in [n]} y_i f(\boldsymbol{X}_i; \boldsymbol{p}^*, \boldsymbol{v}^*) = \frac{\gamma}{m} \cdot m = \gamma$,

which contradicts the optimality of $(\boldsymbol{v}_\gamma, \boldsymbol{p}_\gamma)$ to Problem 5. We conclude that $(\boldsymbol{v}_\gamma, \boldsymbol{p}_\gamma)$ is a solution to Problem 80 for $t = t(\gamma)$, i.e. $(\boldsymbol{v}_\gamma, \boldsymbol{p}_\gamma) = (\boldsymbol{v}_{t(\gamma)}, \boldsymbol{p}_{t(\gamma)})$, where $(\boldsymbol{v}_{t(\gamma)}, \boldsymbol{p}_{t(\gamma)})$ is a solution for Problem 80 with $t = t(\gamma)$. Let $r_{t(\gamma)} := \|\boldsymbol{v}_{t(\gamma)}\|$ and $R_{t(\gamma)} := \|\boldsymbol{p}_{t(\gamma)}\|$. By Lemma A.50 we have

$$(\boldsymbol{v}_\gamma, \boldsymbol{p}_\gamma) = (\boldsymbol{v}_{t(\gamma)}, \boldsymbol{p}_{t(\gamma)}) = \left( \boldsymbol{v}_{(r_{t(\gamma)}, R_{t(\gamma)})}, \boldsymbol{p}_{(r_{t(\gamma)}, R_{t(\gamma)})} \right), \tag{81}$$

and that $r_{t(\gamma)} \to \infty, R_{t(\gamma)} \to \infty$ as $t(\gamma) \to \infty$. Clearly $t(\gamma) \to \infty$ as $\gamma \to \infty$. By Thm. 4.2, The classifier $\text{sign}(f(\boldsymbol{X}; \boldsymbol{v}_{(r,R)}, \boldsymbol{p}_{(r,R)}))$ generalizes well on test data:

$$\mathbb{P}_{(\boldsymbol{X},y) \sim \mathcal{D}}(y \neq \text{sign}(f(\boldsymbol{X}; \boldsymbol{v}_{(r,R)}, \boldsymbol{p}_{(r,R)})))$$
$$= \eta + \exp(-\Omega(d/n^2)) + \exp\left( -\Theta\left( \frac{(1-\zeta)}{\sqrt{\frac{\eta n}{d} + \frac{1}{\rho^2}}} - \frac{\log(d)}{R} \right)^2 \right)$$

In particular, there exists $r_0, R_0$ such that for any $r \geq r_0, R \geq R_0$, the above probability can be upper bound by $\eta + \exp(-\Omega(d/n^2)) + \exp(-\Theta((1/\rho^2 + \eta n/d)^{-1}))$ (see Remark 4.3). Choose large enough $\gamma_0$ such that for any $\gamma \geq \gamma_0$ we have that $r_{t(\gamma)} \geq r_0$ and $R_{t(\gamma)} \geq R_0$. Then we conclude

$$\mathbb{P}_{(\boldsymbol{X},y) \sim \mathcal{D}} (y \neq \text{sign}(f(\boldsymbol{X}; \boldsymbol{v}_\gamma, \boldsymbol{p}_\gamma)))$$
$$= \mathbb{P}_{(\boldsymbol{X},y) \sim \mathcal{D}} \left( y \neq \text{sign} \left( f(\boldsymbol{X}; \boldsymbol{v}_{(r_{t(\gamma)}, R_{t(\gamma)})}, \boldsymbol{p}_{(r_{t(\gamma)}, R_{t(\gamma)})}) \right) \right)$$
$$\leq \eta + \exp(-\Omega(d/n^2)) + \exp(-\Theta((1/\rho^2 + \eta n/d)^{-1})),$$

where the first equality is from Eq. 81, as required. □

### A.2.6 Proof of Thm. 4.6

**Proof Sketch**
First we prove that in this case, only by selecting the noise token for every sample can we achieve the largest margin in the downstream task,

$$\boldsymbol{r}_i^* = \boldsymbol{\xi}_i, \forall i \in [n] \tag{82}$$

Similarly, we define the respective max-margin solution for $\boldsymbol{p}$ and $\boldsymbol{v}$ in this case.

**Definition A.51** (p-SVM, negative case). $\boldsymbol{p}$ should satisfy

$$\boldsymbol{p}_{mm}(\alpha) = \underset{p}{\arg\min} \|\boldsymbol{p}\|$$

subjected to

$$\boldsymbol{p}^\top (\boldsymbol{\xi}_i - \boldsymbol{\mu}_i) \geq 1, \tag{83}$$

for all $1 \leq i \leq n$. $\Xi = 1/\|\boldsymbol{p}_{mm}\|$ is the margin induced by $\boldsymbol{p}_{mm}$.

**Definition A.52** (v-SVM, negative case).
$$\boldsymbol{v}(\boldsymbol{p}) = \operatorname*{argmin}_{\boldsymbol{v} \in \mathbb{R}^d} \|\boldsymbol{v}\| \text{ s.t. } y_i \cdot \boldsymbol{v}^\top \boldsymbol{r}_i \geq 1, \quad \text{ for all } i \in [n]. \tag{84}$$

$\Gamma(\boldsymbol{p}) = 1/\|\boldsymbol{v}(\boldsymbol{p})\|$ is the **label margin** induced by $\boldsymbol{v}$ and $\boldsymbol{p}$. When $\boldsymbol{r}_i = \boldsymbol{\xi}_i, i \in [n]$,

$$\boldsymbol{v}_{mm} = \operatorname*{argmin}_{\boldsymbol{v} \in \mathbb{R}^d} \|\boldsymbol{v}\| \text{ s.t. } y_i \cdot \boldsymbol{v}^\top \boldsymbol{\xi}_i \geq 1, \quad \text{ for all } i \in [n]. \tag{85}$$

$\Gamma = 1/\|\boldsymbol{v}_{mm}\|$ is the label margin induced by $\boldsymbol{v}_{mm}$.

To prove this token selection is optimal, we need to explain that the optimality of the token choice is strict in the sense that mixing other tokens will shrink the label margin. We formalize this into the following proposition:

**Proposition A.53** (Optimal Token Condition). *Suppose that Assumption 4.5 holds, with probability at least $1 - \delta$ on the training dataset, for all $\boldsymbol{p}$, the token selection under $\boldsymbol{p}$ results in a label margin of at most $\Gamma - c \cdot \max_{i \in [n]} (1 - s_{i2})$.*

Then we derive the convergence direction of $\boldsymbol{p}$ and $\boldsymbol{v}$ by Theorem A.27. Note that as $\|\boldsymbol{p}\| \to \infty$, the attention is more focused on the noise token for every training sample. Therefore, the output of signal token is upper bounded by a small value.

Consider a test sample $(\boldsymbol{X}, y)$, $\boldsymbol{X} = (\boldsymbol{\mu}', \boldsymbol{\xi}')$. As $\|\boldsymbol{p}\|$ increasing, the noise token $\boldsymbol{\xi}'$ will will dominate the overall output if $\boldsymbol{p}_{(r,R)}^\top \boldsymbol{\xi}' \geq 0$, which indicates the output of attention layer will close to the noise token, $\boldsymbol{r}' \to \boldsymbol{\xi}'$. Meanwhile, we can prove that $\boldsymbol{p}_{(r,R)}$ and $\boldsymbol{v}_{(r,R)}$ are near orthogonal, so $\boldsymbol{p}_{(r,R)}^\top \boldsymbol{\xi}'$ and $\boldsymbol{v}_{(r,R)}^\top \boldsymbol{\xi}'$ are nearly independent variables subjected to Gaussian distribution. Therefore, the probability that $y_i \boldsymbol{v}_{(r,R)}^\top \boldsymbol{\xi}' < 0$ is at least constant order.

**Optimal Token Condition**
First we find the optimal token selection in this case.

**Proposition A.54** (Optimal Token Condition). *Suppose that Assumption 4.5 holds, with probability at least $1 - \delta$ on the training dataset, for all $\boldsymbol{p}$, the token selection under $\boldsymbol{p}$ results in a label margin of at most $\Gamma - c \cdot \max_{i \in [n]} (1 - s_{i2})$.*

*Proof of Proposition A.53.* Similar as above, we consider the following three situations:

1. $p \neq 0, k - p = 0$. (All wrong token selections come from clean set)

2. $p = 0, k - p \neq 0$. (All wrong token selections come from noisy set)

3. $p \neq 0, k - p \neq 0$. (Wrong token selections are from both sets)

We will discuss each situation specifically and prove that Proposition A.14 holds in every possible case.

**Situation 1:** $p \neq 0, k - p = 0$

First, let's see the condition under the optimal choice of tokens:
*Condition* 15 (Original Condition).
$$y_i \boldsymbol{v}^\top \boldsymbol{\xi}_i \geq 1, i \in [n]$$

Similarly, $\boldsymbol{v}_{mm}$ also satisfies the KKT conditions of the max-margin problem (42) in this case, so we could write $\boldsymbol{v}$ as

$$\boldsymbol{v} = \lambda_1 \boldsymbol{\mu}_1 + \lambda_2 \boldsymbol{\mu}_2 + \sum_{i \in [n]} y_i \theta_i \boldsymbol{\xi}_i. \tag{86}$$

Plugging (86) in the condition 15, we can rewrite these conditions as:

$$\theta_i \cdot \|\boldsymbol{\xi}_i\|^2 + \sum_{i' \neq i} y_i y_{i'} \theta_{i'} \langle \boldsymbol{\xi}_i, \boldsymbol{\xi}_{i'} \rangle \geq 1, i \in [n].$$

Then we introduce a lemma to estimate the parameters of optimal solution under this condition:

**Lemma A.55** (Balanceing noise factor for KKT point). *Suppose that Assumption 4.5 holds, under Condition 15, we have*

$$\max_{i \in [n]} \theta_i \leq \frac{1}{(1 - \kappa)d - 2n\sqrt{d \log(6n^2/\delta)}},$$

$$\min_{i \in [n]} \theta_i \geq \frac{(1 - \kappa)d - 4n\sqrt{d \log(6n^2/\delta)}}{(1 + \kappa)d((1 - \kappa)d - 2n\sqrt{d \log(6n^2/\delta)})}.$$

*Proof of Lemma A.55.* First we prove the upper bound. Denote $j = \underset{i \in [n]}{\mathrm{argmax}}\, \theta_i$, we have

$$y_j \boldsymbol{v}^\top \boldsymbol{\xi}_j = \sum_{i \in [n]} y_i y_j \theta_i \langle \boldsymbol{\xi}_i, \boldsymbol{\xi}_j \rangle = \theta_j \|\boldsymbol{\xi}_j\|_2^2 + \sum_{i \neq j, i \in [n]} y_i y_j \theta_i \langle \boldsymbol{\xi}_i, \boldsymbol{\xi}_j \rangle$$

$$\geq \theta_j \cdot (1 - \kappa)d - n\theta_j \cdot 2\sqrt{d \log(6n^2/\delta)}$$

The last inequality is because Lemma A.65 and the definition of j. Consider the contrary case when $\theta_j > \frac{1}{(1-\kappa)d - 2n\sqrt{d\log(6n^2/\delta)}}$, we have

$$y_j \boldsymbol{v}^\top \boldsymbol{\xi}_j > \frac{1}{(1 - \kappa)d - 2n\sqrt{d \log(6n^2/\delta)}} \cdot ((1 - \kappa)d - n \cdot 2\sqrt{d \log(6n^2/\delta)}) = 1.$$

By the KKT conditions, if $y_j \boldsymbol{v}^\top \boldsymbol{\xi}_j > 1$ then we must have $\theta_j = 0$, and thus we reach a contradiction.

Then we prove the lower bound. For $\forall j \in [n]$ we have

$$1 \leq \theta_j \|\boldsymbol{\xi}_j\|_2^2 + \sum_{i \neq j, i \in [n]} y_i y_j \theta_i \langle \boldsymbol{\xi}_i, \boldsymbol{\xi}_j \rangle \leq \theta_j \cdot (1 + \kappa)d + n \max_{i \in [n]} \theta_i \cdot 2\sqrt{d \log(6n^2/\delta)}$$

$$\leq \theta_j \cdot (1 + \kappa)d + \frac{n}{(1 - \kappa)d - 2n\sqrt{d \log(6n^2/\delta)}} \cdot 2\sqrt{d \log(6n^2/\delta)}.$$

The second inequality is due to Lemma A.65 and the last inequality is from the upper bound we just get. Therefore, we have

$$\theta_j \geq \frac{(1 - \kappa)d - 4n\sqrt{d \log(6n^2/\delta)}}{(1 + \kappa)d((1 - \kappa)d - 2n\sqrt{d \log(6n^2/\delta)})}$$

This completes the proof.

$\square$

As for the signal parameters $\lambda_1$ and $\lambda_2$, to achieve the minimal norm for $\boldsymbol{v}$, it is obvious that $\lambda_1 = \lambda_2 = 0$. Then we can estimate $\|\boldsymbol{v}_{mm}\|$ in this case:

**Lemma A.56** (Norm of $\boldsymbol{v}_{mm}$). *Suppose that Assumption 4.5 holds, with probability at least $1 - \delta$ on the training dataset, for the solution $\boldsymbol{v}_{mm}$ of (42) under the token selection (82), we have*

$$\frac{n}{2d} \leq \|\boldsymbol{v}_{mm}\|^2 \leq \frac{5n}{d}.$$

*This implies*

$$\|\boldsymbol{v}_{mm}\| = \Theta\left(\sqrt{\frac{n}{d}}\right).$$

*Proof of Lemma A.56.* As $\boldsymbol{v}_{mm}$ is the max-margin solution and satisfies KKT condition, it can be represented as

$$\boldsymbol{v}_{mm} = \lambda_1 \boldsymbol{\mu}_1 + \lambda_2 \boldsymbol{\mu}_2 + \sum_{i \in \mathcal{C}} y_i \theta_i \boldsymbol{\xi}_i + \sum_{i \in [n]} y_i \theta_i \boldsymbol{\xi}_i. \tag{87}$$

As there is no constraint on $\lambda_1, \lambda_2$, both of them can take 0 to achieve max-margin. So we could lower bound $\|v_{mm}\|$ as

$$\|v_{mm}\|^2 \geq \sum_{i\in[n]} \theta_i^2 \|\xi_i\|^2 + \sum_{i\in[n]}\sum_{j\in[n]} y_i y_j \theta_i \theta_j \langle \xi_i, \xi_j \rangle \geq O\left(\frac{n^2}{d^{3/2}}\right) \geq \frac{n}{2d}.$$

The second inequality is from Lemma A.55 that $\theta_i = \Theta(1/d)$ for $i \in [n]$ and the last inequality is from Assumption 4.5.

Then to upper bound $\|v_{mm}\|$, consider the following possible solution $\widetilde{v}$

$$\widetilde{v} = \sum_{i\in[n]} 2y_i\xi_i/d.$$

For $i \in [n]$, we have

$$y_i \widetilde{v}^\top r_i = y_i \widetilde{v}^\top \xi_i = 2\|\xi_i\|^2/d + \sum_{j\in[n], j\neq i} 2y_iy_j\langle \xi_i, \xi_j \rangle/d$$

$$\geq 2(1 - \kappa) - 2n\sqrt{\log(6n^2/\delta)/d} \geq 1.$$

The first inequality is from Lemma A.65 and the second inequality is from Assumption 4.5. Therefore, $\widetilde{v}$ is a possible solution of SVM problem A.13 when $p$ converges to $p_{mm}$. So we have

$$\|v_{mm}\|^2 \leq \|\widetilde{v}\|^2 = \sum_{i\in[n]} 4\|\xi_i\|^2/d^2 + \sum_{i\in[n]}\sum_{j\in[n]} 4y_iy_j\langle \xi_i, \xi_j \rangle/d^2 \leq \frac{5n}{d}.$$

The last inequality is from Lemma A.65, Lemma A.68 and Assumption 4.5. Combine the results above, we have $\|v_{mm}\|^2 = \Theta(\frac{n}{d})$.

$\square$

Denote the mixed samples as $k_1, k_2, ..., k_p$. And for every mixed sample $k_i$, we have $r_{k_i} = (1 - \beta_i)\mu_{k_i} + \beta_i \xi_{k_i}$. Without losing generality, we assume that $y_{k_i} = +1$ for all $i \in [p]$. Then the conditions under *Situation 1* become

*Condition* 16 ($p$ clean samples violating optimal token selection).

$$\begin{cases} y_i v^\top \xi_i \geq 1, i \in [n]\backslash[p] \\ v^\top r_{k_i} \geq 1, i \in [p] \end{cases}$$

Denote the max-margin solution under this condition as $v'$ with parameters $\lambda_1', \lambda_2', \theta_i'$. Plugging this representation into the condition 16, we have:

$$\begin{cases} \theta_i' \cdot \|\xi_{i'}\|^2 + \sum_{i'\neq i} y_iy_{i'}\theta_{i'}'\langle \xi_i, \xi_{i'} \rangle \geq 1, i \in [n]\backslash[p] \\ (1 - \beta_i)\lambda_1' \cdot \|\mu_1\|^2 + \beta_i(\theta_{k_i}' \cdot \|\xi_{k_i}\|^2 + \sum_{i'\neq k_i} y_{i'}\theta_{i'}'\langle \xi_{k_i}, \xi_{i'} \rangle) \geq 1, i \in [p] \end{cases}$$

We consider two cases: $\lambda_1'\|\mu_1\|^2 < 1$ and $\lambda_1'\|\mu_1\|^2 \geq 1$. First when $\lambda_1'\|\mu_1\|^2 < 1$, the condition for mixed clean sample becomes:

$$\theta_{k_i}' \cdot \|\xi_{k_i}\|^2 + \sum_{i'\neq k_i} y_{i'}\theta_{i'}'\langle \xi_{k_i}, \xi_{i'} \rangle \geq \frac{1 - (1 - \beta_i)\lambda_1'\|\mu_1\|^2}{\beta_i} > 1,$$

which indicates that the condition for $\theta_{k_i}'$ is strengthened. So mixing 1 more clean sample is equal to strengthening 1 constraint in the original setting. Therefore, mixing p samples will not result in a better solution than only mixing 1 clean sample. Then we can simplify this case to mixing only 1 clean sample and denote this sample as $k_*$, $r_{k_*} = (1 - \beta)\mu_1 + \beta\xi_{k_*}$. Now the condition becomes:

*Condition* 17 (1 clean sample violating optimal token selection).

$$\begin{cases} \theta_i' \cdot \|\xi_{i'}\|^2 + \sum_{i'\neq i} y_iy_{i'}\theta_{i'}'\langle \xi_i, \xi_{i'} \rangle \geq 1, i \in [n]\backslash\{k_*\} \\ (1 - \beta)\lambda_1' \cdot \|\mu_1\|^2 + \beta(\theta_{k_*}' \cdot \|\xi_{k_i}\|^2 + \sum_{i'\neq k_*} y_{i'}\theta_{i'}'\langle \xi_{k_*}, \xi_{i'} \rangle) \geq 1 \end{cases}$$

Similarly, we introduce the following lemma which estimates the parameters in $v'$. We define

$$\alpha = \frac{1 - (1-\beta)\lambda_1'\|\boldsymbol{\mu}_1\|^2}{\beta}$$

for the convenience of the following proof.

**Lemma A.57.** *Suppose that Assumption 4.5 holds, under condition 17, with probability at least $1-\delta$ on the training dataset, we have*

$$\theta_{k_*}' \leq \frac{\alpha}{(1-\kappa)d - 2n\sqrt{d\log(6n^2/\delta)}},$$

$$\theta_{k_*}' \geq \frac{\alpha}{(1+\kappa)d}\left(1 - \frac{2n\sqrt{d\log(6n^2/\delta)}}{(1-\kappa)d - 2n\sqrt{d\log(6n^2/\delta)}}\right),$$

$$\max_{i\in[n]\setminus\{k_*\}} \theta_i' \leq \frac{(1-\kappa)d + 2(\alpha-n)\sqrt{d\log(6n^2/\delta)}}{((1-\kappa)d - 2n\sqrt{d\log(6n^2/\delta)})^2},$$

$$\min_{i\in[n]\setminus\{k_*\}} \theta_i' \geq \frac{1}{(1+\kappa)d}\cdot\left(1 - \frac{2n\alpha\sqrt{d\log(6n^2/\delta)}}{(1-\kappa)d - 2n\sqrt{d\log(6n^2/\delta)}}\right).$$

*Proof of Lemma A.57.* Denote $j = \underset{i\in[n]}{\operatorname{argmax}}\,\theta_i'$, we have

$$y_j {v'}^\top \boldsymbol{\xi}_j = \theta_j'\|\boldsymbol{\xi}_j\|^2 + \sum_{i\in[n],i\neq j} y_i y_j \theta_i'\langle\boldsymbol{\xi}_i,\boldsymbol{\xi}_j\rangle$$

$$\geq \theta_j'(1-\kappa)d - n\max_{i\in[n]}\theta_i'\cdot 2\sqrt{d\log(6n^2/\delta)}$$

$$= \theta_j'((1-\kappa)d - n\cdot 2\sqrt{d\log(6n^2/\delta)}).$$

The first inequality is due to Lemma A.65 and the last equation is from our definition of j. Consider the contrary case when $\theta_j' > \frac{\alpha}{(1-\kappa)d - 2n\sqrt{d\log(6n^2/\delta)}}$, we have

$$y_j {v'}^\top \boldsymbol{\xi}_j > \alpha.$$

By the KKT conditions, if $y_j {v'}^\top \boldsymbol{\xi}_j > \frac{1+\lambda_1'(1-\beta)\|\boldsymbol{\mu}_1\|^2}{\beta}$ then we must have $\theta_j' = 0$, and thus we reach a contradiction. Therefore, $\theta_{k_\star}' \leq \theta_j' \leq \frac{\alpha}{(1-\kappa)d - 2n\sqrt{d\log(6n^2/\delta)}}$. Then denote $j' = \underset{i\in[n],i\neq k_\star}{\operatorname{argmax}}\,\theta_i''$, we have

$$y_{j'} {v'}^\top \boldsymbol{\xi}_{j'} = \theta_{j'}'\|\boldsymbol{\xi}_{j'}\|^2 + \sum_{i\in[n],i\neq j'} y_i y_{j'} \theta_i'\langle\boldsymbol{\xi}_i,\boldsymbol{\xi}_{j'}\rangle$$

$$\geq \theta_{j'}'(1-\kappa)d - n\max_{i\in[n],i\neq j'}\theta_i'\cdot 2\sqrt{d\log(6n^2/\delta)} - \theta_{k_\star}'\sqrt{d\log(6n^2/\delta)}$$

$$\geq \theta_j'((1-\kappa)d - n\cdot 2\sqrt{d\log(6n^2/\delta)}) - \frac{2\alpha\sqrt{d\log(6n^2/\delta)}}{(1-\kappa)d - 2n\sqrt{d\log(6n^2/\delta)}}.$$

The first inequality is from Lemma A.65 and the second inequality is from the upper bound of $\theta_{k_\star}'$ we just get. Consider the case when $\theta_{j'}' > \frac{(1-\kappa)d + 2(\alpha-n)\sqrt{d\log(6n^2/\delta)}}{((1-\kappa)d - 2n\sqrt{d\log(6n^2/\delta)})^2}$, we have

$$y_{j'} {v'}^\top \boldsymbol{\xi}_{j'} > 1.$$

By the complementary slackness condition, if $y_{j'} {v''}^\top \boldsymbol{\xi}_{j'} > 1$ then we must have $\theta_{j'}' = 0$, and thus we reach a contradiction.

Next we estimate the lower bound of $\theta_j'$ when $j \neq k_*$. We have

$$
\begin{aligned}
1 &\leq y_j {\boldsymbol{v}'}^\top \boldsymbol{\xi}_j \\
&= \theta_j' \|\boldsymbol{\xi}_j\|^2 + \sum_{i \in [n], i \neq j} y_i y_j \theta_i' \langle \boldsymbol{\xi}_i, \boldsymbol{\xi}_j \rangle \\
&\leq \theta_j' (1+\kappa)d + n \max_{i \in [n]} \theta_i' \cdot 2\sqrt{d \log(6n^2/\delta)} \\
&\leq \theta_j' (1+\kappa)d + \frac{\alpha}{(1-\kappa)d - 2n\sqrt{d\log(6n^2/\delta)}} \cdot 2n\sqrt{d\log(6n^2/\delta)}
\end{aligned}
$$

The last inequality is from the upper bound of $\theta_{k_*}'$ we just get. Therefore, we have

$$
\theta_j' \geq \frac{1}{(1+\kappa)d} \cdot \left( 1 - \frac{2n\alpha\sqrt{d\log(6n^2/\delta)}}{(1-\kappa)d - 2n\sqrt{d\log(6n^2/\delta)}} \right)
$$

for all $j \in [n]$ and $j \neq k_*$.

Last we lower bound $\theta_{k_*}'$. We have

$$
\begin{aligned}
\alpha &\leq y_k {\boldsymbol{v}''}^\top \boldsymbol{\xi}_{k_*} \\
&= \theta_{k_*}' (1+\kappa)d + n \max_{i \in [n]} \theta_i' \cdot 2\sqrt{d\log(6n^2/\delta)}
\end{aligned}
$$

Similarly, we have

$$
\theta_{k_*}' \geq \frac{\alpha}{(1+\kappa)d} \left( 1 - \frac{2n\sqrt{d\log(6n^2/\delta)}}{(1-\kappa)d - 2n\sqrt{d\log(6n^2/\delta)}} \right).
$$

$\square$

Therefore, we could estimate the difference between $\|\boldsymbol{v}'\|^2$ and $\|\boldsymbol{v}_{mm}\|^2$.

**Lemma A.58.** *Suppose that Assumption 4.5 holds, with probability at least $1 - \delta$ on the training dataset, denote $\boldsymbol{v}$ and $\boldsymbol{v}'$ as the optimal solutions under condition 15 and condition 17 respectively. We have*

$$
\|\boldsymbol{v}'\|_2^2 - \|\boldsymbol{v}_{mm}\|_2^2 \geq \frac{C_1(1-\beta)}{d}.
$$

*where $C_1 = \Theta(1)$ is a constant.*

*Proof of Lemma A.58.* From the first inequality in Condition 17, for $i[n], i \neq k_\star$ we have

$$
\theta_i' \cdot \|\boldsymbol{\xi}_i\|^2 + \sum_{i' \neq i, k_\star} y_i y_{i'} \theta_{i'}' \langle \boldsymbol{\xi}_i, \boldsymbol{\xi}_{i'} \rangle \geq 1 - y_i y_{k_\star} \theta_{k_\star}' \langle \boldsymbol{\xi}_i, \boldsymbol{\xi}_{k_\star} \rangle.
$$

Then we add $y_i y_{k_\star} w \langle \boldsymbol{\xi}_i, \boldsymbol{\xi}_{k_\star} \rangle$ on both sides, where we set $w = \theta_{k_\star}' - \frac{\alpha-1}{(1+\kappa)d - 2\sqrt{d\log(6n^2/\delta)}} \leq \theta_{k_\star}'$.
Then we have

$$
\theta_i' \cdot \|\boldsymbol{\xi}_{i'}\|^2 + \sum_{i' \neq i, k_\star} y_i y_{i'} \theta_{i'}' \langle \boldsymbol{\xi}_i, \boldsymbol{\xi}_{i'} \rangle + y_i y_{k_\star} w \langle \boldsymbol{\xi}_i, \boldsymbol{\xi}_{k_\star} \rangle \geq 1 - y_i y_{k_\star} (\theta_{k_\star}' - w) \langle \boldsymbol{\xi}_i, \boldsymbol{\xi}_{k_\star} \rangle
$$

$$
\begin{aligned}
&\geq 1 - 2(\theta_{k_\star}' - w)\sqrt{d\log(6n^2/\delta)} \\
&= \frac{(1+\kappa)d - 2\alpha\sqrt{d\log(6n^2/\delta)}}{(1+\kappa)d - 2\sqrt{d\log(6n^2/\delta)}}. \quad (88)
\end{aligned}
$$

The second inequality is from Lemma A.65. Now consider a new $\underline{\boldsymbol{v}} = \underline{\lambda}_1 \boldsymbol{\mu}_1 + \underline{\lambda}_2 \boldsymbol{\mu}_2 + \sum_{i \in [n]} y_i \underline{\theta}_i \boldsymbol{\xi}_i$
with

$$
\underline{\lambda}_1 = \lambda_1'; \quad \underline{\lambda}_2 = \lambda_2';
$$

$$
\underline{\theta}_i = \theta_i' / (1 - 2(\theta_{k_\star}' - w)\sqrt{d\log(6n^2/\delta)}) \text{ for } i \in [n], i \neq k_\star
$$

and

$$\underline{\theta}_{k_\star} = \frac{w}{1 - 2(\theta'_{k_\star} - w)\sqrt{d\log(6n^2/\delta)}}.$$

We can prove that $\underline{v}$ satisfies all constraints for $v_{mm}$.

By dividing $1 - 2(\theta'_{k_\star} - w)\sqrt{d\log(6n^2/\delta)}$ on both sides of (88), for $\forall i \in [n], i \neq k_\star$ we have

$$\underline{\theta}_i \cdot \|\boldsymbol{\xi}_i\|^2 + \sum_{i' \neq i} y_i y_{i'} \underline{\theta}_i \langle \boldsymbol{\xi}_i, \boldsymbol{\xi}_{i'} \rangle \geq 1.$$

Then we prove that $\underline{\theta}_{k_\star}\|\boldsymbol{\xi}_{k_\star}\|^2 + \sum_{i \neq k_\star} y_i y_{k_\star} \underline{\theta}_i \langle \boldsymbol{\xi}_i, \boldsymbol{\xi}_{k_\star} \rangle \geq 1$. From the last inequality in Condition 17 we have

$$\theta'_{k_\star} \cdot \|\boldsymbol{\xi}_{k_\star}\|^2 + \sum_{i \neq k_\star} y_{k_\star} y_i \theta'_i \langle \boldsymbol{\xi}_i, \boldsymbol{\xi}_{k_\star} \rangle \geq \alpha.$$

Dividing $1 - 2(\theta'_{k_\star} - w)\sqrt{d\log(6n^2/\delta)}$ on both sides, we get

$$\frac{\theta'_{k_\star}\|\boldsymbol{\xi}_{k_\star}\|^2}{1 - 2(\theta'_{k_\star} - w)\sqrt{d\log(6n^2/\delta)}} + \sum_{i \neq k_\star} y_i y_{k_\star} \underline{\theta}_i \langle \boldsymbol{\xi}_i, \boldsymbol{\xi}_{k_\star} \rangle \geq \frac{\alpha}{1 - 2(\theta'_{k_\star} - w)\sqrt{d\log(6n^2/\delta)}}.$$

Therefore we have

$$\underline{\theta}_{k_\star}\|\boldsymbol{\xi}_{k_\star}\|^2 + \sum_{i \neq k_\star} y_i y_{k_\star} \underline{\theta}_i \langle \boldsymbol{\xi}_i, \boldsymbol{\xi}_{k_\star} \rangle \geq \frac{\alpha - (\theta'_{k_\star} - w)\|\boldsymbol{\xi}_{k_\star}\|^2}{1 - 2(\theta'_{k_\star} - w)\sqrt{d\log(6n^2/\delta)}} \geq \frac{\alpha - (\theta'_{k_\star} - w)(1+\kappa)d}{1 - 2(\theta'_{k_\star} - w)\sqrt{d\log(6n^2/\delta)}} = 1.$$

The second inequality is from Lemma A.65 and the last equality is by our definition $\theta'_{k_\star} - w = \frac{\alpha-1}{(1+\kappa)d - 2\sqrt{d\log(6n^2/\delta)}}$. Thus, $\underline{v}$ is a possible solution under Condition 1 and $\|\underline{v}\| \geq \|v_{mm}\|$.

Next we estimate the difference between $\|v'\|^2$ and $\|\underline{v}\|^2$. The expansion of $\|v'\|^2$ and $\|\underline{v}\|^2$ are:

$$\|v'\|^2 = \lambda_1'^2\|\boldsymbol{\mu}_1\|^2 + \lambda_2'^2\|\boldsymbol{\mu}_2\|^2 + \sum_{i \in [n]} \theta_i'^2\|\boldsymbol{\xi}_i\|^2 + \sum_{i \in [n]} \sum_{j \in [n]} y_i y_j \theta_i' \theta_j' \langle \boldsymbol{\xi}_i, \boldsymbol{\xi}_j \rangle,$$

$$\|\underline{v}\|^2 = \underline{\lambda}_1^2\|\boldsymbol{\mu}_1\|^2 + \underline{\lambda}_2^2\|\boldsymbol{\mu}_2\|^2 + \sum_{i \in [n]} \underline{\theta}_i^2\|\boldsymbol{\xi}_i\|^2 + \sum_{i \in [n]} \sum_{j \in [n]} y_i y_j \underline{\theta}_i \underline{\theta}_j \langle \boldsymbol{\xi}_i, \boldsymbol{\xi}_j \rangle.$$

Similar to the condition (49), we have $\|v'\| \leq 2\|v_{mm}\| = \Theta(\sqrt{n/d})$, which implies that $\alpha = O(\sqrt{n}\log n)$. Otherwise, we have

$$\theta'_{k_\star}\|\boldsymbol{\xi}_{k_\star}\|^2 \geq \alpha - \sum_{i \neq k_\star} y_{k_\star} y_i \theta'_i \langle \boldsymbol{\xi}_i, \boldsymbol{\xi}_{k_\star} \rangle = \Omega(\alpha).$$

It further yields that

$$\|v'\|^2 = \Omega(\frac{n}{d}) + \theta_{k_\star}'^2\|\boldsymbol{\xi}_{k_\star}\|^2 = \Omega(\frac{n}{d} + \frac{\alpha^2}{d}) = \Omega(\frac{n\log^2 n}{d}),$$

which contradicts with $\|v'\| = \Theta(\sqrt{n/d})$.

We decompose the difference between $\|v'\|^2$ and $\|\underline{v}\|^2$ into four terms:

$$\|v'\|^2 - \|\underline{v}\|^2 = \underbrace{(\theta_{k_\star}'^2 - \underline{\theta}_{k_\star}^2)\|\boldsymbol{\xi}_{k_\star}\|^2}_{I_1} + \underbrace{\sum_{i \in [n], i \neq k_\star} (\theta_i'^2 - \underline{\theta}_i^2)\|\boldsymbol{\xi}_i\|^2}_{I_2} - \underbrace{\sum_{i \in [n]} \sum_{j \in [n]} y_i y_j \underline{\theta}_i \underline{\theta}_j \langle \boldsymbol{\xi}_i, \boldsymbol{\xi}_j \rangle}_{I_3}$$

$$+ \underbrace{\sum_{i \in [n]} \sum_{j \in [n]} y_i y_j \theta_i' \theta_j' \langle \boldsymbol{\xi}_i, \boldsymbol{\xi}_j \rangle}_{I_4}.$$

We now estimate $I_1$ to $I_4$ sequentially. For the first term,

$$I_1 \geq (\theta'^2_{k_\star} - \underline{\theta}^2_{k_\star})(1-\kappa)d = (\theta'_{k_\star} - \underline{\theta}_{k_\star})(\theta'_{k_\star} + \underline{\theta}_{k_\star})(1-\kappa)d$$

$$= \frac{(\alpha-1)(1 - 2\theta'_{k_\star}\sqrt{d\log(6n^2/\delta)})}{(1+\kappa)d - 2\sqrt{d\log(6n^2/\delta)}} \cdot \Omega\left(\frac{1}{d}\right) \cdot (1-\kappa)d$$

$$= \Omega\left(\frac{\alpha-1}{d}\right),$$

where the first inequality is from Lemma A.65; the second equality is from Lemma A.57; and the last equality uses the fact that $\alpha = O(\sqrt{n}\log n)$. Then we can further upper bound $\max_{i\in[n],i\neq k_\star} \theta'_i$ as

$$\max_{i\in[n],i\neq k_\star} \theta'_i \leq \frac{(1-\kappa)d + 2(\alpha - n)\sqrt{d\log(6n^2/\delta)}}{((1-\kappa)d - 2n\sqrt{d\log(6n^2/\delta)})^2} = O(\frac{1}{d}). \tag{89}$$

For the second term $I_2$, we have

$$|I_2| \leq \sum_{i\in[n],i\neq k_\star} (\underline{\theta}^2_i - \theta'^2_i)(1+\kappa)d$$

$$\leq \left(\frac{1}{(1 - (\theta'_{k_\star} - w)\sqrt{d\log(6n^2/\delta)})^2} - 1\right) \max_{i\in[n],i\neq k_\star} \theta'^2_i \cdot n(1+\kappa)d$$

$$= \frac{(\alpha-1)\sqrt{d\log(6n^2/\delta)}}{(1+\kappa)d - \sqrt{d\log(6n^2/\delta)}} \cdot O(\frac{n}{d}) = \widetilde{O}\left(\frac{(\alpha-1)n}{d^{3/2}}\right).$$

The second inequality is from Lemma A.57. The first equality is from (89) and the last equality is from Assumption 4.5.

Then we bound $|-I_3 + I_4|$ as:

$$|-I_3 + I_4| \leq \sum_{i\in[n]} \sum_{j\in[n]\setminus\{i\}} |\underline{\theta}_i\underline{\theta}_j - \theta'_i\theta'_j| \cdot |\langle \boldsymbol{\xi}_i, \boldsymbol{\xi}_j \rangle|$$

$$\leq \sum_{i\in[n]\setminus\{k_\star\}} \sum_{j\in[n]\setminus\{k_\star,i\}} |\underline{\theta}_i\underline{\theta}_j - \theta'_i\theta'_j| \cdot |\langle \boldsymbol{\xi}_i, \boldsymbol{\xi}_j \rangle| + 2\sum_{t\in[n]\setminus\{k_\star\}} |\underline{\theta}_{k_\star}\underline{\theta}_t - \theta'_{k_\star}\theta'_t| \cdot |\langle \boldsymbol{\xi}_{k_\star}, \boldsymbol{\xi}_t \rangle|$$

$$\leq n^2\left(\frac{1}{(1 - (\theta'_{k_\star} - w)\sqrt{d\log(6n^2/\delta)})^2} - 1\right) \max_{i\in[n],i\neq k_\star} \theta'^2_i \cdot 2\sqrt{d\log(6n^2/\delta)}$$

$$+ n\left(\theta'_{k_\star} - \frac{\underline{\theta}_{k_\star}}{1 - 2(\theta'_{k_\star} - w)\sqrt{d\log(6n^2/\delta)}}\right) \max_{i\in[n],i\neq k_\star} \theta'_i 4\sqrt{d\log(6n^2/\delta)}$$

$$\leq \frac{(\alpha-1)\sqrt{d\log(6n^2/\delta)}}{(1+\kappa)d - \sqrt{d\log(6n^2/\delta)}} \cdot O(\frac{n^2(1+\kappa)}{d^{3/2}}) + \frac{\alpha-1}{d} \cdot O(\frac{n}{d}) \cdot 2\sqrt{d\log(6n^2/\delta)}$$

$$= O\left(\frac{(\alpha-1)n^2}{d^2} + \frac{(\alpha-1)n}{d^{3/2}}\right).$$

The third inequality is from Lemma A.55 and Lemma A.57; The fourth inequality is from the fact that

$$\theta'_{k_\star} - \frac{\underline{\theta}_{k_\star}}{1 - 2(\theta'_{k_\star} - w)\sqrt{d\log(6n^2/\delta)}} = \frac{\theta'_{k_\star} - \underline{\theta}_{k_\star} - 2\theta'_{k_\star}(\theta'_{k_\star} - w)\sqrt{d\log(6n^2/\delta)}}{1 - 2(\theta'_{k_\star} - w)\sqrt{d\log(6n^2/\delta)}}$$

$$= \frac{\Omega(\frac{\alpha-1}{d}) - O(\frac{\alpha(\alpha-1)}{d^{3/2}})}{1 - 2(\theta'_{k_\star} - w)\sqrt{d\log(6n^2/\delta)}} > 0$$

So we have $\theta'_{k_\star} - \frac{\underline{\theta}_{k_\star}}{1 - 2(\theta'_{k_\star} - w)\sqrt{d\log(6n^2/\delta)}} \leq \theta'_{k_\star} - \underline{\theta}_{k_\star}$; The last equality is from Assumption 4.1.

Combining the above results, we have

$$\|\boldsymbol{v}'\|^2_2 - \|\boldsymbol{v}_{mm}\|^2_2 \geq \Theta\left(\frac{\alpha-1}{d}\right) + O\left(\frac{(\alpha-1)\eta n}{d^{3/2}}\right) \geq \frac{C_1(1-\beta)}{d}.$$

Here $C_1 = \Theta(1)$ is a constant. $\qquad\square$

Then we consider the case when $\lambda_1' \|\boldsymbol{\mu}_1\|^2 \geq 1$. In this case, the condition for mixed clean sample becomes:

$$\theta_{k_i}' \cdot \|\boldsymbol{\xi}_{k_i}\|^2 + \sum_{i' \neq k_i} y_{ki} y_{i'} \theta_{i'}' \langle \boldsymbol{\xi}_{k_i}, \boldsymbol{\xi}_{i'} \rangle \geq \frac{1 - (1 - \beta_i) \lambda_1' \|\boldsymbol{\mu}_1\|^2}{\beta_i},$$

and $\frac{1 - (1 - \beta_i) \lambda_1' \|\boldsymbol{\mu}_1\|^2}{\beta_i} \leq 1$, which indicates that the condition for $\theta_{k_i}'$ is relaxed. So mixing 1 more clean sample is equal to relaxing 1 constraint in the original setting. Therefore, mixing all clean samples will achieve the best result. From the data generalization model, there are $(1 - \eta)n/2 + o(n)$ clean samples with label $+1$ and denote $S_{+1}$ as their set. Now the condition becomes:

*Condition* 18 (All clean samples violating optimal token selection).

$$\begin{cases} \theta_i' \cdot \|\boldsymbol{\xi}_{i'}\|^2 + \sum\limits_{i' \neq i} y_i y_{i'} \theta_{i'}' \langle \boldsymbol{\xi}_i, \boldsymbol{\xi}_{i'} \rangle) \geq 1, i \in [n] \setminus S_{+1} \\ (1 - \beta) \lambda_1' \cdot \|\boldsymbol{\mu}_1\|^2 + \beta(\theta_i' \cdot \|\boldsymbol{\xi}_i\|^2 + \sum\limits_{i' \neq i} y_i y_{i'} \theta_{i'}' \langle \boldsymbol{\xi}_i, \boldsymbol{\xi}_{i'} \rangle) \geq 1, i \in S_{+1} \end{cases}$$

We have another lemma to estimate the scale of parameters in the max-margin solution in this case. Here $\alpha = \frac{1 - (1 - \widetilde{\beta}) \lambda_1' \|\boldsymbol{\mu}_1\|^2}{\widetilde{\beta}}$ and $\widetilde{\beta} = \min\limits_{i \in [n]} \{\beta_i\}$.

**Lemma A.59.** *Suppose that Assumption 4.5 holds, under Condition 18, we have*

$$\max_{i \in [n]} \theta_i' \leq \frac{1}{(1 - \kappa)d - 2n\sqrt{d \log(6n^2/\delta)}},$$

$$\min_{i \in [n]} \theta_i' \geq \frac{(1 - \kappa)d\alpha - 2n\sqrt{d \log(6n^2/\delta)}(\alpha + 1)}{(1 + \kappa)d((1 - \kappa)d - 2n\sqrt{d \log(6n^2/\delta)})}.$$

*Proof of Lemma A.59.* First we prove the upper bound. Denote $j = \operatorname*{argmax}\limits_{i \in [n]} \theta_i$, we have

$$\begin{aligned} y_j \boldsymbol{v}^\top \boldsymbol{\xi}_j &= \sum_{i \in [n]} y_i y_j \theta_i \langle \boldsymbol{\xi}_i, \boldsymbol{\xi}_j \rangle \\ &= \theta_j \|\boldsymbol{\xi}_j\|_2^2 + \sum_{i \neq j, i \in [n]} y_i y_j \theta_i \langle \boldsymbol{\xi}_i, \boldsymbol{\xi}_j \rangle \\ &\geq \theta_j \cdot (1 - \kappa)d - n\theta_j \cdot 2\sqrt{d \log(6n^2/\delta)} \end{aligned}$$

The last inequality is because Lemma A.65 and the definition of j. Consider the contrary case when $\theta_j > \frac{1}{(1 - \kappa)d - 2n\sqrt{d \log(6n^2/\delta)}}$, we have

$$y_j \boldsymbol{v}^\top \boldsymbol{\xi}_j > \frac{1}{(1 - \kappa)d - 2n\sqrt{d \log(6n^2/\delta)}} \cdot ((1 - \kappa)d - n \cdot 2\sqrt{d \log(6n^2/\delta)}) = 1.$$

By the KKT conditions, if $y_j \boldsymbol{v}^\top \boldsymbol{\xi}_j > 1$ then we must have $\theta_j = 0$, and thus we reach a contradiction. Then we prove the lower bound. For $\forall j \in S_{+1}$ we have

$$\begin{aligned} \alpha &\leq \theta_j \|\boldsymbol{\xi}_j\|_2^2 + \sum_{i \neq j, i \in [n]} y_i y_j \theta_i \langle \boldsymbol{\xi}_i, \boldsymbol{\xi}_j \rangle \\ &\leq \theta_j \cdot (1 + \kappa)d + n \max_{i \in [n]} \theta_i \cdot 2\sqrt{d \log(6n^2/\delta)} \\ &\leq \theta_j \cdot (1 + \kappa)d + \frac{n}{(1 - \kappa)d - 2n\sqrt{d \log(6n^2/\delta)}} \cdot 2\sqrt{d \log(6n^2/\delta)}. \end{aligned}$$

The second inequality is due to Lemma A.65 and the last inequality is from the upper bound we just get. Therefore, we have

$$\theta_j \geq \frac{(1 - \kappa)d\alpha - 2n\sqrt{d \log(6n^2/\delta)}(\alpha + 1)}{(1 + \kappa)d((1 - \kappa)d - 2n\sqrt{d \log(6n^2/\delta)})}.$$

This completes the proof

$\square$

Then we can estimate the difference between $\|\boldsymbol{v}'\|^2$ and $\|\boldsymbol{v}_{mm}\|^2$ with the following lemma:

**Lemma A.60.** *Suppose that Assumption 4.5 holds, denote $\boldsymbol{v}$ and $\boldsymbol{v}'$ as the optimal solutions under condition 15 and condition 18 respectively. We have*

$$\|\boldsymbol{v}'\|_2^2 - \|\boldsymbol{v}_{mm}\|_2^2 \geq \frac{C_2(1-\beta)}{\rho^2}.$$

*where $C_2 = \Theta(1)$ is a constant.*

*Proof of Lemma A.60.* Recall the expansion of $\|\boldsymbol{v}_{mm}\|^2$ and $\|\boldsymbol{v}'\|^2$:

$$\|\boldsymbol{v}_{mm}\|^2 = \sum_{i\in[n]} \theta_i^2\|\boldsymbol{\xi}_i\|^2 + \sum_{i\in[n]}\sum_{j\in[n]} y_iy_j\theta_i\theta_j\langle\boldsymbol{\xi}_i,\boldsymbol{\xi}_j\rangle,$$

$$\|\boldsymbol{v}'\|^2 = \lambda_1'^2\|\boldsymbol{\mu}_1\|^2 + \sum_{i\in[n]} \theta_i'^2\|\boldsymbol{\xi}_i\|^2 + \sum_{i\in[n]}\sum_{j\in[n]} y_iy_j\theta_i'\theta_j'\langle\boldsymbol{\xi}_i,\boldsymbol{\xi}_j\rangle.$$

Then we have

$$\|\boldsymbol{v}'\|^2 - \|\boldsymbol{v}_{mm}\|^2 = \underbrace{\lambda_1'^2\|\boldsymbol{\mu}_1\|^2}_{I_1} + \underbrace{\sum_{i\in[n]}(\theta_i'^2 - \theta_i^2)\|\boldsymbol{\xi}_i\|^2}_{I_2} - \underbrace{\sum_{i\in[n]}\sum_{j\in[n]} y_iy_j\theta_i\theta_j\langle\boldsymbol{\xi}_i,\boldsymbol{\xi}_j\rangle}_{I_3}$$

$$+ \underbrace{\sum_{i\in[n]}\sum_{j\in[n]} y_iy_j\theta_i''\theta_j''\langle\boldsymbol{\xi}_i,\boldsymbol{\xi}_j\rangle}_{I_4}.$$

We now estimate $I_1$ to $I_4$ sequentially. Here we use the same notation $\alpha = \frac{1-(1-\widetilde{\beta})\lambda_1'\|\boldsymbol{\mu}_1\|^2}{\widetilde{\beta}}$ and $\widetilde{\beta} = \min_{i\in[n]}\{\beta_i\}$ as in Lemma A.59. First from our assumption $\lambda_1'\|\boldsymbol{\mu}_1\|^2 \geq 1$ we have

$$I_1 = \lambda_1'^2\|\boldsymbol{\mu}_1\|^2 \geq 1/\rho^2.$$

Then for $I_2$, we have

$$|I_2| \leq n(\max_{i\in[n]}\theta_i^2 - \min_{i\in[n]}\theta_i'^2) \cdot (1+\kappa)d$$

$$\leq \left(\frac{1}{((1-\kappa)d - 2n\sqrt{d\log(6n^2/\delta)})^2} - \frac{1}{(1+\kappa)^2d^2}\cdot\left(\alpha - \frac{2n\sqrt{d\log(6n^2/\delta)}}{(1-\kappa)d - 2n\sqrt{d\log(6n^2/\delta)}}\right)^2\right)\cdot(1+\kappa)dn$$

$$= d(1+\kappa)n \cdot \frac{1 - \frac{1}{(1+\kappa)^2d^2}((1-\kappa)d\alpha - 2(\alpha+1)n\sqrt{d\log(6n^2/\delta)})^2}{((1-\kappa)d - 2n\sqrt{d\log(6n^2/\delta)})^2}$$

$$= O\left(\frac{n}{d}\right).$$

The second inequality is from Lemma A.55 and Lemma A.59.

Then we bound $|-I_3 + I_4|$ as:

$$|-I_3 + I_4| \leq \sum_{i\in[n]}\sum_{j\in[n]\setminus\{i\}} (\theta_i'\theta_j' - \theta_i\theta_j)\cdot|\langle\boldsymbol{\xi}_i,\boldsymbol{\xi}_j\rangle|$$

$$\leq (n)^2(\max_{i\in[n]}\theta_i'^2 - \min_{i\in[n]}\theta_i^2)\cdot 2\sqrt{d\log(6n^2/\delta)}$$

$$\leq (n)^2\left[\left(\frac{1}{(1-\kappa)d - 2n\sqrt{d\log(6n^2/\delta)}}\right)^2 - \left(\frac{(1-\kappa)d - 4n\sqrt{d\log(6n^2/\delta)}}{(1+\kappa)d((1-\kappa)d - 2n\sqrt{d\log(6n^2/\delta)})}\right)^2\right]\cdot 2\sqrt{d\log(6n^2/\delta)}$$

$$= \widetilde{O}\left(\frac{\kappa n^2}{d^{3/2}}\right) = O\left(\frac{n^2}{d^2}\right).$$

The third inequality is from Lemma A.55 and A.59; The last two equalities are from Assumption 4.5. Combining the above results, we have

$$\|\boldsymbol{v}'\|_2^2 - \|\boldsymbol{v}_{mm}\|_2^2 \geq \frac{C}{\rho^2} + O\left(\frac{n}{d}\right) \geq \frac{C_2(1-\beta)}{\rho^2}.$$

Here $C_2 = \Theta(1)$ is a constant. $\qquad \square$

Therefore, combining Lemma A.58 and A.60, we have the following statement for the difference between $\|\boldsymbol{v}'\|$ and $\|\boldsymbol{v}_{mm}\|$:

$$\|\boldsymbol{v}'\|_2^2 - \|\boldsymbol{v}_{mm}\|_2^2 \geq \frac{C_3(1-\beta)}{d}. \tag{90}$$

Here $C_3 = \Theta(1)$ is a constant. The inequality is from the SNR condition that $\rho = o(\sqrt{d/n})$.

Now we can prove the main proposition in this scenario.

*Proof of Proposition A.53 in case 1.* From (90) we have

$$\|\boldsymbol{v}''\|_2^2 - \|\boldsymbol{v}\|_2^2 \geq \frac{C_3(1-\beta)}{d} = S(1-\beta)$$

Here we substitute $S = \frac{C_3}{d} \geq 0$ Then we have

$$\Gamma^2 - \Gamma'^2 = \frac{1}{\|\boldsymbol{v}\|^2} - \frac{1}{\|\boldsymbol{v}'\|^2} = \frac{\|\boldsymbol{v}'\|^2 - \|\boldsymbol{v}\|^2}{\|\boldsymbol{v}'\|^2 \cdot \|\boldsymbol{v}\|^2} \geq \frac{S(1-\beta)}{\|\boldsymbol{v}'\|^2 \cdot \|\boldsymbol{v}\|^2}.$$

Therefore,

$$\Gamma - \Gamma' \geq \frac{S(1-\beta)}{(\Gamma + \Gamma')\|\boldsymbol{v}\|^2 \cdot \|\boldsymbol{v}'\|^2} \geq \frac{S(1-\beta)}{2\Gamma\|\boldsymbol{v}\|^2 \cdot \|\boldsymbol{v}'\|^2}.$$

Set $c = \frac{S}{2\Gamma\|\boldsymbol{v}\|^2 \cdot \|\boldsymbol{v}'\|^2} = \frac{S}{2\|\boldsymbol{v}\|\|\boldsymbol{v}'\|^2}$, we have $\Gamma' \leq \Gamma - c(1-\beta)$. And we can upper bound $c$ as

$$c = \frac{S}{2\|\boldsymbol{v}\|\|\boldsymbol{v}'\|^2} \leq \frac{S}{r_{mm}^3} \leq \frac{C_3}{r_{mm}^3 d}.$$

The first inequality is from $\|\boldsymbol{v}'\| \geq \|\boldsymbol{v}\|$ and the second equality is from $S = \frac{C_2}{d}$.

$\qquad \square$

**Situation 2:** $p = 0, k - p \neq 0$

Then we consider the case when all wrong token selections come from noisy set. Same as above, denote the mixed samples as $k_1, k_2, ..., k_{k-p}$. And for every mixed sample $k_i$, we have $\boldsymbol{r}_{k_i} = (1 - \beta_i)\boldsymbol{\mu}_{k_i} + \beta_i\boldsymbol{\xi}_{k_i}$. Without losing generality, we assume that $y_{k_i} = +1$ for all $i \in [k - p]$, so the corresponding signal token is $\boldsymbol{\mu}_2$. Then the conditions under *Situation 2* become

*Condition* 19 (Change k-p noisy samples).

$$\begin{cases} y_i \boldsymbol{v}^\top \boldsymbol{\xi}_i \geq 1, i \in [n]\backslash[k-p] \\ \boldsymbol{v}^\top \boldsymbol{r}_{k_i} \geq 1, i \in [k-p] \end{cases}$$

Denote the max-margin solution under this condition as $\boldsymbol{v}'$ with parameters $\lambda_1', \lambda_2', \theta_i'$, we can interpret the condition for parameters:

$$\begin{cases} \theta_i' \cdot \|\boldsymbol{\xi}_{i'}\|^2 + \sum\limits_{i' \neq i} y_i y_{i'} \theta_{i'}' \langle \boldsymbol{\xi}_i, \boldsymbol{\xi}_{i'} \rangle \geq 1, i \in [n]\backslash[k-p] \\ (1 - \beta_i)\lambda_2' \cdot \|\boldsymbol{\mu}_2\|^2 + \beta_i(\theta_{k_i}' \cdot \|\boldsymbol{\xi}_{k_i}\|^2 + \sum\limits_{i' \neq k_i} y_{k_i} y_{i'} \theta_{i'}' \langle \boldsymbol{\xi}_{k_i}, \boldsymbol{\xi}_{i'} \rangle) \geq 1, i \in [k-p] \end{cases}$$

Compare with Codition 16, the only difference is that we substitute $\lambda_1'\|\boldsymbol{\mu}_1\|^2$ with $\lambda_2'\|\boldsymbol{\mu}_2\|^2$. From the symmetry, we can see that the two conditions are actually the same. Thereofre, we can follow the proof of Situation 1 to prove for Proposition A.53 under this situation.

**Situation 3:** $p \neq 0, k - p \neq 0$

Last we consider the case when wrong tokens come from both clean and noisy sets. Denote the mixed clean samples as $k_1, k_2, ..., k_p$ and the mixed noisy samples as $q_1, q_2, ..., q_{k-p}$. Without losing generality, we assume that $y_{k_i} = +1$ for $i \in [p]$ and $y_{q_i} = -1$ for $i \in [k - p]$, which indicates that their signal tokens are all $\boldsymbol{\mu}_1$. Then the conditions under *Situation 2* become

*Condition* 20 (p clean samples and k-p noisy samples violating optimal token selection).

$$
\begin{cases}
y_i \boldsymbol{v}^\top \boldsymbol{\xi}_i \geq 1, i \in [n] \backslash [k] \\
\boldsymbol{v}^\top \boldsymbol{r}_{k_i} \geq 1, i \in [p] \\
-\boldsymbol{v}^\top \boldsymbol{r}_{q_i} \geq 1, i \in [k - p]
\end{cases}
$$

Denote the max-margin solution under this condition as $\boldsymbol{v}''$ with parameters $\lambda_1'', \lambda_2'', \theta_i''$, we can interpret the condition for parameters:

$$
\begin{cases}
\theta_i'' \cdot \|\boldsymbol{\xi}_{i'}\|^2 + \sum\limits_{i' \neq i} y_i y_{i'} \theta_{i'}'' \langle \boldsymbol{\xi}_i, \boldsymbol{\xi}_{i'} \rangle) \geq 1, i \in [n] \backslash [k] \\
(1 - \beta_i) \lambda_1'' \cdot \|\boldsymbol{\mu}_1\|^2 + \beta_i (\theta_{k_i}'' \cdot \|\boldsymbol{\xi}_{k_i}\|^2 + \sum\limits_{i' \neq k_i} y_{k_i} y_{i'} \theta_{i'}'' \langle \boldsymbol{\xi}_{k_i}, \boldsymbol{\xi}_{i'} \rangle) \geq 1, i \in [p] \\
-(1 - \beta_i) \lambda_1'' \cdot \|\boldsymbol{\mu}_1\|^2 - \beta_i (\theta_{q_i}'' \cdot \|\boldsymbol{\xi}_{q_i}\|^2 + \sum\limits_{i' \neq q_i} y_{q_i} y_{i'} \theta_{i'}'' \langle \boldsymbol{\xi}_{q_i}, \boldsymbol{\xi}_{i'} \rangle) \geq 1, i \in [k - p]
\end{cases}
$$

We consider three cases: $\lambda_1'' \|\boldsymbol{\mu}_1\|^2 \geq 1$, $1 > \lambda_1'' \|\boldsymbol{\mu}_1\|^2 \geq -1$ and $\lambda_1'' \|\boldsymbol{\mu}_1\|^2 < -1$.

- $\lambda_1'' \|\boldsymbol{\mu}_1\|^2 \geq 1$

  First when $\lambda_1'' \|\boldsymbol{\mu}_1\|^2 \geq 1$, we have $\frac{1 - (1 - \beta_i) \lambda_1' \|\boldsymbol{\mu}_1\|^2}{\beta_i} \leq 1$, which indicates that the condition for mixed clean samples' parameter $\theta_{k_i}'$ is relaxed. Meanwhile, for the mixed noisy samples we have

  $$
  -\theta_{q_i}'' \cdot \|\boldsymbol{\xi}_{q_i}\|^2 + \sum\limits_{i' \neq q_i} y_{q_i} y_{i'} \theta_{i'}'' \langle \boldsymbol{\xi}_{q_i}, \boldsymbol{\xi}_{i'} \rangle \geq \frac{1 + (1 - \beta_i) \lambda_1'' \|\boldsymbol{\mu}_1\|^2}{\beta_i} \geq 1,
  $$

  which indicates that the condition is strengthened. Therefore, this case is an extension of the second case of Situation 1 with strengthening some constraints. These constraints will not result in a better solution than Situation 1. The following proof is the same as Situation 1 and we omit it for convenience.

- $1 > \lambda_1'' \|\boldsymbol{\mu}_1\|^2 \geq -1$

  In this case, the constraints for both mixed clean and noisy samples are strengthened. So this can be taken as an extension of the first case in Situation 1 with strengthening some constraints. The following proof is the same as Situation 1 and we omit it for convenience.

- $\lambda_1'' \|\boldsymbol{\mu}_1\|^2 < -1$

  In this case, the constraints are strengthened for mixed clean samples while relaxed for the mixed noisy samples. So we consider it as the extension of Situation 2 when $\lambda_1' \|\boldsymbol{\mu}_1\|^2 < -1$ with strengthening some constraints. The following proof is the same as Situation 2 and we omit it for convenience.

Therefore, we complete the proof for all possible situations. $\qquad\square$

**Training and Test error analysis**

From Proposition A.53 we can derive the convergence direction of $\boldsymbol{p}$ and $\boldsymbol{v}$, i.e. $\boldsymbol{p}_{mm}$ and $\boldsymbol{v}_{mm}$. Note that Theorem A.27 does not depend on the selection of optimal tokens, so it still holds in this case when optimal tokens are noise tokens for all samples. We restate it here for convenience:

**Theorem A.61.** *Suppose that Assumption 4.5 holds, with probability at least $1 - \delta$ on the training dataset, we have*

- *the margin induced by $\boldsymbol{p}_{(r, R)}/R$ in p-SVM is at least $(1 - \zeta)\Xi$, where*

  $$
  \zeta = \frac{\log(4\sqrt{(1 + \kappa)d} \|\boldsymbol{v}_{mm}\|^3 d\rho^2)}{R\Xi}.
  $$

- *the label margin induced by $\boldsymbol{v}_{(r,R)}/r$ in $\nu$-SVM is at least $(1 - \gamma)\Gamma$, where $\gamma = \frac{2\sqrt{(1+\kappa)d}}{\Gamma \exp((1-\zeta)R\Xi)}$.*

Then we could estimate the test error in this case. From Theorem A.61 we have

$$\boldsymbol{p}_{(r,R)}^{\top}(\boldsymbol{\xi}_i - \boldsymbol{\mu}_i) \geq (1 - \zeta)R\Xi, \forall i \in [n] \tag{91}$$

$$y_i \boldsymbol{v}_{(r,R)}^{\top}\boldsymbol{\xi}_i \geq (1 - \gamma)\Gamma r, \forall i \in [n]. \tag{92}$$

Here $\zeta, \gamma, \Xi, \Gamma$ are the same as the definition in Theorem A.61. Similarly, we have the following lemma for $\zeta, \gamma$.

**Lemma A.62.** *Suppose that Assumption 4.5 holds, with probability at least $1 - \delta$ on the training dataset, consider the same setting in Theorem A.27, we have $\zeta < 0.2$ and $\gamma < 0.1$.*

*Proof of Lemma A.62.* First we upper bound $\|\boldsymbol{p}_{mm}\|$. Consider the following possible solution $\widetilde{\boldsymbol{p}}$:

$$\widetilde{\boldsymbol{p}} = \sum_{i \in [n]} 2\frac{\boldsymbol{\xi}_i}{d}. \tag{93}$$

We then proved that $\widetilde{\boldsymbol{p}}$ satisfies (83). For $\forall k \in [n]$, we have

$$\widetilde{\boldsymbol{p}}^{\top}(\boldsymbol{\xi}_k - \boldsymbol{\mu}_k) = \sum_{i \in [n]} 2\frac{\langle \boldsymbol{\xi}_i, \boldsymbol{\xi}_k \rangle}{d} \geq 2(1-\kappa) + \sum_{i \in [n], i \neq k} 2\frac{\langle \boldsymbol{\xi}_i, \boldsymbol{\xi}_k \rangle}{d}$$

$$\geq 2(1-\kappa) + \frac{2n\sqrt{d\log(6n^2/\delta)}}{d} \geq 1.$$

The first and second inequalities are from Lemma A.65; The last inequality is from Assumption 4.5.

Therefore, the max-margin solution $\boldsymbol{p}_{mm}$ must have no greater norm than $\widetilde{\boldsymbol{p}}$. So we can upper bound $\boldsymbol{p}_{mm}$ as

$$\|\boldsymbol{p}_{mm}\|^2 \leq \|\widetilde{\boldsymbol{p}}\|^2 = \frac{4}{d^2}\Big(\sum_{i \in [n]} \|\boldsymbol{\xi}_i\|^2 + \sum_{i,j \in [n], i \neq j} \langle \boldsymbol{\xi}_i, \boldsymbol{\xi}_j \rangle\Big)$$

$$\leq \frac{4}{d^2}\big((1 + \kappa)nd + 2n^2\sqrt{d\log(6n^2/\delta)}\big) \leq \frac{5n}{d}.$$

The second inequality is from Lemma A.65; The last inequality is from the definition of $d$ in Assumption 4.5.

Then from the definition of $\zeta$ in Theorem A.27, we have

$$\zeta = \frac{\log(4\sqrt{(1 + \kappa)d}\|\boldsymbol{v}_{mm}\|^3 d\rho^2)}{R\Xi} \leq C_1 \frac{\sqrt{n/d}}{R}\log(4\sqrt{(1 + \kappa)d}\|\boldsymbol{v}_{mm}\|^3 d\rho^2)$$

$$\leq C_2 \frac{\sqrt{n/d}}{R}\log\left(\frac{n^3}{d}\right) < 0.2.$$

Here $C_1, C_2 = \Theta(1)$. The first inequality is from $\Xi^{-1} = \|\boldsymbol{p}_{mm}\| \leq \sqrt{5n/d}$; The second inequality is from the upper bound of $\|\boldsymbol{v}_{mm}\|$ in Lemma A.56 and the last inequality is from the definition of $R$ in Assumption 4.5. And for $\gamma$, we have

$$\gamma = \frac{2M}{\Gamma \exp((1 - \zeta)R\Xi)} = C_1' \frac{M\|\boldsymbol{v}_{mm}\|}{\exp(R/\|\boldsymbol{v}_{mm}\|)} \leq C_2' \frac{\sqrt{d \cdot (n/d)}}{\exp(R/\sqrt{n/d})} < 0.1.$$

Here $C_1', C_2' = \Theta(1)$. The first inequality is from the lower and upper bound of $\|\boldsymbol{v}_{mm}\|$ in Lemma A.16 and the last inequality is from the definition of $R$ in Assumption 4.1. $\square$

Then we have the following lemma to estimate the innerproduct of $\boldsymbol{p}_{(r,R)}$ and signal token:

**Lemma A.63.** *Suppose that Assumption 4.5 holds, with probability at least* $1 - \delta$ *on the training dataset, we have*

$$|\langle \boldsymbol{p}_{(r,R)}, \boldsymbol{\mu}_j \rangle| \le 0.9(1 - \zeta)R\xi$$

*for* $j \in \{1, 2\}$.

*Proof of Lemma A.63.* First we use contradiction to prove for the lower bound. Assume that $|\langle \boldsymbol{p}_{(r,R)}, \boldsymbol{\mu}_j \rangle| > 0.9(1 - \zeta)R\Xi$. We can estimate $\|\boldsymbol{p}_{(r,R)}\|$ as

$$\|\boldsymbol{p}_{(r,R)}\|^2 > (0.9(1 - \zeta)R\Xi)^2/\rho^2 > (0.5\Xi^2/\rho^2) \cdot R^2 \ge (0.1d/n\rho^2) \cdot R^2 > R^2.$$

The second inequality is from Lemma A.62 ; The third inequality is from $\Xi^2 = \|\boldsymbol{p}_{mm}\|^{-2} \ge d/(5n)$; The last inequality is from our SNR condition $\rho = o(\sqrt{d/n})$. This leads to a contradiction.

$\square$

From Lemma A.31, we can denote $\boldsymbol{v}_{(r,R)}$ as

$$\boldsymbol{v}_{(r,R)} = \lambda_1 \boldsymbol{\mu}_1 + \lambda_2 \boldsymbol{\mu}_2 + \sum_{i \in [n]} y_i \theta_i \boldsymbol{\xi}_i.$$

Denote $\boldsymbol{v}_{\boldsymbol{\xi}} = \sum_{i \in [n]} y_i \theta_i \boldsymbol{\xi}_i$ as the noise part of $\boldsymbol{v}_{(r,R)}$. Then we prove that $\boldsymbol{p}_{(r,R)}, \boldsymbol{v}_{\boldsymbol{\xi}}$ are near orthogonal

**Lemma A.64.** *Suppose that Assumption 4.5 holds, with probability at least* $1 - \delta$ *on the training dataset, we have*

$$|\langle \boldsymbol{p}_{(r,R)}, \boldsymbol{v}_{\boldsymbol{\xi}} \rangle| \le c$$

*for some constant* $c \in (0, 1)$.

*Proof of Lemma A.64.* First plugging in the parameters in $\boldsymbol{v}_{\boldsymbol{\xi}}$ we have

$$
\begin{aligned}
\langle \boldsymbol{p}_{(r,R)}, \boldsymbol{v}_{\boldsymbol{\xi}} \rangle &= \sum_{i \in [n]} y_i \theta_i \boldsymbol{p}_{(r,R)}^\top \boldsymbol{\xi}_i \\
&= \sum_{y_i = +1} \theta_i \boldsymbol{p}_{(r,R)}^\top \boldsymbol{\xi}_i - \sum_{y_i = -1} \theta_i \boldsymbol{p}_{(r,R)}^\top \boldsymbol{\xi}_i \\
&\le (n_{11} + n_{21})(\max_i \theta_i)(R\Xi + O(R\rho)) - (n_{12} + n_{22})(\min_i \theta_i)((1 - \zeta)R\Xi - O(R\rho)) \\
&\le \underbrace{(n/2)(\max_i \theta_i - \min_i \theta_i)R\Xi}_{I_1} + \underbrace{O(\sqrt{n})(\max_i \theta_i)R\Xi}_{I_2} + \underbrace{n(\max_i \theta_i)(\zeta R\Xi + O(R\rho))}_{I_3}.
\end{aligned}
$$

The first inequality is from Theorem A.61 that $(1 - \zeta)R\Xi \le \boldsymbol{p}_{(r,R)}^\top(\boldsymbol{\xi}_i - \boldsymbol{\mu}_i) \le R\Xi$ and $\boldsymbol{p}_{(r,R)}^\top \boldsymbol{\mu}_i = O(R\rho)$ and the second inequality is from Lemma A.68. Then we bound $I_1 \sim I_3$ respectively. For $I_1$, we need to first bound $\theta_i$. From Theorem A.61 we have

$$(1 - \gamma)\Gamma r \le y_i \boldsymbol{v}_{(r,R)}^\top \boldsymbol{\xi}_i \le \Gamma r, \forall i \in [n].$$

Denote $j = \arg\max_i \theta_i$, we have

$$y_j \boldsymbol{v}_{(r,R)}^\top \boldsymbol{\xi}_j \ge \theta_j \|\boldsymbol{\xi}_j\|^2 + n\theta_j \sqrt{d \log(6n^2/\delta)} \ge \theta_j((1 - \kappa)d + n\sqrt{d \log(6n^2/\delta)}).$$

Therefore, we can upper bound $\theta_j$ as

$$\theta_j \le \frac{y_j \boldsymbol{v}_{(r,R)}^\top \boldsymbol{\xi}_i}{(1 - \kappa)d + n\sqrt{d \log(6n^2/\delta)}} \le \frac{\Gamma r}{(1 - \kappa)d + n\sqrt{d \log(6n^2/\delta)}}. \tag{94}$$

Then we can lower bound $\theta_i$ as

$$y_i \boldsymbol{v}_{(r,R)}^\top \boldsymbol{\xi}_i \le \theta_i \|\boldsymbol{\xi}_i\|^2 + n\theta_j \sqrt{d \log(6n^2/\delta)} \le (1 + \kappa)d\theta_i + \frac{\Gamma r n \sqrt{d \log(6n^2/\delta)}}{(1 - \kappa)d + n\sqrt{d \log(6n^2/\delta)}}.$$

Therefore,

$$\theta_i \geq \frac{(1-\gamma)(1-\kappa)\Gamma r d - \gamma \Gamma r n \sqrt{d \log(6n^2/\delta)}}{(1+\kappa)d(1-\kappa)d + n\sqrt{d \log(6n^2/\delta)}}.$$

So we can estimate $I_1$ as

$$I_1 \leq (nR\Xi/2) \cdot \left( \frac{\Gamma r}{(1-\kappa)d + n\sqrt{d \log(6n^2/\delta)}} - \frac{(1-\gamma)(1-\kappa)\Gamma r d - \gamma \Gamma r n \sqrt{d \log(6n^2/\delta)}}{(1+\kappa)d(1-\kappa)d + n\sqrt{d \log(6n^2/\delta)}} \right)$$

$$\leq R\sqrt{nd}/2 \cdot \Gamma r \cdot \left( \frac{1 - \frac{(1-\gamma)(1-\kappa)}{1+\kappa} + \frac{\gamma n \log(6n^2/\delta)}{(1+\kappa)d}}{(1-\kappa)d + n\sqrt{d \log(6n^2/\delta)}} \right)$$

$$\leq Rr(\kappa + \gamma).$$

The second inequality is from $\Xi = \|\boldsymbol{p}_{mm}\| = \Theta(\sqrt{d/n})$ and the last inequality is from $\Gamma = \|\boldsymbol{v}_{mm}\|^{-1} = \Theta(\sqrt{d/n})$.

Then we bound $I_2$. From (94) we have $\max_i \theta_i = \Theta(\Gamma r/d)$. Therefore,

$$I_2 \leq O(\sqrt{n})\Theta(\Gamma r/d)R\Xi \leq Rr \cdot O(1/\sqrt{n}).$$

The last inequality is from $\Gamma, \Xi = \Theta(\sqrt{d/n})$.

Last we bound $I_3$ as

$$I_3 = n\Theta(\Gamma r/d)(\zeta R\Xi + O(R\rho))$$

$$\leq \Theta(r\sqrt{n/d})(\log(4\sqrt{(1+\kappa)d}\|\boldsymbol{v}_{mm}\|^3 d\rho^2) + O(R\rho))$$

$$\leq Rr \cdot O(\rho\sqrt{n/d}).$$

The first inequality is from $\Gamma, \Xi = \Theta(\sqrt{d/n})$ and the last inequality is from Assumption 4.5.

Combining the results above, we have

$$\langle \boldsymbol{p}_{(r,R)}, \boldsymbol{v}_{\boldsymbol{\xi}} \rangle \leq I_1 + I_2 + I_3 \leq Rr \cdot O(\sqrt{1/n} + \rho\sqrt{n/d}) \leq c$$

for sufficiently large $d$ and $n$. Here the last inequality comes from Assumption 4.5. $\qquad\square$

With the lemmas above, we could prove for the main theorem

*Proof of Theorem 4.6.* First we show that the model can perfectly classify all training samples. From Theorem A.27, we have

$$y_i \boldsymbol{v}_{(r,R)}^\top \boldsymbol{r}_i = y_i \beta_i \boldsymbol{v}_{(r,R)}^\top \boldsymbol{\xi}_i + y_i(1-\beta_i)\boldsymbol{v}_{(r,R)}^\top \boldsymbol{\mu}_i \geq \beta_i(1-\gamma)\Gamma r - 0.9(1-\beta_i)(1-\gamma)\Gamma r > 0,$$

for $\forall i \in [n]$. The last inequality is from Lemma A.62. Thus $y_i = \text{sign}(f(\boldsymbol{X}_i; \boldsymbol{v}_{(r,R)}, \boldsymbol{p}_{(r,R)}))$ for all $i \in [n]$.

Then we bound the test error. This is equivalent to estimate $y \cdot f(\boldsymbol{v}_{(r,R)}, \boldsymbol{p}_{(r,R)}; \boldsymbol{X})$ and we could write it as

$$y \cdot f(\boldsymbol{v}_{(r,R)}, \boldsymbol{p}_{(r,R)}; \boldsymbol{X}) = y \cdot \frac{\exp(\langle \boldsymbol{p}_{(r,R)}, \boldsymbol{\mu}' \rangle)\boldsymbol{v}_{(r,R)}^\top \boldsymbol{\mu}' + \exp(\langle \boldsymbol{p}_{(r,R)}, \boldsymbol{\xi}' \rangle)\boldsymbol{v}_{(r,R)}^\top \boldsymbol{\xi}'}{\exp(\langle \boldsymbol{p}_{(r,R)}, \boldsymbol{\mu}' \rangle) + \exp(\langle \boldsymbol{p}_{(r,R)}, \boldsymbol{\xi}' \rangle)}.$$

We first upper bound the term $y \cdot \exp(\langle \boldsymbol{p}_{(r,R)}, \boldsymbol{\mu}' \rangle)\boldsymbol{v}_{(r,R)}^\top \boldsymbol{\mu}'$. From Theorem A.61, the non-optimality of $i$-th sample is

$$1 - \beta_i = \frac{\exp(\langle \boldsymbol{p}_{(r,R)}, \boldsymbol{\mu}_i \rangle)}{\exp(\langle \boldsymbol{p}_{(r,R)}, \boldsymbol{\mu}_i \rangle) + \exp(\langle \boldsymbol{p}_{(r,R)}, \boldsymbol{\xi}_i \rangle)} \leq \frac{1}{1 + \exp((1-\zeta)\Xi R)} \quad \text{for all } i \in [n].$$

The last inequality is from the first statement in Theorem A.61. Consider the sample that contains the same signal token as $\boldsymbol{\mu}'$, we have

$$(1-\beta_i)\boldsymbol{v}_{(r,R)}^\top \boldsymbol{\mu}_i = \frac{\exp(\langle \boldsymbol{p}_{(r,R)}, \boldsymbol{\mu}_i \rangle)\boldsymbol{v}_{(r,R)}^\top \boldsymbol{\mu}_i}{\exp(\langle \boldsymbol{p}_{(r,R)}, \boldsymbol{\mu}_i \rangle) + \exp(\langle \boldsymbol{p}_{(r,R)}, \boldsymbol{\xi}_i \rangle)}.$$

Therefore,

$$y \cdot \exp(\langle \boldsymbol{p}_{(r,R)}, \boldsymbol{\mu}' \rangle) \boldsymbol{v}_{(r,R)}^\top \boldsymbol{\mu}' \leq \exp(\langle \boldsymbol{p}_{(r,R)}, \boldsymbol{\mu}_i \rangle) |\boldsymbol{v}_{(r,R)}^\top \boldsymbol{\mu}_i| \leq \frac{\exp(\langle \boldsymbol{p}_{(r,R)}, \boldsymbol{\mu}_i \rangle) + \exp(\langle \boldsymbol{p}_{(r,R)}, \boldsymbol{\xi}_i \rangle)}{1 + \exp((1-\zeta)\Xi R)} \cdot |\boldsymbol{v}_{(r,R)}^\top \boldsymbol{\mu}_i|$$

$$\leq \frac{2\exp(\langle \boldsymbol{p}_{(r,R)}, \boldsymbol{\xi}_i \rangle)}{\exp((1-\zeta)\Xi R)} \cdot |\boldsymbol{v}_{(r,R)}^\top \boldsymbol{\mu}_i| \leq \frac{2\exp(\Xi R)}{\exp((1-\zeta)\Xi R)} \cdot |\boldsymbol{v}_{(r,R)}^\top \boldsymbol{\mu}_i|$$

$$\leq 2\exp(\zeta \Xi R) \cdot \rho r = (4\sqrt{(1+\kappa)d} \|\boldsymbol{v}_{mm}\|^3 d\rho^2) \cdot \rho r \leq C n^{3/2} \rho^3 r \tag{95}$$

for some constant $C > 0$. Here the third inequality is from $\boldsymbol{p}_{(r,R)}^\top (\boldsymbol{\xi}_i - \boldsymbol{\mu}_i) \geq 0$; The fourth inequality is from the fact that $\langle \boldsymbol{p}_{(r,R)}, \boldsymbol{\xi}_i \rangle \leq \Xi R$ and the last inequality is from $\|\boldsymbol{v}_{(r,R)}\| \leq r, \|\boldsymbol{\mu}_i\| \leq \rho$. Then we can bound the test error as

$$\mathbb{P}(y \cdot f(\boldsymbol{v}_{(r,R)}, \boldsymbol{p}_{(r,R)}; \boldsymbol{X}) \leq 0) = \mathbb{P}(y \cdot \exp(\langle \boldsymbol{p}_{(r,R)}, \boldsymbol{\mu}' \rangle) \boldsymbol{v}_{(r,R)}^\top \boldsymbol{\mu}' + y \cdot \exp(\langle \boldsymbol{p}_{(r,R)}, \boldsymbol{\xi}' \rangle) \boldsymbol{v}_{(r,R)}^\top \boldsymbol{\xi}' \leq 0)$$

$$\geq \mathbb{P}(y \cdot \exp(\langle \boldsymbol{p}_{(r,R)}, \boldsymbol{\xi}' \rangle) \boldsymbol{v}_{(r,R)}^\top \boldsymbol{\xi}' \leq -C n^{3/2} \rho^3 r)$$

$$\geq \frac{1}{4} \mathbb{P}\left( y \boldsymbol{v}_{\boldsymbol{\xi}}^\top \boldsymbol{\xi}' \leq -e^{-R/C} \cdot C n^{3/2} \rho^3 r \mid \langle \boldsymbol{p}_{(r,R)}/R, \boldsymbol{\xi}' \rangle \in [1/C, C] \right)$$

$$\geq \frac{1}{4}\left( \frac{1}{2} - \frac{cC + C\exp(-R/C)n^{3/2}\rho^3}{\sqrt{2\pi(1-c^2)}} \right) \geq \frac{1}{16}.$$

The first inequality is from (95); the second inequality use the fact that there exists a constant $C > 0$ such that $\mathbb{P}(N(0,1) \in [1/C, C]) \geq 1/4$; the third inequality comes from Lemma A.69 and the last inequality uses Assumption 4.5. $\qquad \square$

.

## A.3  Supplement Lemmas

Here we list some technical lemmas for the main proof.

**Lemma A.65.** *(Properties of Training Data) Suppose that $\delta > 0$. Then exists some constant $c_D > 0$ (that depends just on the number of tokens $T$) and $\kappa \leq c_D\sqrt{\log(n/\delta)/d} = \widetilde{O}(1/\sqrt{d})$ such that with probability at least $1 - \delta$, we have*

$$(1-\kappa)d \leq \|\boldsymbol{\xi}_{i,\tau}\|_2^2 \leq (1+\kappa)d, \forall i \in [n], \tau \in \{2, \ldots, T\}$$

$$|\langle \boldsymbol{\xi}_{i,\tau}, \boldsymbol{\xi}_{j,\tau'} \rangle| \leq c_D\sqrt{d\log(n/\delta)} \ \forall i, j \in [n], \tau, \tau' \in \{2, \ldots, T\} \ s.t.(i,\tau) \neq (j, \tau').$$

*Proof of Lemma A.65.* Note that $\mathbb{E}[\|\boldsymbol{\xi}_{i,\tau}\|^2] = d$, then by Bernstein's inequality (see Theorem 2.8.1 in Vershynin [35]), with probability at least $1 - \delta/(3n)$ we have

$$|\|\boldsymbol{\xi}_{i,\tau}\|_2^2 - d| \leq c_1 \cdot \sqrt{d\log(n/\delta)},$$

where $c_1$ is some universal constant. Therefore, for $\kappa \leq c_1\sqrt{\log(n/\delta)/d}$ we have that

$$(1-\kappa)d \leq \|\boldsymbol{\xi}_{i,\tau}\|_2^2 \leq (1+\kappa)d.$$

Moreover, $\langle \boldsymbol{\xi}_{i,\tau}, \boldsymbol{\xi}_{j,\tau'} \rangle$ has mean zero for any $i, j \in [n], \tau, \tau' \in \{2, \ldots, T\}$ such that $(i,\tau) \neq (j,\tau')$. By Bernstein's inequality, with probability at least $1 - \delta/(3T^2 n^2)$ we have

$$|\langle \boldsymbol{\xi}_{i,\tau}, \boldsymbol{\xi}_{j,\tau'} \rangle| \leq c_2\sqrt{d\log(n/\delta)},$$

where $c_2$ is some universal constant. Applying a union bound and setting $c_D = \max(c_1, c_2)$ completes the proof. $\qquad \square$

Following Lemma A.65 we conclude the next remark:

*Remark* A.66. (Properties of New Test Sample) Let $(\boldsymbol{X} = (\boldsymbol{\mu}_k, \boldsymbol{\xi}_2, \ldots, \boldsymbol{\xi}_\tau), y) \sim \mathcal{D}$. Then exists universal constant $c_D$ such that for any $C_1 > 0$, with probability at least $1 - n\exp(-d/C_1^2 c_D^2 n)$, we have

$$|\langle \boldsymbol{\xi}_\tau, \boldsymbol{\xi}_{i,\tau'}\rangle| \leq \frac{d}{C_1}$$

for any $i \in [n]$ and $\tau, \tau' \in \{2, \ldots, T\}$.

*Proof.* Similarly to the proof of Lemma A.65, exists some constant $c_D$ such that with probability at least $1 - \delta$ we have that

$$|\langle \boldsymbol{\xi}_{i,\tau}, \boldsymbol{\xi}_{j,\tau'}\rangle| \leq c_D\sqrt{d\log(n/\delta)}$$

Setting $\delta = n\exp(-d/c_D^2 C_1^2)$ completes the proof. $\qquad\square$

**Lemma A.67** (Properties of Combined Noise Tokens). *Suppose that* $\delta > 0$ *and* $\kappa' = O(\sqrt{\log(6n/\delta)/d}) = \widetilde{O}(1/\sqrt{d})$ *.If Lemma A.65 holds, we have*

$$(1 - \kappa')d/T \leq \|\overline{\boldsymbol{\xi}}_i\|_2^2 \leq (1 + \kappa')d$$
$$|\langle \overline{\boldsymbol{\xi}}_i, \overline{\boldsymbol{\xi}}_j\rangle| \leq 2T^2\sqrt{d\log(6n^2/\delta)}$$

*for any* $i, j \in [n]$.

*Proof of Lemma A.34.* Note that $\overline{\boldsymbol{\xi}}_i = \sum_{\tau=2}^T t_{i,\tau}\boldsymbol{\xi}_{i,\tau}$ and $\sum_{\tau=2}^T t_{i,\tau} = 1$. Lemma A.65 holds for each noise token $\boldsymbol{\xi}_{i,\tau}$, so for the composed noise token we have,

$$
\begin{aligned}
\|\overline{\boldsymbol{\xi}}_i\|_2^2 &= \|\sum_{\tau=2}^T t_{i,\tau}\boldsymbol{\xi}_{i,\tau}\|_2^2 \\
&\geq \sum_{\tau=2}^T t_{i,\tau}^2 \|\boldsymbol{\xi}_{i,\tau}\|_2^2 - \sum_{\tau_1=2}^T \sum_{\tau_2\neq\tau_1} |t_{\tau_1}t_{\tau_2}\langle\boldsymbol{\xi}_{i,\tau_1},\boldsymbol{\xi}_{i,\tau_2}\rangle| \\
&\geq (1-\kappa)d\sum_{\tau=2}^T t_{i,\tau}^2 - 2\sqrt{d\log(6n^2/\delta)}\sum_{\tau_1=2}^T \sum_{\tau_2\neq\tau_1} |t_{\tau_1}t_{\tau_2}| \\
&\geq (1-\kappa)d/T - 2\sqrt{d\log(6n^2/\delta)} \cdot T^2 O(1) \\
&\geq (1-\kappa')d/T.
\end{aligned}
$$

The first inequality is from triangle inequality; The second inequality is from Lemma A.65; The third inequality is from Cauchy–Schwarz inequality that $\sum_{\tau=2}^T t_{i,\tau}^2 \cdot T \geq (\sum_{\tau=2}^T t_{i,\tau})^2 = 1$ and $t_{\tau_1}, t_{\tau_2} = O(1)$; The last inequality is from the definition $\kappa'$.

To upper bound $\|\overline{\boldsymbol{\xi}}_i\|_2^2$, we have

$$
\begin{aligned}
\|\overline{\boldsymbol{\xi}}_i\|_2^2 &= \|\sum_{\tau=2}^T t_{i,\tau}\boldsymbol{\xi}_{i,\tau}\|_2^2 \\
&\leq \sum_{\tau=2}^T t_{i,\tau}^2 \|\boldsymbol{\xi}_{i,\tau}\|_2^2 + \sum_{\tau_1=2}^T \sum_{\tau_2\neq\tau_1} |t_{\tau_1}t_{\tau_2}\langle\boldsymbol{\xi}_{i,\tau_1},\boldsymbol{\xi}_{i,\tau_2}\rangle| \\
&\leq (1+\kappa)d\sum_{\tau=2}^T t_{i,\tau}^2 + 2\sqrt{d\log(6n^2/\delta)}\sum_{\tau_1=2}^T \sum_{\tau_2\neq\tau_1} |t_{\tau_1}t_{\tau_2}| \\
&\leq (1+\kappa)d + 2\sqrt{d\log(6n^2/\delta)} \cdot T^2 O(1) \\
&\leq (1+\kappa')d.
\end{aligned}
$$

The first two inequalities are similar as above; The third inequality is from $t_{i,\tau}^2 \leq t_{i,\tau}$ for $t_{i,\tau} \in [0,1]$, so $\sum_{\tau=2}^T t_{i,\tau}^2 \leq \sum_{\tau=2}^T t_{i,\tau} = 1$; The last inequality is from the definition of $\kappa'$.

Last we consider the innerproduct of composed noise tokens. For $\forall i, j \in [n], i \neq j$ we have

$$|\langle \bar{\boldsymbol{\xi}}_i, \bar{\boldsymbol{\xi}}_j \rangle| = |\sum_{\tau_1=2}^{T} \sum_{\tau_2=2}^{T} t_{i,\tau_1} t_{j,\tau_2} \langle \boldsymbol{\xi}_{i,\tau_1}, \boldsymbol{\xi}_{j,\tau_2} \rangle|$$

$$\leq 2\sqrt{d \log(6n^2/\delta)} \cdot |\sum_{\tau_1=2}^{T} \sum_{\tau_2=2}^{T} t_{i,\tau_1} t_{j,\tau_2}|$$

$$\leq 2T^2 \sqrt{d \log(6n^2/\delta)}.$$

The first inequality is from triangle inequality and the second inequality is due to $t_{i,\tau_1}, t_{j,\tau_2} \in [0,1]$. This completes the proof. $\qquad\square$

**Lemma A.68.** *With probability at least $1 - 6\delta$,*

$$\left||\mathcal{C}| - n(1-\eta)\right| \leq \sqrt{n \log(\frac{1}{\delta})}; \quad \left||\mathcal{N}| - n\eta\right| \leq \sqrt{n \log(\frac{1}{\delta})};$$

$$\left||\mathcal{C}_i| - \frac{n(1-\eta)}{2}\right| \leq \sqrt{n \log(\frac{1}{\delta})}; \quad \left||\mathcal{N}_i| - \frac{n\eta}{2}\right| \leq \sqrt{n \log(\frac{1}{\delta})}, \quad i = 1, 2.$$

*Proof.* Note that $|\mathcal{C}| \sim \text{Binom}(n, 1-\eta)$. Applying Hoeffding's inequality, we have

$$\mathbb{P}(\left||\mathcal{C}| - (1-\eta)n\right| > t) \leq 2 \exp(-\frac{2t^2}{n}).$$

Let $t = \sqrt{n \log(1/\delta)}$. We have that with probability at least $1 - \delta$,

$$\left||\mathcal{C}| - (1-\eta)n\right| \leq \sqrt{n \log(\frac{1}{\delta})}.$$

Similarly, note that $|\mathcal{N}| \sim \text{Binom}(n, \eta), |\mathcal{C}_1| \sim \text{Binom}(n, (1-\eta)/2), |\mathcal{C}_2| \sim \text{Binom}(n, (1-\eta)/2), |\mathcal{N}_1| \sim \text{Binom}(n, \eta/2)$ and $|\mathcal{N}_2| \sim \text{Binom}(n, \eta/2)$, we have that each of the following events holds with probability at least $1 - \delta$:

$$\left||\mathcal{C}| - n(1-\eta)\right| \leq \sqrt{n \log(\frac{1}{\delta})}; \quad \left||\mathcal{N}| - n\eta\right| \leq \sqrt{n \log(\frac{1}{\delta})};$$

$$\left||\mathcal{C}_i| - n(1-\eta)/2\right| \leq \sqrt{n \log(\frac{1}{\delta})}, \quad i = 1, 2;$$

$$\left||\mathcal{N}_i| - n\eta/2\right| \leq \sqrt{n \log(\frac{1}{\delta})}, \quad i = 1, 2.$$

$\qquad\square$

**Lemma A.69.** *Suppose $X \sim N(0, \boldsymbol{I}_d)$, and $\boldsymbol{v}, \boldsymbol{p} \in \mathbb{R}^d$ are two vectors with $\|\boldsymbol{v}\| = \|\boldsymbol{p}\| = 1, \boldsymbol{v}^\top \boldsymbol{p} \leq c$ for some constant $c \in (0, 1)$. Given some constant $C > 1$, for $z < 0$,*

$$\mathbb{P}(\boldsymbol{v}^\top X < z | \boldsymbol{p}^\top X \in [1/C, C]) \geq \frac{1}{2} - \frac{1}{\sqrt{2\pi}} \frac{cC - z}{\sqrt{1 - c^2}}.$$

*Proof of Lemma A.69.* Denote $x_v = v^\top X \sim N(0, 1), x_p = \boldsymbol{p}^\top X \sim N(0, 1)$. Then we have $x_v, x_p \sim \mathcal{N}(0, 1)$. Denote the covariance between $x_v, x_p$ by $c_0$, then we have

$$c_0 = \text{Cov}(x_v, x_p) = \boldsymbol{v}^\top \text{Cov}(X) \boldsymbol{p} = \boldsymbol{v}^\top \boldsymbol{p} \leq c.$$

Note that

$$x_v \overset{d}{=} c_0 x_p + \sqrt{1 - c_0^2} r,$$

where $r \sim N(0, 1)$ is independent of $x_p$. It follows that

$$\mathbb{P}(x_v < z | x_p \in [\frac{1}{C}, C]) = \mathbb{P}(r < \frac{z - c_0 x_p}{\sqrt{1 - c_0^2}} | x_p \in [\frac{1}{C}, C]) \geq \mathbb{P}(r < \frac{z - cC}{\sqrt{1 - c^2}}) \geq \frac{1}{2} - \frac{1}{\sqrt{2\pi}} \frac{cC - z}{\sqrt{1 - c^2}}.$$

$\qquad\square$

## A.4 Additional Experiments

In this section, we present additional experiments, including various architectures (different from Eq. (3)), alternative relationships between parameters (e.g., deviating from Assumption 3.1), and gradient descent with weight decay, and more. Additionally, all experiments were conducted on a single NVIDIA T4 GPU with 16GB memory. Each individual run was completed in under a minute. The total compute cost for all experiments presented in the paper and appendix is negligible. No parallelization or distributed training was used.

In Figure 3 we use the same settings and parameters as in the main paper (i.e. as in Figure 2), but with a smaller step size. In contrast to Theorem 3.3 and Figure 2, benign overfitting occurs after about 150 iterations. This provides empirical validation for Remark 5.1.

In Figure 4, we explore the relationship between the number of samples $n$ and the input dimension $d$. We observe that even when $d \approx n$, the model exhibits benign overfitting. For intermediate values of $d$, the model demonstrates harmful overfitting, where the test error increases. For smaller values of $d$, the model fails to fit the data.

In Figure 5, we examine **self-attention with respect to the first token**, as in [7]. Here, the model exhibits benign overfitting with behavior similar to that observed under self-attention with respect to tunable token (Remark 5.1). Additionally, the attention probabilities are consistent with those in the tunable token attention model (see the resemblance to Figure 2).

In Figure 6 and 7, we set the number of tokens to $T = 5$. The results show benign overfitting after two iterations. Furthermore, for clean samples, the softmax probability of the signal token, $s_{j,1}^t$, dominates the overall attention. In contrast, for noisy samples, the softmax probabilities of the noise tokens, $\sum_{\tau=2}^{T} s_{j,\tau}^t$, dominate. This align with Thm. 3.3.

In Figure 8, we use the same settings and parameters as in Figure 2, but with **Gaussian initialization** instead of zero initialization. The results remain consistent, providing empirical validation for Remark 3.2 that states that zero initialization is without loss of generality.

In figure 9, we consider a **four-layer attention model** defined as $f(\boldsymbol{X}) = f_1(f_2(f_3(f_4(\boldsymbol{X}))))$, where $f_i : \mathbb{R}^{(T+1) \times d} \to \mathbb{R}^{(T+1) \times d}$ for $i \in \{2, 3, 4\}$ is defined in Eq. (1) and $f_1 : \mathbb{R}^{(T+1) \times d} \to \mathbb{R}$ is defined in Eq. (3). The results show benign overfitting after roughly 20 iterations. Moreover, the softmax probabilities in the first layer align with the behavior observed in the single-layer model.

In Figure 10, we consider a **four-head attention model** which concatenates the results from all heads as the attention output. The results show benign overfitting after roughly 2 iterations. Moreover, the softmax probabilities in the first layer align with the behavior observed in the single-layer model.

In Figure 11, we examine **GD with weight decay**, which encourages norm minimization (or margin maximization). Here, benign overfitting occurs after approximately 150 iterations. Also here the attention mechanism continues to separate signal tokens from noise tokens.

To further validate our theoretical findings, we conducted additional experiments on real-world datasets, including MNIST and CIFAR-10. In both cases, we trained a one-layer Transformer model ($d = 1024$) to perform binary classification. Since the signal-to-noise ratio (SNR) is fixed for each dataset, we varied the training sample size $n$ to examine how train and test accuracy evolve with $n$. The results in table 2 and 3 indicate that while training accuracy remains near 100%, the test accuracy improves as $n$ increases. This corresponds to our SNR threshold $\Theta(1/\sqrt{n})$ that determines the transition between benign and harmful overfitting.

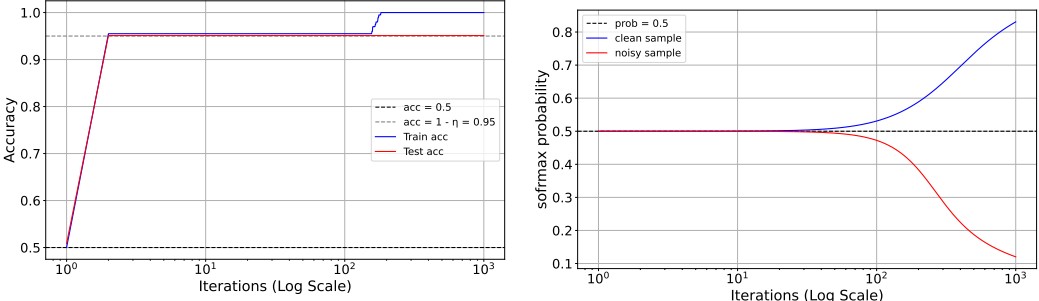

(a) train and test accuracy          (b) attention weights on signal token

Figure 3: The left panel shows train and test accuracies during training with a small step size. The clean training samples are correctly classified already after one iteration, but in contrast to Theorem 3.3 and Figure 2, benign overfitting occurs after about 150 iterations. In the right panel, we see that the attention starts separating signal and noise tokens shortly before benign overfitting occurs. Parameters: $n = 200, d = 40000, T = 2, \beta = 0.0001, \rho = 30, \eta = 0.05$, test sample size $= 2000$.

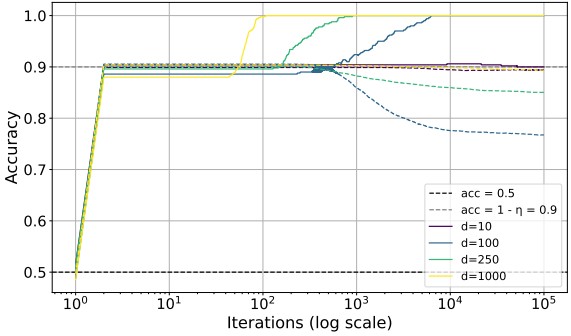

Figure 4: Comparing train (solid lines) and test (dashed lines) accuracies with different dimensions. Here, we see that for small $d$ (purple line), the model is unable to fit the data (at least in the first $10^5$ first iterations), and both the train and test accuracies are at the noise-rate level. For intermediate values of $d$ (green and blue lines), the model exhibits harmful overfitting, and for larger $d$ (yellow line) the model exhibits benign overfitting. We note that benign overfitting occurs here for $d = 2n \ll n^2$, which suggests that the assumptions on $d$ in our theorems are loose. Parameters: $n = 500, \beta = 0.02, T = 5, \rho = 30, \eta = 0.1$, test sample size $= 10000$.

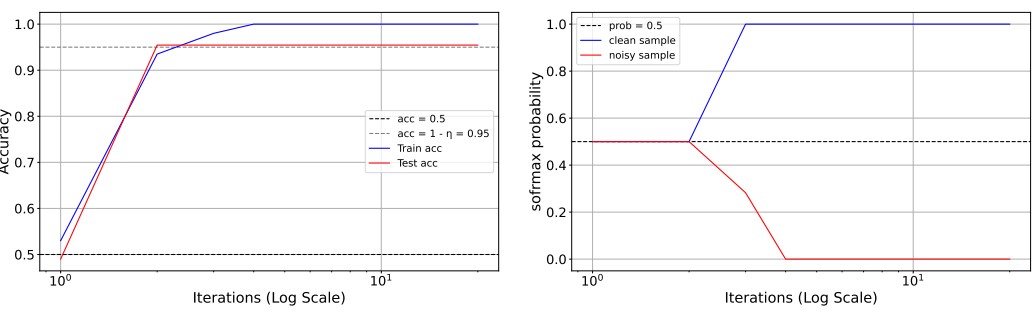

(a) train and test accuracy          (b) attention weights on signal token

Figure 5: **Self-attention experiments**. The model: $\boldsymbol{X} \to \boldsymbol{v}^\top \boldsymbol{X}^T \mathbb{S}(\boldsymbol{X} \boldsymbol{W} \boldsymbol{x}^{(1)})$, same as Vasudeva et al. [7]. The left panel shows the train and test accuracies during training. It shows that benign overfitting also occurs after 2 iterations. In the right panel, we show the softmax probability of the signal token for clean and noisy samples (average of the softmax probabilities $s_{j,1}^t$ over $\mathcal{C}$ and $\mathcal{N}$ respectively). We see that after 2 iterations, the attention focuses on signal tokens for clean examples, and on noise tokens for noisy examples. This indicates that our results also capture the behavior in a self-attention mechanism. Parameters: $n = 200, d = 40000, \text{T=2}, \beta = 0.025, \rho = 20, \eta = 0.05$, test sample size $= 2000$.

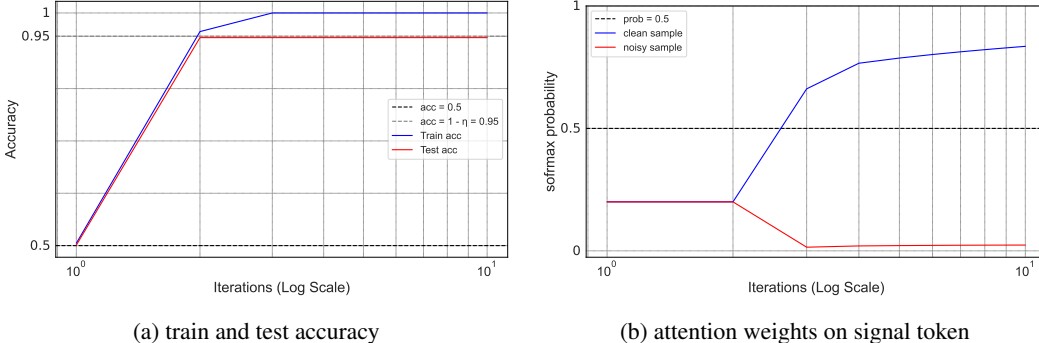

(a) train and test accuracy        (b) attention weights on signal token

Figure 6: The left panel shows train and test accuracies during training. It shows that benign overfitting occurs after 2 iterations. In the right panel, we see that after 2 iterations, the attention focuses on signal tokens for clean examples, and on noise tokens for noisy examples. Parameters: $n = 200, d = 40000, T = 5, \beta = 0.1, \rho = 40, \eta = 0.05$, test sample size $= 2000$.

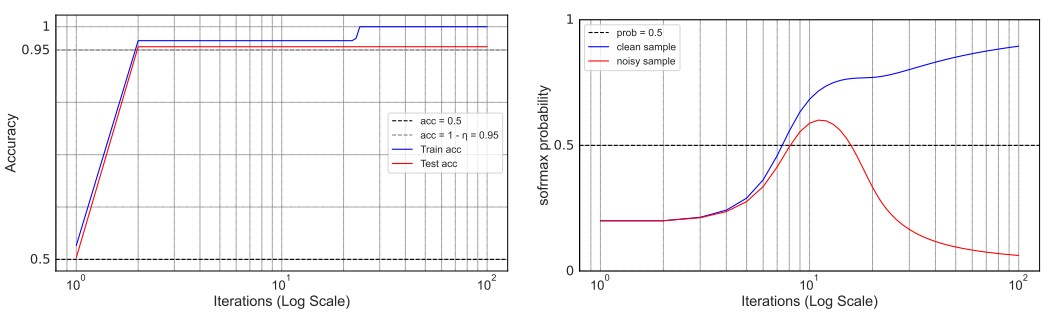

(a) train and test accuracy        (b) attention weights on signal token

Figure 7: The left panel shows train and test accuracies during training with a small step size. The clean training samples are correctly classified already after one iteration, but benign overfitting occurs after about 22 iterations. In the right panel, we see that the attention starts separating signal and noisy tokens shortly before benign overfitting occurs. Parameters: $n = 200, d = 40000, T = 5, \beta = 0.003, \rho = 50, \eta = 0.05$, test sample size $= 2000$.

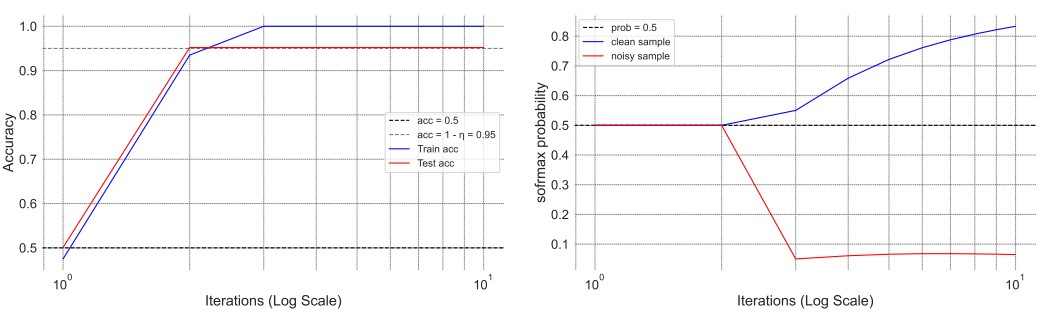

(a) train and test accuracy        (b) attention weights in signal token

Figure 8: The left panel shows the train and test accuracies during training (with **Gaussian initialization**, where each entry has variance 0.01). As in Figure 2, It shows that benign overfitting occurs after 2 iterations. After the first iteration, the model correctly classifies the clean training examples, but not the noisy ones. In the right panel, we show the softmax probability of the signal token for clean and noisy samples (average of the softmax probabilities $s_{j,1}^t$ over $\mathcal{C}$ and $\mathcal{N}$ respectively). We see that after 2 iterations, the attention focuses on signal tokens for clean examples, and on noise tokens for noisy examples. This aligns with Theorem 3.3. Parameters: $n = 200, d = 40000, \beta = 0.025, \rho = 30, \eta = 0.05$, test sample size $= 2000$.

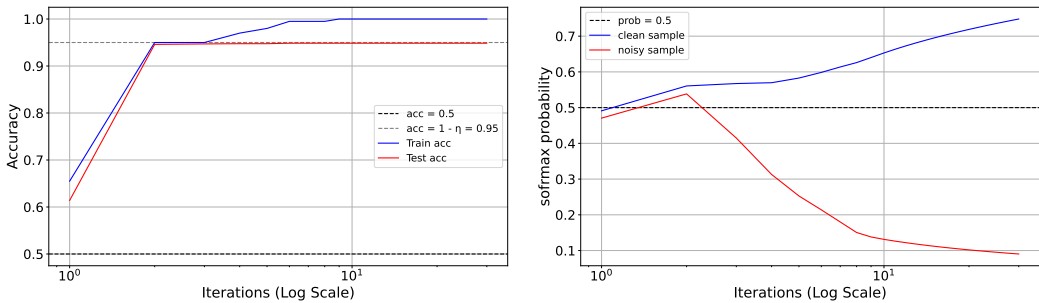

(a) train and test accuracy

(b) attention weights on signal token

Figure 9: **Multi-layer experiments**. The left panel shows the train and test accuracies during training in a 4-layer single-head attention model. It shows that benign overfitting occurs after roughly 20 iterations. After the first iteration, the model correctly classifies the clean training examples, but not the noisy ones. In the right panel, we show the softmax probability of the signal token for clean and noisy samples (average of the softmax probabilities $s_{j,1}^t$ over $\mathcal{C}$ and $\mathcal{N}$ respectively) in the first layer. We see that the attention focuses on signal tokens for clean examples, and on noise tokens for noisy examples. This indicates that our results essentially capture the behavior also in multi-layer models. Parameters: $n = 200, d = 10000, T = 2, \beta = 0.025, \rho = 20, \eta = 0.05$, test sample size $= 2000$.

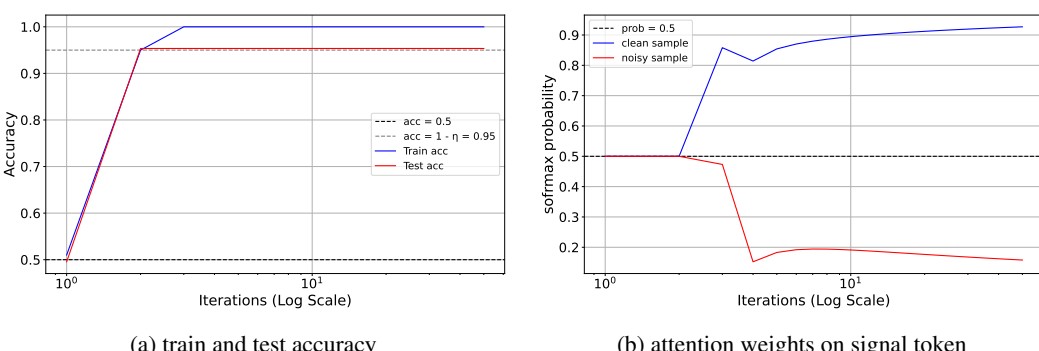

(a) train and test accuracy

(b) attention weights on signal token

Figure 10: **Multi-head experiments.** The left panel shows train and test accuracies during training in a 4-head attention model. The clean training samples are correctly classified already after one iteration, and benign overfitting occurs after 2 iterations. In the right panel, we see that after 2 iterations, the attention focuses on signal tokens for clean examples, and on noise tokens for noisy examples. Parameters: $n = 200, d = 10000, T = 2, \beta = 0.3, \rho = 15, \eta = 0.05$, test sample size $= 2000$.

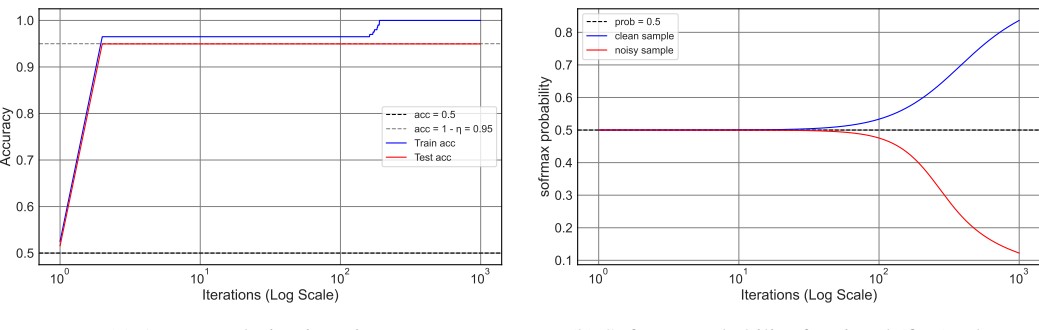

(a) Accuracy during iterations

(b) Softmax probability for signal (first) token

Figure 11: The left panel shows train and test accuracies during training with GD with **weight decay**. The clean training samples are correctly classified already after one iteration, and benign overfitting occurs after about 150 iterations. In the right panel, we see that the attention starts separating signal and noise tokens shortly before benign overfitting occurs. Parameters: weight decay $= 0.01$, $n = 200, d = 40000, T = 2, \beta = 0.0001, \rho = 30, \eta = 0.05$, test sample size $= 2000$.

| Eval | Training Size $n$ | | | |
|---|---|---|---|---|
| | 80 | 400 | 800 | 2000 |
| Train acc | 100% | 100% | 99.9% | 99.9% |
| Test acc (on clean data) | 87.6% | 91.9% | 94.1% | 95.8% |

Table 2: Training and test accuracy on label-noisy ($\eta = 0.1$) MNIST dataset for 500 iterations in one-layer, two-head Transformers.

| Eval | Training Size $n$ | | | |
|---|---|---|---|---|
| | 40 | 400 | 4000 | 40000 |
| Train acc | 100% | 100% | 100% | 100% |
| Test acc (on clean data) | 77.6% | 80.9% | 86.6% | 88.9% |

Table 3: Training and test accuracy on label-noisy ($\eta = 0.05$) CIFAR-10 dataset for 500 iterations in one-layer, four-head Transformers.

