# OpenReview forum: "Benign Overfitting in Single-Head Attention"
_NeurIPS.cc/2025/Conference — NeurIPS 2025 poster_

### Official Review · Reviewer_zipP · 2025-06-24

**Clarity:** 4
**Significance:** 2
**Originality:** 3
**Rating:** 4
**Confidence:** 4

**Summary:**

This paper studies the benign overfitting phenomenon in single head attention. Specifically, when signal-to-noise ratio (SNR) is larger than $\Omega(1/\sqrt{n})$, the test loss will be small after training.
To show that the requirement on SNR is tight, this paper provides a harmful overfitting result when SNR is smaller than $O(1/\sqrt{n})$ in the max-margin scenario.
The experimental results verifies the theoretical results.

**Questions:**

Q1: Line 172, this paper requires $d \ge C \cdot n^2 \log (n / \delta) / \eta^2$, then as $\eta \rightarrow 0$, we have $d \rightarrow \infty$. Is the method in this paper able to analyze the situation when $\eta = 0$ (no lable flipping) ?


Q2: The condition of benign overfitting is determined by SNR and n. Would it be related to label flipping rate $\eta$ ?

In my intuition, the $\Omega(1/\sqrt{n})$ requirement may become more specific if $\eta$ is considered.


Q3: This paper only validate the theoretical results on sythetic datasets.It is recommended that the authors consider more experiments on real datasets, such as MNIST and CIFAR10.

**Ethical Concerns:**

["NO or VERY MINOR ethics concerns only"]

**Final Justification:**

This work studies benign overfitting for Transformer from a distinctive perspective compared to other works. I am glad to see the experiments on MNIST and CIFAR, and the relaxed assumption on $d$.

**Limitations:**

yes

**Quality:**

3

**Strengths And Weaknesses:**

strengths:

1. This paper demonstrates the importance of SNR for generalisation through rigorous theoretical analysis, and characterizes the boundary between benign overfitting and harmful overfitting.

2. The theoretical result are verified by the experiments (especially the heatmap).

3. The writing is clear and easy to follow, all the assumptions and theorem are well structured.

weaknesses:

1. The assumptions are oversimplified: The original attention mechanism consists of the QKV matrices, but this paper reduces them to two vectors, making it misaligned with reality.

2. Lack of training dynamics: This paper only consider two step of GD in Section 3. It is better to consider a smaller step size to characterize the dynamics of $v$ and $p$.

3. The benign overfitting phenomenon has been studied in [1] and [2], thus this paper seems not novel.

4. In terms of gradient descent (Section 3), no harmful overfitting results are provided in this paper.

references

[1] Jiang, J., Huang, W., Zhang, M., Suzuki, T., & Nie, L. Unveil benign overfitting for transformer in vision: Training dynamics, convergence, and generalization. In NeurIPS 2024.

[2] Sakamoto, K., & Sato, I. Benign or not-benign overfitting in token selection of attention mechanism. In ICML 2025.

---

> ### Author Rebuttal · Authors · 2025-07-31
>
> We thank the reviewer for their thoughtful feedback. Below, we respond to each concern and question in order.
>
> **W1:**  First, we chose to combine the key and query matrices to simplify the analysis, which is already technically involved. We note that this simplification is very common in prior works studying attention mechanisms. For example, [1,2,3,4,5] (as well as many other papers) also used a combined key-query matrix.  Then, following many prior works on classification (or regression) tasks in transformer, we introduce a classification token (CLS token or prompt), whose corresponding output is used for prediction, which leads us to Eq. 2, that is, $f(X; W, v) = v^\top X^\top \mathbb{S}(X W^\top q)$, that also has been studied in prior works (e.g., [1,2,6] and more). We then follow the approach of [5], which showed that optimizing W with a fixed q is equivalent to optimizing q and removing W, as done in our paper. We note that in Section 3 of that paper, they are optimizing both v and q, exactly as we do. We also note that our model in Eq. 3 can be obtained by fixing $q=(1,0,\ldots,0)^\top$ in Eq. 2, and then the vector $p$ from Eq. 3 is simply the first column of $W$ in Eq. 2.
>
> **W2:** We agree that considering a smaller step size and analyzing the trajectory for more than two steps is interesting. However, analyzing even two steps of GD in our setting is challenging. This analysis gives us many insights into how attention handles noisy label data, how the attention focuses on the signal token for clean samples and on the noisy tokens for noisy samples, sufficient and necessary conditions to allow benign overfitting, and the relation between different parameters. It also provides insights into how the trajectory looks after more steps of GD (see Remark 5.1, line 371), which we validate in our experiments (Figure 3 in the appendix). Our experimental results suggest that our theoretical findings extend beyond the two-step setting, demonstrating consistency with longer training and a broader range of learning rates. While a full trajectory analysis is an interesting direction, our results already highlight fundamental phenomena.
>
> Moreover, we prove that min-norm/max-margin solutions also exhibit benign overfitting, where these solutions roughly correspond to the convergence of GD with (a small) weight decay. Indeed, GD with weight decay tends (after enough iterations) to solutions that minimize the loss while preserving a small weight norm. We also refer the reviewer to lines 232-238, which provide additional motivation for analyzing min-norm/max-margin solutions.
>
> **W3:** Benign overfitting has been studied across a variety of architectures, including CNNs, MLPs, and linear models. Given the widespread interest in Transformers, we believe it is natural and valuable for multiple studies, including ours, to explore benign overfitting in this context. We view this as a sign of growing interest in the topic, and believe our contributions offer a significant and distinctive perspective.
>
> Although [7], [6], and our paper all study the benign overfitting phenomenon in attention mechanisms, there are several fundamental differences in setting, assumptions, and theoretical scope. Below we provide a detailed comparison to clarify the novelty of our contribution.
>
> - Jiang et al. [7] do not incorporate label noise in their analysis. The presence of label noise is not a minor extension but a \textbf{crucial element} in understanding whether interpolation is compatible with generalization. In fact, label noise is central to the definition of benign overfitting in the literature and appears in almost all relevant works, with only a few exceptions such as [8], and [9].
> Without label noise, many existing results on benign overfitting can be trivially explained through standard uniform convergence arguments. For example, consider the classical result of Bartlett et al. [10] on benign overfitting in linear regression, where they proved under what conditions on the data covariance matrix minimum-norm interpolators exhibit benign overfitting. In linear regression, generalization in a realizable setting with a sufficiently large dataset follows from standard norm-based uniform convergence bounds. Hence, [10]’s result is challenging and interesting mainly because it considers label noise.
>
> - The work of Sakamoto et al. [6] (which was developed in parallel to our work) fixes the attention head $v$ and performs gradient descent only on the attention weights. They also make assumptions on the correlation between the fixed attention head $v$ and the signal token to ensure benign overfitting happens. Their assumption on $v$ holds with exponentially small probability (in the dimension $d$) if $v$ is drawn from a spherically symmetric distribution. In contrast, our setting allows both the attention vector $p$ and the prediction head $v$ to be fully trainable, and we analyze their joint optimization dynamics.
>
> Finally, both works show that there exists a training step where the model achieves a low test loss and zero training error, but they do not characterize behavior beyond that point. In contrast, we go further by analyzing the behavior of the min-norm/max-margin solution, which can be viewed as a fully converged solution, since it corresponds to the convergence of GD with (a small) weight decay. We prove that the min-norm or max-margin interpolator also exhibits benign overfitting, and thus our results describe not just a transient stage but also the asymptotic behavior of a trained model.
>
> **W4:** We can show harmful overfitting results in the first two steps (when SNR = $o(1/\sqrt{n})$). Roughly speaking, the attention will be focused on the noisy tokens, both for clean and noisy examples, which leads to a small training error but poor generalization. We will add a remark about that in the revised version. Moreover, our experiments (specifically Figure 1) indicate that for SNR = $o(1/\sqrt{n})$, harmful overfitting persists even after many training steps.
>
> **Q1:** Our GD analysis focuses on the case where there is at least one noisy sample, which occurs with high probability as long as $\eta \gg 1/n$, where $n$ is the number of training samples. When there are no noisy examples, the problem becomes significantly simpler and has already been addressed in prior work, such as the GD result of Jiang et al.. More broadly, the typical regime studied in the benign overfitting literature is when the label flipping rate $\eta$ is large enough (typically, a constant), so that noisy samples exist in the training set (with high probability).
>
> **Q2:** In our work, we assume the label flipping rate is bounded by a certain constant $1/C$ (Assumptions 3.1 and 4.1, Item 4) to ensure that the label flipping rate is relatively low so the noisy samples will not outweigh clean samples in deciding the convergence direction of model parameters. This assumption is typical in the study of benign overfitting with label flipping noise (e.g. [6, 12, 13]) and we follow this line of work to study it in attention mechanisms. We agree that increasing $\eta$ will add difficulty to achieve benign overfitting, and the SNR threshold might also grow. We leave it as an intriguing topic to study in the future.
>
> **Q3:** To further validate our theoretical findings, we conducted additional experiments on real-world datasets, including MNIST and CIFAR-10. In both cases, we trained a one-layer Transformer model ($d=1024$) to perform binary classification. Since the signal-to-noise ratio (SNR) is fixed for each dataset, we varied the training sample size $n$ to examine how train and test accuracy evolve with $n$. In both cases, the results indicate that while training accuracy remains near 100\%, the test accuracy improves as $n$ increases. This corresponds to our SNR threshold $\Theta(1/\sqrt{n})$ that determines the transition between benign and harmful overfitting.
>
> MNIST, $\eta=0.1$, 500 itr, 2 heads:
> | Sample size $n$ | 80    | 400   | 800   | 2000  |
> | --------------- | ----- | ----- | ----- | ----- |
> | Train accuracy  | 100%  | 100%  | 99.9% | 99.9% |
> | Test accuracy (on clean data)   | 87.6% | 91.9% | 94.1% | 95.8% |
>
> CIFAR-10, $\eta=0.05$, 500 itr, 4 heads:
> | Sample size $n$ | 40    | 400   | 4000  | 40000 |
> | --------------- | ----- | ----- | ----- | ----- |
> | Train accuracy  | 100%  | 100%  | 100%  | 100%  |
> | Test accuracy (on clean data)   | 77.6% | 80.9% | 86.6% | 88.9% |
>
> [1] Tarzanagh et al. Max-margin token selection in the attention mechanism. NeurIPS 23.
>
> [2] Vasudeva et al. "Implicit bias and fast convergence rates for self-attention."
>
> [3] Johannes Von Oswald et al. Transformers learn in-context by gradient descent.
>
> [4] Ahn et al. Transformers learn to implement preconditioned gradient descent for in-context learning.
>
> [5] Zhang et al. Trained transformers learn linear models in-context.
>
> [6] Sakamoto et al, I. Benign or not-benign overfitting in token selection of attention mechanism. In ICML 2025.
>
> [7] Jiang et al. Unveil benign overfitting for transformer in vision.
>
> [8] Cao et al. Benign overfitting in two-layer convolutional neural networks.
>
> [9] Wang et al. Benign overfitting in binary classification of gaussian mixtures.
>
> [10] Bartlett P.L. et al. Benign overfitting in linear regression.
>
> [11] Shang et al. Initialization Matters: On the Benign Overfitting of Two-Layer ReLU CNN with Fully Trainable Layers
>
> [12] Kou et al. Benign overfitting in two-layer relu convolutional neural networks
>
> [13] Meng et al. Benign overfitting in two-layer relu convolutional neural networks for xor data

---

> > ### Comment · Reviewer_zipP · 2025-08-01
> >
> > Thank you for the response. I have another question:
> >
> > What is the reason for $d \ge \tilde{\Omega}(n^2 / \eta^2)$ (Line. 172)?
> >
> > I understand that $d \ge \tilde{\Omega}(n^2)$ is enough to ensure $\Vert \xi_i \Vert_2^2 >> | \langle \xi_i, \xi_j \rangle |$ by Bernstein’s inequality, but what about the $1 / \eta^2$ factor.

---

> > > ### Author Response · Authors · 2025-08-03
> > >
> > > This technical assumption (the $1/\eta^2$ factor) explicitly appears in line 600 of our proof. Intuitively, the additional $1/\eta^2$ factor helps us to show that, after two iterations of gradient descent (GD), the attention can shift from primarily focusing on the signal token (as initially happens for both clean and noisy examples) to correctly emphasizing the noisy tokens for the noisy samples.
> > >
> > > More concretely, after one iteration of GD, attention focuses mostly on the signal token, both for clean and noisy samples, causing misclassification for noisy samples. To correct this behavior in the next iteration, the contribution of noisy samples, whose number scales with $\eta$, must become sufficiently significant. Therefore, as $\eta$ becomes smaller (fewer noisy samples), we require a larger dimension $d$ to amplify their effect, ensuring their contribution to gradients remains significant compared to other terms.
> > >
> > > Following your question, we carefully revisited our proof and found that the original assumption can be relaxed through a slightly more refined argument. Below, we describe the necessary adjustments, which are somewhat technical.
> > >
> > > In lines 580-607, we aimed to show that the attention vector at time $t=2$, denoted as $p_{t=2}$ focused on the noisy tokens for noisy samples. Specifically, for any noisy example $i$, we analyzed the quantity $p_2^{\top} (x_{j,\tau}-x_{j,1})$, which captures the attention difference attributed to the noisy tokens. In line 593, we lower bound this term by the sum of four components: two positive and two negative.
> > >
> > > In line 594, we conservatively dropped the first positive term and argued that the second positive term--scaling as $ \eta d c_{\rho}^2 $, where $c_{\rho}$ reflects the signal-to-noise ratio--dominates the two negative terms, which scale as $ M_c d c_{\rho}^2 $ and $ n \sqrt{d} $, respectively. Here, $ M_c $ is an upper bound on the loss for clean samples and is relatively small. To ensure that $ \eta d c_{\rho}^2 \gg n \sqrt{d} $, we previously required the assumption $ d = \tilde{\Omega}(n^2/\eta^2) $.
> > >
> > > To relax this assumption, we now utilize the first positive term that was previously omitted. As shown explicitly in line 593, this term scales with $ d $ and dominates the negative term $ n \sqrt{d} $ under the weaker condition $ d = \tilde{\Omega}(n^2) $. We will clarify this refinement in the revised version of our paper.
> > >
> > > We thank the reviewer for pointing this out and prompting us to refine this aspect of the analysis.

---

> > > > ### Comment · Reviewer_zipP · 2025-08-04
> > > >
> > > > Thank you for your detailed rebuttal and additional response! The responses address my concerns.
> > > >
> > > > This work studies benign overfitting for Transformer from a distinctive perspective compared to other works. I am glad to see the experiments on MNIST and CIFAR, and the relaxed assumption on $d$.
> > > >
> > > > I encourage the authors to include the experiments on real datasets and the relaxed assumptions in the revised version.
> > > >
> > > > I am updating the score from 3 to 4.

---

### Official Review · Reviewer_Ta2Q · 2025-06-28

**Clarity:** 4
**Significance:** 2
**Originality:** 3
**Rating:** 5
**Confidence:** 4

**Summary:**

This paper studies benign overfitting in a model for single head softmax attention. The attention layer is adapted to perform binary classification by adopting a vector valued value matrix and considering its sign. The data consists of $n$ points, or inputs, where input consists of $T$ vectors or tokens: one of these tokens correlates with an output label paired with the input while the rest are gaussian noise. The attention layer is trained by tuning the output value vector along with a prompt vector inside the softmax function and benign overfitting is established in a certain regime and under a variety of training processes.

**Questions:**

- Is there a typo where $W \in R^{T \times d}$ should be  $W \in R^{d \times T}$
- On the model:

     i) In equation 2 why do we suddenly ignore / remove the $S(XWX^T)$ term?

     ii) It seems a little odd to me that on the one hand you argue that $W$ and $q$ are fixed relative to one another and therefore you can simplify $Wq$ to $q$, but on the other you train $v$ and $q$ together. If $q$ is a prompt while $W$ and $v$ are parameters then does it not seem more relevant perhaps to consider them trained via separate processes?

- Assumptions

   i) The required dimension of the tokens seems very large, i.e., $d$ needs to scale like $n^2$. What is the reason for this in this setting and could it be relaxed? For example, in some of the works you cite for linear classifiers and ReLU networks then analogous results can be proved for $d$ scaling like $n$.

  ii) What breaks down if the number of tokens is not bounded by a constant? Intuitively a larger $T$ feels like it would help one achieve benign overfitting?

- General comments: for any data set you might sample is it always true that a solution pair $(v,q)$ exist that solve the constraints of your optimization problem?

- Theorem 3.3

   i) Item 3 confuses me a bit: shouldn't for clean data the result imply that the attention mass is focused on the first token while for the noisy data points the mass on the first token should be very small and the mass distributed amongst the other near orthogonal tokens? For large $T$ I don't see how the result states this. Maybe this is why you only allow for a few tokens?

  ii) What happens if you keep training for more steps?


- Can you comment on any differences between benign overfitting across the different models people have studied, i.e., linear, shallow feedforward, CNN and single head attention? In particular, are some models in some sense more or less biased towards the benign overfitting in some or any sense you can identify?

**Ethical Concerns:**

["NO or VERY MINOR ethics concerns only"]

**Final Justification:**

Overall, and despite the limitations with regards to the setting and assumptions, I think this paper is a good early step towards a better mathematical understanding of how benign overfitting can occur in attention.

**Limitations:**

Not applicable.

**Paper Formatting Concerns:**

None noticed.

**Quality:**

4

**Strengths And Weaknesses:**

The paper is well written and of high quality. The assumptions made are clearly stated and remarks are provided throughout to add interpretation and highlight key aspects of the work, as such I rate the clarity also highly. The technical contributions are the study of benign overfitting in the context of an attention layer, which to the best of my knowledge is new, but under pretty standard data assumptions. Therefore, and although not exceptional, I think the originality of the paper is good. Overall I think this is a good technically solid paper. If I had to identify the weakest dimension of the paper I might say significance: in particular, the paper seems to adapt single head attention to binary classification in a manner which feels a little contrived. The wider implications of the results also do not seem particularly well discussed.

---

> ### Author Rebuttal · Authors · 2025-07-31
>
> We thank the reviewer for their thoughtful feedback. Below, we respond to each concern and question in order.
>
> **Q1:** Thank you for pointing this out. There is indeed a typo in the expression, although it should be $W \in \mathbb{R}^{d \times d}$. We will fix it in our revised version.
>
> **Q2:**
>
> **2.1:** The output of the softmax function in Eq. 1 is a matrix in $\mathbb{R}^{(T+1)\times(T+1)}$, representing interactions between all pairs of tokens. However, for classification (or regression) purposes, the model must produce a scalar as an output. To achieve this, both in theory and practice (e.g. [1], see Figure 1), it is common to introduce a classification token (CLS token or prompt), whose corresponding output is used for prediction. Thus, in Eq. 2, we focus on the row corresponding to the "classification" token, which leads directly to Equation 3.
>
> [1] Dosovitskiy et al. "An image is worth 16x16 words: Transformers for image recognition at scale."
>
> **2.2:** If q is a fixed vector (i.e. corresponds to a fixed prompt), then only $W$ and $v$ need to be trained. In this case, we follow the approach of [2], which showed that optimizing W with a fixed q is equivalent to optimizing q and removing W, as done in our paper. We note that in Section 3 of that paper, they are optimizing both v and q, exactly as we do. We agree that this approach can be viewed as a kind of simplification.
>  We also note that our model in Eq. 3 can be obtained by fixing $\mathbf{q}=(1,0,\ldots,0)^\top$ in Eq. 2, and then the vector $\mathbf{p}$ from Eq. 3 is simply the first column of $W$ in Eq. 2.
>
> [2] Tarzanagh et al. Max-margin token selection in attention mechanism.
>
> **Q3:**
>
> **Q3.1:**  The $\Omega(n^2)$ dimension requirement arises from our need to ensure sufficient separation between token inner products and large dimension will help the model to overfit all training data more easily. This assumption is crucial for our theoretical analysis of token selection and margin gaps, and is also common in prior works on benign overfitting (e.g., [3, 4, 5]).
>
> Nevertheless, we acknowledge that refining the lower bound on $d$ with respect to $n$ is an interesting direction, as explored in works like [6]. However, such relaxations are typically more tractable in linear models or ReLU networks, where feature mappings are fixed and the inductive biases are simpler. In contrast, the softmax attention mechanism involves data-dependent and dynamically evolving token weights, making the analysis more delicate. We have also done relevant experiments (see Figure 4 in the Appendix) and note that benign overfitting occurs for $d = 2n \ll n^2$, which suggests that the assumptions on $d$ in our theorems are loose.
> We view our current dimensionality assumption as a step toward a better understanding of benign overfitting in attention models. Investigating whether this condition can be relaxed while preserving the phenomenon would be a valuable direction for future work.
>
> [3] Frei S., Chatterji N., Bartlett P. Benign Overfitting without Linearity: Neural Network Classifiers Trained by Gradient Descent for Noisy Linear Data.
>
> [4] Kou Y., Chen Z., Chen Y., Gu Q. Benign overfitting in two-layer relu convolutional neural networks.
>
> [5] Jiang J., Huang W., Zhang M., Suzuki T., Nie L. Unveil benign overfitting for transformer in vision.
>
> [6] Karhadkar K., George E., Murray M., Montúfar G., Needell D.
> Benign overfitting in leaky ReLU networks with moderate input dimension.
>
> **Q3.2:** While increasing $T$ introduces more tokens to interpolate, it also raises the risk of the attention mechanism focusing on high-scoring noise tokens due to softmax sensitivity. Besides, the definition of signal-to-noise ratio also needs to be redefined when the token number increases dramatically. In our current analysis, we fix $T=\Theta(1)$ to ensure analytical tractability to study the token-selection mechanism in the attention model. Extending our results to the large-$T$ regime, where a phase transition might occur, is an interesting direction for future work.
>
> **Q4:** In our data model and architecture, with high probability over the training data, we can construct a pair $(v, p)$ that interpolates the training data and satisfies our constraints, and we even showed that GD reaches such a solution (Thm. 3.3).
>
> **Q5:**
>
> **5.1:** Item 3 in Thm. 3.3 shows that the attention mass for noisy examples is focused on the noisy tokens as expected. We therefore believe that the reviewer's concern is more about the clean examples, which indeed yield a relatively small lower bound on the softmax probability of the first token, especially when the number of tokens $T$ is large. This lower bound is actually more closely related to the SNR threshold. Our proof suggests that for clean examples, when the SNR is large (i.e. $c_\rho$ is large), the attention head $v$ has a significant component in the direction of the first (signal) token, compared to the direction of the noisy tokens. In such cases, even if the softmax probability of the signal token is not very large, the model still prioritizes the signal token. Conversely, when the SNR is smaller, we can obtain a stronger lower bound on the softmax probability of the first token for clean examples, as stated in Remark 3.4 and formally proved in the appendix.
>
> **5.2:** While our current analysis of GD  is limited to two steps, we believe that benign overfitting persists well beyond this point: First, our experimental results suggest that the theoretical findings extend to longer training, showing consistent behavior with our analysis even after many GD steps.
>
> Second, we proved that min-norm/max-margin solutions also exhibit benign overfitting, when these solutions roughly correspond to the convergence of GD with (a small) weight decay. Indeed, it is natural to expect that GD with weight decay tends (after enough iterations) to a solution that minimizes the loss while preserving a small weight norm. We also refer the reviewer to lines 232-238, which provide additional motivation for analyzing min-norm/max-margin solutions.
>
> Taken together, these points reinforce our belief that benign overfitting persists beyond the two-step regime.
>
> **Q6:**
> - **Linear vs single-head attention**: Benign overfitting was first studied in the context of linear models (Bartlett et al.[7]). However, for linear models and MLPs the data has a different structure (without tokens) and the implicit bias is better understood in these cases.
>
> - **CNN vs single-head attention:** Compared with two-layer CNN with ReLU$^q$ activation (Cao et al.[8]), when the signal-to-noise ratio is small (SNR$\le 1$), our results show that the single-head attention model requires a smaller number of samples to achieve benign overfitting, which reflects the advantage of the attention mechanism.
> Compared with two-layer CNN with ReLU activation (Kou et al.[9]), they only analyze the case where SNR is small ($\|\mu\|/\sqrt{d} = O(1/n^{1/2})$) as they impose the lower bound of $d$ is related with signal strength $\|\mu\|$. So their results cannot apply to the case when SNR increases while our results show that attention model can generalize well without an upper bound for SNR due to its token-selection mechanism.
>
>
> [7] Bartlett P.L., Long P.M., Lugosi G., Tsigler A. Benign overfitting in linear regression.
>
> [8] Cao Y., Chen Z., Belkin M., Gu Q. Benign overfitting in two-layer convolutional neural networks
>
> [9] Kou Y., Chen Z., Chen Y., Gu Q. Benign overfitting in two-layer relu convolutional neural networks

---

> > ### Comment · Reviewer_Ta2Q · 2025-08-04
> > **Thanks for answering my questions**
> >
> > I will keep my current score.

---

### Official Review · Reviewer_wMV9 · 2025-06-29

**Clarity:** 2
**Significance:** 2
**Originality:** 3
**Rating:** 4
**Confidence:** 3

**Summary:**

This paper investigates benign overfitting in a single-head softmax attention model trained with noisy labels. The authors prove that when the signal-to-noise ratio is $\Theta(1/\sqrt{n})$, the model can interpolate noisy training data while generalizing well to test data. This is shown both for early-stage gradient descent after two iterations and for max-margin solutions. They provide tight conditions on the SNR for benign versus harmful overfitting and support the theory with synthetic experiments.

**Questions:**

1. Can you extend the gradient descent analysis beyond two iterations? Does it eventually converge to the max-margin solution, or does it behave differently?
2. In the multi-token setting (e.g., multiple noise tokens), how robust is the token selection pattern? Does the model consistently attend to one dominant noise token for noisy samples, or does attention fragment across them?
3. What fundamentally distinguishes the attention model in your setup from a standard linear classifier trained on convex combinations of signal and noise tokens?

**Ethical Concerns:**

["NO or VERY MINOR ethics concerns only"]

**Final Justification:**

This paper provides a clear and rigorous theoretical study of benign overfitting in single-head attention models under label noise, supported by synthetic experiments. While the setting is simplified, the work addresses an underexplored architecture in the benign overfitting literature and offers interpretable conditions on the signal-to-noise ratio. The rebuttal clarified the technical challenges in extending the analysis to more iterations and in handling richer multi-token dynamics, and helped distinguish the role of attention from simpler linear models. Although the scope remains somewhat limited, the contribution is meaningful for understanding overfitting behavior in attention mechanisms, and the technical execution is solid. I am therefore raising my score to 4.

**Limitations:**

This paper offers a rigorous and technically interesting analysis of benign overfitting in single-head softmax attention models under noise. However, the theoretical results are limited to early-stage training and rely on strong simplifying assumptions about both data and architecture. While the paper improves our theoretical understanding, its insights into realistic attention models remain preliminary. Stronger justification of the attention mechanism's unique role, extended training dynamics, or theoretical treatment of more general token structures would significantly enhance the impact of this work.

**Paper Formatting Concerns:**

No.

**Quality:**

2

**Strengths And Weaknesses:**

Strengths:
1. The paper addresses benign overfitting in attention-based models, a relatively underexplored setting compared to linear regression, kernel models, and CNNs. This aligns with growing interest in understanding generalization in Transformers.

Weaknesses:
1. The theoretical results for gradient descent stop after just two iterations, and do not analyze convergence behavior or connection to the max-margin solution. As a result, the relationship between early dynamics and implicit bias remains unclear.
2. Although each input has $n$ tokens, only one is a signal token and the rest are orthogonal noise. In clean samples, attention rapidly concentrates on the signal; in noisy samples, it shifts to a single dominant noise token. This dynamic resembles a two-mode selection process, so the attention mechanism effectively reduces to selecting between signal and noise, limiting the richness of attention interactions typically seen in real-world applications.
3. Signal and noise vectors are assumed orthogonal, noise tokens are Gaussian, and all vectors have fixed norms. These stylized assumptions are typical in theoretical work on overparameterized models but limit practical relevance.
4. While the authors emphasize that softmax attention leads to token selection, it remains somewhat ambiguous how this differs fundamentally from a model that linearly interpolates signal and noise under logistic loss. In the current setup, the attention head largely acts as a soft selector, and it’s unclear what inductive bias is unique to attention beyond this selection.

---

> ### Author Rebuttal · Authors · 2025-07-31
>
> We thank the reviewer for their thoughtful feedback. Below we respond to each concern and question in order.
>
> **W1:** See the answer for Q1.
>
> **W2:** Indeed, under our setup, the attention mechanism exhibits a two-mode behavior, selecting between signal and noise tokens in clean and noisy samples respectively.
> This token-selection behavior is a reflection of an important inductive bias in softmax attention. It aligns with recent empirical findings in real-world models, where attention often becomes highly sparse or concentrates on a few dominant tokens([1, 2, 3]).
>
> Besides, the real-world attention settings are highly complex and currently out of reach for comprehensive theoretical analysis. We were inspired by image data to design this simplified but still representative data model to capture essential aspects of token selection while remaining analytically tractable. Similar distributions are common in theoretical studies of the benign overfitting phenomenon (see [4, 5, 6, 7])
>
> Finally, even in this simplified setting, the two-mode behavior poses significant technical challenges. Our analysis requires careful handling of randomness across noise tokens and a detailed understanding of how parameter norms relate to generalization in the overparameterized regime. While we agree that richer token interactions are important in practical applications, our focus here is to isolate and understand this core mechanism. We believe that this is a meaningful step toward more general theoretical frameworks, and we view extending the analysis to broader data distributions as an important direction for future work.
>
>
> [1] Ataee Tarzanagh D., Li Y., Zhang X., Oymak S. Max-margin token selection in attention mechanism
>
> [2] Jelassi S., Sander M., Li Y. Vision transformers provably learn spatial structure.
>
> [3] Wang Z., Wei S., Hsu D., Lee J. Transformers Provably Learn Sparse Token Selection While Fully-Connected Nets Cannot
>
> [4] Cao Y., Chen Z., Belkin M., Gu Q. Benign overfitting in two-layer convolutional neural networks
>
> [5] Kou Y., Chen Z., Chen Y., Gu Q. Benign overfitting in two-layer relu convolutional neural networks
>
> [6] Meng X., Zou D., Cao Y. Benign overfitting in two-layer relu convolutional neural networks for xor data
>
> [7] Jiang J., Huang W., Zhang M., Suzuki T., and Nie L. Unveil benign overfitting for transformer in vision: Training dynamics, convergence, and generalization.
>
> **W3:** We adopt this specific signal–noise data distribution to enable a clear and tractable theoretical analysis. This setup is motivated by image data, where inputs are composed of distinct tokens and only some of these tokens are related to the image's class label.
>
> As the reviewer mentioned, such a distribution is typical in theoretical studies on benign overfitting in convolutional networks (where the input has several patches) and transformers (where the input has several tokens). For convolutional neural networks (CNN), we are aware of three papers that studied benign overfitting [8,9,10], and all of them consider a data distribution where there is a single signal patch and a single noise patch, and the signal is either $\pm \mu$ for some vector $\mu$ [8,9], or has an XOR pattern [10]. Moreover, in [8,10] the noise patch is exactly orthogonal to the signal patch. The prior work [11], which studied benign overfitting in transformers (albeit without label noise), considered a distribution similar to ours: one signal token and several noise tokens, where the signal is either $\mu_+$ or $\mu_{-}$ for orthogonal $\mu_+,\mu_-$, and the noise tokens are orthogonal to the signal.
> Hence, in our paper we follow a common setting, which is also the focus of previous works on related topics.
>
> [8] Cao Y., Chen Z., Belkin M., Gu Q. Benign overfitting in two-layer convolutional neural networks
>
> [9] Kou Y., Chen Z., Chen Y., Gu Q. Benign overfitting in two-layer relu convolutional neural networks
>
> [10] Meng X., Zou D., Cao Y. Benign overfitting in two-layer relu convolutional neural networks for xor data
>
> [11] Jiang J., Huang W., Zhang M., Suzuki T., and Nie L. Unveil benign overfitting for transformer in vision: Training dynamics, convergence, and generalization.
>
> **W4:** See the Answer for Q3.
>
> **Q1:** In general, there is no known characterization of the implicit bias in softmax transformers that applies to our setting, so it remains unclear whether GD converges to the max-margin solution. This is indeed a very interesting open direction. We note that min-norm/max-margin solutions roughly correspond to the convergence of GD with (a small) weight decay. Indeed,
> GD with weight decay tends to prefer solutions that minimize the loss while preserving a small weight norm.
> We also refer the reviewer to lines 232-238, which provide additional motivation for analyzing min-norm/max-margin solutions, even in settings where the implicit bias of GD does not necessarily lead to a min-norm solution (e.g., Savarese et al. [28], Ongie et al. [29], Ergen and Pilanci [30], Hanin [31], Debarre et al. [ 32], Boursier and Flammarion [33]).
>
> While we don't know how to extend the analysis beyond a constant number of steps. We note that analyzing even two steps of GD in our setting is challenging. This analysis gives us many insights into how attention handles noisy label data, how the attention focuses on the signal token for clean samples and on the noisy tokens for noisy samples, sufficient and necessary conditions to allow benign overfitting, and the relation between different parameters. It also provides insights into how the trajectory looks after more steps of GD (see Remark 5.1, line 371), which we validate in our experiments (Figure 3 in the appendix). Our experimental results suggest that our theoretical findings extend beyond the two-step setting, demonstrating consistency with longer training and a broader range of learning rates. While a full trajectory analysis is an interesting direction, our results already highlight fundamental phenomena.
>
> **Q2:** In our current setup, even with multiple noise tokens, attention mechanism consistently concentrates on a single dominant noise token in the noisy samples. This is due to the softmax amplification and randomness in token scores. Since all noise tokens are sampled independently from the same Gaussian distribution, their dot products with the learned query vector exhibit random fluctuations. As a result, one noise token typically achieves a noticeably higher score than the others, and the softmax function amplifies this gap exponentially, which leads to sharply concentrated attention on that token.
>
> However, which specific noise token is selected does not significantly affect generalization, as long as the model avoids assigning high attention weight to the signal token in noisy samples. Therefore, in our theoretical analysis, we focus on the aggregate attention weight assigned to all noise tokens, rather than tracking the exact distribution across them. This abstraction allows for a cleaner and more interpretable characterization of benign overfitting behavior in the attention model.
>
> **Q3:** First, we note that our motivation in this work is to understand benign overfitting in attention mechanisms. Thus, our focus is on analyzing the behavior of our model, rather than understanding whether the same data distribution is also learnable with other models.
>
> Regarding the question, while the attention output is indeed a convex combination of token embeddings, our model differs fundamentally from a standard linear classifier trained on convex combinations of signal and noise tokens in several key ways.
>  First, in our attention model, the combination weights (attention scores) are learned via a softmax over dot products, and are therefore **data-dependent and nonlinear** in the output. This differs from a linear classifier, where the convex combination is fixed and independent of the data. Second, the attention model does not assume that the position of the signal token is fixed. Hence, it is more flexible compared to non-attention models like linear models and MLPs.

---

> > ### Comment · Reviewer_wMV9 · 2025-08-05
> >
> > Thank you to the authors for the detailed response. I now have a clearer understanding of the challenges involved in extending the gradient descent analysis beyond two steps. While such an extension would strengthen the connection between training dynamics and benign overfitting, I acknowledge the technical difficulty and appreciate the partial results provided. The authors also clarified the multi-token setting and the role of attention in token selection under noise, which helped address some concerns I had about the effective simplicity of the model. Based on these clarifications, I am raising my score to 4 (borderline accept).

---

> > > ### Author Response · Authors · 2025-08-07
> > >
> > > Thank you for your positive feedback and for raising the score! We are happy to provide further clarifications if there are any additional questions.

---

### Official Review · Reviewer_pRmd · 2025-07-03

**Clarity:** 2
**Significance:** 2
**Originality:** 3
**Rating:** 5
**Confidence:** 3

**Summary:**

The authors study benign overfitting for a single softmax attention model. Using a specific data distribution (Section 2.1), a simplified attention model (Section 2.2)  and specific assumptions (Assumptions 3.1) which include a given signal-to-noise ratio, they show that benign overfitting occurs at the 2nd iteration of gradient descent. This is the main result and verified empirically for synthetic data from the specified data distribution. Complementary results for a max-margin learning process are developed and proven.

**Questions:**

- To what extent does the studied data distribution restrict the value and significance of the findings?
- The simplified attention model includes a combined key-query. Given that having these separate or combined displayed different optimization characteristics in [5], is this choice worth commenting on in the context of the current paper?

**Ethical Concerns:**

["NO or VERY MINOR ethics concerns only"]

**Final Justification:**

This is a step towards understanding benign overfitting in attention models. Theoretical results are provided for a restricted setting, but this is supplemented with empirical results in a more relaxed setting. A complex problem is tackled carefully and methodically.

**Limitations:**

yes

**Quality:**

3

**Strengths And Weaknesses:**

Strengths:
- This seems to be a solid theoretical analysis of benign overfitting in the studied context.
- New perspectives on the conditions required for benign overfitting to occur for a single attention head.
- Empirical results confirming theoretical predictions for the specific setting: both in the main paper, and the additional results in the appendix, Section A.4.

Weaknesses:
- In addition to the assumptions listed in 3.1. which can be considered ‘appropriate conditions’, the specific data distribution studied is very specific: two signals orthogonal, and noise also orthogonal with the signal. While this is a valid setting to investigate, it is not clear from the paper how this restricts the applicability of the analysis, or how significant the findings are. The study context itself can be better motivated.
- 74 pages of supplementary material is excessive – very little of this was reviewed.
- I struggled to follow the proofs, and was unable to do so in the time available.

Minor comments:
- Fig 2: check top/bottom vs left/right panels; ‘exsists’ line 487
- Even if obvious, please consider adding the longer form of abbreviations on first use (GD, SVM etc).

---

> ### Author Rebuttal · Authors · 2025-07-31
>
> We thank the reviewer for their thoughtful feedback. Below we respond to each concern and question in order.
>
> **W1**: We adopt this specific signal–noise data distribution to enable a clear and tractable theoretical analysis. This setup is motivated by image data, where inputs are composed of distinct tokens and only some of these tokens are related to the image's class label.
>
> Such a distribution is typical in theoretical studies on benign overfitting in convolutional networks (where the input has several patches) and transformers (where the input has several tokens). For convolutional neural networks (CNN), we are aware of three papers that studied benign overfitting [1,2,3], and all of them consider a data distribution where there is a single signal patch and a single noise patch, and the signal is either $\pm \mu$ for some vector $\mu$ [1,2], or has an XOR pattern [3]. Moreover, in [1,3] the noise patch is exactly orthogonal to the signal patch. The prior work [4], which studied benign overfitting in transformers (albeit without label noise), considered a distribution similar to ours: one signal token and several noise tokens, where the signal is either $\mu_+$ or $\mu_{-}$ for orthogonal $\mu_+,\mu_-$, and the noise tokens are orthogonal to the signal.
> Hence, in our paper we follow a common setting, which is also the focus of previous works on related topics.
>
> [1] Cao Y., Chen Z., Belkin M., Gu Q. Benign overfitting in two-layer convolutional neural networks
>
> [2] Kou Y., Chen Z., Chen Y., Gu Q. Benign overfitting in two-layer relu convolutional neural networks
>
> [3] Meng X., Zou D., Cao Y. Benign overfitting in two-layer relu convolutional neural networks for xor data
>
> [4] Jiang J., Huang W., Zhang M., Suzuki T., and Nie L. Unveil benign overfitting for transformer in vision: Training dynamics, convergence, and generalization.
>
> **Q1:** see the Answer for W1
>
> **Q2:**  We chose to combine the key and query representations to simplify the analysis, which is already technically involved. We note that this simplification is common in prior works studying attention mechanisms. For example, [6,7,8,9,10] (as well as many other papers) also used a combined key-query matrix.
>
> As the reviewer noted, Tarzanagh et al. [5] explore optimization behavior under separate key and query matrices, but their analysis focuses only on the optimization of the softmax weights, with the attention head (denoted by $v$ in both papers) held fixed. In contrast, our setup involves optimizing both the softmax weights and the attention heads, so their technique does not directly apply to our setting. We will add a comment on this issue in the final version.
>
> [5] Davoud Ataee Tarzanagh, Yingcong Li, Christos Thrampoulidis, and Samet Oymak. Transformers as support vector machines.
>
> [6] Davoud Ataee Tarzanagh, Yingcong Li, Xuechen Zhang, and Samet Oymak. Max-margin token selection in the attention mechanism.
>
> [7] Vasudeva, Bhavya, Puneesh Deora, and Christos Thrampoulidis. "Implicit bias and fast convergence rates for self-attention."
>
> [8] Johannes Von Oswald et al. Transformers learn in-context by
> gradient descent.
>
> [9] Kwangjun Ahn, Xiang Cheng, Hadi Daneshmand, and Suvrit Sra. Transformers learn to implement preconditioned gradient descent for in-context learning.
>
> [10] Ruiqi Zhang, Spencer Frei, and Peter L. Bartlett. Trained transformers learn linear models in-context.

---

> > ### Comment · Reviewer_pRmd · 2025-08-06
> >
> > Thank you for the clarification - the choice for the specific data distribution and setup is now better motivated. I am still curious: while this is clearly a valid setting, do you think your results are only specifically valid if these assumptions hold? I understand that you are able to prove your finding for a very specific setting. Is your expectation that similar trends will hold in more relaxed settings? Or are these expected to be restricted to your analysis setup?

---

> > > ### Author Response · Authors · 2025-08-07
> > >
> > > Thank you for the follow-up question. We do believe that the core phenomena identified in our work extend beyond the specific assumptions used in our theoretical analysis. In Appendix A.4, we present a series of experiments:
> > >
> > > - Using various architectures, such as self-attention, multi-head attention, and multi-layer attention (Figures 5,9,10), we show that the models still exhibit benign overfitting. We find that all of these models behave similarly to our single-head attention; in particular, the softmax probabilities corresponding to the CLS token and optimization dynamics align with the behavior observed in the single-head attention.
> > >
> > > - We also conduct an experiment using GD with weight decay, which encourages norm minimization (Figure 11), and a small step size experiment (Figure 3).  In both cases, we observe results consistent with our main findings.
> > >
> > >
> > > - Additionally, we present experiments with alternative relationships between parameters. For instance, regarding the dimension constraint $d = \Omega(n^2)$ used in our proofs (which is a common assumption in theoretical benign overfitting results, as we discuss in lines 178-183), we conduct experiments with much smaller dimensions (see Figure 4) and note that benign overfitting occurs for $d = 2n \ll n^2$. This suggests that the high-dimensionality assumption in our theory is conservative, and that the phenomenon is robust to relaxations in this and other modeling choices.
> > >
> > > Finally, in our response to Reviewer zipP (specifically, see question 3), we provide empirical results on the MNIST and CIFAR-10 datasets.
> > >
> > > We view our current assumptions and results as a step towards providing rigorous proofs for understanding benign overfitting in attention models. Investigating whether these conditions can be relaxed while preserving the phenomenon would be a valuable direction for future work.

---

> > > > ### Comment · Reviewer_pRmd · 2025-08-09
> > > >
> > > > Got it - thanks. It would be interesting to see how this work develops. Raising my score from a 4 to a 5.

---

### Decision · Program_Chairs · 2025-09-17

**Decision:**

Accept (poster)

**Comment:**

This paper studies
benign overfitting in a single-head softmax attention model.

The reviewers appreciate the **strengths** of the paper:
(pRmd) solid theoretical analysis, new perspectives on the conditions, empirical results confirming theoretical predictions,
(wMV9) addresses a relatively under explored setting (attention-based models), aligns with growing interest in Transformers,
(Ta2Q) well-written, clearly stated assumptions, originality is good, technically solid,
(zipP) rigorous theoretical analysis verified by the experiments, clear writing, well-structured (assumptions and theorem).

The reviewers also find the **weaknesses** of the paper:
(pRmd) specific data distribution,
(wMV9) theoretical results for GD stop after just two iterations, no convergence analysis or connection to the max-margin solution, (binary signal/noise settings) limiting the richness of attention interaction
(Ta2Q) weak significance (binary signal/noise settings),
(zipP) oversimplified assumption, lack of training dynamics, not novel.


**After the discussion**, many concerns seem resolved and clarified (e.g, experiment on MNIST/CIFAR; relaxed assumption on $d$), and the reviewers agreed upon accepting the paper as it is a good early step despite the limited setting and assumptions.